# ON THE FEATURE LEARNING IN DIFFUSION MODELS

**Andi Han**[*]     **Wei Huang**[*]     **Yuan Cao**[†]     **Difan Zou**[‡]

[*]RIKEN AIP (`andi.han@riken.jp`, `wei.huang.vr@riken.jp`). Equal contribution.
[†]Department of Statistics and Actuarial Science, University of Hong Kong (`yuancao@hku.hk`)
[‡]Department of Computer Science and Institute of Data Science, University of Hong Kong
(`dzou@cs.hku.hk`)

## ABSTRACT

The predominant success of diffusion models in generative modeling has spurred significant interest in understanding their theoretical foundations. In this work, we propose a feature learning framework aimed at analyzing and comparing the training dynamics of diffusion models with those of traditional classification models. Our theoretical analysis demonstrates that diffusion models, due to the denoising objective, are encouraged to learn more balanced and comprehensive representations of the data. In contrast, neural networks with a similar architecture trained for classification tend to prioritize learning specific patterns in the data, often focusing on easy-to-learn components. To support these theoretical insights, we conduct several experiments on both synthetic and real-world datasets, which empirically validate our findings and highlight the distinct feature learning dynamics in diffusion models compared to classification.

## 1 INTRODUCTION

Diffusion models (Ho et al., 2020; Song et al., 2021) have emerged as a powerful class of generative models for content synthesis and have demonstrated state-of-the-art generative performance in a variety of domains, such as computer vision (Dhariwal & Nichol, 2021; Peebles & Xie, 2023), acoustic (Kong et al., 2021; Chen et al., 2021) and biochemical (Hoogeboom et al., 2022; Watson et al., 2023). Recently, many works have employed (pre-trained) diffusion models to extract useful representations for tasks other than generative modelling, and demonstrated surprising capabilities in classical tasks such as image classification with little-to-no tuning (Mukhopadhyay et al., 2023; Xiang et al., 2023; Li et al., 2023a; Clark & Jaini, 2024; Yang & Wang, 2023; Jaini et al., 2024). Compared to discriminative models trained with supervised learning, diffusion models not only are able to achieve comparable recognition performance (Li et al., 2023a), but also demonstrate exceptional out-of-distribution transferablity (Li et al., 2023a; Jaini et al., 2024) and improved classification robustness (Chen et al., 2024c).

The significant representation learning power suggests diffusion models are able to extract meaningful features from training data. Indeed, the core of diffusion models is to estimate the data distribution through progressively denoising noisy inputs over several iterative steps. This inherently views data distribution as a composition of multiple latent features and therefore learning the data distribution corresponds to learning the underlying features. Nevertheless, it remains unclear

*how feature learning emerges during the training of diffusion models and whether the feature learning process is different to supervised learning.*

Regardless of the ground-breaking success of diffusion models, the theoretical understanding is still in its infancy. Existing analysis on diffusion models has mostly focused on theoretical guarantees in terms of distribution estimation and sampling convergence. Several works have derived statistical estimation errors between distribution generated by diffusion models to ground-truth distribution (Oko et al., 2023; Zhang et al., 2024; Chen et al., 2023a), showing that diffusion models achieve a minimax optimal rate under certain assumptions on the true density (Oko et al., 2023; Zhang et al., 2024). Algorithmically, Li et al. (2023c); Han et al. (2024) studied the estimation error of diffusion models trained with gradient descent using kernel methods. Shah et al. (2023); Gatmiry et al. (2024);

Chen et al. (2024d) introduced algorithms based on diffusion models for learning Gaussian mixture models. In addition, given access to sufficiently accurate score estimation, Lee et al. (2022; 2023); Chen et al. (2023b); Li et al. (2023b) proved the convergence guarantees of sampling in (score-based) diffusion models. Despite showing provable guarantees for diffusion models, existing theories are limited to the generative aspects of diffusion models, namely distribution learning and sampling. To the best of our knowledge, *no theoretical analysis* is performed to elucidate the *feature learning process* in diffusion models.

**Notations.** We make use of the following notations throughout the paper. We use $\|\cdot\|$ to denote $L_2$ norm for vectors and Frobenius norm for matrices, unless mentioning otherwise. We use $O(\cdot), \Omega(\cdot), \Theta(\cdot), o(\cdot), \omega(\cdot)$ for the big-O, big-Omega, big-Theta, small-o, small-omega notations. We write $\widetilde{O}(\cdot)$ to hide (poly)logarithmic factors and similar notations hold for $\widetilde{\Omega}(\cdot)$ and $\widetilde{\Theta}(\cdot)$. For a binary condition $\mathcal{C}$, we let $\mathbb{1}(\mathcal{C}) = 1$ if $\mathcal{C}$ is true and $\mathbb{1}(\mathcal{C}) = 0$ otherwise.

## 1.1 OUR MAIN RESULTS

In this work, we develop a theoretical framework that studies feature learning dynamics of diffusion models and compares with classification. Inspired by the image data structure, we employ a multi-patch data distribution $\mathbf{x} = [\boldsymbol{\mu}_y, \boldsymbol{\xi}]$ for both classification and diffusion model training. We consider a binary-class data setup with $y = \pm 1$ as the data label and $\boldsymbol{\mu}_1, \boldsymbol{\mu}_{-1} \in \mathbb{R}^d$ are two fixed orthogonal vectors, i.e., $\boldsymbol{\mu}_1 \perp \boldsymbol{\mu}_{-1}$, representing the signal. On the other hand, $\boldsymbol{\xi}$ is the label-independent noise, which is randomly sampled from a Gaussian distribution with standard deviation $\sigma_\xi$.

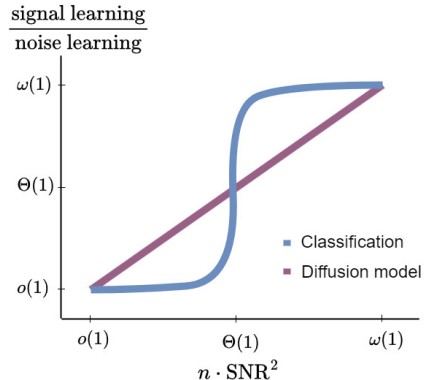

Figure 1: Illustration of the ratio of signal learning to noise learning when varying $n \cdot \mathrm{SNR}^2$, where $\mathrm{SNR} := \|\boldsymbol{\mu}\|/(\sigma_\xi\sqrt{d})$. We show diffusion model tends to study more balanced signal and noise while classification has a sharp phase transition and tends to focus on learning either signal or noise.

In order to elucidate the difference of feature learning dynamics for the two tasks, we adopt a *two-layer convolutional neural network* with *quadratic activation*. For diffusion model, we consider a weight-sharing setting for the first and second layer, which is commonly considered for analyzing autoencoders (Nguyen, 2021; Cui & Zdeborová, 2024). For classification, we fix the second layer weights to be $\pm 1$, following Cao et al. (2022); Kou et al. (2023). In other words, the classifier can be viewed as attaching a fixed linear head to the intermediate layer of the diffusion model. Given a training dataset of $n$ samples from the multi-patch data distribution, we use *gradient descent* to minimize the empirical logistic loss for classification and the DDPM loss (Ho et al., 2020) with expectation over the diffusion noise.

Under the above settings, we investigate the differences of feature learning dynamics (Allen-Zhu & Li, 2023; Cao et al., 2022; Zou et al., 2023; Huang et al., 2023b; Jiang et al., 2024; Huang et al., 2024a; Lu et al., 2024; Meng et al., 2024) between diffusion model and classification. We quantify the feature learning in terms of signal learning and noise learning, measured through the alignment between the network weights $\mathbf{w}$ to the directions of signal/noise respectively, i.e., $|\langle \mathbf{w}, \boldsymbol{\mu}_y \rangle|, |\langle \mathbf{w}, \boldsymbol{\xi} \rangle|$. We present the following (informal) results for the two learning paradigms.

**Theorem 1.1** (Informal). *Let* $\mathrm{SNR} := \|\boldsymbol{\mu}\|/(\sigma_\xi\sqrt{d})$ *be the signal-to-noise ratio. We can show*

- *For **diffusion model**, $|\langle \mathbf{w}, \boldsymbol{\mu}_y \rangle|, |\langle \mathbf{w}, \boldsymbol{\xi} \rangle|$ exhibit linear growth initially and there exists a stationary point along the path of the training dynamics that satisfies $|\langle \mathbf{w}, \boldsymbol{\mu}_y \rangle|/|\langle \mathbf{w}, \boldsymbol{\xi} \rangle| = \Theta(n \cdot \mathrm{SNR}^2)$.*

- *For **classification**, $|\langle \mathbf{w}, \boldsymbol{\mu}_y \rangle|, |\langle \mathbf{w}, \boldsymbol{\xi} \rangle|$ exhibit exponential growth initially and when $n \cdot \mathrm{SNR}^2 \geq \beta$ for some constant $\beta > 1$, $|\langle \mathbf{w}, \boldsymbol{\mu}_y \rangle|/|\langle \mathbf{w}, \boldsymbol{\xi} \rangle| = \omega(1)$, and when $n \cdot \mathrm{SNR}^2 < 1/\beta$, $|\langle \mathbf{w}, \boldsymbol{\mu}_y \rangle|/|\langle \mathbf{w}, \boldsymbol{\xi} \rangle| = o(1)$.*

Theorem 1.1 highlights differences in the feature learning process between diffusion models and classification. Especially in the regime where $n \cdot \mathrm{SNR}^2 = \Theta(1)$, classification is sensitive to changes in SNR and tends to learn either the signal $\boldsymbol{\mu}_y$ or the noise $\boldsymbol{\xi}$. In contrast, diffusion model learns both signal and noise to the same order. Such a claim is visualized in Figure 1.

We believe our framework represents the *first* attempt to systematically investigate feature learning within diffusion models, potentially uncovering novel insights into the intriguing properties of diffusion models, including but not limited to critical window (Sclocchi et al., 2024; Li & Chen, 2024), shape bias (Jaini et al., 2024), classification robustness (Chen et al., 2024c), feature composition and dependence (Okawa et al., 2024; Yang et al., 2025; Han et al., 2025).

## 1.2 RELATED WORK

**Theoretical analysis of diffusion model.** Existing theoretical guarantees for diffusion models focus on distribution estimation and sampling. For distribution estimation, Oko et al. (2023) proved that diffusion models achieve a nearly minimax optimal estimation error where the true density is defined over a bounded Besov space. Zhang et al. (2024) extended the minimax optimality to more general sub-Gaussian densities with sufficient smoothness. When the density is supported on a low-dimensional subspace, diffusion models avoid curse of dimensionality with an estimation rate depending only on the intrinsic dimension (Oko et al., 2023; Chen et al., 2023a). Furthermore, Shah et al. (2023); Gatmiry et al. (2024); Chen et al. (2024d) introduced algorithms based on diffusion models for learning a mixture of Gaussians. Other works provided guarantees of diffusion model trained by gradient descent (Li et al., 2023c; Han et al., 2024; Wang et al., 2024). For sampling, Lee et al. (2022; 2023); Chen et al. (2023b); Li et al. (2023b) have shown (score-based) diffusion models converge polynomially under sufficiently accurate score estimation. Recent studies also aimed to accelerate the convergence via strategies such as consistency training (Song et al., 2023; Li et al., 2024b), advanced design of the reverse transition kernel (Huang et al., 2024b), higher-order approximation (Li et al., 2024a) and parallelization (Chen et al., 2024a; Gupta et al., 2024). In addition, Li & Chen (2024) theoretically verified critical windows of feature emergence during the sampling process, provided accurate score estimation.

**Theoretical analysis on (denoising) autoencoders.** Diffusion models can be viewed as multi-level denoising autoencoders (Xiang et al., 2023). While there is extensive research on the theoretical guarantees of autoencoders without denoising, most studies focus on linear autoencoders (Kunin et al., 2019; Oftadeh et al., 2020; Steck, 2020; Bao et al., 2020). In contrast, only a limited number of works analyze non-linear autoencoders, primarily in the lazy training regime (Nguyen et al., 2021) or the mean-field regime (Nguyen, 2021). Additionally, the training dynamics of non-linear autoencoders have been investigated under population gradient descent (Shevchenko et al., 2023; Kögler et al., 2024) and online gradient descent (Refinetti & Goldt, 2022). On the other hand, the training dynamics of denoising autoencoders have been studied in the context of linear networks (Pretorius et al., 2018) and in the high-dimensional asymptotic limit (Cui & Zdeborová, 2024).

**Diffusion model for representation learning.** Pre-trained diffusion models are shown to learn powerful representation, which is useful for downstream tasks such as classification (Mukhopadhyay et al., 2023; Xiang et al., 2023; Li et al., 2023a; Clark & Jaini, 2024; Yang & Wang, 2023), semantic segmentation (Baranchuk et al., 2022; Zhao et al., 2023; Yang & Wang, 2023). Moreover, many works have found intriguing properties of diffusion models used as classifier, including its ability to understand shape bias (Jaini et al., 2024) and improved adversarial robustness (Chen et al., 2024c). For more detailed exposition, we refer to the recent survey on this matter (Fuest et al., 2024).

## 2 PROBLEM SETTING

This section introduces the problem settings for both diffusion model and classification, including the data model, neural network functions as well as training objectives and algorithm.

**Definition 2.1** (Data distribution). *Each data sample consists of two patches, as $\mathbf{x} = [\mathbf{x}^{(1)\top}, \mathbf{x}^{(2)\top}]^\top$, where each patch is generated as follows:*

- *Sample $y \in \{-1, 1\}$ uniformly with $\mathbb{P}(y = -1) = \mathbb{P}(y = 1) = 1/2$.*

- *Given two orthogonal signal vectors $\boldsymbol{\mu}_1, \boldsymbol{\mu}_{-1}$, with $\boldsymbol{\mu}_1 \perp \boldsymbol{\mu}_{-1}$, we set $\mathbf{x}^{(1)} = \boldsymbol{\mu}_y$, i.e., $\mathbf{x}^{(1)} = \boldsymbol{\mu}_1$ if $y = 1$ and $\mathbf{x}^{(1)} = \boldsymbol{\mu}_{-1}$ if $y = -1$. For simplicity, we assume $\|\boldsymbol{\mu}_1\| = \|\boldsymbol{\mu}_{-1}\| = \|\boldsymbol{\mu}\|$.*

- *Set $\mathbf{x}^{(2)} = \boldsymbol{\xi}$ where $\boldsymbol{\xi} \sim \mathcal{N}(0, \sigma_\xi^2(\mathbf{I} - \boldsymbol{\mu}_1 \boldsymbol{\mu}_1^\top \|\boldsymbol{\mu}_1\|^{-2} - \boldsymbol{\mu}_{-1} \boldsymbol{\mu}_{-1}^\top \|\boldsymbol{\mu}_{-1}\|^{-2}))$.*

This multi-patch data model reflects the structure of image data, where each image consists of multiple patches, and only a subset of the patches are relevant to the class label, while the rest contribute as background noise. This data model has been employed in several existing studies (Allen-Zhu & Li, 2023; Cao et al., 2022; Kou et al., 2023; Meng et al., 2024; Zou et al., 2023). A difference in our model is the use of two orthogonal signal vectors, in contrast to previous works, that employ a single signal vector of the form $y\boldsymbol{\mu}$. Additionally, while our analysis focuses on a two-patch setting for simplicity, it can be readily extended to multi-patch data. We let $\mathrm{SNR} := \|\boldsymbol{\mu}\| / (\sigma_\xi \sqrt{d})$ denote the signal-to-noise ratio.

**Neural network functions.** We study two-layer convolutional-type neural networks for both diffusion model and classification. For **diffusion model**, we consider neural network with quadratic activation and shared first-layer and second-layer weights:

$$\boldsymbol{f}(\mathbf{W}, \mathbf{x}) = \left[ \boldsymbol{f}_1(\mathbf{W}, \mathbf{x}^{(1)})^\top, \boldsymbol{f}_2(\mathbf{W}, \mathbf{x}^{(2)})^\top \right]^\top \in \mathbb{R}^{2d},$$

$$\text{where} \quad \boldsymbol{f}_p(\mathbf{W}, \mathbf{x}^{(p)}) = \frac{1}{\sqrt{m}} \sum_{r=1}^m \langle \mathbf{w}_r, \mathbf{x}^{(p)} \rangle^2 \mathbf{w}_r, \quad p = 1, 2$$

where $m$ denotes the network width and $r$ represents the neuron index.

For **classification**, we consider a similar neural network with quadratic activation where second-layer weights are fixed to be $\pm 1$ (instead of $\mathbf{w}_r$):

$$f(\mathbf{W}, \mathbf{x}) = F_1(\mathbf{W}_1, \mathbf{x}) - F_{-1}(\mathbf{W}_{-1}, \mathbf{x}),$$

$$\text{where} \quad F_j(\mathbf{W}, \mathbf{x}) = \frac{1}{m} \sum_{r=1}^m \langle \mathbf{w}_{j,r}, \mathbf{x}^{(1)} \rangle^2 + \frac{1}{m} \sum_{r=1}^m \langle \mathbf{w}_{j,r}, \mathbf{x}^{(2)} \rangle^2.$$

We remark that the use of polynomial activation, such as quadratic, cubic and ReLU with polynomial smoothing is not uncommon in existing theoretical works (Cao et al., 2022; Jelassi & Li, 2022; Zou et al., 2023; Huang et al., 2023a; Meng et al., 2023). The aim is to better elucidate the separation between signal and noise learning dynamics.

**Training objectives and algorithm.** For **diffusion model**, we employ the objective of denoising diffusion probabilistic model (DDPM) (Ho et al., 2020). We let $\mathbf{x}_0 = [\mathbf{x}^{(1)}, \mathbf{x}^{(2)}]^\top \in \mathbb{R}^{2d}$ to denote input image. For a given diffusion time step $t \in [0, T]$, we sample $\mathbf{x}_t = \alpha_t \mathbf{x}_0 + \beta_t \boldsymbol{\epsilon}_t$ for $\boldsymbol{\epsilon}_t \sim \mathcal{N}(0, \mathbf{I})$ and a pre-determined noise schedule coefficients $\{\alpha_t, \beta_t\}_{t=0}^T$. The aim of diffusion models is to estimate the mean of the posterior distribution of the noise $\boldsymbol{\epsilon}_t$ conditioned on $\mathbf{x}_t$. This is achieved by training a neural network $f$ to predict the noise added at each step $t$. The DDPM loss is given by $\mathbb{E}_{\mathbf{x}_0, \boldsymbol{\epsilon}_t, t} \| f(\mathbf{x}_t) - \boldsymbol{\epsilon}_t \|^2$ up to some re-scaling (Ho et al., 2020). We consider a finite-sample setup given by the training images $\{\mathbf{x}_i\}_{i=1}^n$ sampled according to Definition 2.1 and thus the empirical DDPM loss at time step $t$ becomes

$$L_F(\mathbf{W}_t) = \frac{1}{2n} \sum_{i=1}^n \mathbb{E}_{\boldsymbol{\epsilon}_{t,i}} \| \boldsymbol{f}(\mathbf{W}_t, \mathbf{x}_{t,i}) - \boldsymbol{\epsilon}_{t,i} \|^2 = \frac{1}{2n} \sum_{i=1}^n \mathbb{E}_{\boldsymbol{\epsilon}_{t,i}} \| \boldsymbol{f}(\mathbf{W}_t, \alpha_t \mathbf{x}_{0,i} + \beta_t \boldsymbol{\epsilon}_{t,i}) - \boldsymbol{\epsilon}_{t,i} \|^2,$$

where we let $\mathbf{x}_{0,i} = \mathbf{x}_i$ and $\mathbf{x}_{t,i} = \alpha_t \mathbf{x}_{0,i} + \beta_t \boldsymbol{\epsilon}_{t,i}$. Here, we decouple the training of neural network at each diffusion time step with separate weight parameters, a strategy also adopted in (Shah et al., 2023) for simplicity of analysis.

Unlike (Han et al., 2024), where each sample $i$ is associated with a single noise $\boldsymbol{\epsilon}_{t,i} \sim \mathcal{N}(0, \mathbf{I})$, we here consider taking the expectation over the noise distribution, which aligns with the practical setting where multiple noises are sampled for each input data. We use gradient descent to train diffusion model starting from random Gaussian initialization $\mathbf{w}_{r,t}^0 \sim \mathcal{N}(0, \sigma_0^2 \mathbf{I})$ as $\mathbf{w}_{r,t}^{k+1} = \mathbf{w}_{r,t}^k - \eta \nabla_{\mathbf{w}_{r,t}} L_F(\mathbf{W}_t^k)$, where the superscript $k$ is the iteration index.

For **classification**, we minimize the empirical logistic loss over the training data $\{\mathbf{x}_i, y_i\}_{i=1}^n$,

$$L_S(\mathbf{W}) = \frac{1}{n} \sum_{i=1}^n \ell\big(y_i f(\mathbf{W}, \mathbf{x}_i)\big), \quad \ell(z) = \log\big(1 + \exp(-z)\big).$$

The same as diffusion model, we use gradient descent to train the neural network starting from random Gaussian initialization $\mathbf{w}_{j,r}^0 \sim \mathcal{N}(0, \sigma_0^2 \mathbf{I})$.

## 3 MAIN RESULTS

Our main results are based on the following conditions.

**Condition 3.1.** *Suppose the following holds.*

1. *Dimension $d$ is sufficiently large with $d = \widetilde{\Omega}(n^7 m^5)$.*

2. *The sample size $n$ satisfies $n = \widetilde{\Omega}(1)$.*

3. *The standard deviation of initialization $\sigma_0$ is chosen such that $\widetilde{O}(n^2 m \sigma_\xi^{-1} d^{-1}) \leq \sigma_0 \leq \widetilde{O}\big(\min\{m^{-1/6} d^{-1/6} \sigma_\xi^{1/3} n^{-1/3}, m^{-1/6} d^{-7/12} \sigma_\xi^{-1/3} n^{1/3}, d^{-3/4} \sigma_\xi^{-1} n\}\big).$*

4. *The learning rate $\eta$ satisfies $\eta \leq \widetilde{O}\big(\min\{nm\sigma_0 \sigma_\xi^{-1} d^{-1/2}, nm\sigma_\xi^{-2} d^{-1}\}\big).$*

5. *The signal strength satisfies $\|\boldsymbol{\mu}\| = \Theta(1)$ and noise variation $\sigma_\xi$ satisfies $\widetilde{O}(\max\{n^{5/2} m^{7/4} d^{-5/8}, nm^{1/6} d^{-1}\}) \leq \sigma_\xi \leq \widetilde{O}(d^{-1/4}).$*

6. *The noise coefficients for diffusion model satisfy $\alpha_t, \beta_t = \Theta(1)$.*

Condition 3.1 requires $d$ to be sufficiently large to ensure learning in an over-parameterized setting. Furthermore, we require the sample size to be lower bounded by a constant subject to logarithmic factors. The upper bound on the initialization $\sigma_0$ is to ensure random initialization does not significantly affect the signal and noise learning dynamics. The lower bound on $\sigma_0$ is required to bound the noise inner product at initialization for properly minimizing the training loss of classification. The learning rate $\eta$ is chosen sufficiently small for the convergence analysis for the classification. Lastly for diffusion model, we consider the constant order for $\|\boldsymbol{\mu}\|$ and further restrict the range of $\sigma_\xi$. Despite these conditions, our setting covers a broad range of $n \cdot \mathrm{SNR}^2$, i.e., $\widetilde{O}(nd^{-1/2}) \leq n \cdot \mathrm{SNR}^2 \leq \widetilde{O}(\min\{n^{-4} m^{-7/2} d^{1/4}, n^{-1} m^{-1/3} d\})$. We also consider constant order of $\alpha_t, \beta_t$ to avoid degeneracy in learning dynamics.

We present the main results for diffusion model (Theorem 3.1) and classification (Theorem 3.2).

**Theorem 3.1** (Diffusion model). *Under Condition 3.1, suppose $m = \Theta(1)$. With probability at least $1 - \delta$ (for any $\delta > 0$), there exists a stationary point $\mathbf{W}_t^*$ along the training trajectory of diffusion model, i.e., $\nabla_{\mathbf{w}_{r,t}} L_F(\mathbf{W}_t^*) = 0$ that satisfies (1) $\langle \mathbf{w}_{r,t}^*, \boldsymbol{\mu}_j \rangle = \Theta(\langle \mathbf{w}_{r',t}^*, \boldsymbol{\mu}_{j'} \rangle)$, (2) $\langle \mathbf{w}_{r,t}^*, \boldsymbol{\xi}_i \rangle = \Theta(\langle \mathbf{w}_{r',t}^*, \boldsymbol{\xi}_{i'} \rangle)$, and (3) for all $j = \pm 1, r \in [m], i \in [m]$,*

$$|\langle \mathbf{w}_{r,t}^*, \boldsymbol{\mu}_j \rangle| / |\langle \mathbf{w}_{r,t}^*, \boldsymbol{\xi}_i \rangle| = \Theta(n \cdot \mathrm{SNR}^2),$$

*with $\langle \mathbf{w}_{r,t}^*, \boldsymbol{\mu}_j \rangle = \Theta(1)$ if $n \cdot \mathrm{SNR}^2 = \Omega(1)$, and $\langle \mathbf{w}_{r,t}^*, \boldsymbol{\xi}_i \rangle = \Theta(1)$ if $n^{-1} \cdot \mathrm{SNR}^{-2} = \Omega(1)$.*

Theorem 3.1 states that diffusion model training encourages balanced signal and noise learning, i.e., the neurons share the same order in the directions of signals and noise. Notably, the ratio between signal and noise learning is governed by the SNR, with a stationary magnitude as $n \cdot \mathrm{SNR}^2$.

**Theorem 3.2** (Classification). *Let $T_\mu = \widetilde{\Theta}(\eta^{-1} m \|\boldsymbol{\mu}\|^{-2})$ and $T_\xi = \widetilde{\Theta}(\eta^{-1} nm\sigma_\xi^{-2} d^{-1})$ and $\delta > 0$. Under Condition 3.1, suppose $m = \Omega(\log(n/\delta))$. There exist two absolute constants $\overline{C} > \underline{C} > 0$ such that with probability at least $1 - \delta$, it satisfies that:*

- *When $n \cdot \mathrm{SNR}^2 \geq \overline{C}$, there exists $0 \leq k \leq T_\mu$ such that $L_S(\mathbf{W}^k) \leq 0.1$ and*

$$\max_r |\langle \mathbf{w}_{j,r}^k, \boldsymbol{\mu}_j \rangle| \geq 2, \forall j = \pm 1, \qquad \max_{j,r,i} |\langle \mathbf{w}_{j,r}^k, \boldsymbol{\xi}_i \rangle| = o(1).$$

- When $n \cdot \mathrm{SNR}^2 \leq \underline{C}$, there exists $0 \leq k \leq T_\xi$ such that $L_S(\mathbf{W}^k) \leq 0.1$ and

$$\max_r |\langle \mathbf{w}_{y_i,r}^k, \boldsymbol{\xi}_i \rangle| \geq 1, \ \forall i \in [n], \qquad \max_{j,r,y} |\langle \mathbf{w}_{j,r}^k, \boldsymbol{\mu}_y \rangle| = o(1).$$

Theorem 3.2 establishes a sharp phase transition between signal and noise learning under classification training. The transition is precisely determined by $n \cdot \mathrm{SNR}^2$. That is, when $n \cdot \mathrm{SNR}^2 \geq \overline{C}$ for some constant $\overline{C} > 0$, the neural network learns signal to achieve small training loss. On the contrary, when $n \cdot \mathrm{SNR}^2 \leq \underline{C}$ for some constant $\underline{C} \in (0, \overline{C})$, the neural network overfits noise to achieve convergence. With standard techniques, such as in (Cao et al., 2022), we can show signal and noise learning corresponds to the regime of benign and harmful overfitting respectively. To the best of our knowledge, this is the first result that shows separation under the constant of $n \cdot \mathrm{SNR}^2$.

**Diffusion model learns balanced features while classification learn dominant features.** Comparing the learning outcomes of diffusion model and classification, we reveal a critical difference that *diffusion models learn more balanced features depending on the SNR conditions, while classification is prone to learning either signal or noise predominately*. This can be best understood in the case of $n \cdot \mathrm{SNR}^2 = \Theta(1)$. By Theorem 3.2, we have either signal learning or noise dominating the learning process in classification, while Theorem 3.1 suggests signal and noise learning are in the same order in diffusion models. The theoretical findings corroborate the empirical observations that the neural network trained for classification is prone to overly rely on learning a specific pattern that is easier to learn, a process known as shortcut learning (Geirhos et al., 2020). Meanwhile, diffusion models tend to learn low-frequency, global patterns (Jaini et al., 2024), which helps to improve the classification robustness (Chen et al., 2024b;c).

# 4 PROOF OVERVIEW

This section outlines the proof roadmap for the main results. For *diffusion model*, the mean-squared loss, the joint training of two layers as well as learning in the direction of initialization, pose significant challenges for the analysis. We adopt a two-stage analysis and characterize the stationary points based on the derived results at the end of the second stage. For *classification*, the two-stage analysis is similar as in (Cao et al., 2022; Kou et al., 2023) where the first stage learns signal or noise vector sufficiently fast and the second stage shows convergence in the training loss where the learned scale difference in the first stage is maintained. However for classification analysis, we highlight two critical differences compared to existing works (Cao et al., 2022; Kou et al., 2023; Meng et al., 2024), i.e., a constant $n \cdot \mathrm{SNR}^2$ condition and quadratic activation.

## 4.1 DIFFUSION MODEL

We first simplify the DDPM loss by taking the expectation with respect to the added diffusion noise:

$$L_F(\mathbf{W}_t) = d + \frac{1}{2n} \sum_{i=1}^n \sum_{p=1}^2 \Big( \underbrace{\frac{1}{m} \mathbb{E}_{\boldsymbol{\epsilon}_{t,i}} \big\| \sum_{r=1}^m \langle \mathbf{w}_{r,t}, \mathbf{x}_{t,i}^{(p)} \rangle^2 \mathbf{w}_{r,t} \big\|^2}_{I_1} - \underbrace{\frac{4\alpha_t \beta_t}{\sqrt{m}} \sum_{r=1}^m \|\mathbf{w}_{r,t}\|^2 \langle \mathbf{w}_{r,t}, \mathbf{x}_{0,i}^{(p)} \rangle}_{I_2} \Big),$$

where for $p = 1, 2$, $\mathbf{x}_{t,i}^{(p)} = \alpha_t \mathbf{x}_{0,i}^{(p)} + \beta_t \boldsymbol{\epsilon}_{t,i}^{(p)}$, with $\mathbf{x}_{0,i}^{(1)} = \boldsymbol{\mu}_{y_i}$ and $\mathbf{x}_{0,i}^{(2)} = \boldsymbol{\xi}_i$ and $\boldsymbol{\epsilon}_{t,i}^{(1)}, \boldsymbol{\epsilon}_{t,i}^{(2)} \sim \mathcal{N}(0, \mathbf{I})$. We further simplify $I_1$ in Lemma D.2 (in Appendix). We remark that $I_1$ corresponds to a regularization term that regulates the magnitude and alignment of neurons, while $I_2$ corresponds to the main learning term. We highlight that apart from the signal and noise directions, the learning term $I_2$ also includes the initialization direction $\mathbf{w}_{r,t}^0$, which further complicates the analysis.

**First stage.** In the first stage, where all the key quantities, including signal and noise inner products, weight norms and cross-neuron inner products remain close to their respective initialization, we can show the growth of the signal and noise inner products is approximately linear:

$$\begin{aligned}
\langle \mathbf{w}_{r,t}^{k+1}, \boldsymbol{\mu}_j \rangle &= \langle \mathbf{w}_{r,t}^k, \boldsymbol{\mu}_j \rangle + \Theta(\eta \|\mathbf{w}_{r,t}^k\|^2 \|\boldsymbol{\mu}\|^2) \\
\langle \mathbf{w}_{r,t}^{k+1}, \boldsymbol{\xi}_i \rangle &= \langle \mathbf{w}_{r,t}^k, \boldsymbol{\xi}_i \rangle + \Theta(\eta n^{-1} \|\mathbf{w}_{r,t}^k\|^2 \|\boldsymbol{\xi}_i\|^2)
\end{aligned} \tag{1}$$

In addition, the change of $\mathbf{w}_{r,t}^k$ along direction $\mathbf{w}_{r',t}^0$ can be properly controlled such that the scale of key quantities remain unaffected and the simplification in (1) is valid throughout the first stage.

The updates in (1) immediately suggest that once the growth terms of the inner products dominate their initialization, we obtain $|\langle \mathbf{w}_{r,t}^k, \boldsymbol{\mu}_j \rangle| / |\langle \mathbf{w}_{r,t}^k, \boldsymbol{\xi}_i \rangle| = \Theta(n \cdot \mathrm{SNR}^2)$. This marks the end of the first stage, as described in the following lemma.

**Lemma 4.1.** *Under Condition 3.1, there exists an iteration $T_1 = \max\{T_\mu, T_\xi\}$, where $T_\mu = \widetilde{\Theta}(\sqrt{m}\sigma_0^{-1}d^{-1}\|\boldsymbol{\mu}\|^{-1}\eta^{-1})$ and $T_\xi = \widetilde{\Theta}(n\sqrt{m}\sigma_0^{-1}\sigma_\xi^{-1}d^{-3/2}\eta^{-1})$ such that for all $k \leq T_1$, $\|\mathbf{w}_{r,t}^k\|^2 = \Theta(\sigma_0^2 d), \langle \mathbf{w}_{r,t}^k, \mathbf{w}_{r,t}^0 \rangle = \Theta(\sigma_0^2 d)$ for all $r \in [m], j = \pm 1, i \in [n]$. Furthermore, we can show for all $j, j' = \pm 1, r, r' \in [m], i, i' \in [n]$,*

- $\langle \mathbf{w}_{r,t}^{T_1}, \boldsymbol{\mu}_j \rangle = \Theta(\langle \mathbf{w}_{r',t}^{T_1}, \boldsymbol{\mu}_{j'} \rangle), \langle \mathbf{w}_{r,t}^{T_1}, \boldsymbol{\xi}_i \rangle = \Theta(\langle \mathbf{w}_{r',t}^{T_1}, \boldsymbol{\xi}_{i'} \rangle)$, and

- $|\langle \mathbf{w}_{r,t}^{T_1}, \boldsymbol{\mu}_j \rangle| / |\langle \mathbf{w}_{r,t}^{T_1}, \boldsymbol{\xi}_i \rangle| = \Theta(n \cdot \mathrm{SNR}^2)$,

Lemma 4.1 verifies that at the end of the first stage, all the neurons are concentrated and the ratio is precisely determined by $n \cdot \mathrm{SNR}^2$. This is critically different compared to classification where signal and noise learning exhibits exponential growth as we show later and thus shows a clear scale difference at the end of the first stage, even when $n \cdot \mathrm{SNR}^2 = \Theta(1)$.

**Second stage.** The second stage aims to characterize when the dominant terms in the gradients along the key directions become no longer dominant. To this end, we decompose the gradient into

$$\langle \nabla_{\mathbf{w}_{r,t}} L(\mathbf{W}_t^k), \boldsymbol{\mu}_j \rangle = -\Theta(\|\mathbf{w}_{r,t}^k\|^2 \|\boldsymbol{\mu}\|^2) + E_{r,t,\mu_j}^k,$$

$$\langle \nabla_{\mathbf{w}_{r,t}} L(\mathbf{W}_t^k), \boldsymbol{\xi}_i \rangle = -\Theta(n^{-1}\|\mathbf{w}_{r,t}^k\|^2 \|\boldsymbol{\xi}_i\|^2) + E_{r,t,\xi_i}^k,$$

$$\langle \nabla_{\mathbf{w}_{r,t}} L(\mathbf{W}_t^k), \mathbf{w}_{r,t}^0 \rangle = -\Theta\left( \left( \langle \mathbf{w}_{r,t}^k, \boldsymbol{\mu}_j + \overline{\boldsymbol{\xi}} \rangle - \|\mathbf{w}_{r,t}^k\|^4 \right) \langle \mathbf{w}_{r,t}^k, \mathbf{w}_{r,t}^0 \rangle + \|\mathbf{w}_{r,t}^k\|^2 \langle \mathbf{w}_{r,t}^0, \boldsymbol{\mu}_j + \overline{\boldsymbol{\xi}} \rangle \right) + E_{r,t,w^0}^k,$$

where $E_{r,t,\mu_j}^k, E_{r,t,\xi_i}^k, E_{r,t,w^0}^k$ are the residual terms of the gradients and we let $\overline{\boldsymbol{\xi}} = \frac{1}{n}\sum_{i=1}^n \boldsymbol{\xi}_i$. The following lemma shows before $E_{r,t,\mu_j}^k, E_{r,t,\xi_i}^k, E_{r,t,w^0}^k$ reach order as the dominant terms, the ratio of signal and noise inner products are preserved.

**Lemma 4.2.** *There exists an iteration $T_2 > T_1$ with $T_2 = \Theta(\max\{\eta^{-1}\sigma_0^{-2}d^{-1}, \eta^{-1}n\sigma_0^{-2}\sigma_\xi^2\})$ such that for all $j = \pm 1, r \in [m], i \in [n]$ (1) if $n \cdot \mathrm{SNR}^2 = \Omega(1)$, $\langle \mathbf{w}_{r,t}^{T_2}, \boldsymbol{\mu}_j \rangle = \Theta(1)$ and if $n^{-1} \cdot \mathrm{SNR}^{-2} = \Omega(1)$, $\langle \mathbf{w}_{r,t}^{T_2}, \boldsymbol{\xi}_i \rangle = \Theta(1)$; (2) $E_{r,t,\mu_j}^{T_2} = \Theta(\|\mathbf{w}_{r,t}^{T_2}\|^2 \|\boldsymbol{\mu}\|^2)$, $E_{r,t,\xi_i} = \Theta(n^{-1}\|\mathbf{w}_{r,t}^{T_2}\|^2 \|\boldsymbol{\xi}_i\|^2), E_{r,t,w^0}^{T_2} = \Theta((\langle \mathbf{w}_{r,t}^k, \boldsymbol{\mu}_j + \overline{\boldsymbol{\xi}} \rangle - \|\mathbf{w}_{r,t}^k\|^4)\langle \mathbf{w}_{r,t}^k, \mathbf{w}_{r,t}^0 \rangle + \|\mathbf{w}_{r,t}^k\|^2 \langle \mathbf{w}_{r,t}^0, \boldsymbol{\mu}_j + \overline{\boldsymbol{\xi}} \rangle)$ and (3) for all $T_1 \leq k \leq T_2$, we have*

- $\langle \mathbf{w}_{r,t}^k, \boldsymbol{\mu}_j \rangle = \Theta(\langle \mathbf{w}_{r',t}^k, \boldsymbol{\mu}_{j'} \rangle), \langle \mathbf{w}_{r,t}^k, \boldsymbol{\xi}_i \rangle = \Theta(\langle \mathbf{w}_{r',t}^k, \boldsymbol{\xi}_{i'} \rangle),$

- $|\langle \mathbf{w}_{r,t}^k, \boldsymbol{\mu}_j \rangle| / |\langle \mathbf{w}_{r,t}^k, \boldsymbol{\xi}_i \rangle| = \Theta(n \cdot \mathrm{SNR}^2)$,

Lemma 4.2 characterizes $T_2$ as the point where the dominant terms of the gradients in the first stage become comparable to the residual terms. Meanwhile, we show the scale of signal and noise inner products escape from initialization and reach constant order. Throughout the second stage, the concentration of neurons are preserved and the ratio of signal to noise learning is dictated by $n \cdot \mathrm{SNR}^2$. Finally, we identify there exists a stationary point that satisfies the conditions at the end of the second stage (Lemma 4.2).

**Theorem 4.1** (Informal). *There exists a stationary point $\mathbf{W}_t^*$, i.e., $\nabla_{\mathbf{w}_{r,t}} L(\mathbf{W}_t^*) = 0$ such that the conditions at $T_2$ (in Lemma 4.2) are satisfied, and in particular $|\langle \mathbf{w}_{r,t}^*, \boldsymbol{\mu}_j \rangle| / |\langle \mathbf{w}_{r,t}^*, \boldsymbol{\xi}_i \rangle| = \Theta(n \cdot \mathrm{SNR}^2)$ for all $j = \pm 1, r \in [m], i \in [n]$.*

### 4.2 CLASSIFICATION

For classification, we first let $\mathcal{S}_y := \{i \in [n] : y_i = y\}$ for $y = \pm 1$ and $\ell_i'^k = \ell'(y_i f(\mathbf{W}^k, \mathbf{x}))$. We can rewrite the gradient descent updates in terms of the signal and noise inner products:

$$\langle \mathbf{w}_{j,r}^{k+1}, \boldsymbol{\mu}_y \rangle = \langle \mathbf{w}_{j,r}^k, \boldsymbol{\mu}_y \rangle - \frac{\eta |\mathcal{S}_y|}{nm} \ell_i'^k \langle \mathbf{w}_{j,r}^k, \boldsymbol{\mu}_y \rangle jy\|\boldsymbol{\mu}\|^2 = (1 - \frac{\eta |\mathcal{S}_y|\|\boldsymbol{\mu}\|^2}{nm}\ell_i'^k jy)\langle \mathbf{w}_{j,r}^k, \boldsymbol{\mu}_y \rangle, \quad (2)$$

$$\langle \mathbf{w}_{j,r}^{k+1}, \boldsymbol{\xi}_i \rangle = \langle \mathbf{w}_{j,r}^k, \boldsymbol{\xi}_i \rangle - \frac{\eta}{nm}\ell_i'^k \langle \mathbf{w}_{j,r}^k, \boldsymbol{\xi}_i \rangle \|\boldsymbol{\xi}_i\|^2 jy_i - \frac{\eta}{nm}\sum_{i' \neq i}\ell_{i'}'^k \langle \mathbf{w}_{j,r}^k, \boldsymbol{\xi}_{i'} \rangle jy_{i'} \langle \boldsymbol{\xi}_{i'}, \boldsymbol{\xi}_i \rangle, \quad (3)$$

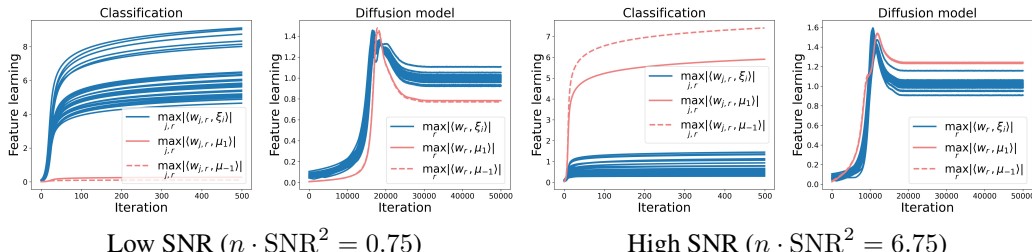

Figure 2: Experiments on the synthetic dataset with both low SNR ($n \cdot \mathrm{SNR}^2 = 0.75$) and high SNR ($n \cdot \mathrm{SNR}^2 = 6.75$). In the low SNR setting, we see noise learning quickly dominates signal learning for the classification task and in the high SNR setting, signal learning quickly dominates noise learning. Meanwhile diffusion model converges to a stationary point that with signal-to-noise learning ratio respects the order of $n \cdot \mathrm{SNR}^2$. More experimental results on additional SNR values are in Appendix A.2.

for all $j, y = \pm 1, r \in [m], i \in [n]$. For the signal inner product, its update suggests that for any $j = \pm 1$, $\mathbf{w}_{j,r}$ specializes the learning of $\boldsymbol{\mu}_j$ due to that $\ell_i'^k < 0$, $|\langle \mathbf{w}_{j,r}^{k+1}, \boldsymbol{\mu}_y \rangle| = (1 - \frac{\eta |\mathcal{S}_y| \|\boldsymbol{\mu}\|^2}{nm} \ell_i'^k jy) |\langle \mathbf{w}_{j,r}^k, \boldsymbol{\mu}_y \rangle| > |\langle \mathbf{w}_{j,r}^k, \boldsymbol{\mu}_y \rangle|$ only when $j = y$. For the noise inner product, the growth is dominated by the second term where $|\langle \boldsymbol{\xi}_i, \boldsymbol{\xi}_{i'} \rangle| = \widetilde{O}(d^{-1/2}) \|\boldsymbol{\xi}_i\|^2$ is negligible. Thus, we show $|\langle \mathbf{w}_{j,r}^{k+1}, \boldsymbol{\xi}_i \rangle|$ grows only for $j = y_i$. In contrast, for $j = -y_i$, its magnitude cannot increase relative to the scale of initialization. Next, we decompose the analysis into two stages.

**First stage.** In the first stage before the maximum of signal and noise inner product reaches constant order, the loss derivatives can be lower bounded by an absolute constant, i.e., $|\ell_i'^k| \geq C_\ell$, for all $k \leq T_1$. As a result, both signal and noise inner product can grow exponentially and the relative growth rates are determined by $n \cdot \mathrm{SNR}^2$. A constant order of difference in the growth rates is sufficient to ensure a scale separation in the signal and noise learning at the end of the first stage, as shown in the following Lemma.

**Lemma 4.3.** *Under Condition 3.1: (1) When $n \cdot \mathrm{SNR}^2 = \Omega(1)$, there exists $T_1 = \widetilde{\Theta}(\eta^{-1} m \|\boldsymbol{\mu}\|^{-2})$, such that $\frac{1}{m} \sum_{r=1}^m |\langle \mathbf{w}_{j,r}^{T_1}, \boldsymbol{\mu}_j \rangle| \geq 2$ for all $j = \pm 1$ and $\max_{j,r,i} |\langle \mathbf{w}_{j,r}^{T_1}, \boldsymbol{\xi}_i \rangle| = o(1)$. (2) When $n^{-1} \cdot \mathrm{SNR}^{-2} = \Omega(1)$, there exists $T_1 = \widetilde{\Theta}(\eta^{-1} nm\sigma_\xi^{-2} d^{-1})$ such that $\frac{1}{m} \sum_{r=1}^m |\langle \mathbf{w}_{y_i,r}^{T_1}, \boldsymbol{\xi}_i \rangle| \geq 4$ for all $i \in [n]$ and $\max_{j,r,y} |\langle \mathbf{w}_{j,r}^{T_1}, \boldsymbol{\mu}_y \rangle| = o(1)$.*

**Remark 4.1.** *Different to existing analysis that only shows maximum inner product reaches constant order (Cao et al., 2022; Huang et al., 2023a), we also show the average inner product reach constant order at the same time. Such a stronger result is required for the analysis under the constant order of $n \cdot \mathrm{SNR}^2$, which reduces the required iteration number in the second stage by an order of $m$.*

**Second stage.** In the second stage, we show the loss converges while the scale separation established in Lemma 4.3 is maintained. Because $n \cdot \mathrm{SNR}^2$ can be a constant, we require to carefully bound the loss derivatives in the second stage particularly for establishing the upper bound for $|\langle \mathbf{w}_{j,r}^k, \boldsymbol{\xi}_i \rangle|$ when $n \cdot \mathrm{SNR}^2 = \Omega(1)$. The naïve bound $\max_i |\ell_i'^k| \leq \max_i |\ell_i^k| \leq nL_S(\mathbf{W}^k)$ used in (Cao et al., 2022) no longer works as it introduces an additional factor of $n$. To provide a tighter bound, we show the ratio of loss derivatives in the case of $n \cdot \mathrm{SNR}^2 = \Omega(1)$, i.e., $|\ell_i'^k|/|\ell_{i'}'^k| \leq C_1$ for all $i, i' \in [n]$ with $y_i = y_{i'}$, $k \geq T_1$, where $C_1 > 0$ is a constant. This is possible because the network output is dominated by the signal, which is shared across samples with the same label. This allows to bound $\max_i |\ell_i'^k| = \Theta(|\mathcal{S}_{y_{i*}}|^{-1} \sum_{i \in \mathcal{S}_{y_{i*}}} |\ell_i'^k|) \leq \Theta(L_S(\mathbf{W}^k))$.

## 5 EXPERIMENTS

We conduct both synthetic and real-world experiments to verify our theoretical claims.

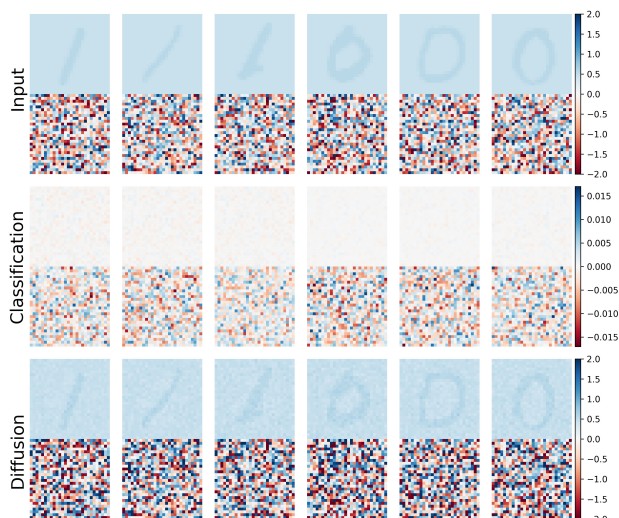 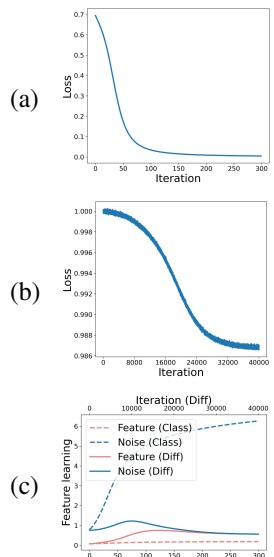

Figure 3: Experiments on Noisy-MNIST with $\widetilde{\mathrm{SNR}} = 0.1$. (First row): Test Noisy-MNIST images; (Second row): Illustration of input gradient, i.e., $\nabla_{\mathbf{x}} F_{+1}(\mathbf{W}, \mathbf{x})$ when $y = 1$ and $\nabla_{\mathbf{x}} F_{-1}(\mathbf{W}, \mathbf{x})$ when $y = 0$. (Third row): denoised image from diffusion model. In this low-SNR case, we see classification tends to predominately learn noise while diffusion learns both signals and noise.

Figure 4: Experiments on Noisy-MNIST with $\widetilde{\mathrm{SNR}} = 0.1$. (a) Train loss for classification. (b) Train loss for diffusion model. (c) Feture learning dynamics.

## 5.1 SYNTHETIC EXPERIMENT

**Setup.** We follow Definition 2.1 to generate a synthetic dataset for both diffusion model and classification. Specifically, we set data dimension $d = 1000$ and let $\boldsymbol{\mu}_1 = [\mu, 0, \cdots, 0] \in \mathbb{R}^d$ and $\boldsymbol{\mu}_{-1} = [0, \mu, 0, \cdots, 0] \in \mathbb{R}^d$. We sample the noise patch $\boldsymbol{\xi}_i \sim \mathcal{N}(0, \mathbf{I}_d)$, $i \in [n]$ (i.e., $\sigma_\xi = 1$). We set sample size and network width to be $n = 30$ and $m = 20$ and initialize the weights to be Gaussian with a standard deviation $\sigma_0 = 0.001$. We vary the choice of $\mu$ to create two problem settings: (1) low SNR with $\mu = 5$, which leads to $n \cdot \mathrm{SNR}^2 = 0.75$ and (2) high SNR with $\mu = 15$, which leads to $n \cdot \mathrm{SNR}^2 = 6.75$. We use the same two-layer networks introduced in Section 2. For classification, we set a learning rate of $\eta = 0.1$ and train for 500 iterations. For diffusion model, we minimize the DDPM loss by averaging over the diffusion noise, following the standard training of diffusion model. In particular, for each sample, we samples $n_\epsilon = 2000$ noise at each iteration and the loss is calculated by taking an average over the noise. For the noise coefficients, we consider a time $t = 0.2$ and set $\alpha_t = \exp(-t) = 0.82$ and $\beta_t = \sqrt{1 - \exp(-2t)} = 0.57$. For diffusion model, we set $\eta = 0.5$ and train for 40000 iterations.

**Results.** In Figure 2, we compare signal and noise learning dynamics–visualized through maximum signal and noise inner product–between classification and diffusion model. In Appendix A.1, we also include training loss convergence for both the tasks as well as training and test accuracy for classification. For classification, the training loss converges while diffusion model recovers only a stationary point.

In terms of feature learning, noise learning in classification quickly dominates signal learning by exhibiting a significant larger growth in the first stage (up to around 20 iterations). This ensures that noise learning stabilizes at a constant order while signal learning remains relatively small. In the second stage, training loss converges and signal and noise learning exhibits logarithmic growth. For diffusion model, in the first stage, where training loss does not materially change, both signal and noise learning increase linearly. In the second stage where loss significantly decreases, signal and noise learning start to grow exponentially and in the final stage, due to the weight regularization terms, noise and signal reach a stationary point that preserves the scale of $n \cdot \mathrm{SNR}^2$.

## 5.2 REAL-WORLD EXPERIMENT

**Setup.** We also conduct experiments on the MNIST dataset (Lecun et al., 1998) to support our theory. In order to better control the signal-to-noise ratio, we create a *Noisy*-MNIST dataset, where we treat each original MNIST image as a clean signal patch and concatenate a standard Gaussian noise patch with the same size, i.e., $28 \times 28$. In addition, we scale the signal patch by a constant denoted as $\widetilde{\text{SNR}}$. Because the noise scale is fixed, higher $\widetilde{\text{SNR}}$ corresponds to higher SNR. Some sample images with $\widetilde{\text{SNR}} = 0.1$ are shown in the first row of Figure 3. We select 50 samples each from digit 0 and 1 respectively (i.e., $n = 100$). We consider the same neural networks as in the synthetic example, where we set $m = 100$ and initialize the weights with $\sigma_0 = 0.01$. For diffusion model, we choose the same $\alpha_t, \beta_t$ as in the synthetic experiment. In the main paper, we present the results for $\widetilde{\text{SNR}} = 0.1$, which corresponds to a low SNR setting.

**Results.** Figure 4(a,b) shows that both classification and diffusion model converge in loss. Additionally, Figure 4(c) plots the signal and noise learning dynamics. Because each image is composed of unique signal $\boldsymbol{\mu}_i$ and noise patch $\boldsymbol{\xi}_i$ for $i \in [n]$, we measure the signal and noise learning by computing $\frac{1}{n} \sum_{i=1}^n \max_r |\langle \mathbf{w}_r, \boldsymbol{\mu}_i \rangle|$ and $\frac{1}{n} \sum_{i=1}^n \max_r |\langle \mathbf{w}_r, \boldsymbol{\xi}_i \rangle|$ respectively. We notice that due to the low SNR, noise learning in classification dominates signal learning at convergence while diffusion model learns more balanced features. This corroborates our theoretical findings.

To visualize the patterns learned by the neural networks, for classification, we adopt an approach similar to Grad-CAM (Selvaraju et al., 2020) by analyzing the gradient of output with respect to the input. Specifically, for samples of digit 0, we plot the gradient of negative function output, $\nabla_{\mathbf{x}} F_{-1}(\mathbf{W}, \mathbf{x})$, while for digit 1, we plot $\nabla_{\mathbf{x}} F_{+1}(\mathbf{W}, \mathbf{x})$. As shown in the second row of Figure 3, the gradients of six test images indicate that classification primarily learns the noise rather than the signal patch. For diffusion model, we first add diffusion noise to the input images and use the network to predict the added noise. Then we reconstruct the input with the formula $\hat{\mathbf{x}}_0 = (\mathbf{x}_t - \beta_t \hat{\boldsymbol{\epsilon}}(\mathbf{x}_t))/\alpha_t$, where $\hat{\boldsymbol{\epsilon}}(\mathbf{x}_t)$ denotes the predicted diffusion noise. The third row of Figure 3 shows that the diffusion model learns both the signal and noise. In Appendix A.3, we present results for a high-SNR setting with $\widetilde{\text{SNR}} = 0.5$, where we observe the reverse pattern: classification predominately captures signal rather than noise while diffusion model continues to balance the learning of both signal and noise. Additionally, Appendix A.5 presents experiments on all 10 digits of the MNIST dataset, verifying the observed distinctions in feature learning between diffusion models and classification.

## 6 CONCLUSION

This work introduces a theoretical framework for analyzing the feature learning dynamics in diffusion models, taking an initial step toward understanding the representation learning in diffusion models. Our findings demonstrate that diffusion models inherently promote a more balanced feature learning, in contrast to classification models, which tend to prioritize certain features over others. This suggests that classification models may be more sensitive to variations in the signal-to-noise ratio compared to diffusion models. Consequently, this may provide an explanation for the inherent adversarial robustness of diffusion models in downstream applications, such as classification (Li et al., 2023a; Chen et al., 2024c;b), as perturbations are less likely to significantly affect the learned representations compared to classification models.

Although our study focuses on a two-patch data setup, the proposed framework can be extended to accommodate more complex data settings. For example, our analysis can be extended to multi-feature data distributions, where certain features appear more frequently (Zou et al., 2023) or have larger norms than others (Lu et al., 2024). Such extensions could provide deeper insights into the mechanisms of feature learning in more realistic scenarios. We hypothesize that, despite the infrequent occurrence or smaller norm of certain features, diffusion models can effectively learn them due to the nature of the denoising objective. This insight has significant implications for downstream tasks, such as out-of-distribution classification, where rare or weak features may be the primary distinguishing factors.

We believe our framework holds broader potential beyond the scope of this work and can be adapted to analyze conditional and latent diffusion models, elucidate the mechanisms of various training objectives and optimizers, and examine other generative paradigms, such as flow matching.

ACKNOWLEDGEMENTS

We would like to thank the anonymous reviewers and area chairs for their helpful comments. Wei Huang is supported by JSPS KAKENHI Grant Number 24K20848. Yuan Cao is supported in part by NSFC 12301657 and Hong Kong ECS award 27308624. Difan Zou is supported in part by NSFC 62306252, Hong Kong ECS award 27309624, Guangdong NSF 2024A1515012444, and the central fund from HKU IDS.

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

APPENDIX CONTENTS

## A    ADDITIONAL EXPERIMENTAL RESULTS

This section includes additional experiment results.

### A.1    SUPPLEMENTARY RESULTS FOR SYNTHETIC EXPERIMENT

We first include the convergence in loss plots as well as accuracy for classification under the two SNR conditions considered in the main experiment. The (in-distribution) test accuracy is computed with 3000 test samples. We see both classification and diffusion model are able to converge in loss, although diffusion model only finds a stationary point. In the low SNR setting, classification is able to perfectly fit the training samples with a 100% classification accuracy. However because it primarily focuses on learning noise, the generalization is poor with a test accuracy of around 50%. For the high-SNR case, both training and test sets can be perfectly classified due to the signal learning.

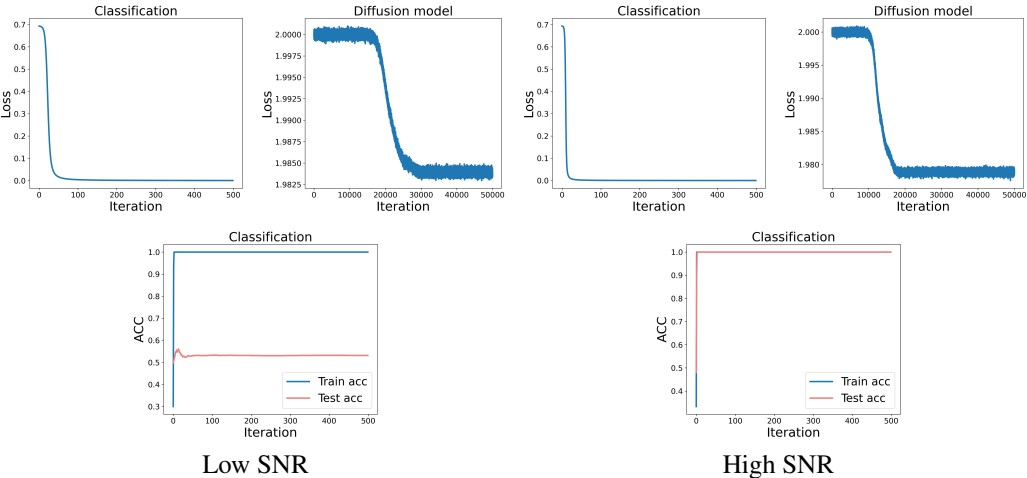

Figure 5: Experiments on the synthetic dataset with both low SNR ($n \cdot \mathrm{SNR}^2 = 0.75$) and high SNR ($n \cdot \mathrm{SNR}^2 = 6.75$).

### A.2    FEATURE LEARNING COMPARISON UNDER VARYING SNRS

In this section, we compare the feature learning dynamics of classification and diffusion models on additional settings of SNR. Apart from the $n \cdot \mathrm{SNR}^2 = 0.75$ and $n \cdot \mathrm{SNR}^2 = 6.75$ as shown in the main text, we additionally test on (1) $n \cdot \mathrm{SNR}^2 = 1.92$, (2) $n \cdot \mathrm{SNR}^2 = 3$ (3) $n \cdot \mathrm{SNR}^2 = 4.32$. The feature learning dynamics under the corresponding SNR settings are shown in Figure 6.

From the figures, we can see that classification indeed is more sensitive to the SNR scale, where it easily overfit to either signal or noise (except for the case where $n \cdot \mathrm{SNR}^2 = 3$ where classification learns signal and noise to approximately the same scale). On the other hand, we can verify that at stationarity, diffusion model learns in a more balanced scale for signal and noise.

### A.3    HIGH SNR SETTING ON NOISY-MNIST

Here we include experiment results when $\widetilde{\mathrm{SNR}} = 0.5$, which corresponds to the high SNR setting. The experiment settings are exactly the same as in the main experiment. Figure 8 shows both classification and diffusion model converge in terms of objective. In addition, we see the high SNR encourages classification to learn primarily the signal while ignoring the noise. In contrast, diffusion model still learns both signal and noise to relatively the same order. Figure 7 suggests that classification learns more signal compared to noise while diffusion model still learns more balanced signal and noise. We also plot classification accuracy for both the low and high SNR cases. In the low-SNR case, because classification predominately learns noise, the generalization is poor with test accuracy around 50%. Conversely in the high-SNR case, where the model is able to learn signals, the classification demonstrates effective generalization with nearly 100% test accuracy.

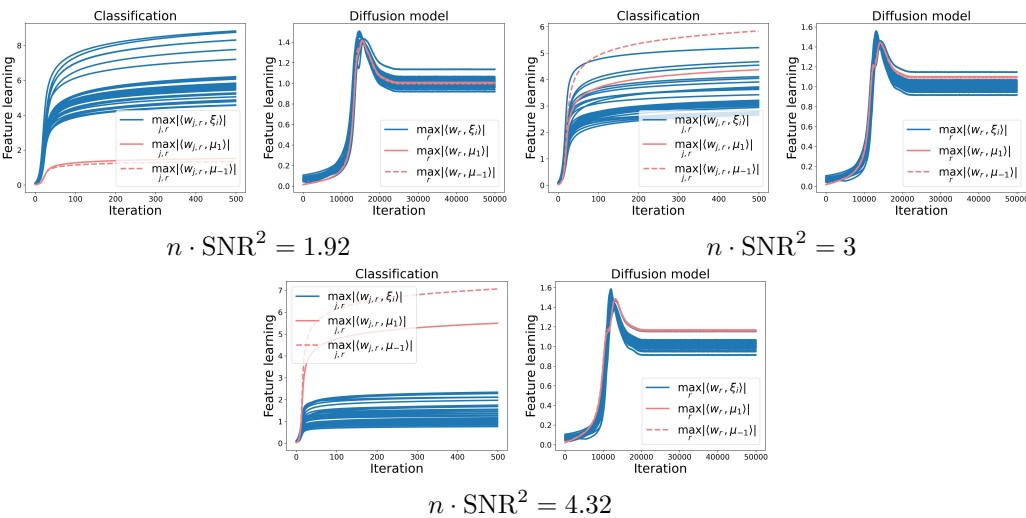

Figure 6: Experiments on the synthetic dataset with varying SNRs.

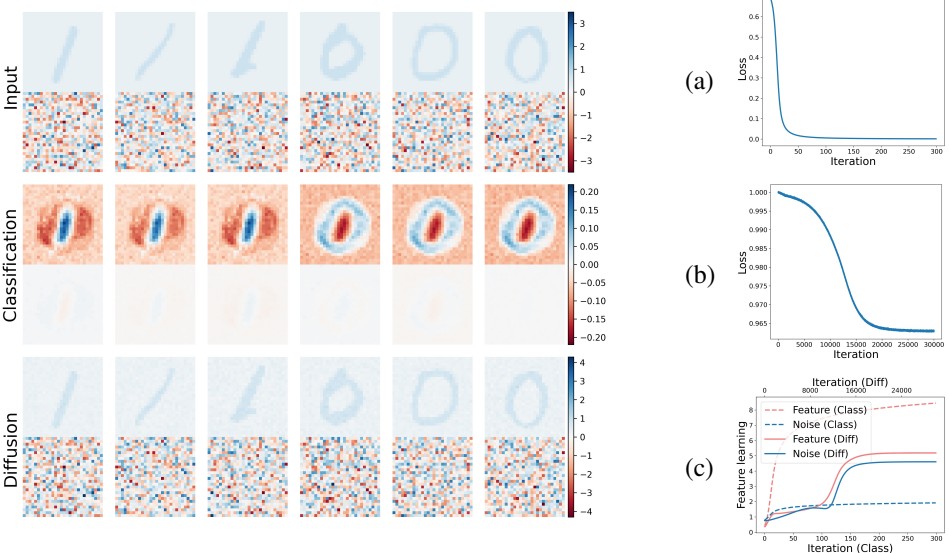

Figure 7: Experiments on Noisy-MNIST with $\widetilde{\mathrm{SNR}} = 0.5$. (First row): Test Noisy-MNIST images; (Second row): Illustration of input gradient, i.e., $\nabla_{\mathbf{x}} F_{+1}(\mathbf{W}, \mathbf{x})$ when $y = 1$ and $\nabla_{\mathbf{x}} F_{-1}(\mathbf{W}, \mathbf{x})$ when $y = 0$. (Third row): denoised image from diffusion model. In this low-SNR case, we see classification tends to predominately learn noise while diffusion learns both signal and noise.

Figure 8: Experiments on Noisy-MNIST with $\widetilde{\mathrm{SNR}} = 0.5$. (a) Train loss for classification. (b) Train loss for diffusion model. (c) Feature learning dynamics.

## A.4 EXPERIMENTS WITH ADDITIONAL DIFFUSION TIME STEP

Here we also test on additional diffusion time step for learning on noisy-MNIST dataset. In particular, we consider $t = 0.8$, which gives $\alpha_t = \exp(-t) = 0.45$ and $\beta_t = \sqrt{1 - \exp(-2t)} = 0.89$. We include the illustrations of denoised images as well as loss convergence and feature learning dynamics in Figure 10, 11, 12, 13. We see despite with a larger scale of added diffusion noise, diffusion model still learn both signals and noise unlike for the case of classification.

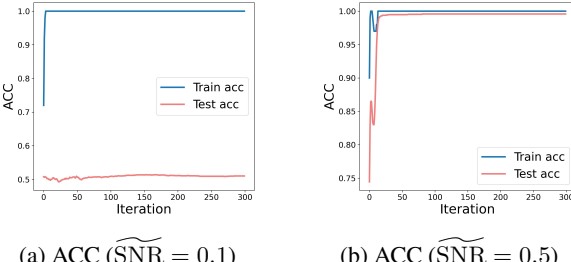

(a) ACC ($\widetilde{\text{SNR}} = 0.1$)  (b) ACC ($\widetilde{\text{SNR}} = 0.5$)

Figure 9: Classification accuracy on (a) low-SNR and (b) high-SNR noisy MNIST datasets. This demonstrates that when classification focuses on learning noise (as in the low-SNR case), the test accuracy hovers around 50%, thus suggesting failure to generalize. In contrast, when classification focuses on learning signals (as in the high-SNR case), classification generalizes effectively, achieving near-perfect accuracy.

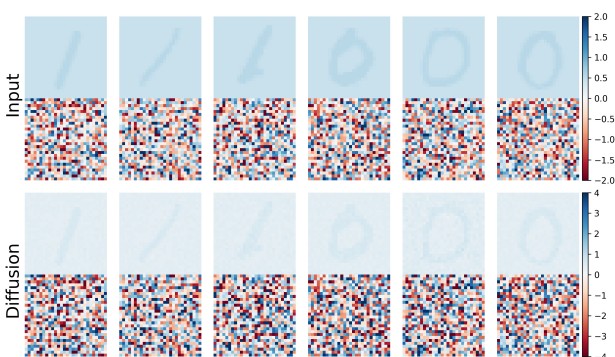
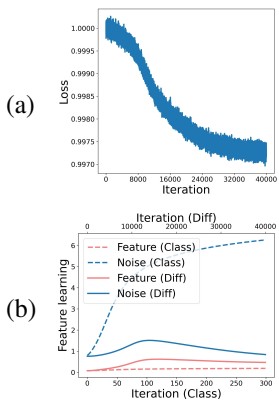

Figure 10: Additional experiments on Noisy-MNIST with $\widetilde{\text{SNR}} = 0.1$ and diffusion $t = 0.8$. (First row): Test Noisy-MNIST images; (Second row): denoised image from diffusion model. We see diffusion still learns both signals and noise even with large diffusion time step.

Figure 11: Additional experiments on Noisy-MNIST with $\widetilde{\text{SNR}} = 0.1$ and $= t = 0.8$. (a) Train loss for diffusion model. (c) Feature learning dynamics.

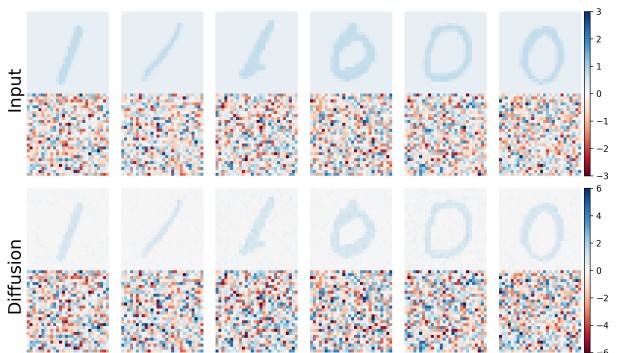
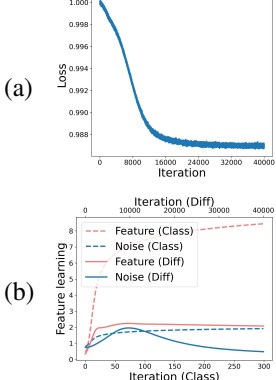

Figure 12: Additional experiments on Noisy-MNIST with $\widetilde{\text{SNR}} = 0.5$ and diffusion $t = 0.8$. (First row): Test Noisy-MNIST images; (Second row): denoised image from diffusion model. We see diffusion still learns both signals and noise even with large diffusion time step.

Figure 13: Additional experiments on Noisy-MNIST with $\widetilde{\text{SNR}} = 0.5$ and $= t = 0.8$. (a) Train loss for diffusion model. (c) Feature learning dynamics.

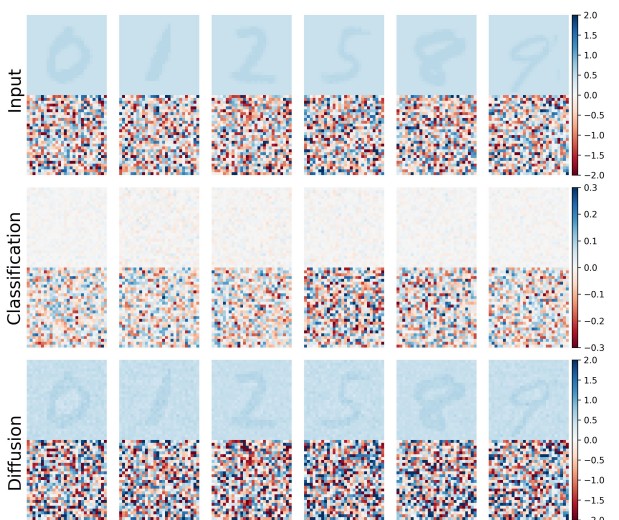 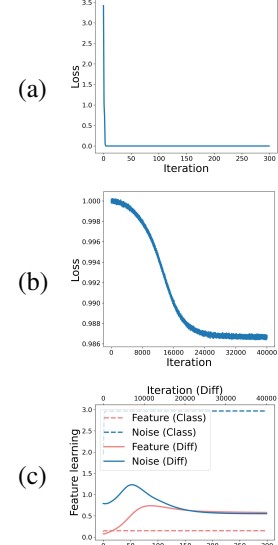

Figure 14: Experiments on 10-class Noisy-MNIST with $\widetilde{\mathrm{SNR}} = 0.1$. (First row): Test Noisy-MNIST images; (Second row): Illustration of gradient of output (for the true class) with respect to the input. (Third row): denoised image from diffusion model. In this low-SNR case, we see classification tends to predominately learn noise while diffusion learns both signals and noise.

Figure 15: Experiments on 10-class Noisy-MNIST with $\widetilde{\mathrm{SNR}} = 0.1$. (a) Train loss for classification. (b) Train loss for diffusion model. (c) Feature learning dynamics.

## A.5 ON THE FEATURE LEARNING WITH 10-CLASS MNIST

In the main paper, we only conduct experiments on Noisy-MNIST restricted to two classes. In this section, we experiment over the 10-class MNIST dataset, which contains more features and is more challenging for both diffusion model and classification.

We adopt the same data processing pipelines as in Section 5.2 except that for each class, we select 10 images. We set the scaled SNR $\widetilde{\mathrm{SNR}} = 0.1$, consistent with the main paper. While the diffusion model remains unchanged, the classification model requires modification. Specifically, the second layer's weight matrix has dimensions $m \times 10$, with entries fixed uniformly to values in $\{-1, 1\}$. Furthermore, we employ cross-entropy loss for training the classification model.

We plot the visualization of feature learning in Figure 14. We observe that, even with additional features and labels, the similar learning patterns are observed, i.e., diffusion model learns both signals and noise in order to reconstruct the input distribution while classification model learns primarily noise for loss minimization. From Figure 15(c), we notice that diffusion model learns features to relatively the same scale while for classification, the growth of feature learning is dominated by noise learning.

## B    PRELIMINARY LEMMAS

Recall we define $\mathcal{S}_1 = \{i \in [n] : y_i = 1\}$ and $\mathcal{S}_{-1} = \{i \in [n] : y_i = -1\}$.

**Lemma B.1.** *Given arbitrary $\delta > 0$, with probability at least $1 - \delta$, we have*

$$\frac{n}{2}\big(1 - \widetilde{O}(n^{-1/2})\big) \leq |\mathcal{S}_1|, |\mathcal{S}_{-1}| \leq \frac{n}{2}\big(1 + \widetilde{O}(n^{-1/2})\big)$$

*Proof of Lemma B.1.* The proof is the same as in (Cao et al., 2022; Kou et al., 2023) and we include here for completeness. Because $|\mathcal{S}_1| = \sum_{i=1}^n \mathbb{1}(y_i = 1)$ and $|\mathcal{S}_{-1}| = \sum_{i=1}^n \mathbb{1}(y_i = -1)$ and $\mathbb{P}(y_i = 1) = \mathbb{P}(y_i = -1) = 1/2$ for all $i \in [n]$, then $\mathbb{E}|\mathcal{S}_1| = \mathbb{E}|\mathcal{S}_{-1}| = n/2$. By Hoeffding's inequality, for arbitrary $a > 0$,

$$\mathbb{P}(||\mathcal{S}_{\pm 1}| - n/2| \geq a) \leq 2\exp(-2a^2 n^{-1}).$$

Setting $a = \sqrt{n \log(4/\delta)/2}$ and taking union bound, we have with probability at least $1 - \delta$,

$$\left||\mathcal{S}_{\pm 1}| - \frac{n}{2}\right| \leq \sqrt{\frac{n \log(4/\delta)}{2}}.$$

Hence the proof is complete. □

**Lemma B.2.** *Given arbitrary $\delta > 0$, with probability at least $1 - \delta$,*

$$\sigma_\xi^2 d(1 - \widetilde{O}(d^{-1/2})) \leq \|\boldsymbol{\xi}_i\|^2 \leq \sigma_\xi^2 d(1 + \widetilde{O}(d^{-1/2}))$$
$$|\langle \boldsymbol{\xi}_i, \boldsymbol{\xi}_{i'} \rangle| \leq 2\sigma_\xi^2 \sqrt{d \log(4n^2/\delta)}$$

*for all $i, i' \in [n]$.*

*Proof of Lemma B.2.* The proof is the same as in (Cao et al., 2022; Kou et al., 2023) and we include here for completeness. By Bernstein's inequality, with probability at least $1 - \delta/(2n)$, we have

$$|\|\boldsymbol{\xi}_i\|^2 - \sigma_\xi^2 d| = O(\sigma_\xi^2 \sqrt{d \log(4n/\delta)}),$$

which shows the first result.

For the second claim, we can show by Bernstein's inequality, with probability at least $1 - \delta/(2n^2)$ that for any $i \neq i'$

$$|\langle \boldsymbol{\xi}_i, \boldsymbol{\xi}_{i'} \rangle| \leq 2\sigma_\xi^2 \sqrt{d \log(4n^2/\delta)}$$

Then we apply union bound to show the results hold for all $i, i' \in [n]$. □

## C    CLASSIFICATION

We track the inner product dynamics during the training of supervised classification to elucidate the signal learning and noise learning. We first write the gradient descent dynamics as follows.

$$\mathbf{w}_{j,r}^{k+1} = \mathbf{w}_{j,r}^k - \eta \nabla_{\mathbf{w}_{j,r}} L_S(\mathbf{W}^k)$$
$$= \mathbf{w}_{j,r}^k - \frac{\eta}{nm} \sum_{i=1}^n \ell_i'^k \langle \mathbf{w}_{j,r}^k, \mathbf{x}_i^{(1)} \rangle j y_i \mathbf{x}_i^{(1)} - \frac{\eta}{nm} \sum_{i=1}^n \ell_i'^k \langle \mathbf{w}_{j,r}^k, \boldsymbol{\xi}_i \rangle j y_i \boldsymbol{\xi}_i$$
$$= \mathbf{w}_{j,r}^k - \frac{\eta}{nm} \sum_{i \in \mathcal{S}_1} \ell_i'^k \langle \mathbf{w}_{j,r}^k, \boldsymbol{\mu}_1 \rangle j \boldsymbol{\mu}_1 + \frac{\eta}{nm} \sum_{i \in \mathcal{S}_{-1}} \ell_i'^k \langle \mathbf{w}_{j,r}^k, \boldsymbol{\mu}_{-1} \rangle j \boldsymbol{\mu}_{-1} - \frac{\eta}{nm} \sum_{i=1}^n \ell_i'^k \langle \mathbf{w}_{j,r}^k, \boldsymbol{\xi}_i \rangle j y_i \boldsymbol{\xi}_i$$

Here we restate the Condition 3.1 specific for the case of supervised classification.

**Condition C.1.** *Suppose that*

1. *Dimension $d$ satisfies $d = \widetilde{\Omega}(\max\{n^2 m \sigma_\xi^{-1} \|\boldsymbol{\mu}\|, n^4 m\})$.*

2. *Training sample and network width satisfy $m = \Omega(\log(n/\delta)), n = \Omega(\log(m/\delta))$.*

3. *The initialization variation $\sigma_0$ satisfies $\widetilde{O}(n^2 m \sigma_\xi^{-1} d^{-1}) \le \sigma_0 \le \widetilde{O}(\min\{\|\boldsymbol{\mu}\|^{-1}, \sigma_\xi^{-1} d^{-1/2}\})$.*

4. *The learning rate satisfies $\eta \le \widetilde{O}(\min\{m\|\boldsymbol{\mu}\|^{-2}, nm\sigma_0\sigma_\xi^{-1}d^{-1/2}, nm\sigma_\xi^{-2}d^{-1}\})$*

We make the particular remarks as follows. The lower bound on $m = \widetilde{\Omega}(1)$ is to ensure the initialization is concentrated and thus provides a lower bound on the maximum and average inner product. The lower bound on $n = \widetilde{\Omega}(1)$ is required such that $|\mathcal{S}_1|, |\mathcal{S}_{-1}| = \Theta(n)$ and $\widetilde{O}(n^{-1/2})$ remains small. The lower bound on $\sigma_0$ is required for the noise memorization setting where we need to control the lower bound for the noise inner product at initialization. Thus to ensure the lower bound $\sigma_0$ is valid, we require further conditions on the dimension $d$ apart from $d = \widetilde{\Omega}(n^2)$.

## C.1 Useful lemmas

We first provide a lemma that bound the inner product at initialization.

**Lemma C.1** (Cao et al. (2022)). *Suppose $\delta > 0$ and that $d = \Omega(\log(mn/\delta)), m = \Omega(\log(1/\delta))$, then with probability at least $1 - \delta$,*

$$|\langle \mathbf{w}_{j,r}^0, \boldsymbol{\mu}_{j'}\rangle| \le \sqrt{2\log(8m/\delta)}\sigma_0\|\boldsymbol{\mu}\|$$
$$|\langle \mathbf{w}_{j',r}^0, \boldsymbol{\xi}_i\rangle| \le 2\sqrt{\log(8mn/\delta)}\sigma_0\sigma_\xi\sqrt{d}$$

*for all $j, j' \in \{\pm 1\}, r \in [m], i \in [n]$. In addition,*

$$\max_{r\in[m]} |\langle \mathbf{w}_{j,r}^0, \boldsymbol{\mu}_{j'}\rangle| \ge \sigma_0\|\boldsymbol{\mu}\|/2,$$
$$\max_{r\in[m]} |\langle \mathbf{w}_{j,r}^0, \boldsymbol{\xi}_i\rangle| \ge \sigma_0\sigma_\xi\sqrt{d}/4$$

*for all $j, j' \in \{\pm 1\}, i \in [n]$.*

We decompose the weights into its signal components and noise components.

**Lemma C.2.** *The weight can be decomposed as*

$$\mathbf{w}_{j,r}^k = \mathbf{w}_{j,r}^0 + \zeta_1^k \boldsymbol{\mu}_1 + \zeta_{-1}^k \boldsymbol{\mu}_{-1} + \sum_{i=1}^n \rho_{j,r,i}^k \|\boldsymbol{\xi}_i\|^{-2}\boldsymbol{\xi}_i$$

*where the noise coefficients $\rho_{j,r,i}^k$ satisfy $\rho_{j,r,i}^0 = 0$ and*

$$\rho_{j,r,i}^{k+1} = \rho_{j,r,i}^k - \frac{\eta}{nm}\ell_i'^k \langle \mathbf{w}_{j,r}^k, \boldsymbol{\xi}_i\rangle j y_i \|\boldsymbol{\xi}_i\|^2$$

*for all $j = \pm 1$, $r \in [m]$ and $i \in [n]$.*

*Proof of Lemma C.2.* The proof follows from (Cao et al., 2022; Kou et al., 2023). First, we recall the gradient descent update as

$$\mathbf{w}_{j,r}^{k+1} = \mathbf{w}_{j,r}^k - \frac{\eta}{nm}\sum_{i\in\mathcal{S}_1} \ell_i'^k \langle \mathbf{w}_{j,r}^k, \boldsymbol{\mu}_1\rangle j\boldsymbol{\mu}_1 + \frac{\eta}{nm}\sum_{i\in\mathcal{S}_{-1}} \ell_i'^k \langle \mathbf{w}_{j,r}^k, \boldsymbol{\mu}_{-1}\rangle j\boldsymbol{\mu}_{-1} - \frac{\eta}{nm}\sum_{i=1}^n \ell_i'^k \langle \mathbf{w}_{j,r}^k, \boldsymbol{\xi}_i\rangle j y_i \boldsymbol{\xi}_i$$

$$= \mathbf{w}_{j,r}^0 - \frac{\eta}{nm}\sum_{s=0}^k \sum_{i\in\mathcal{S}_1} \ell_i'^k \langle \mathbf{w}_{j,r}^s, \boldsymbol{\mu}_1\rangle j\boldsymbol{\mu}_1 + \frac{\eta}{nm}\sum_{s=0}^k \sum_{i\in\mathcal{S}_{-1}} \ell_i'^k \langle \mathbf{w}_{j,r}^s, \boldsymbol{\mu}_{-1}\rangle j\boldsymbol{\mu}_{-1}$$

$$- \frac{\eta}{nm}\sum_{s=0}^k \sum_{i=1}^n \ell_i'^k \langle \mathbf{w}_{j,r}^s, \boldsymbol{\xi}_i\rangle j y_i \boldsymbol{\xi}_i.$$

By the data model, we have with probability 1, the vectors are linearly independent and thus the decomposition is unique with

$$\rho_{j,r,i}^k = -\frac{\eta}{nm}\sum_{s=0}^k \ell_i'^k \langle \mathbf{w}_{j,r}^s, \boldsymbol{\xi}_i\rangle j y_i \|\boldsymbol{\xi}_i\|^2$$

Then writing out the iterative update for $\rho_{j,r,i}^k$ completes the proof. □

**Lemma C.3.** *Let $x \sim \mathcal{N}(0, \sigma^2)$. Then $\mathbb{P}(|x| \leq c) \leq \mathrm{erf}\left(\frac{c}{\sqrt{2}\sigma}\right) \leq \sqrt{1 - \exp(-\frac{2c^2}{\pi\sigma^2})}$.*

*Proof of Lemma C.3.* The probability density function for $x$ is given by

$$f(x) = \frac{1}{\sqrt{2\pi}\sigma} \exp(-\frac{x^2}{2\sigma^2}).$$

Then we know that

$$\mathbb{P}(|x| \leq c) = \frac{1}{\sqrt{2\pi}\sigma} \int_{-c}^{c} \exp(-\frac{x^2}{2\sigma^2}) dx.$$

By the definition of $\mathrm{erf}$ function

$$\mathrm{erf}(c) = \frac{2}{\sqrt{\pi}} \int_0^c \exp(-x^2) dx,$$

and variable substitution yields

$$\mathrm{erf}(\frac{c}{\sqrt{2}\sigma}) = \frac{1}{\sqrt{2\pi}\sigma} \int_0^c \exp(-\frac{x^2}{2\sigma^2}) dx.$$

Therefore, we first conclude $\mathbb{P}(|x| \leq c) = 2\mathrm{erf}(\frac{c}{\sqrt{2}\sigma})$. Next, by the inequality $\mathrm{erf}(x) \leq \sqrt{1 - \exp(-4x^2/\pi)}$, we obtain the desired result. $\qquad\square$

## C.2    SCALE OF INNER PRODUCTS

We first derive a global bound for the growth of inner products until convergence. To this end, we let $T^* = \eta^{-1}\mathrm{poly}(\|\boldsymbol{\mu}\|^{-1}, \sigma_\xi^{-2}d^{-1}, \sigma_0^{-1}, n, m, d)$ be the maximum number of iterations considered and let $\alpha = 2\log(T^*)$. We also denote $\beta := 3\max_{j,r,i,y}\{|\langle \mathbf{w}_{j,r}^0, \boldsymbol{\mu}_y \rangle|, |\langle \mathbf{w}_{j,r}^0, \boldsymbol{\xi}_i \rangle|\}$. Then from Lemma C.1 and from Condition C.1, we can bound

$$3\max\{\sigma_0\|\boldsymbol{\mu}\|/2, \sigma_0\sigma_\xi\sqrt{d}/4\} \leq \beta \leq 1/C \tag{4}$$

for some sufficiently large constant $C > 0$.

**Proposition C.1.** *Under Condition C.1, for all $0 \leq k \leq T^*$, we can bound*

$$|\langle \mathbf{w}_{j,r}^k, \boldsymbol{\mu}_j \rangle|, |\langle \mathbf{w}_{y_i,r}^k, \boldsymbol{\xi}_i \rangle|, |\rho_{y_i,r,i}^k| \leq \alpha, \tag{5}$$

$$|\langle \mathbf{w}_{j,r}^k, \boldsymbol{\mu}_{-j} \rangle| \leq \beta, \tag{6}$$

$$|\langle \mathbf{w}_{-y_i,r}^k, \boldsymbol{\xi}_i \rangle|, |\rho_{-y_i,r,i}^k| \leq \beta + 12\sqrt{\frac{\log(4n^2/\delta)}{d}}n\alpha \tag{7}$$

*for all $i \in [n]$, $r \in [m]$ and $j = \pm 1$.*

We will prove the bound by induction and we first derive several intermediate lemmas as follows.

**Lemma C.4.** *Suppose results in Proposition C.1 hold at iteration $k$, then we have $F_j(\mathbf{W}_j^k, \mathbf{x}_i) \leq 0.5$ for all $i \in [n]$, $j \neq y_i$.*

*Proof of Lemma C.4.* Recall that

$$\begin{aligned}
F_j(\mathbf{W}_j^k, \mathbf{x}_i) &= \frac{1}{m}\sum_{r=1}^m \left(\langle \mathbf{w}_{j,r}^k, \mathbf{x}_i^{(1)} \rangle^2 + \langle \mathbf{w}_{j,r}^k, \mathbf{x}_i^{(2)} \rangle^2\right) \\
&= \frac{1}{m}\sum_{r=1}^m \left(\langle \mathbf{w}_{j,r}^k, \boldsymbol{\mu}_{y_i} \rangle^2 + \langle \mathbf{w}_{j,r}^k, \boldsymbol{\xi}_i \rangle^2\right) \\
&\leq \beta^2 + \left(\beta + 12\sqrt{\frac{\log(4n^2/\delta)}{d}}n\alpha\right)^2 \\
&\leq 0.5
\end{aligned}$$

where the second last inequality is by (6) and (7). The last inequality is by Condition C.1 such that $\beta \leq 1/C \leq 0.25$ and $d \geq 144n^2\alpha^2\log(4n^2/\delta)$. $\qquad\square$

**Lemma C.5.** *Suppose results in Proposition C.1 hold at iteration $k$, then we have*

$$|\langle \mathbf{w}_{j,r}^k - \mathbf{w}_{j,r}^0, \boldsymbol{\xi}_i \rangle - \rho_{j,r,i}^k| \le 4\sqrt{\frac{\log(4n^2/\delta)}{d}} n\alpha,$$

*for all $j = \pm 1, r \in [m], i \in [n]$.*

*Proof.* By Lemma C.2, we recall the decomposition as

$$\mathbf{w}_{j,r}^k = \mathbf{w}_{j,r}^0 + \zeta_1^k \boldsymbol{\mu}_1 + \zeta_{-1}^k \boldsymbol{\mu}_{-1} + \sum_{i=1}^n \rho_{j,r,i}^k \|\boldsymbol{\xi}_i\|^{-2} \boldsymbol{\xi}_i.$$

By the orthogonality, we can show

$$\langle \mathbf{w}_{j,r}^k, \boldsymbol{\xi}_i \rangle = \langle \mathbf{w}_{j,r}^0, \boldsymbol{\xi}_i \rangle + \rho_{j,r,i}^k + \sum_{i \ne i'} \rho_{j,r,i'}^k \|\boldsymbol{\xi}_{i'}\|^{-2} \langle \boldsymbol{\xi}_i, \boldsymbol{\xi}_{i'} \rangle$$

By Lemma B.2 and suppose $d = \Omega(\log(n/\delta))$, then $|\langle \boldsymbol{\xi}_i, \boldsymbol{\xi}_{i'} \rangle| \|\boldsymbol{\xi}_i\|^{-2} \le 4\sqrt{\log(4n^2/\delta)d^{-1}}$. Thus we have

$$|\langle \mathbf{w}_{j,r}^k - \mathbf{w}_{j,r}^0, \boldsymbol{\xi}_i \rangle - \bar{\rho}_{j,r,i}^k| \le 4\sqrt{\frac{\log(4n^2/\delta)}{d}} n\alpha,$$

where we use the upper bound on $|\bar{\rho}_{j,r,i}^k| \le \alpha$. $\qquad\square$

**Lemma C.6.** *For any $r \in [m], j, y = \pm 1$, we have $\mathrm{sign}(\langle \mathbf{w}_{j,r}^0, \boldsymbol{\mu}_y \rangle) = \mathrm{sign}(\langle \mathbf{w}_{j,r}^k, \boldsymbol{\mu}_y \rangle)$ for all $0 \le k \le T^*$.*

*Proof of Lemma C.6.* We prove the results by induction. First, it is clear at $k = 0$, the results are satisfied. Then suppose there exists an iteration $\widetilde{k}$ such that $\mathrm{sign}(\langle \mathbf{w}_{j,r}^k, \boldsymbol{\mu}_y \rangle) = \mathrm{sign}(\langle \mathbf{w}_{j,r}^0, \boldsymbol{\mu}_y \rangle)$ holds for all $k \le \widetilde{k} - 1$, we show the sign invariance also holds at $\widetilde{k}$. Recall the gradient descent update as

$$\mathbf{w}_{j,r}^{k+1} = \mathbf{w}_{j,r}^k - \frac{\eta}{nm} \sum_{i \in \mathcal{S}_1} \ell_i'^k \langle \mathbf{w}_{j,r}^k, \boldsymbol{\mu}_1 \rangle j \boldsymbol{\mu}_1 + \frac{\eta}{nm} \sum_{i \in \mathcal{S}_{-1}} \ell_i'^k \langle \mathbf{w}_{j,r}^k, \boldsymbol{\mu}_{-1} \rangle j \boldsymbol{\mu}_{-1}$$

$$- \frac{\eta}{nm} \sum_{i=1}^n \ell_i'^k \langle \mathbf{w}_{j,r}^k, \boldsymbol{\xi}_i \rangle j y_i \boldsymbol{\xi}_i.$$

Then the update of the inner product is

$$\langle \mathbf{w}_{j,r}^{\widetilde{k}}, \boldsymbol{\mu}_y \rangle = \langle \mathbf{w}_{j,r}^{\widetilde{k}-1}, \boldsymbol{\mu}_y \rangle - \frac{\eta}{nm} \sum_{i \in \mathcal{S}_y} \ell_i'^{\widetilde{k}-1} \langle \mathbf{w}_{j,r}^{\widetilde{k}-1}, \boldsymbol{\mu}_y \rangle j y \|\boldsymbol{\mu}\|^2$$

$$= \left(1 - \frac{\eta}{nm} j y \sum_{i \in \mathcal{S}_y} \ell_i'^{\widetilde{k}-1} \|\boldsymbol{\mu}\|^2\right) \langle \mathbf{w}_{j,r}^{\widetilde{k}-1}, \boldsymbol{\mu}_y \rangle$$

By the condition that $\eta \le C^{-1} m \|\boldsymbol{\mu}\|^{-2}$ for sufficiently large constant $C$, we have $|\frac{\eta}{nm} j y \sum_{i \in \mathcal{S}_y} \ell_i'^{\widetilde{k}-1} \|\boldsymbol{\mu}\|^2| < 1$. Thus we can guarantee the $\mathrm{sign}(\langle \mathbf{w}_{j,r}^{\widetilde{k}}, \boldsymbol{\mu}_y \rangle) = \mathrm{sign}(\langle \mathbf{w}_{j,r}^{\widetilde{k}-1}, \boldsymbol{\mu}_y \rangle) = \mathrm{sign}(\langle \mathbf{w}_{j,r}^0, \boldsymbol{\mu}_y \rangle)$. $\qquad\square$

*Proof of Proposition C.1.* We prove the results by induction. For $\rho_{j,r,i}^k$, we prove a stronger result that $|\rho_{y_i,r,i}^k| \le 0.9\alpha \le \alpha$ and $|\rho_{-y_i,r,i}^k| \le 0.6\beta + 8\sqrt{\frac{\log(4n^2/\delta)}{d}} n\alpha$. First it is clear at $t = 0$, the results are satisfied based on the definition of $\beta$ and $\alpha \ge \beta$. Now suppose that there exists $\widetilde{T} \le T^*$ such that results hold for all $0 \le k \le \widetilde{T} - 1$. We wish to show the results also hold for $k = \widetilde{T}$.

First recall the gradient descent update as

$$\mathbf{w}_{j,r}^{k+1} = \mathbf{w}_{j,r}^k - \frac{\eta}{nm} \sum_{i \in \mathcal{S}_1} \ell_i'^k \langle \mathbf{w}_{j,r}^k, \boldsymbol{\mu}_1 \rangle j \boldsymbol{\mu}_1 + \frac{\eta}{nm} \sum_{i \in \mathcal{S}_{-1}} \ell_i'^k \langle \mathbf{w}_{j,r}^k, \boldsymbol{\mu}_{-1} \rangle j \boldsymbol{\mu}_{-1}$$

$$- \frac{\eta}{nm} \sum_{i=1}^{n} \ell_i'^k \langle \mathbf{w}_{j,r}^k, \boldsymbol{\xi}_i \rangle j y_i \boldsymbol{\xi}_i.$$

Then based on the orthogonal data modelling assumption, we have for $y \neq j$, i.e., $y = -j$,

$$\langle \mathbf{w}_{j,r}^{k+1}, \boldsymbol{\mu}_{-j} \rangle = \langle \mathbf{w}_{j,r}^k, \boldsymbol{\mu}_{-j} \rangle + \frac{\eta}{nm} \sum_{i \in \mathcal{S}_{-j}} \ell_i'^k \langle \mathbf{w}_{j,r}^k, \boldsymbol{\mu}_{-j} \rangle \|\boldsymbol{\mu}\|^2$$

$$= \Big( 1 - \frac{\eta \|\boldsymbol{\mu}\|^2}{nm} \sum_{i \in \mathcal{S}_{-j}} |\ell_i'^k| \Big) \langle \mathbf{w}_{j,r}^k, \boldsymbol{\mu}_{-j} \rangle$$

where the second equality is by $\ell_i'^k < 0$ for all $i, k$. From Lemma C.6, we have $\text{sign}(\langle \mathbf{w}_{j,r}^{k+1}, \boldsymbol{\mu}_{-j} \rangle) = \text{sign}(\langle \mathbf{w}_{j,r}^k, \boldsymbol{\mu}_{-j} \rangle)$ and thus

$$|\langle \mathbf{w}_{j,r}^{\widetilde{T}}, \boldsymbol{\mu}_{-j} \rangle| \leq \left| \Big( 1 - \frac{\eta \|\boldsymbol{\mu}\|^2}{nm} \sum_{i \in \mathcal{S}_{-j}} |\ell_i'^{\widetilde{T}-1}| \Big) \right| \left| \langle \mathbf{w}_{j,r}^{\widetilde{T}-1}, \boldsymbol{\mu}_{-j} \rangle \right| \leq \left| \langle \mathbf{w}_{j,r}^{\widetilde{T}-1}, \boldsymbol{\mu}_{-j} \rangle \right| \leq \beta$$

On the other hand, for $y = j$, we have

$$\langle \mathbf{w}_{j,r}^{k+1}, \boldsymbol{\mu}_j \rangle = \langle \mathbf{w}_{j,r}^k, \boldsymbol{\mu}_j \rangle - \frac{\eta}{nm} \sum_{i \in \mathcal{S}_j} \ell_i'^k \langle \mathbf{w}_{j,r}^k, \boldsymbol{\mu}_j \rangle \|\boldsymbol{\mu}\|^2$$

$$= \langle \mathbf{w}_{j,r}^k, \boldsymbol{\mu}_j \rangle + \frac{\eta \|\boldsymbol{\mu}\|^2}{nm} \sum_{i \in \mathcal{S}_j} |\ell_i'^k| \langle \mathbf{w}_{j,r}^k, \boldsymbol{\mu}_j \rangle$$

Next, we notice that

$$|\ell_i'^k| = \frac{1}{1 + \exp \big( F_{y_i}(\mathbf{W}_{y_i}^k, \mathbf{x}_i) - F_{-y_i}(\mathbf{W}_{-y_i}^k, \mathbf{x}_i) \big)}$$

$$\leq \exp \big( - F_{y_i}(\mathbf{W}_{y_i}^k, \mathbf{x}_i) + F_{-y_i}(\mathbf{W}_{-y_i}^k, \mathbf{x}_i) \big)$$

$$\leq \exp \big( - F_{y_i}(\mathbf{W}_{y_i}^k, \mathbf{x}_i) + 0.5 \big)$$

$$= \exp \big( - \frac{1}{m} \sum_{r=1}^{m} \big( \langle \mathbf{w}_{y_i,r}^k, \boldsymbol{\mu}_{y_i} \rangle^2 + \langle \mathbf{w}_{y_i,r}^k, \boldsymbol{\xi}_i \rangle^2 \big) + 0.5 \big) \tag{8}$$

where the last inequality is by Lemma C.4. Let $k_{j,r}$ be the last time $k \leq T^*$ that $|\langle \mathbf{w}_{j,r}^k, \boldsymbol{\mu}_j \rangle| \leq 0.5\alpha$. Then we have

$$\langle \mathbf{w}_{j,r}^{\widetilde{T}}, \boldsymbol{\mu}_j \rangle = \langle \mathbf{w}_{j,r}^{k_{j,r}}, \boldsymbol{\mu}_j \rangle + \underbrace{\frac{\eta \|\boldsymbol{\mu}\|^2}{nm} |\ell_i'^{k_{j,r}}| \langle \mathbf{w}_{j,r}^{k_{j,r}}, \boldsymbol{\mu}_j \rangle}_{A_1} + \underbrace{\frac{\eta \|\boldsymbol{\mu}\|^2}{nm} \sum_{k_{j,r} < k \leq \widetilde{T}-1} |\ell_i'^k| \langle \mathbf{w}_{j,r}^k, \boldsymbol{\mu}_j \rangle}_{A_2}.$$

Without loss of generality, we suppose $\langle \mathbf{w}_{j,r}^0, \boldsymbol{\mu}_j \rangle \geq 0$, then by Lemma C.6, $\langle \mathbf{w}_{j,r}^k, \boldsymbol{\mu}_j \rangle \geq 0$ for all $k \geq 0$. Then we can bound

$$|A_1| \leq \frac{\eta \|\boldsymbol{\mu}\|^2}{nm} 0.5\alpha \leq 0.25\alpha$$

where the last inequality is by the condition that $\eta \leq nm \|\boldsymbol{\mu}\|^{-2}/2$. Furthermore,

$$|A_2| \leq \frac{\eta \|\boldsymbol{\mu}\|^2}{nm} \sum_{k_{j,r} < k \leq \widetilde{T}-1} \exp(-F_{y_i}(\mathbf{W}_{y_i}^k, \mathbf{x}_i) + 0.5) \langle \mathbf{w}_{j,r}^k, \boldsymbol{\mu}_j \rangle$$

$$\leq \frac{2\eta \|\boldsymbol{\mu}\|^2 \alpha}{nm} T^* \exp(-\alpha^2/4)$$

$$= \frac{2\eta \|\boldsymbol{\mu}\|^2 \alpha}{nm} T^* \exp(-\log(T^*))$$

$$\leq 0.25\alpha$$

where the first inequality is by (8) and the second inequality is by upper bound on $\langle \mathbf{w}_{j,r}^k, \boldsymbol{\mu}_j \rangle \leq \alpha$ for all $k \leq \widetilde{T} - 1$. The equality is by the definition of $\alpha = 2\log(T^*)$ and the last inequality is by the condition $\eta \leq nm\|\boldsymbol{\mu}\|^{-2}/8$. Thus, we can show

$$\langle \mathbf{w}_{j,r}^{\widetilde{T}}, \boldsymbol{\mu}_j \rangle \leq 0.5\alpha + 0.25\alpha + 0.25\alpha = \alpha.$$

Next for the noise growth, from Lemma C.2, we have for $y_i \neq j$

$$\rho_{-y_i,r,i}^{\widetilde{T}} = \rho_{-y_i,r,i}^{\widetilde{T}-1} + \frac{\eta}{nm} \ell_i'^k \langle \mathbf{w}_{-y_i,r}^{\widetilde{T}-1}, \boldsymbol{\xi}_i \rangle \|\boldsymbol{\xi}_i\|^2. \tag{9}$$

When $|\rho_{-y_i,r,i}^{\widetilde{T}-1}| \leq 1.5\big(0.3\beta + 4\sqrt{\frac{\log(4n^2/\delta)}{d}}n\alpha\big)$, we have

$$|\rho_{-y_i,r,i}^{\widetilde{T}}| \leq |\rho_{-y_i,r,i}^{\widetilde{T}-1}| + \frac{2\eta\sigma_\xi^2 d\alpha}{nm} \leq |\rho_{-y_i,r,i}^{\widetilde{T}-1}| + 0.15\beta \leq 0.6\beta + 8\sqrt{\frac{\log(4n^2/\delta)}{d}}n\alpha$$

where the second inequality is by triangle inequality and $|\ell_i'^k| \leq 1$ and Lemma B.2. The third inequality is by the lower bound on $\beta$ in (4) and the condition that $\eta \leq 0.05nm\sigma_0\sigma_\xi^{-1}d^{-1/2}\alpha$.

Further, because $|\langle \mathbf{w}_{-y_i,r}^0, \boldsymbol{\xi}_i \rangle| \leq 0.3\beta$, when $1.5\big(0.3\beta + 4\sqrt{\frac{\log(4n^2/\delta)}{d}}n\alpha\big) \leq |\rho_{-y_i,r,i}^{\widetilde{T}-1}| \leq 0.6\beta + 8\sqrt{\frac{\log(4n^2/\delta)}{d}}n\alpha$, we can show from Lemma C.5 that if $\rho_{-y_i,r,i}^{\widetilde{T}-1} > 0$, then

$$\frac{1}{3}\rho_{-y_i,r,i}^{\widetilde{T}-1} \leq \langle \mathbf{w}_{-y_i,r}^{\widetilde{T}-1}, \boldsymbol{\xi}_i \rangle \leq \frac{4}{3}\rho_{-y_i,r,i}^{\widetilde{T}-1}$$

Then (9) suggests

$$\rho_{-y_i,r,i}^{\widetilde{T}} \leq \big(1 - \frac{\eta\|\boldsymbol{\xi}_i\|^2}{3nm}|\ell_i'^k|\big)\rho_{-y_i,r,i}^{\widetilde{T}-1} \leq \rho_{-y_i,r,i}^{\widetilde{T}-1} \leq 0.6\beta + 8\sqrt{\frac{\log(4n^2/\delta)}{d}}n\alpha$$

If $\rho_{-y_i,r,i}^{\widetilde{T}-1} < 0$, then

$$\frac{4}{3}\rho_{-y_i,r,i}^{\widetilde{T}-1} \leq \langle \mathbf{w}_{-y_i,r}^{\widetilde{T}-1}, \boldsymbol{\xi}_i \rangle \leq \frac{1}{3}\rho_{-y_i,r,i}^{\widetilde{T}-1}$$

Then (9) suggests

$$\rho_{-y_i,r,i}^{\widetilde{T}} \geq \big(1 - \frac{\eta\|\boldsymbol{\xi}_i\|^2}{3nm}|\ell_i'^k|\big)\rho_{-y_i,r,i}^{\widetilde{T}-1} \geq \rho_{-y_i,r,i}^{\widetilde{T}-1} \geq -0.6\beta - 8\sqrt{\frac{\log(4n^2/\delta)}{d}}n\alpha$$

Thus this completes the proof that $|\rho_{-y_i,r,i}^{\widetilde{T}}| \leq 0.6\beta + 8\sqrt{\frac{\log(4n^2/\delta)}{d}}n\alpha$.

Finally, by Lemma C.5 we have for all $k \geq 0$

$$|\langle \mathbf{w}_{-y_i,r}^k, \boldsymbol{\xi}_i \rangle| \leq |\langle \mathbf{w}_{-y_i,r}^0, \boldsymbol{\xi}_i \rangle| + |\rho_{-y_i,r,i}^k| + 4\sqrt{\frac{\log(4n^2/\delta)}{d}}n\alpha \leq 0.9\beta + 12\sqrt{\frac{\log(4n^2/\delta)}{d}}n\alpha$$

which proves the upper bound for $|\langle \mathbf{w}_{-y_i,r}^{\widetilde{T}}, \boldsymbol{\xi}_i \rangle|$ and $|\rho_{-y_i,r,i}^{\widetilde{T}}|$.

Next, from Lemma C.2, we have for $y_i = j$,

$$\rho_{y_i,r,i}^{k+1} = \rho_{y_i,r,i}^k - \frac{\eta}{nm} \ell_i'^k \langle \mathbf{w}_{y_i,r}^k, \boldsymbol{\xi}_i \rangle \|\boldsymbol{\xi}_i\|^2. \tag{10}$$

Let $\tilde{k}_{r,i}$ be the last time $k < T^*$ that $|\rho_{y_i,r,i}^k| \leq 0.6\alpha$. Then it can be verified that for $k \geq \tilde{k}_{r,i}$,

$$|\langle \mathbf{w}_{y_i,r}^k, \boldsymbol{\xi}_i \rangle| \geq |\rho_{y_i,r,i}^k| - |\langle \mathbf{w}_{y_i,r}^0, \boldsymbol{\xi}_i \rangle| - 4\sqrt{\frac{\log(4n^2/\delta)}{d}}n\alpha \geq 0.5\alpha$$

where the first inequality is by Lemma C.5 and the last inequality is by $|\langle \mathbf{w}_{y_i,r}^0, \boldsymbol{\xi}_i \rangle| + 4\sqrt{\frac{\log(4n^2/\delta)}{d}}n\alpha \leq 1 \leq 0.1\alpha$.

We now expand (10) as

$$\rho_{y_i,r,i}^{\widetilde{T}} = \rho_{y_i,r,i}^{\tilde{k}_{r,i}} + \underbrace{\frac{\eta}{nm}|\ell_i'^{\tilde{k}_{r,i}}||\langle \mathbf{w}_{y_i,r}^{\tilde{k}_{r,i}}, \boldsymbol{\xi}_i\rangle\|\boldsymbol{\xi}_i\|^2}_{A_3} + \underbrace{\frac{\eta}{nm}\sum_{\tilde{k}_{r,i}<k\leq\widetilde{T}-1}|\ell_i'^k||\langle \mathbf{w}_{y_i,r}^k, \boldsymbol{\xi}_i\rangle\|\boldsymbol{\xi}_i\|^2}_{A_4}$$

Then we can bound

$$|A_3| \leq \frac{2\eta\sigma_\xi^2 d}{nm}|\langle \mathbf{w}_{y_i,r}^{\tilde{k}_{r,i}}, \boldsymbol{\xi}_i\rangle| \leq \frac{2\eta\sigma_\xi^2 d}{nm}\left(|\langle \mathbf{w}_{y_i,r}^0, \boldsymbol{\xi}_i\rangle| + 0.6\alpha + 4\sqrt{\frac{\log(4n^2/\delta)}{d}}n\alpha\right)$$

$$\leq \frac{2\eta\sigma_\xi^2 d}{nm}0.7\alpha$$

$$\leq 0.15\alpha$$

where the first inequality is by $|\ell_i'^k| \leq 1$ and Lemma B.2 with $d = \Omega(\log(n/\delta))$ and the second inequality is by Lemma C.5. The last inequality is by the condition $\eta \leq C^{-1}nm\sigma_\xi^{-2}d^{-1}$ for sufficiently large constant $C$.

In addition, we bound

$$|A_4| \leq \frac{2\eta\sigma_\xi^2 d\alpha}{nm}\sum_{k_{j,r}<k\leq\widetilde{T}-1}\exp(-F_{y_i}(\mathbf{W}_{y_i}^k, \mathbf{x}_i) + 0.5)$$

$$\leq \frac{4\eta\sigma_\xi^2 d\alpha}{nm}T^*\exp(-\alpha^2/4)$$

$$\leq \frac{4\eta\sigma_\xi^2 d\alpha}{nm}$$

$$\leq 0.15\alpha$$

where the first inequality is by (8) and the second inequality is by $|\langle \mathbf{w}_{y_i,r}^k, \boldsymbol{\xi}_i\rangle| \geq 0.6\alpha - 0.1\alpha = 0.5\alpha$. The last inequality is by the condition $\eta \leq C^{-1}nm\sigma_\xi^{-2}d^{-1}$ for sufficiently large constant $C$.

Combining the bound on $|A_3|$ and $|A_4|$, we have

$$|\rho_{y_i,r,i}^{\widetilde{T}}| \leq 0.6\alpha + 0.15\alpha + 0.15\alpha = 0.9\alpha.$$

Lastly, we bound

$$|\langle \mathbf{w}_{y_i,r}^{\widetilde{T}}, \boldsymbol{\xi}_i\rangle| \leq |\langle \mathbf{w}_{y_i,r}^0, \boldsymbol{\xi}_i\rangle| + |\rho_{y_i,r,i}^{\widetilde{T}}| + 4\sqrt{\frac{\log(4n^2/\delta)}{d}}n\alpha \leq 0.3\beta + 4\sqrt{\frac{\log(4n^2/\delta)}{d}}n\alpha + 0.9\alpha$$

$$\leq \alpha.$$

This shows the upper bound as $|\langle \mathbf{w}_{y_i,r}^{\widetilde{T}}, \boldsymbol{\xi}_i\rangle|, |\rho_{y_i,r,i}^{\widetilde{T}}| \leq \alpha.$ $\qquad\square$

We require the following lemma that lower bound the loss derivatives in the first stage before the inner products reach constant order.

**Lemma C.7.** *If* $\max_{r,i,y}\{\langle \mathbf{w}_{j,r}^k, \boldsymbol{\mu}_y\rangle, \langle \mathbf{w}_{j,r}^k, \boldsymbol{\xi}_i\rangle\} = O(1)$*, there exists a constant* $C_\ell > 0$ *such that* $|\ell_i'^k| \geq C_\ell$ *for all* $i \in [n]$.

*Proof of Lemma C.7.* If $\max_{r,i,y}\{\langle \mathbf{w}_{j,r}^k, \boldsymbol{\mu}_y\rangle, \langle \mathbf{w}_{j,r}^k, \boldsymbol{\xi}_i\rangle\} = O(1)$, we can bound for all $j = \pm 1$

$$F_j(\mathbf{W}_j^k, \mathbf{x}_i) = \frac{1}{m}\sum_{r=1}^m\left(\langle \mathbf{w}_{j,r}^k, \boldsymbol{\mu}_{y_i}\rangle^2 + \langle \mathbf{w}_{j,r}^k, \boldsymbol{\xi}_i\rangle^2\right) \leq O(1)$$

Therefore, we can bound $|\ell_i'^k| = (1 + \exp(F_{y_i}(\mathbf{W}_{y_i}^k, \mathbf{x}_i) - F_{-y_i}(\mathbf{W}_{-y_i}^k, \mathbf{x}_i)))^{-1} \geq \Omega(1).$ $\qquad\square$

We also prove the following upper bound on the gradient norm.

**Lemma C.8** (Proof of Lemma C.8). *Under Condition C.1, for $0 \leq k \leq T^*$, we can bound*

$$\|\nabla L_S(\mathbf{W}^k)\|^2 = O(\max\{\|\boldsymbol{\mu}\|^2, \sigma_\xi^2 d\}) L_S(\mathbf{W}^k)$$

*Proof of Lemma C.8.* The proof adopts a similar argument as in (Cao et al., 2022, Lemma C.7) and we include here for completeness. We first bound

$$\|\nabla f(\mathbf{W}^k, \mathbf{x}_i)\| \leq \frac{2}{m} \sum_{j,r} \left\| \langle \mathbf{w}_{j,r}^k, \boldsymbol{\mu}_{y_i} \rangle \boldsymbol{\mu}_{y_i} + \langle \mathbf{w}_{j,r}^k, \boldsymbol{\xi}_i \rangle \boldsymbol{\xi}_i \right\|$$

$$\leq \frac{2}{m} \sum_r |\langle \mathbf{w}_{y_i,r}^k, \boldsymbol{\mu}_{y_i} \rangle| \|\boldsymbol{\mu}\| + \frac{2}{m} \sum_r |\langle \mathbf{w}_{y_i,r}^k, \boldsymbol{\xi}_i \rangle| \|\boldsymbol{\xi}_i\|$$

$$+ \frac{2}{m} \sum_r |\langle \mathbf{w}_{-y_i,r}^k, \boldsymbol{\mu}_{y_i} \rangle| \|\boldsymbol{\mu}\| + \frac{2}{m} \sum_r |\langle \mathbf{w}_{-y_i,r}^k, \boldsymbol{\xi}_i \rangle| \|\boldsymbol{\xi}_i\|$$

$$\leq \frac{2}{m} \sum_{r=1}^m \left( |\langle \mathbf{w}_{y_i,r}^k, \boldsymbol{\mu}_{y_i} \rangle| + |\langle \mathbf{w}_{y_i,r}^k, \boldsymbol{\xi}_i \rangle| \right) \max\{\|\boldsymbol{\mu}\|, 2\sigma_\xi \sqrt{d}\}$$

$$+ \frac{2}{m} \sum_{r=1}^m \left( |\langle \mathbf{w}_{-y_i,r}^k, \boldsymbol{\mu}_{y_i} \rangle| + |\langle \mathbf{w}_{-y_i,r}^k, \boldsymbol{\xi}_i \rangle| \right) \max\{\|\boldsymbol{\mu}\|, 2\sigma_\xi \sqrt{d}\}$$

$$\leq 2 \left( \sqrt{F_{y_i}(\mathbf{W}_{y_i}^k, \mathbf{x}_i)} + \sqrt{F_{-y_i}(\mathbf{W}_{-y_i}^k, \mathbf{x}_i)} \right) \max\{\|\boldsymbol{\mu}\|, 2\sigma_\xi \sqrt{d}\}$$

$$\leq 2 \left( \sqrt{F_{y_i}(\mathbf{W}_{y_i}^k, \mathbf{x}_i)} + 1 \right) \max\{\|\boldsymbol{\mu}\|, 2\sigma_\xi \sqrt{d}\}$$

where the third inequality is by Lemma B.2 and the fourth inequality is by Jensen's inequality and the last inequality is by Lemma C.4 that $F_{-y_i}(\mathbf{W}_{-y_i}^k, \mathbf{x}_i)$ for all $i \in [n]$. Then we have

$$- \ell'(y_i f(\mathbf{W}^k, \mathbf{x}_i)) \|\nabla f(\mathbf{W}^k, \mathbf{x}_i)\|^2$$

$$\leq -\ell' \left( F_{y_i}(\mathbf{W}_{y_i}^k, \mathbf{x}_i) - 0.5 \right) \left( 2 \left( \sqrt{F_{y_i}(\mathbf{W}_{y_i}^k, \mathbf{x}_i)} + 1 \right) \max\{\|\boldsymbol{\mu}\|, 2\sigma_\xi \sqrt{d}\} \right)^2$$

$$= -4\ell' \left( F_{y_i}(\mathbf{W}_{y_i}^k, \mathbf{x}_i) - 0.5 \right) \left( \sqrt{F_{y_i}(\mathbf{W}_{y_i}^k, \mathbf{x}_i)} + 1 \right)^2 \max\{\|\boldsymbol{\mu}\|^2, 4\sigma_\xi^2 d\}$$

$$\leq \max_{z>0} \{-4\ell'(z - 0.5)(\sqrt{z} + 1)^2\} \max\{\|\boldsymbol{\mu}\|^2, 4\sigma_\xi^2 d\}$$

$$= O(\max\{\|\boldsymbol{\mu}\|^2, \sigma_\xi^2 d\})$$

where the last equality is by $\max_{z>0}\{-4\ell'(z-0.5)(\sqrt{z}+1)^2\} < \infty$ because $\ell'$ has an exponentially decaying tail. Then we can bound

$$\|\nabla L_S(\mathbf{W}^k)\|^2 \leq \left( \frac{1}{n} \sum_{i=1}^n \ell'(y_i f(\mathbf{W}^k, \mathbf{x}_i)) \|\nabla f(\mathbf{W}^k, \mathbf{x}_i)\| \right)^2$$

$$\leq \left( \frac{1}{n} \sum_{i=1}^n \sqrt{-O(\max\{\|\boldsymbol{\mu}\|^2, \sigma_\xi^2 d\}) \ell'(y_i f(\mathbf{W}^k, \mathbf{x}_i))} \right)^2$$

$$\leq O(\max\{\|\boldsymbol{\mu}\|^2, \sigma_\xi^2 d\}) \frac{1}{n} \sum_{i=1}^n -\ell'(y_i f(\mathbf{W}^k, \mathbf{x}_i))$$

$$\leq O(\max\{\|\boldsymbol{\mu}\|^2, \sigma_\xi^2 d\}) L_S(\mathbf{W}^k)$$

where the third inequality is by Cauchy-Schwartz inequality and the last inequality is by $-\ell' \leq \ell$ for cross-entropy loss. □

## C.3 Signal learning

We first analyze the setting, where $n \cdot \text{SNR}^2 \geq C'$ for some constant $C' > 0$, which allows signal learning to dominate noise memorization, thus reaching benign overfitting.

For the purpose of signal learning, we derive an anti-concentration result that provides a lower bound for signal inner product at initialization.

**Lemma C.9.** *Suppose $\delta > 0$ and $m = \Omega(\log(1/\delta))$. Then with probability at least $1 - \delta$, we have for all $j, y = \pm 1$*

$$\sigma_0 \|\boldsymbol{\mu}\|/2 \leq \frac{1}{m} \sum_{r=1}^n |\langle \mathbf{w}_{j,r}^0, \boldsymbol{\mu}_y \rangle| \leq \sigma_0 \|\boldsymbol{\mu}\|$$

*Proof of Lemma C.16.* First notice that for any $j = \pm 1$, $\langle \mathbf{w}_{j,r}^0, \boldsymbol{\mu}_y \rangle \sim \mathcal{N}(0, \sigma_0^2 \|\boldsymbol{\mu}\|^2)$ and thus we have $\mathbb{E}[|\langle \mathbf{w}_{j,r}^0, \boldsymbol{\mu}_y \rangle|] = \sigma_0 \|\boldsymbol{\mu}\| \sqrt{2/\pi}$. By sub-Gaussian tail bound, with probability at least $1 - \delta/8$, for any $j, y = \pm 1$

$$\left| \frac{1}{m} \sum_{r=1}^m |\langle \mathbf{w}_{j,r}^0, \boldsymbol{\mu}_y \rangle| - \sigma_0 \|\boldsymbol{\mu}\| \sqrt{2/\pi} \right| \leq \sqrt{\frac{2 \sigma_0^2 \|\boldsymbol{\mu}\|^2 \log(8/\delta)}{m}}$$

Choosing $m = \Omega(\log(1/\delta))$, we have

$$\sigma_0 \|\boldsymbol{\mu}\| \sqrt{2/\pi} 0.99 \leq \frac{1}{m} \sum_{r=1}^n |\langle \mathbf{w}_{j,r}^0, \boldsymbol{\mu}_y \rangle| \leq \sigma_0 \|\boldsymbol{\mu}\| \sqrt{2/\pi} 1.01.$$

Then we have $\sigma_0 \|\boldsymbol{\mu}\|/2 \leq \frac{1}{m} \sum_{r=1}^n |\langle \mathbf{w}_{j,r}^0, \boldsymbol{\xi}_i \rangle| \leq \sigma_0 \|\boldsymbol{\mu}\|$. Finally taking the union bound for all $j, y = \pm 1$ completes the proof. □

We have established several preliminary lemmas that hold with high probability, including Lemma B.1, Lemma B.2, Lemma C.1, Lemma C.9. We let $\mathcal{E}_{\text{prelim}}$ be the event such that all the results in these lemmas hold for a given $\delta$. Then by applying union bound, we have $\mathbb{P}(\mathcal{E}_{\text{prelim}}) \geq 1 - 4\delta$. The subsequent analysis are conditioned on the event $\mathcal{E}_{\text{prelim}}$.

### C.3.1 FIRST STAGE

In the first stage where $\max_{r,i,y}\{\langle \mathbf{w}_{j,r}^k, \boldsymbol{\mu}_y \rangle, \langle \mathbf{w}_{j,r}^k, \boldsymbol{\xi}_i \rangle\} = O(1)$, we show in Lemma C.7 that we can lower bound the loss derivatives by a constant $C_\ell$, i.e., $|\ell_i'^k| \geq C_\ell$, for all $i \in [n], k \leq T_1$.

**Theorem C.1.** *Under Condition C.1, suppose $n \cdot \text{SNR}^2 \geq C'$ for some $C' \geq 0$. Then there exists a time $T_1 = \widetilde{\Theta}(\eta^{-1} m \|\boldsymbol{\mu}\|^{-2})$, such that (1) $\max_r |\langle \mathbf{w}_{j,r}^{T_1}, \boldsymbol{\mu}_j \rangle| \geq 2$, for all $j = \pm 1$, (2) $\frac{1}{m} \sum_{r=1}^m |\langle \mathbf{w}_{j,r}^{T_1}, \boldsymbol{\mu}_j \rangle| \geq 2$, for all $j = \pm 1$ (3) $\max_{r,i} |\langle \mathbf{w}_{y_i,r}^{T_1}, \boldsymbol{\xi}_i \rangle| = \widetilde{O}(n^{-1/2})$.*

*Proof of Theorem C.1.* We first upper bound the growth of noise by analyzing inner product dynamics

$$\langle \mathbf{w}_{y_i,r}^k, \boldsymbol{\xi}_i \rangle = \langle \mathbf{w}_{y_i,r}^{k-1}, \boldsymbol{\xi}_i \rangle - \frac{\eta}{nm} \sum_{i'=1}^n \ell_{i'}'^{k-1} \langle \mathbf{w}_{j,r}^{k-1}, \boldsymbol{\xi}_{i'} \rangle \langle \boldsymbol{\xi}_{i'}, \boldsymbol{\xi}_i \rangle$$

$$= \langle \mathbf{w}_{y_i,r}^{k-1}, \boldsymbol{\xi}_i \rangle - \frac{\eta}{nm} \ell_i'^k \langle \mathbf{w}_{y_i,r}^{k-1}, \boldsymbol{\xi}_i \rangle \|\boldsymbol{\xi}_i\|^2 - \frac{\eta}{nm} \sum_{i' \neq i} \ell_{i'}'^{k-1} \langle \mathbf{w}_{y_i,r}^{k-1}, \boldsymbol{\xi}_{i'} \rangle \langle \boldsymbol{\xi}_{i'}, \boldsymbol{\xi}_i \rangle$$

This suggests

$$|\langle \mathbf{w}_{y_i,r}^k, \boldsymbol{\xi}_i \rangle| \leq |\langle \mathbf{w}_{y_i,r}^{k-1}, \boldsymbol{\xi}_i \rangle| + \frac{\eta}{nm} |\ell_i'^k| |\langle \mathbf{w}_{y_i,r}^{k-1}, \boldsymbol{\xi}_i \rangle| \|\boldsymbol{\xi}_i\|^2 + \frac{\eta}{nm} \sum_{i' \neq i} |\ell_{i'}'^k| |\langle \mathbf{w}_{y_i,r}^{k-1}, \boldsymbol{\xi}_{i'} \rangle| |\langle \boldsymbol{\xi}_{i'}, \boldsymbol{\xi}_i \rangle|$$

$$\tag{11}$$

Next, from Lemma C.7 and Lemma B.2, we have for any $i' \neq i \in [n]$ and $k \leq T_1$,

$$\frac{|\ell_{i'}'^k| \cdot |\langle \boldsymbol{\xi}_i, \boldsymbol{\xi}_{i'} \rangle|}{|\ell_i'^k| \cdot \|\boldsymbol{\xi}_i\|^2} \leq \frac{2\sigma_\xi^2 \sqrt{d \log(4n^2/\delta)}}{C_\ell 0.99 \sigma_\xi^2 d} = 2.1 C_\ell^{-1} \sqrt{\frac{\log(4n^2/\delta)}{d}}$$

where we use the lower and upper bound on loss derivatives during the first stage, as well as Lemma B.2. Then taking the maximum of (11) over the neurons and samples, we let $B^k := \max_{r,i} |\langle \mathbf{w}_{y_i,r}^k, \boldsymbol{\xi}_i \rangle|$ and obtain

$$B^k \leq B^{k-1} + \frac{\eta}{nm} \Big( 1 + 2.1 C_\ell^{-1} n \sqrt{\frac{\log(4n^2/\delta)}{d}} \Big) |\ell_i'^k| \|\boldsymbol{\xi}_i\|^2 B^{k-1}$$

$$\leq \Big(1 + \frac{1.01\eta\|\boldsymbol{\xi}_i\|^2}{nm}\Big)B^{k-1}$$

$$\leq \Big(1 + \frac{1.02\eta\sigma_\xi^2 d}{nm}\Big)^k B^0$$

where the second inequality is by $d = \widetilde{\Omega}(n^2)$ sufficiently large and $|\ell_i'^k| \leq 1$. The third inequality is by Lemma B.2.

We then consider the propagation of $\langle \mathbf{w}_{j,r}^k, \boldsymbol{\mu}_y \rangle$. From the gradient update we can show for $j = y$,

$$\langle \mathbf{w}_{j,r}^{k+1}, \boldsymbol{\mu}_j \rangle = \langle \mathbf{w}_{j,r}^k, \boldsymbol{\mu}_j \rangle - \frac{\eta}{nm}\sum_{i\in\mathcal{S}_j}\ell_i'^k\langle \mathbf{w}_{j,r}^k, \boldsymbol{\mu}_j\rangle\|\boldsymbol{\mu}\|^2$$

$$\geq \langle \mathbf{w}_{j,r}^k, \boldsymbol{\mu}_j \rangle + \frac{\eta C_\ell|\mathcal{S}_1|\|\boldsymbol{\mu}\|^2}{nm}\langle \mathbf{w}_{j,r}^k, \boldsymbol{\mu}_j\rangle$$

$$\geq \Big(1 + 0.49\frac{\eta C_\ell\|\boldsymbol{\mu}\|^2}{m}\Big)\langle \mathbf{w}_{j,r}^k, \boldsymbol{\mu}_j\rangle$$

where the first inequality is by loss derivative lower bound and the the second inequality is by Lemma B.1 and $n = \widetilde{\Omega}(1)$ sufficiently large. This implies that

$$|\langle \mathbf{w}_{j,r}^k, \boldsymbol{\mu}_j \rangle| \geq \Big(1 + 0.49\frac{\eta C_\ell\|\boldsymbol{\mu}\|^2}{m}\Big)|\langle \mathbf{w}_{j,r}^{k-1}, \boldsymbol{\mu}_j \rangle| \geq \Big(1 + 0.49\frac{\eta C_\ell\|\boldsymbol{\mu}\|^2}{m}\Big)^k|\langle \mathbf{w}_{j,r}^0, \boldsymbol{\mu}_j \rangle|$$

Applying Lemma C.1 and Lemma C.9, we have for all $j = \pm 1$,

$$\max_r |\langle \mathbf{w}_{j,r}^k, \boldsymbol{\mu}_j \rangle| \geq \Big(1 + 0.49\frac{\eta C_\ell\|\boldsymbol{\mu}\|^2}{m}\Big)^k \sigma_0\|\boldsymbol{\mu}\|/2$$

$$\frac{1}{m}\sum_{r=1}^m |\langle \mathbf{w}_{j,r}^k, \boldsymbol{\mu}_j \rangle| \geq \Big(1 + 0.49\frac{\eta C_\ell\|\boldsymbol{\mu}\|^2}{m}\Big)^k \sigma_0\|\boldsymbol{\mu}\|/2$$

Consider

$$T_1 = \log(4m\sigma_0^{-1}\|\boldsymbol{\mu}\|^{-1})/\log\Big(1 + 0.49\frac{\eta C_\ell\|\boldsymbol{\mu}\|^2}{m}\Big) = \Theta(\eta^{-1}m\|\boldsymbol{\mu}\|^{-2}\log(4m\sigma_0^{-1}\|\boldsymbol{\mu}\|^{-1}))$$

for $\eta$ sufficiently small. Then we can verify that for $j = \pm 1$, we have

$$\max_r |\langle \mathbf{w}_{j,r}^{T_1}, \boldsymbol{\mu}_j \rangle| \geq 2, \quad \frac{1}{m}\sum_{r=1}^m |\langle \mathbf{w}_{j,r}^{T_1}, \boldsymbol{\mu}_j \rangle| \geq 2,$$

Now under the SNR condition, we can bound the growth of noise as

$$B^{T_1} \leq \Big(1 + 1.01\frac{\eta\sigma_\xi^2 d}{nm}\Big)^{T_1} 2\sigma_0\sigma_\xi\sqrt{d}\sqrt{\log(8mn/\delta)}$$

$$= \exp\Big(\frac{\log(1 + 1.02\frac{\eta\sigma_\xi^2 d}{nm})}{\log(1 + 0.49\frac{\eta C_\ell\|\boldsymbol{\mu}\|^2}{m})}\log(4\sigma_0^{-1}\|\boldsymbol{\mu}\|^{-1})\Big)2\sigma_0\sigma_\xi\sqrt{d}\sqrt{\log(8mn/\delta)}$$

$$\leq \exp\Big((2.1/C_\ell n^{-1}\mathrm{SNR}^{-2} + \widetilde{O}(n\mathrm{SNR}^2\eta))\log(4\sigma_0^{-1}\|\boldsymbol{\mu}\|^{-1})\Big)2\sigma_0\sigma_\xi\sqrt{d}\sqrt{\log(8mn/\delta)}$$

$$\leq \exp\Big((2.1/C_\ell n^{-1}\mathrm{SNR}^{-2} + 0.01)\log(4\sigma_0^{-1}\|\boldsymbol{\mu}\|^{-1})\Big)2\sigma_0\sigma_\xi\sqrt{d}\sqrt{\log(8mn/\delta)}$$

$$\leq 8\mathrm{SNR}^{-1}\sqrt{\log(8mn/\delta)}$$

$$= \widetilde{O}(n^{-1/2})$$

where the first inequality is by Lemma C.1 and the second inequality is by Taylor expansion around $\eta = 0$. The third inequality is by choosing $\eta$ sufficiently small and the fourth inequality is by the SNR condition that $n \cdot \mathrm{SNR}^2 \geq C' \geq 2.5C_\ell^{-1}$. $\qquad\square$

### C.3.2 SECOND STAGE

First, at the end of first stage, we have

- $\max_r |\langle \mathbf{w}_{j,r}^{T_1}, \boldsymbol{\mu}_j \rangle| \geq 2$ for all $j = \pm 1$.

- $\frac{1}{m} \sum_{r=1}^m |\langle \mathbf{w}_{j,r}^{T_1}, \boldsymbol{\mu}_j \rangle| \geq 2$ for all $j = \pm 1$.

- $\max_{r,i} |\langle \mathbf{w}_{y_i,r}^{T_1}, \boldsymbol{\xi}_i \rangle| = \widetilde{O}(n^{-1/2})$

- $\max_{r,i} |\langle \mathbf{w}_{-y_i,r}^{T_1}, \boldsymbol{\xi}_i \rangle| \leq \beta + 12\sqrt{\frac{\log(4n^2/\delta)}{d}} n\alpha.$

Next we define

$$\mathbf{w}_{j,r}^* = \mathbf{w}_{j,r}^0 + 2\log(4/\epsilon)\mathrm{sign}(\langle \mathbf{w}_{j,r}^0, \boldsymbol{\mu}_j \rangle)\frac{\boldsymbol{\mu}_j + \boldsymbol{\mu}_{-j}}{\|\boldsymbol{\mu}\|^2}$$

We first show the monotonicity of signal inner product in the second stage.

**Lemma C.10.** *Under the same conditions as in Theorem C.1, we have for all $j = \pm 1, r \in [m]$, $T_1 \leq k \leq T$, $|\langle \mathbf{w}_{j,r}^k, \boldsymbol{\mu}_j \rangle| \geq |\langle \mathbf{w}_{j,r}^{T_1}, \boldsymbol{\mu}_j \rangle| \geq 2$.*

*Proof of Lemma C.10.* From the update of signal inner product, we have for all $j = \pm 1, r \in [m]$, $T_1 \leq k \leq T$

$$\langle \mathbf{w}_{j,r}^{k+1}, \boldsymbol{\mu}_j \rangle = \langle \mathbf{w}_{j,r}^k, \boldsymbol{\mu}_j \rangle - \frac{\eta}{nm} \sum_{i \in \mathcal{S}_j} \ell_i'^k \langle \mathbf{w}_{j,r}^k, \boldsymbol{\mu}_j \rangle \|\boldsymbol{\mu}\|^2$$

$$= \left(1 - \frac{\eta\|\boldsymbol{\mu}\|^2}{nm} \sum_{i \in \mathcal{S}_j} \ell_i'^k\right) \langle \mathbf{w}_{j,r}^k, \boldsymbol{\mu}_j \rangle.$$

Thus $|\langle \mathbf{w}_{j,r}^k, \boldsymbol{\mu}_j \rangle| \geq |\langle \mathbf{w}_{j,r}^{k-1}, \boldsymbol{\mu}_j \rangle| \geq |\langle \mathbf{w}_{j,r}^{T_1}, \boldsymbol{\mu}_j \rangle| \geq 2$ for all $j = \pm 1, r \in [m], T_1 \leq k \leq T$. $\quad\square$

We then bound the distance between $\mathbf{W}^{T_1}$ to $\mathbf{W}^*$.

**Lemma C.11.** *Under Condition C.1, we can bound $\|\mathbf{W}^{T_1} - \mathbf{W}^*\| = O(\sqrt{m}\log(1/\epsilon)\|\boldsymbol{\mu}\|^{-1}).$*

*Proof of Lemma C.11.* Let $\mathbf{P}_{\boldsymbol{\xi}}$ be the projection matrix to the direction of $\boldsymbol{\xi}$, i.e., $\mathbf{P}_{\boldsymbol{\xi}} = \frac{\boldsymbol{\xi}\boldsymbol{\xi}^\top}{\|\boldsymbol{\xi}\|^2}$. Then we can represent

$$\mathbf{w}_{j,r}^k - \mathbf{w}_{j,r}^0 = \mathbf{P}_{\boldsymbol{\mu}_1}(\mathbf{w}_{j,r}^k - \mathbf{w}_{j,r}^0) + \mathbf{P}_{\boldsymbol{\mu}_{-1}}(\mathbf{w}_{j,r}^k - \mathbf{w}_{j,r}^0) + \sum_{i=1}^n \mathbf{P}_{\boldsymbol{\xi}_i}(\mathbf{w}_{j,r}^k - \mathbf{w}_{j,r}^0)$$

$$+ \left(\mathbf{I} - \mathbf{P}_{\boldsymbol{\mu}_1} - \mathbf{P}_{\boldsymbol{\mu}_{-1}} - \sum_{i=1}^n \mathbf{P}_{\boldsymbol{\xi}_i}\right)(\mathbf{w}_{j,r}^k - \mathbf{w}_{j,r}^0).$$

By the scale difference at $T_1$ and the fact that gradient descent only updates in the direction of $\boldsymbol{\mu}_j$, $j = \pm 1$ and $\boldsymbol{\xi}_i$, we can bound

$\|\mathbf{W}^{T_1} - \mathbf{W}^0\|^2$

$$\leq \sum_{j=\pm 1, r \in [m]} \left(\frac{\langle \mathbf{w}_{j,r}^{T_1} - \mathbf{w}_{j,r}^0, \boldsymbol{\mu}_1 \rangle^2}{\|\boldsymbol{\mu}\|^2} + \frac{\langle \mathbf{w}_{j,r}^{T_1} - \mathbf{w}_{j,r}^0, \boldsymbol{\mu}_{-1} \rangle^2}{\|\boldsymbol{\mu}\|^2} + \sum_{i=1}^n \frac{\langle \mathbf{w}_{j,r}^{T_1} - \mathbf{w}_{j,r}^0, \boldsymbol{\xi}_i \rangle^2}{\|\boldsymbol{\xi}_i\|^2}\right)$$

$$+ \sum_{j=\pm 1, r \in [m]} \left\|\left(\mathbf{I} - \mathbf{P}_{\boldsymbol{\mu}_1} - \mathbf{P}_{\boldsymbol{\mu}_{-1}} - \sum_{i=1}^n \mathbf{P}_{\boldsymbol{\xi}_i}\right)(\mathbf{w}_{j,r}^{T_1} - \mathbf{w}_{j,r}^0)\right\|^2$$

$$\leq 2m\left(\frac{2\max_r \langle \mathbf{w}_{j,r}^{T_1}, \boldsymbol{\mu}_j \rangle^2}{\|\boldsymbol{\mu}\|^2} + \frac{2\langle \mathbf{w}_{j,r}^{T_1}, \boldsymbol{\mu}_{-j} \rangle^2 + 2\langle \mathbf{w}_{j,r}^0, \boldsymbol{\mu}_{-j} \rangle^2 + 2\langle \mathbf{w}_{j,r}^0, \boldsymbol{\mu}_j \rangle^2}{\|\boldsymbol{\mu}\|^2}\right)$$

$$+ \sum_{i=1}^{n} \frac{2\langle \mathbf{w}_{j,r}^{T_1}, \boldsymbol{\xi}_i \rangle^2 + 2\langle \mathbf{w}_{j,r}^0, \boldsymbol{\xi}_i \rangle^2}{\|\boldsymbol{\xi}_i\|^2} \Bigg) + \sum_{j=\pm 1, r \in [m]} \left\| \Big( \mathbf{I} - \mathbf{P}_{\boldsymbol{\mu}_1} - \mathbf{P}_{\boldsymbol{\mu}_{-1}} - \sum_{i=1}^{n} \mathbf{P}_{\boldsymbol{\xi}_i} \Big) (\mathbf{w}_{j,r}^{T_1} - \mathbf{w}_{j,r}^0) \right\|^2$$

$$\leq O(m\|\boldsymbol{\mu}\|^{-2})$$

where we have use the scale difference at $T_1$. Therefore,

$$\|\mathbf{W}^{T_1} - \mathbf{W}^*\| \leq \|\mathbf{W}^{T_1} - \mathbf{W}^0\| + \|\mathbf{W}^0 - \mathbf{W}^*\|$$
$$\leq O(\sqrt{m}\|\boldsymbol{\mu}\|^{-1}) + O(\sqrt{m}\log(1/\epsilon)\|\boldsymbol{\mu}\|^{-1})$$
$$\leq O(\sqrt{m}\log(1/\epsilon)\|\boldsymbol{\mu}\|^{-1})$$

where we use the definition of $\mathbf{W}^*$. $\qquad\square$

**Lemma C.12.** *Under Condition C.1, we have for all $T_1 \leq k \leq T^*$*

$$\|\mathbf{W}^k - \mathbf{W}^*\|^2 - \|\mathbf{W}^{k+1} - \mathbf{W}^*\|^2 \geq 2\eta L_S(\mathbf{W}^t) - \eta\epsilon$$

*Proof of Lemma C.12.* The proof is similar as in Cao et al. (2022). We first show a lower bound on $y_i \langle \nabla f(\mathbf{W}^t, \mathbf{x}_i), \mathbf{W}^* \rangle$ for any $i \in [n]$ for all $T_1 \leq k \leq T^*$.

$$y_i \langle \nabla f(\mathbf{W}^k, \mathbf{x}_i), \mathbf{W}^* \rangle = \frac{1}{m} \sum_{j,r} j y_i \langle \mathbf{w}_{j,r}^k, \boldsymbol{\mu}_{y_i} \rangle \langle \boldsymbol{\mu}_{y_i}, \mathbf{w}_{j,r}^* \rangle + \frac{1}{m} \sum_{j,r} j y_i \langle \mathbf{w}_{j,r}^k, \boldsymbol{\xi}_i \rangle \langle \boldsymbol{\xi}_i, \mathbf{w}_{j,r}^* \rangle$$

$$= \frac{1}{m} \sum_{r=1}^{m} \langle \mathbf{w}_{y_i,r}^k, \boldsymbol{\mu}_{y_i} \rangle \langle \mathbf{w}_{y_i,r}^*, \boldsymbol{\mu}_{y_i} \rangle - \frac{1}{m} \sum_{r=1}^{m} \langle \mathbf{w}_{-y_i,r}^k, \boldsymbol{\mu}_{y_i} \rangle \langle \mathbf{w}_{-y_i,r}^*, \boldsymbol{\mu}_{y_i} \rangle$$

$$+ \frac{1}{m} \sum_{j,r} j y_i \langle \mathbf{w}_{j,r}^k, \boldsymbol{\xi}_i \rangle \langle \boldsymbol{\xi}_i, \mathbf{w}_{j,r}^0 \rangle$$

$$= \underbrace{\frac{1}{m} \sum_{r=1}^{m} |\langle \mathbf{w}_{y_i,r}^k, \boldsymbol{\mu}_{y_i} \rangle| 2\log(4/\epsilon)}_{A_5} + \underbrace{\frac{1}{m} \sum_{r=1}^{m} \langle \mathbf{w}_{y_i,r}^k, \boldsymbol{\mu}_{y_i} \rangle \langle \mathbf{w}_{y_i,r}^0, \boldsymbol{\mu}_{y_i} \rangle}_{A_6}$$

$$\underbrace{- \frac{1}{m} \sum_{r=1}^{m} \langle \mathbf{w}_{-y_i,r}^k, \boldsymbol{\mu}_{y_i} \rangle \langle \mathbf{w}_{-y_i,r}^*, \boldsymbol{\mu}_{y_i} \rangle}_{A_7} + \underbrace{\frac{1}{m} \sum_{j,r} j y_i \langle \mathbf{w}_{j,r}^k, \boldsymbol{\xi}_i \rangle \langle \boldsymbol{\xi}_i, \mathbf{w}_{j,r}^0 \rangle}_{A_8}$$

where the second equality is by definition of $\mathbf{W}^*$. The third equality is by Lemma C.6. We next bound

$$|A_6| \leq \sigma_0 \|\boldsymbol{\mu}\| \sqrt{2\log(8m/\delta)}\alpha = \widetilde{O}(\sigma_0 \|\boldsymbol{\mu}\|)$$

$$|A_7| \leq \frac{1}{m} \sum_{r=1}^{m} |\mathbf{w}_{-y_i,r}^k, \boldsymbol{\mu}_{y_i}| \big( |\langle \mathbf{w}_{-y_i,r}^0, \boldsymbol{\mu}_{y_i} \rangle| + 2\log(2/\epsilon) \big) = \widetilde{O}(\sigma_0 \|\boldsymbol{\mu}\|)$$

$$|A_8| \leq \widetilde{O}(\sigma_0 \sigma_\xi \sqrt{d})$$

where we use the global bound on the inner product by $\widetilde{O}(1)$. Next, by Theorem C.1 and Lemma C.10, we can show $\frac{1}{m} \sum_{r=1}^{m} |\langle \mathbf{w}_{y_i,r}^k, \boldsymbol{\mu}_{y_i} \rangle| \geq 2$ for all $i \in [n]$ and we can lower bound $A_5 \geq 4\log(4/\epsilon)$ and thus

$$y_i \langle \nabla f(\mathbf{W}^k, \mathbf{x}_i), \mathbf{W}^* \rangle \geq 4\log(4/\epsilon) - 2\log(4/\epsilon) = 2\log(4/\epsilon) \qquad (12)$$

where we bound $|A_6| + |A_7| + |A_8| \leq 2\log(4/\epsilon)$ under Condition C.1.

Further, we derive

$$\|\mathbf{W}^k - \mathbf{W}^*\|^2 - \|\mathbf{W}^{k+1} - \mathbf{W}^*\|^2$$
$$= 2\eta \langle \nabla L_S(\mathbf{W}^k), \mathbf{W}^k - \mathbf{W}^* \rangle - \eta^2 \|\nabla L_S(\mathbf{W}^k)\|^2$$
$$= \frac{2\eta}{n} \sum_{i=1}^{n} \ell_i'^k y_i \big( 2f(\mathbf{W}^k, \mathbf{x}_i) - \langle \nabla f(\mathbf{W}^k, \mathbf{x}_i), \mathbf{W}^* \rangle \big) - \eta^2 \|\nabla L_S(\mathbf{W}^k)\|^2$$

$$\geq \frac{2\eta}{n} \sum_{i=1}^{n} \ell_i'^k \left(2y_i f(\mathbf{W}^k, \mathbf{x}_i) - 2\log(2/\epsilon)\right) - \eta^2 \|\nabla L_S(\mathbf{W}^k)\|^2$$

$$\geq \frac{4\eta}{n} \sum_{i=1}^{n} \left(\ell(y_i f(\mathbf{W}^k, \mathbf{x}_i)) - \epsilon/4\right) - \eta^2 \|\nabla L_S(\mathbf{W}^k)\|^2$$

$$\geq 2\eta L_S(\mathbf{W}^k) - \eta\epsilon$$

where the first inequality is by (12) and the second inequality is by convexity of cross-entropy function and the last inequality is by Lemma C.8. □

Before proving the second stage convergence, we require the following lemma in order to bound the ratio of loss derivatives among different samples.

**Lemma C.13** (Kou et al. (2023)). *Let $g(z) = \ell'(z) = -(1 + \exp(z))^{-1}$. Then for any $z_2 - c \geq z_1 \geq -1$ where $c \geq 0$, we have $g(z_1)/g(z_2) \leq \exp(c)$.*

**Theorem C.2.** *Under the same settings as in Theorem C.1, let $T = T_1 + \lfloor \frac{\|\mathbf{W}^{T_1} - \mathbf{W}^*\|^2}{\eta\epsilon} \rfloor = T_1 + O(\eta^{-1}\epsilon^{-1}m\|\boldsymbol{\mu}\|^{-2})$. Then we have*

- *there exists $T_1 \leq k \leq T$ such that $L_S(\mathbf{W}^k) \leq 0.1$.*

- $\max_{j,r,i} |\langle \mathbf{w}_{j,r}^k, \boldsymbol{\xi}_i \rangle| = o(1)$ *for all $T_1 \leq k \leq T$.*

- $\max_r |\langle \mathbf{w}_{j,r}^k, \boldsymbol{\mu}_j \rangle| \geq 2$ *for all $j = \pm 1, T_1 \leq k \leq T$.*

*Proof of Theorem C.2.* By Lemma C.12, for any $T_1 \leq k \leq T$, we have

$$\|\mathbf{W}^k - \mathbf{W}^*\|^2 - \|\mathbf{W}^{k+1} - \mathbf{W}^*\|^2 \geq 2\eta L_S(\mathbf{W}^k) - \eta\epsilon$$

for all $s \leq k$. Then summing over the inequality gives

$$\frac{1}{T - T_1 + 1} \sum_{k=T_1}^{T} L_S(\mathbf{W}^k) \leq \frac{\|\mathbf{W}^{T_1} - \mathbf{W}^*\|^2}{2\eta(T - T_1 + 1)} + \frac{\epsilon}{2} \leq \epsilon$$

where the last inequality is by the choice $T = T_1 + \lfloor \frac{\|\mathbf{W}^{T_1} - \mathbf{W}^*\|^2}{\eta\epsilon} \rfloor = T_1 + \Omega(\eta^{-1}\epsilon^{-1}m\log(1/\epsilon)\|\boldsymbol{\mu}\|^{-2})$. Then we can claim that there exists a $k \in [T_1, T]$ such that $L_S(\mathbf{W}^k) \leq \epsilon$. Setting $\epsilon = 0.1$ shows the desired convergence.

Next, we show the upper bound on $\max_{j,r,i} |\langle \mathbf{w}_{j,r}^k, \boldsymbol{\xi}_i \rangle|$ for all $k \in [T_1, T]$. Notice that by Proposition C.1, we already have $\max_{j,r} |\langle \mathbf{w}_{-y_i,r}^k, \boldsymbol{\xi}_i \rangle| \leq \vartheta$, where we let $\vartheta := 3\max\{\max_{r,i} |\langle \mathbf{w}_{y_i,r}^{T_1}, \boldsymbol{\xi}_i \rangle|, \beta, 4\sqrt{\frac{\log(4n^2/\delta)}{d}} n\alpha\}$. Then we only focus on bounding $\max_{y_i,i} |\langle \mathbf{w}_{j,r}^k, \boldsymbol{\xi}_i \rangle|$. From the scale difference at $T_1$, we know that $\vartheta = \widetilde{O}(\max\{n^{-1/2}, \sigma_0 \sigma_\xi \sqrt{d}, \sigma_0 \|\boldsymbol{\mu}\|, nd^{-1/2}\}) = o(1)$. Next, we can bound

$$\sum_{k=T_1}^{T} L_S(\mathbf{W}^k) \leq \frac{\|\mathbf{W}^{T_1} - \mathbf{W}^*\|^2}{\eta} = O(\eta^{-1}m\log(1/\epsilon)\|\boldsymbol{\mu}\|^{-2}) \tag{13}$$

where we use Lemma C.11 for the last equality.

Then, we first prove $\max_{r,i} |\rho_{y_i,r,i}^k| \leq 2\vartheta$ for all $T_1 \leq k \leq T$. First it is easy to see that at $T_1$, we have

$$\max_{r,i} |\rho_{y_i,r,i}^{T_1}| \leq \max_{r,i} |\langle \mathbf{w}_{y_i,r}^{T_1}, \boldsymbol{\xi}_i \rangle| + \max_{r,i} |\langle \mathbf{w}_{y_i,r}^0, \boldsymbol{\xi}_i \rangle| + 4\sqrt{\frac{\log(4n^2/\delta)}{d}} n\alpha \leq \vartheta \leq 2\vartheta$$

Then suppose there $\widetilde{T} \in [T_1, T]$ such that $\max_{r,i} |\rho_{y_i,r,i}^{T_1}| \leq 2\vartheta$ for all $k \in [T_1, \widetilde{T} - 1]$. Now we let $\phi^k := \max_{r,i} |\rho_{y_i,r,i}^k|$ and thus by the update of noise coefficient

$$\phi^{k+1} \leq \phi^k + \frac{\eta}{nm} |\ell_i'^k| \left(\phi^k + \beta/3 + 4\sqrt{\frac{\log(4n^2/\delta)}{d}} n\alpha\right) \|\boldsymbol{\xi}_i\|^2$$

$$\leq \phi^k + \frac{\eta}{nm} \max_i |\ell_i'^k| \Big( \phi^k + \beta/3 + 4\sqrt{\frac{\log(4n^2/\delta)}{d}} n\alpha \Big) O(\sigma_\xi^2 d).$$

where we use Lemma C.5 in the first inequality. Then taking the summation from $T_1$ to $\widetilde{T}$ gives

$$\phi^{\widetilde{T}} \leq \phi^{T_1} + \frac{\eta}{nm} \sum_{k=T_1}^{\widetilde{T}-1} \max_i |\ell_i'^k| O(\sigma_\xi^2 d)\vartheta \tag{14}$$

where the first inequality is by the induction condition. Next, the aim is bound $\sum_{k=T_1}^{\widetilde{T}-1} \max_i |\ell_i'^k|$. First, for any $i, i' \in [n]$ such that $y_i = y_{i'}$, we can bound for all $T_1 \leq k \leq \widetilde{T} - 1$

$$y_i f(\mathbf{W}^k, \mathbf{x}_i) - y_{i'} f(\mathbf{W}^k, \mathbf{x}_{i'})$$
$$= F_{y_i}(\mathbf{W}_{y_i}^k, \mathbf{x}_i) - F_{-y_i}(\mathbf{W}_{-y_i}^k, \mathbf{x}_i)) - F_{y_{i'}}(\mathbf{W}_{y_{i'}}^k, \mathbf{x}_{i'}) + F_{-y_{i'}}(\mathbf{W}_{-y_{i'}}^k, \mathbf{x}_{i'}))$$
$$\leq \frac{1}{m} \sum_{r=1}^m \big( \langle \mathbf{w}_{y_i,r}^k, \boldsymbol{\mu}_{y_i} \rangle^2 + \langle \mathbf{w}_{y_i,r}^k, \boldsymbol{\xi}_i \rangle^2 \big) - \frac{1}{m} \sum_{r=1}^m \big( \langle \mathbf{w}_{y_i,r}^k, \boldsymbol{\mu}_{y_i} \rangle^2 + \langle \mathbf{w}_{y_i,r}^k, \boldsymbol{\xi}_{i'} \rangle^2 \big) + 1/C_1$$
$$= \frac{1}{m} \sum_{r=1}^m \big( \langle \mathbf{w}_{y_i,r}^k, \boldsymbol{\xi}_i \rangle^2 - \langle \mathbf{w}_{y_i,r}^k, \boldsymbol{\xi}_{i'} \rangle^2 \big) + 1/C_1$$
$$\leq \max_{r,i} \langle \mathbf{w}_{y_i,r}^k, \boldsymbol{\xi}_i \rangle^2 + 1/C_1$$
$$\leq \max_{r,i} \big( |\rho_{y_i,r,i}^k| + \max_{r,i} |\langle \mathbf{w}_{y_i,r}^0, \boldsymbol{\xi}_i \rangle| + 4\sqrt{\frac{\log(4n^2/\delta)}{d}} n\alpha \big)^2$$
$$\leq 6\vartheta^2 \leq \vartheta$$

where in the first inequality we notice that $F_{-y_i}(\mathbf{W}_{-y_i}^k, \mathbf{x}_i)) \geq 0$, $y_i = y_{i'}$ and we recall that $F_{-y_i}(\mathbf{W}_j^k, \mathbf{x}_i) \leq \beta^2 + \big(\beta + 12\sqrt{\frac{\log(4n^2/\delta)}{d}} n\alpha\big)^2 = 1/C_1$ for some sufficiently large constant $C_1 > 0$. The second last inequality is by induction condition and the last inequality is by choosing $\vartheta \leq 1/6$. Then we can bound the ratio of loss derivatives (based on Lemma C.13) that

$$|\ell_{i'}'^k|/|\ell_i'^k| \leq \exp\big( y_i f(\mathbf{W}^k, \mathbf{x}_i) - y_{i'} f(\mathbf{W}^k, \mathbf{x}_{i'}) \big) \leq \exp(\vartheta)$$

This suggests $1 - O(\vartheta) \leq |\ell_{i'}'^k|/|\ell_i'^k| \leq 1 + O(\vartheta)$ for all $i, i' \in [n]$, $T_1 \leq k \leq \widetilde{T} - 1$. Then let $i^* = \arg\max_i |\ell_i'^k|$, we have

$$\sum_{T_1}^T \max_i |\ell_i'^k| = \sum_{T_1}^T \Theta\big(\frac{1}{|\mathcal{S}_{y_{i^*}}|} \sum_{i \in \mathcal{S}_{y_{i^*}}} |\ell_i'^k|\big) \leq \sum_{T_1}^T \Theta\big(\frac{1}{|\mathcal{S}_{y_{i^*}}|} \sum_{i \in \mathcal{S}_{y_{i^*}}} \ell_i^k\big) \leq \sum_{T_1}^T \Theta\big(\frac{n}{|\mathcal{S}_{y_{i^*}}|} L_S(\mathbf{W}^k)\big)$$
$$= \widetilde{O}(\eta^{-1} m \log(1/\epsilon) \|\boldsymbol{\mu}\|^{-2}) \tag{15}$$

where the first inequality is by $|\ell'| \leq \ell$ and the last equality is from (13) and $|\mathcal{S}_{y_{i^*}}| \geq 0.49n$ (based on Lemma B.1).

This allows to bound (14) as

$$\phi^{\widetilde{T}} \leq \phi^{T_1} + \frac{\eta}{nm} \sum_{s=T_1}^{\widetilde{T}-1} \max_i |\ell_i'^k| O(\sigma_\xi^2 d)\vartheta$$
$$\leq \phi^{T_1} + O(n^{-1}\sigma_\xi^2 d \log(1/\epsilon) \|\boldsymbol{\mu}\|^{-2}) \cdot \vartheta$$
$$\leq \vartheta + O(n^{-1} \mathrm{SNR}^{-2} \log(1/\epsilon)) \cdot \vartheta$$
$$\leq 2\vartheta$$

and the second inequality is by (15) and the last inequality is by setting $\epsilon = 0.1$ and $n \cdot \mathrm{SNR}^2 \geq C'$ for sufficiently large constant $C'$. Thus, we have $\max_{r,i} |\langle \mathbf{w}_{y_i,r}^{\widetilde{T}}, \boldsymbol{\xi}_i \rangle| \leq \max_{r,i} |\rho_{y_i,r,i}^{\widetilde{T}}| + \beta + 4\sqrt{\frac{\log(4n^2/\delta)}{d}} n\alpha \leq 3\vartheta = o(1)$. The lower bound on signal inner product is directly from Lemma C.10. $\qquad\square$

### C.4 Noise memorization

We also analyze the setting where $n^{-1}\text{SNR}^{-2} \geq C'$ for some constant $C' > 0$, which allows the noise memorization to dominate signal learning, thus reaching harmful overfitting.

We first require the following anti-concentration result for the noise inner product, which is required to ensure the sign invariance of the inner product along training.

**Lemma C.14.** *Suppose $\delta > 0$ and $\sigma_0 \geq \Omega(\log(n^2/\delta)n^2 m\alpha d^{-1}\sigma_\xi^{-1})$, we have for all $j = \pm 1, r \in [m], i \in [n]$, $|\langle \mathbf{w}_{j,r}^0, \boldsymbol{\xi}_i \rangle| \geq 8\sqrt{\frac{\log(4n^2/\delta)}{d}}n\alpha$.*

*Proof of Lemma C.14.* For any $j = \pm 1, r \in [m], i \in [n]$, we have $\langle \mathbf{w}_{j,r}^0, \boldsymbol{\xi}_i \rangle \sim \mathcal{N}(0, \sigma_0^2 \|\boldsymbol{\xi}_i\|^2)$. Then applying Lemma C.3 by setting RHS to $\delta/(2mn)$ and $c = 8\sqrt{\frac{\log(4n^2/\delta)}{d}}n\alpha$, we require

$$d^2 \geq 42\log(4n^2/\delta)n^2\alpha^2\sigma_0^{-2}\sigma_\xi^{-2}/\log(\frac{4m^2n^2}{4m^2n^2 - \delta^2})$$

where we use Lemma B.2 that $\|\boldsymbol{\xi}_i\|^2 \geq 0.99\sigma_\xi^2 d$. Finally noticing that $1/\log(4m^2n^2/(4m^2n^2 - \delta^2)) \leq \Theta(m^2n^2)$ and taking the union bound completes the proof. $\square$

An immediate consequence of Lemma C.14 is the following result that allows to derive the sign invariance for $\langle \mathbf{w}_{y_i,r,i}^k, \boldsymbol{\xi}_i \rangle$ for all iterations.

**Lemma C.15.** *Under Condition C.1, for any $i \in [n], r \in [m]$, we have $\text{sign}(\langle \mathbf{w}_{y_i,r}^k, \boldsymbol{\xi}_i \rangle) = \text{sign}(\rho_{y_i,r,i}^k) = \text{sign}(\langle \mathbf{w}_{y_i,r}^0, \boldsymbol{\xi}_i \rangle)$ for all $0 \leq k \leq T^*$.*

*Proof of Lemma C.15.* First by Lemma C.14 and Lemma C.5, we can bound if $\langle \mathbf{w}_{y_i,r}^0, \boldsymbol{\xi}_i \rangle \geq 0$,

$$\rho_{y_i,r,i}^k + \frac{1}{2}\langle \mathbf{w}_{y_i,r}^0, \boldsymbol{\xi}_i \rangle \leq \langle \mathbf{w}_{y_i,r}^k, \boldsymbol{\xi}_i \rangle \leq \rho_{y_i,r,i}^k + \frac{3}{2}\langle \mathbf{w}_{y_i,r}^0, \boldsymbol{\xi}_i \rangle$$

and if $\langle \mathbf{w}_{y_i,r}^0, \boldsymbol{\xi}_i \rangle \leq 0$,

$$\rho_{y_i,r,i}^k + \frac{3}{2}\langle \mathbf{w}_{y_i,r}^0, \boldsymbol{\xi}_i \rangle \leq \langle \mathbf{w}_{y_i,r}^k, \boldsymbol{\xi}_i \rangle \leq \rho_{y_i,r,i}^k + \frac{1}{2}\langle \mathbf{w}_{y_i,r}^0, \boldsymbol{\xi}_i \rangle$$

Next we use induction to show the sign invariance. First it is clear when $k = 0$, the sign invariance is trivially satisfied. At $k = 1$, we have by the iterative update of the coefficients,

$$\rho_{y_i,r,i}^1 = \rho_{y_i,r,i}^0 + \frac{\eta}{nm}|\ell_i'^0|\langle \mathbf{w}_{y_i,r}^0, \boldsymbol{\xi}_i \rangle\|\boldsymbol{\xi}_i\|^2 = \frac{\eta}{nm}|\ell_i'^0|\langle \mathbf{w}_{y_i,r}^0, \boldsymbol{\xi}_i \rangle\|\boldsymbol{\xi}_i\|^2$$

and thus $\text{sign}(\rho_{y_i,r,i}^1) = \text{sign}(\langle \mathbf{w}_{y_i,r}^0, \boldsymbol{\xi}_i \rangle)$. Further, by Lemma C.5, and without loss of generality that $\langle \mathbf{w}_{y_i,r}^0, \boldsymbol{\xi}_i \rangle \geq 0$, we have

$$\langle \mathbf{w}_{y_i,r}^1, \boldsymbol{\xi}_i \rangle \geq \rho_{y_i,r,i}^1 + \langle \mathbf{w}_{y_i,r}^0, \boldsymbol{\xi}_i \rangle - 4\sqrt{\frac{\log(4n^2/\delta)}{d}}n\alpha \geq \rho_{y_i,r,i}^1 + \frac{1}{2}\langle \mathbf{w}_{y_i,r}^0, \boldsymbol{\xi}_i \rangle \geq 0.$$

Similar argument also holds for $\langle \mathbf{w}_{y_i,r}^0, \boldsymbol{\xi}_i \rangle < 0$. Then we show at $k = 1$, $\text{sign}(\rho_{y_i,r,i}^1) = \text{sign}(\langle \mathbf{w}_{y_i,r}^1, \boldsymbol{\xi}_i \rangle) = \text{sign}(\langle \mathbf{w}_{y_i,r}^0, \boldsymbol{\xi}_i \rangle)$. Suppose there exists a time $\widetilde{T}$ such that for all $k \leq \widetilde{T} - 1$, the sign invariance holds. Then for $k = \widetilde{T}$, suppose $\text{sign}(\langle \mathbf{w}_{y_i,r}^{\widetilde{T}-1}, \boldsymbol{\xi}_i \rangle) = \text{sign}(\rho_{y_i,r,i}^{\widetilde{T}-1}) = \text{sign}(\langle \mathbf{w}_{y_i,r}^0, \boldsymbol{\xi}_i \rangle) = +1$,

$$\rho_{y_i,r,i}^{\widetilde{T}} = \rho_{y_i,r,i}^{\widetilde{T}-1} + \frac{\eta}{nm}|\ell_i'^{\widetilde{T}-1}|\langle \mathbf{w}_{y_i,r}^{\widetilde{T}-1}, \boldsymbol{\xi}_i \rangle\|\boldsymbol{\xi}_i\|^2$$

$$\geq \rho_{y_i,r,i}^{\widetilde{T}-1} + \frac{\eta}{nm}|\ell_i'^{\widetilde{T}-1}|(\rho_{y_i,r,i}^{\widetilde{T}-1} + \langle \mathbf{w}_{y_i,r}^0, \boldsymbol{\xi}_i \rangle - 4\sqrt{\frac{\log(4n^2/\delta)}{d}}n\alpha)\|\boldsymbol{\xi}_i\|^2$$

$$\geq \rho_{y_i,r,i}^{\widetilde{T}-1} + \frac{\eta}{nm}|\ell_i'^{\widetilde{T}-1}|(\rho_{y_i,r,i}^{\widetilde{T}-1} + \frac{1}{2}\langle \mathbf{w}_{y_i,r}^0, \boldsymbol{\xi}_i \rangle)\|\boldsymbol{\xi}_i\|^2$$

$$\geq 0$$

Further,

$$\langle \mathbf{w}_{y_i,r}^{\widetilde{T}}, \boldsymbol{\xi}_i \rangle \geq \rho_{y_i,r,i}^{\widetilde{T}} + \langle \mathbf{w}_{y_i,r}^0, \boldsymbol{\xi}_i \rangle - 4\sqrt{\frac{\log(4n^2/\delta)}{d}} n\alpha \geq \rho_{y_i,r,i}^{\widetilde{T}} + \frac{1}{2}\langle \mathbf{w}_{y_i,r}^0, \boldsymbol{\xi}_i \rangle \geq 0.$$

and thus completes the induction that $\mathrm{sign}(\langle \mathbf{w}_{y_i,r}^{\widetilde{T}}, \boldsymbol{\xi}_i \rangle) = \mathrm{sign}(\rho_{y_i,r,i}^{\widetilde{T}}) = \mathrm{sign}(\langle \mathbf{w}_{y_i,r}^0, \boldsymbol{\xi}_i \rangle)$. Similar argument holds when $\mathrm{sign}(\langle \mathbf{w}_{y_i,r}^0, \boldsymbol{\xi}_i \rangle) = -1$. □

We also derive the following concentration result for the average noise inner product at initialization.

**Lemma C.16.** *Suppose $\delta > 0$ and $m = \Omega(\log(n/\delta))$. Then with probability at least $1 - \delta$, we have for all $j = \pm 1, i \in [n]$*

$$\sigma_0 \sigma_\xi \sqrt{d}/2 \leq \frac{1}{m}\sum_{r=1}^{n} |\langle \mathbf{w}_{j,r}^0, \boldsymbol{\xi}_i \rangle| \leq \sigma_0 \sigma_\xi \sqrt{d}$$

*Proof of Lemma C.16.* First notice that for any $i \in [n]$, $\langle \mathbf{w}_{j,r}^0, \boldsymbol{\xi}_i \rangle \sim \mathcal{N}(0, \sigma_0^2\|\boldsymbol{\xi}_i\|^2)$ and thus we have $\mathbb{E}[|\langle \mathbf{w}_{j,r}^0, \boldsymbol{\xi}_i \rangle|] = \sigma_0\|\boldsymbol{\xi}_i\|\sqrt{2/\pi}$. By sub-Gaussian tail bound, with probability at least $1 - \delta/(2n)$, for any $i \in [n]$

$$\left| \frac{1}{m}\sum_{r=1}^{m} |\langle \mathbf{w}_{j,r}^0, \boldsymbol{\xi}_i \rangle| - \sigma_0\|\boldsymbol{\xi}_i\|\sqrt{2/\pi} \right| \leq \sqrt{\frac{2\sigma_0^2\|\boldsymbol{\xi}_i\|^2 \log(4n/\delta)}{m}}$$

Choosing $m = \Omega(\log(n/\delta))$, we have

$$\sigma_0\|\boldsymbol{\xi}_i\|\sqrt{2/\pi}0.99 \leq \frac{1}{m}\sum_{r=1}^{n} |\langle \mathbf{w}_{j,r}^0, \boldsymbol{\xi}_i \rangle| \leq \sigma_0\|\boldsymbol{\xi}_i\|\sqrt{2/\pi}1.01.$$

Because from Lemma B.2, we have $0.99\sigma_\xi\sqrt{d} \leq \|\boldsymbol{\xi}_i\| \leq 1.01\sigma_\xi\sqrt{d}$ by choosing $d = \widetilde{\Omega}(1)$ sufficiently large. Then we have $\sigma_0\sigma_\xi\sqrt{d}/2 \leq \frac{1}{m}\sum_{r=1}^{n} |\langle \mathbf{w}_{j,r}^0, \boldsymbol{\xi}_i \rangle| \leq \sigma_0\sigma_\xi\sqrt{d}$. Finally taking the union bound for all $j = \pm 1, i \in [n]$ completes the proof. □

We have established several preliminary lemmas that hold with high probability, including Lemma B.1, Lemma B.2, Lemma C.1, Lemma C.14, Lemma C.16. We let $\mathcal{E}_{\mathrm{prelim}}$ be the event such that all the results in these lemmas hold for a given $\delta$. Then by applying union bound, we have $\mathbb{P}(\mathcal{E}_{\mathrm{prelim}}) \geq 1 - 5\delta$. The subsequent analysis are conditioned on the event $\mathcal{E}_{\mathrm{prelim}}$.

### C.4.1 FIRST STAGE

**Theorem C.3.** *Under Condition C.1, suppose $n^{-1} \cdot \mathrm{SNR}^{-2} \geq C'$ for some constant $C' > 0$. Then there exists a time $T_1 = \widetilde{\Theta}(\eta^{-1}nm\sigma_\xi^{-2}d^{-1})$, such that (1) $\max_r |\langle \mathbf{w}_{y_i,r}^{T_1}, \boldsymbol{\xi}_i \rangle| \geq 2$ for all $i \in [n]$, (2) $\frac{1}{m}\sum_{r=1}^{m} |\langle \mathbf{w}_{y_i,r}^{T_1}, \boldsymbol{\xi}_i \rangle| \geq 4$ for all $i \in [n]$ and (3) $\max_{j,r,y} |\langle \mathbf{w}_{j,r}^{T_1}, \boldsymbol{\mu}_y \rangle| = \widetilde{O}(n^{-1/2})$.*

*Proof of Theorem C.3.* We first bound the growth of signal as follows. From the gradient descent update, we have

$$\begin{aligned}
|\langle \mathbf{w}_{j,r}^k, \boldsymbol{\mu}_j \rangle| &= |\langle \mathbf{w}_{j,r}^{k-1}, \boldsymbol{\mu}_j \rangle| + \frac{\eta|\mathcal{S}_j|}{nm}|\langle \mathbf{w}_{j,r}^{k-1}, \boldsymbol{\mu}_j \rangle|\|\boldsymbol{\mu}\|^2 \\
&\leq \left(1 + 0.51\frac{\eta\|\boldsymbol{\mu}\|^2}{m}\right)|\langle \mathbf{w}_{j,r}^{k-1}, \boldsymbol{\mu}_j \rangle| \\
&\leq \left(1 + 0.51\frac{\eta\|\boldsymbol{\mu}\|^2}{m}\right)^k|\langle \mathbf{w}_{j,r}^0, \boldsymbol{\mu}_j \rangle|
\end{aligned} \tag{16}$$

where the first inequality is by $|\ell_i'^k| \leq 1$ and the second inequality is by Lemma B.1 with $n = \widetilde{\Omega}(1)$ sufficiently large.

On the other hand, for the growth of noise, we have from the inner product update, for any $i \in [n]$

$$\langle \mathbf{w}_{y_i,r}^k, \boldsymbol{\xi}_i \rangle = \langle \mathbf{w}_{y_i,r}^{k-1}, \boldsymbol{\xi}_i \rangle - \frac{\eta}{nm} \sum_{i'=1}^n \ell_{i'}'^{k-1} \langle \mathbf{w}_{j,r}^{k-1}, \boldsymbol{\xi}_{i'} \rangle \langle \boldsymbol{\xi}_{i'}, \boldsymbol{\xi}_i \rangle$$

$$= \left(1 - \frac{\eta}{nm} \ell_i'^k \|\boldsymbol{\xi}_i\|^2\right) \langle \mathbf{w}_{y_i,r}^{k-1}, \boldsymbol{\xi}_i \rangle - \frac{\eta}{nm} \sum_{i' \neq i} \ell_{i'}'^{k-1} \langle \mathbf{w}_{y_i,r}^{k-1}, \boldsymbol{\xi}_{i'} \rangle \langle \boldsymbol{\xi}_{i'}, \boldsymbol{\xi}_i \rangle$$

Then this suggests

$$|\langle \mathbf{w}_{y_i,r}^k, \boldsymbol{\xi}_i \rangle| \geq \left(1 - \frac{\eta}{nm} \ell_i'^k \|\boldsymbol{\xi}_i\|^2\right) |\langle \mathbf{w}_{y_i,r}^{k-1}, \boldsymbol{\xi}_i \rangle| - \frac{\eta}{nm} \sum_{i' \neq i} |\ell_{i'}'^{k-1}| \cdot |\langle \mathbf{w}_{y_i,r}^{k-1}, \boldsymbol{\xi}_{i'} \rangle| \cdot |\langle \boldsymbol{\xi}_{i'}, \boldsymbol{\xi}_i \rangle| \quad (17)$$

We first prove for any $i \in [n]$, $\max_r |\langle \mathbf{w}_{y_i,r}^{k+1}, \boldsymbol{\xi}_i \rangle| \geq \max_r |\langle \mathbf{w}_{y_i,r}^k, \boldsymbol{\xi}_i \rangle| \geq \max_r |\langle \mathbf{w}_{y_i,r}^0, \boldsymbol{\xi}_i \rangle|$ for all $k \leq T_1$. We prove such a result by induction. It is clear that at $k = 0$, the result is satisfied. Now suppose there exists an iteration $\tilde{k}$ such that

$$\max_r |\langle \mathbf{w}_{y_i,r}^k, \boldsymbol{\xi}_i \rangle| \geq \max_r |\langle \mathbf{w}_{y_i,r}^0, \boldsymbol{\xi}_i \rangle| \geq \sigma_0 \sigma_\xi \sqrt{d}/4$$

for all $k \leq \tilde{k} - 1$, where the last inequality is by Lemma C.1. Then we can bound based on Lemma C.7 and Lemma B.2, we have for any $i' \neq i \in [n]$ and

$$\frac{n|\ell_{i'}'^{\tilde{k}-1}| \cdot |\langle \boldsymbol{\xi}_i, \boldsymbol{\xi}_{i'} \rangle| \cdot |\langle \mathbf{w}_{y_i,r}^{\tilde{k}-1}, \boldsymbol{\xi}_{i'} \rangle|}{|\ell_i'^{\tilde{k}-1}| \cdot \|\boldsymbol{\xi}_i\|^2 \max_r |\langle \mathbf{w}_{y_i,r}^{\tilde{k}-1}, \boldsymbol{\xi}_i \rangle|} \leq \frac{2\sigma_\xi^2 \sqrt{d \log(4n^2/\delta)}}{C_\ell 0.99 \sigma_\xi^2 d} n\alpha \sigma_0^{-1} \sigma_\xi^{-1} d^{-1/2}$$

$$= 8.4 C_\ell^{-1} n\alpha \frac{\sqrt{\log(4n^2/\delta)}}{d\sigma_0 \sigma_\xi}$$

$$\leq 0.01 \quad (18)$$

where we use the lower and upper bound on loss derivatives during the first stage, as well as Lemma B.2 and Lemma C.1. The last inequality is by $\sigma_0 \geq 840 n C_\ell^{-1} d^{-1} \sigma_\xi^{-1} \alpha \sqrt{\log(4n^2/\delta)}$. Then we have

$$\max_r |\langle \mathbf{w}_{y_i,r}^{\tilde{k}}, \boldsymbol{\xi}_i \rangle| \geq \left(1 - \frac{\eta}{nm} \ell_i'^{\tilde{k}-1} \|\boldsymbol{\xi}_i\|^2\right) \max_r |\langle \mathbf{w}_{y_i,r}^{\tilde{k}-1}, \boldsymbol{\xi}_i \rangle| - \frac{\eta}{nm} \sum_{i' \neq i} |\ell_{i'}'^{\tilde{k}-1}| \cdot |\langle \mathbf{w}_{y_i,r}^{\tilde{k}-1}, \boldsymbol{\xi}_{i'} \rangle| \cdot |\langle \boldsymbol{\xi}_{i'}, \boldsymbol{\xi}_i \rangle|$$

$$\geq \left(1 + \frac{\eta}{nm} 0.99 |\ell_i'^{\tilde{k}-1}| \|\boldsymbol{\xi}_i\|^2\right) \max_r \left|\langle \mathbf{w}_{y_i,r}^{\tilde{k}-1}, \boldsymbol{\xi}_i \rangle\right|$$

$$\geq \max_r \left|\langle \mathbf{w}_{y_i,r}^{\tilde{k}-1}, \boldsymbol{\xi}_i \rangle\right|$$

$$\geq \max_r |\langle \mathbf{w}_{y_i,r}^0, \boldsymbol{\xi}_i \rangle|$$

Let $B_i^k := \max_r |\langle \mathbf{w}_{y_i,r}^k, \boldsymbol{\xi}_i \rangle|$ and we obtain for any $k \leq T_1$,

$$B_i^k \geq \left(1 + \frac{\eta}{nm} 0.99 |\ell_i'^{\tilde{k}-1}| \|\boldsymbol{\xi}_i\|^2\right) B_i^{k-1} \geq \left(1 + \frac{\eta \sigma_\xi^2 d}{nm} 0.98 C_\ell\right) B_i^{k-1}$$

$$\geq \left(1 + \frac{\eta \sigma_\xi^2 d}{nm} 0.98 C_\ell\right)^k B_i^0$$

$$\geq \left(1 + \frac{\eta \sigma_\xi^2 d}{nm} 0.98 C_\ell\right)^k \sigma_0 \sigma_\xi \sqrt{d}/4$$

where we use (18), which holds for iteration $k$ and Lemma C.1. Consider

$$T_1 = \log(8\sigma_0^{-1}\sigma_\xi^{-1}d^{-1/2})/\log\left(1 + \frac{\eta \sigma_\xi^2 d}{nm} 0.98 C_\ell\right) = \Theta(\eta^{-1}nm\sigma_\xi^{-2}d^{-1}\log(8\sigma_0^{-1}\sigma_\xi^{-1}d^{-1/2}))$$

for $\eta$ sufficiently small. Then it can be shown that

$$B_i^{T_1} = \max_r |\langle \mathbf{w}_{y_i,r}^{T_1}, \boldsymbol{\xi}_i \rangle| \geq 2$$

In addition, we show the average also grows to a constant order with a similar argument. In particular, from (17), we have

$$\frac{1}{m}\sum_{r=1}^m |\langle \mathbf{w}_{y_i,r}^k, \boldsymbol{\xi}_i\rangle| \geq \left(1 - \frac{\eta}{nm}\ell_i'^k \|\boldsymbol{\xi}_i\|^2\right)\frac{1}{m}\sum_{r=1}^m |\langle \mathbf{w}_{y_i,r}^{k-1}, \boldsymbol{\xi}_i\rangle|$$

$$- \frac{\eta}{nm}\sum_{i'\neq i} |\ell_{i'}'^{k-1}| \cdot \frac{1}{m}\sum_{r=1}^m |\langle \mathbf{w}_{y_i,r}^{k-1}, \boldsymbol{\xi}_{i'}\rangle| \cdot |\langle \boldsymbol{\xi}_{i'}, \boldsymbol{\xi}_i\rangle|$$

Using a similar induction argument, we can show

$$\frac{1}{m}\sum_{r=1}^m |\langle \mathbf{w}_{y_i,r}^k, \boldsymbol{\xi}_i\rangle| \geq \frac{1}{m}\sum_{r=1}^m |\langle \mathbf{w}_{y_i,r}^{k-1}, \boldsymbol{\xi}_i\rangle| \geq \frac{1}{m}\sum_{r=1}^m |\langle \mathbf{w}_{y_i,r}^0, \boldsymbol{\xi}_i\rangle| \geq \sigma_0\sigma_\xi\sqrt{d}/2$$

for all $k \leq T_1$, where the last inequality follows from Lemma C.16. Then we can show at $T_1$,

$$\frac{1}{m}\sum_{r=1}^m |\langle \mathbf{w}_{y_i,r}^{T_1}, \boldsymbol{\xi}_i\rangle| \geq \left(1 + \frac{\eta\sigma_\xi^2 d}{nm}0.98 C_\ell\right)^{T_1} \sigma_0\sigma_\xi\sqrt{d}/2 \geq 4.$$

In the meantime, (16) allows to bound the growth of signal learning as for any $j = \pm 1$,

$$\max_r |\langle \mathbf{w}_{j,r}^{T_1}, \boldsymbol{\mu}_j\rangle|$$

$$\leq \left(1 + 0.51\frac{\eta\|\boldsymbol{\mu}\|^2}{m}\right)^{T_1} \sqrt{2\log(8m/\delta)}\sigma_0\|\boldsymbol{\mu}\|$$

$$= \exp\left(\frac{\log(1 + 0.51\frac{\eta\|\boldsymbol{\mu}\|^2}{m})}{\log(1 + 0.98\frac{\eta\sigma_\xi^2 dC_\ell}{nm})}\log\left(8\sigma_0^{-1}\sigma_\xi^{-1}d^{-1/2}\right)\right)\sqrt{2\log(8m/\delta)}\sigma_0\|\boldsymbol{\mu}\|$$

$$\leq \exp\left(\left(0.53C_\ell^{-1}n\mathrm{SNR}^2 + \widetilde{O}(n^{-1}\mathrm{SNR}^{-2}\eta)\right)\log\left(8\sigma_0^{-1}\sigma_\xi^{-1}d^{-1/2}\right)\right)\sqrt{2\log(8m/\delta)}\sigma_0\|\boldsymbol{\mu}\|$$

$$\leq 8\sqrt{2\log(8m/\delta)}\mathrm{SNR}$$

$$= \widetilde{O}(n^{-1/2})$$

where the first inequality is by Lemma C.1 and the second inequality is by Taylor expansion around $\eta = 0$. The third inequality is by choosing $\eta$ sufficiently small and based on the condition that $n^{-1}\mathrm{SNR}^{-2} \geq 0.55 C_\ell^{-1}$. The last equality is by the SNR condition. $\qquad\square$

### C.4.2 SECOND STAGE

We choose $\mathbf{W}^*$ to be

$$\mathbf{w}_{j,r}^* = \mathbf{w}_{j,r}^0 + 2\log(4/\epsilon)\sum_{i=1}^n \mathbb{1}(y_i = j)\mathrm{sign}(\langle \mathbf{w}_{j,r}^0, \boldsymbol{\xi}_i\rangle)\frac{\boldsymbol{\xi}_i}{\|\boldsymbol{\xi}_i\|^2}$$

First we show the invariance of sign of noise inner product after the first stage.

**Lemma C.17.** *Under the same settings as in Theorem C.3, we have* $\max_r |\langle \mathbf{w}_{y_i,r}^k, \boldsymbol{\xi}_i\rangle| \geq 1$ *and* $\frac{1}{m}\sum_{r=1}^m |\langle \mathbf{w}_{y_i,r}^k, \boldsymbol{\xi}_i\rangle| \geq 2$ *for all* $T_1 \leq k \leq T^*$ *and any* $i \in [n]$.

*Proof of Lemma C.17.* In addition to the two results, we also prove $\max_r |\rho_{y_i,r,i}^k| \geq 1.5$ and $\frac{1}{m}\sum_{r=1}^m |\rho_{y_i,r,i}^k| \geq 3$. We prove these results by induction. First, it is clear that at $k = T_1$, the bound regarding inner products are trivially satisfied by Theorem C.3. Then by Lemma C.5, we have

$$\max_r |\rho_{y_i,r,i}^{T_1}| \geq \max_r |\langle \mathbf{w}_{y_i,r}^{T_1}, \boldsymbol{\xi}_i\rangle| - \beta - 4\sqrt{\frac{\log(4n^2/\delta)}{d}}n\alpha \geq 2 - 0.5 = 1.5$$

$$\frac{1}{m}\sum_{r=1}^m |\rho_{y_i,r,i}^{T_1}| \geq \frac{1}{m}\sum_{r=1}^m |\langle \mathbf{w}_{y_i,r}^{T_1}, \boldsymbol{\xi}_i\rangle| - \beta - 4\sqrt{\frac{\log(4n^2/\delta)}{d}}n\alpha \geq 4 - 1 = 3$$

where the last inequalities are by Condition C.1 for sufficiently large constant $C$.

Now suppose there exists a time $T_1 \leq \widetilde{T} \leq T^*$ such that the results hold for all $k \leq \widetilde{T} - 1$. Then at $k = \widetilde{T}$, recall the coefficient update as

$$\rho_{y_i,r,i}^{\widetilde{T}} = \rho_{y_i,r,i}^{\widetilde{T}-1} + \frac{\eta}{nm}|\ell_i'^{\widetilde{T}-1}||\langle \mathbf{w}_{y_i,r}^{\widetilde{T}-1}, \boldsymbol{\xi}_i\rangle|\|\boldsymbol{\xi}_i\|^2 \tag{19}$$

If $\langle \mathbf{w}_{y_i,r}^0, \boldsymbol{\xi}_i\rangle > 0$, by Lemma C.15 we have $\langle \mathbf{w}_{y_i,r}^{\widetilde{T}-1}, \boldsymbol{\xi}_i\rangle, \rho_{y_i,r,i}^{\widetilde{T}-1} > 0$. Then

$$\rho_{y_i,r,i}^{\widetilde{T}} = \rho_{y_i,r,i}^{\widetilde{T}-1} + \frac{\eta}{nm}|\ell_i'^{\widetilde{T}-1}||\langle \mathbf{w}_{y_i,r}^{\widetilde{T}-1}, \boldsymbol{\xi}_i\rangle|\|\boldsymbol{\xi}_i\|^2$$

$$\geq \rho_{y_i,r,i}^{\widetilde{T}-1} + \frac{\eta}{nm}|\ell_i'^{\widetilde{T}-1}|(\rho_{y_i,r,i}^{\widetilde{T}-1} + \langle \mathbf{w}_{y_i,r}^0, \boldsymbol{\xi}_i\rangle - 4\sqrt{\frac{\log(4n^2/\delta)}{d}}n\alpha)\|\boldsymbol{\xi}_i\|^2$$

$$\geq \rho_{y_i,r,i}^{\widetilde{T}-1} + \frac{\eta}{nm}|\ell_i'^{\widetilde{T}-1}|(\rho_{y_i,r,i}^{\widetilde{T}-1} + \frac{1}{2}\langle \mathbf{w}_{y_i,r}^0, \boldsymbol{\xi}_i\rangle)\|\boldsymbol{\xi}_i\|^2.$$

Then taking maximum over $r$,

$$\max_r |\rho_{y_i,r,i}^{\widetilde{T}}| \geq \max_r |\rho_{y_i,r,i}^{\widetilde{T}-1}| + \frac{\eta\|\boldsymbol{\xi}_i\|^2}{2nm}|\ell_i'^{\widetilde{T}-1}|\max_r |\rho_{y_i,r,i}^{\widetilde{T}-1}| \geq \max_r |\rho_{y_i,r,i}^{\widetilde{T}-1}| \geq 1.5$$

where the first inequality follows from $\langle \mathbf{w}_{y_i,r}^0, \boldsymbol{\xi}_i\rangle/2 \leq 0.5 \leq \max_r |\rho_{y_i,r,i}^{\widetilde{T}-1}|/2$ based on Condition C.1. Similarly, when $\langle \mathbf{w}_{y_i,r}^0, \boldsymbol{\xi}_i\rangle < 0$, we can obtain the same result. Then, we have

$$\max_r |\langle \mathbf{w}_{y_i,r}^{\widetilde{T}}, \boldsymbol{\xi}_i\rangle| \geq \max_r |\rho_{y_i,r,i}^{\widetilde{T}}| - \beta - 4\sqrt{\frac{\log(4n^2/\delta)}{d}}n\alpha \geq 1.5 - 0.5 = 1.$$

Furthermore, we prove the results for the average quantities in a similar manner. First, from the coefficient update, and by Lemma C.15, $\text{sign}(\rho_{y_i,r,i}^{\widetilde{T}-1}) = \text{sign}(\langle \mathbf{w}_{y_i,r}^{\widetilde{T}-1}, \boldsymbol{\xi}_i\rangle)$ and thus taking the average of absolute value on both sides of (19), we get

$$\frac{1}{m}\sum_{r=1}^m |\rho_{y_i,r,i}^{\widetilde{T}}| = \frac{1}{m}\sum_{r=1}^m |\rho_{y_i,r,i}^{\widetilde{T}-1}| + \frac{\eta}{nm}|\ell_i'^{\widetilde{T}-1}|\frac{1}{m}\sum_{r=1}^m |\langle \mathbf{w}_{y_i,r}^{\widetilde{T}-1}, \boldsymbol{\xi}_i\rangle|\|\boldsymbol{\xi}_i\|^2$$

$$\geq \frac{1}{m}\sum_{r=1}^m |\rho_{y_i,r,i}^{\widetilde{T}-1}| + \frac{\eta}{nm}|\ell_i'^{\widetilde{T}-1}|(\frac{1}{m}\sum_{r=1}^m |\rho_{y_i,r,i}^{\widetilde{T}-1}| - \beta - 4\sqrt{\frac{\log(4n^2/\delta)}{d}}n\alpha)\|\boldsymbol{\xi}_i\|^2$$

$$\geq \frac{1}{m}\sum_{r=1}^m |\rho_{y_i,r,i}^{\widetilde{T}-1}| + \frac{\eta}{2nm}|\ell_i'^{\widetilde{T}-1}|\frac{1}{m}\sum_{r=1}^m |\rho_{y_i,r,i}^{\widetilde{T}-1}|\|\boldsymbol{\xi}_i\|^2$$

$$\geq \frac{1}{m}\sum_{r=1}^m |\rho_{y_i,r,i}^{\widetilde{T}-1}| \geq 3$$

where we use $|a + b| = |a| + |b|$ when $\text{sign}(a) = \text{sign}(b)$. Then, we have

$$\frac{1}{m}\sum_{r=1}^m |\langle \mathbf{w}_{y_i,r}^{\widetilde{T}}, \boldsymbol{\xi}_i\rangle| \geq \frac{1}{m}\sum_{r=1}^m |\rho_{y_i,r,i}^{\widetilde{T}}| - \beta - 4\sqrt{\frac{\log(4n^2/\delta)}{d}}n\alpha \geq 3 - 1 = 2.$$

where the inequality is by Condition C.1. $\qquad \square$

**Lemma C.18.** *Under Condition C.1, we have* $\|\mathbf{W}^{T_1} - \mathbf{W}^*\| = O(\sqrt{nm}\log(1/\epsilon)\sigma_\xi^{-1}d^{-1/2})$.

*Proof of Lemma C.18.* The proof follows similarly as in Lemma C.11. Let $\mathbf{P}_{\boldsymbol{\xi}}$ be the projection matrix to the direction of $\boldsymbol{\xi}$, i.e., $\mathbf{P}_{\boldsymbol{\xi}} = \frac{\boldsymbol{\xi}\boldsymbol{\xi}^\top}{\|\boldsymbol{\xi}\|^2}$. Then we can represent

$$\mathbf{w}_{j,r}^k - \mathbf{w}_{j,r}^0 = \mathbf{P}_{\boldsymbol{\mu}_1}(\mathbf{w}_{j,r}^k - \mathbf{w}_{j,r}^0) + \mathbf{P}_{\boldsymbol{\mu}_{-1}}(\mathbf{w}_{j,r}^k - \mathbf{w}_{j,r}^0) + \sum_{i=1}^n \mathbf{P}_{\boldsymbol{\xi}_i}(\mathbf{w}_{j,r}^k - \mathbf{w}_{j,r}^0)$$

$$+ \left(\mathbf{I} - \mathbf{P}_{\boldsymbol{\mu}_1} - \mathbf{P}_{\boldsymbol{\mu}_{-1}} - \sum_{i=1}^{n} \mathbf{P}_{\boldsymbol{\xi}_i}\right)(\mathbf{w}_{j,r}^k - \mathbf{w}_{j,r}^0).$$

By the scale difference at $T_1$ and the fact that gradient descent only updates in the direction of $\boldsymbol{\mu}_j$, $j = \pm 1$ and $\boldsymbol{\xi}_i$, we can bound

$$\|\mathbf{W}^{T_1} - \mathbf{W}^0\|^2$$

$$\leq \sum_{j=\pm 1, r \in [m]} \left(\frac{\langle \mathbf{w}_{j,r}^{T_1} - \mathbf{w}_{j,r}^0, \boldsymbol{\mu}_1 \rangle^2}{\|\boldsymbol{\mu}\|^2} + \frac{\langle \mathbf{w}_{j,r}^{T_1} - \mathbf{w}_{j,r}^0, \boldsymbol{\mu}_{-1} \rangle^2}{\|\boldsymbol{\mu}\|^2} + \sum_{i=1}^{n} \frac{\langle \mathbf{w}_{j,r}^{T_1} - \mathbf{w}_{j,r}^0, \boldsymbol{\xi}_i \rangle^2}{\|\boldsymbol{\xi}_i\|^2}\right)$$

$$+ \sum_{j=\pm 1, r \in [m]} \left\|\left(\mathbf{I} - \mathbf{P}_{\boldsymbol{\mu}_1} - \mathbf{P}_{\boldsymbol{\mu}_{-1}} - \sum_{i=1}^{n} \mathbf{P}_{\boldsymbol{\xi}_i}\right)(\mathbf{w}_{j,r}^{T_1} - \mathbf{w}_{j,r}^0)\right\|^2$$

$$\leq 2m \left(\frac{2\langle \mathbf{w}_{j,r}^{T_1}, \boldsymbol{\mu}_j \rangle^2 + 2\langle \mathbf{w}_{j,r}^{T_1}, \boldsymbol{\mu}_{-j} \rangle^2 + 2\langle \mathbf{w}_{j,r}^0, \boldsymbol{\mu}_{-j} \rangle^2 + 2\langle \mathbf{w}_{j,r}^0, \boldsymbol{\mu}_j \rangle^2}{\|\boldsymbol{\mu}\|^2}\right.$$

$$\left. + n \max_{j,r} \frac{2\langle \mathbf{w}_{j,r}^{T_1}, \boldsymbol{\xi}_i \rangle^2 + 2\langle \mathbf{w}_{j,r}^0, \boldsymbol{\xi}_i \rangle^2}{\|\boldsymbol{\xi}_i\|^2}\right) + \sum_{j=\pm 1, r \in [m]} \left\|\left(\mathbf{I} - \mathbf{P}_{\boldsymbol{\mu}_1} - \mathbf{P}_{\boldsymbol{\mu}_{-1}} - \sum_{i=1}^{n} \mathbf{P}_{\boldsymbol{\xi}_i}\right)(\mathbf{w}_{j,r}^{T_1} - \mathbf{w}_{j,r}^0)\right\|^2$$

$$\leq O(mn\sigma_\xi^{-2} d^{-1})$$

where we use the scale difference at $T_1$. Therefore,

$$\|\mathbf{W}^{T_1} - \mathbf{W}^*\| \leq \|\mathbf{W}^{T_1} - \mathbf{W}^0\| + \|\mathbf{W}^0 - \mathbf{W}^*\|$$
$$\leq O(\sqrt{mn}\sigma_\xi^{-1} d^{-1/2}) + O(\sqrt{nm} \log(1/\epsilon)\sigma_\xi^{-1} d^{-1/2})$$
$$\leq O(\sqrt{nm} \log(1/\epsilon)\sigma_\xi^{-1} d^{-1/2})$$

where we use the definition of $\mathbf{W}^*$. $\qquad\square$

**Lemma C.19.** *Under Condition C.1, we have for all $T_1 \leq k \leq T^*$*

$$\|\mathbf{W}^k - \mathbf{W}^*\|^2 - \|\mathbf{W}^{k+1} - \mathbf{W}^*\|^2 \geq 2\eta L_S(\mathbf{W}^t) - \eta\epsilon$$

*Proof of Lemma C.19.* The proof follows from similar arguments as for Lemma C.12. We first obtain a lower bound on $y_i \langle \nabla f(\mathbf{W}^t, \mathbf{x}_i), \mathbf{W}^* \rangle$ for any $i \in [n]$ for all $T_1 \leq k \leq T^*$.

$$y_i \langle \nabla f(\mathbf{W}^k, \mathbf{x}_i), \mathbf{W}^* \rangle = \frac{1}{m} \sum_{j,r} j y_i \langle \mathbf{w}_{j,r}^k, \boldsymbol{\mu}_{y_i} \rangle \langle \boldsymbol{\mu}_{y_i}, \mathbf{w}_{j,r}^* \rangle + \frac{1}{m} \sum_{j,r} j y_i \langle \mathbf{w}_{j,r}^k, \boldsymbol{\xi}_i \rangle \langle \boldsymbol{\xi}_i, \mathbf{w}_{j,r}^* \rangle$$

$$= \frac{1}{m} \sum_{j,r} j y_i \langle \mathbf{w}_{j,r}^k, \boldsymbol{\mu}_{y_i} \rangle \langle \boldsymbol{\mu}_{y_i}, \mathbf{w}_{j,r}^0 \rangle + \frac{1}{m} \sum_{j,r} j y_i \langle \mathbf{w}_{j,r}^k, \boldsymbol{\xi}_i \rangle \langle \boldsymbol{\xi}_i, \mathbf{w}_{j,r}^0 \rangle$$

$$+ \frac{1}{m} \sum_{j=\pm 1} \sum_{r=1}^{m} \sum_{i'=1}^{n} j y_i \langle \mathbf{w}_{j,r}^k, \boldsymbol{\xi}_i \rangle \mathbb{1}(j = y_{i'}) \frac{\langle \boldsymbol{\xi}_i, \boldsymbol{\xi}_{i'} \rangle}{\|\boldsymbol{\xi}_{i'}\|^2} 2\log(4/\epsilon)$$

$$= \underbrace{\frac{1}{m} \sum_{r=1}^{m} |\langle \mathbf{w}_{y_i,r}^k, \boldsymbol{\xi}_i \rangle| 2\log(4/\epsilon)}_{A_9} + \underbrace{\frac{1}{m} \sum_{j,r} \sum_{i' \neq i} \langle \mathbf{w}_{y_i,r}^k, \boldsymbol{\xi}_i \rangle 2\log(4/\epsilon) \frac{\langle \boldsymbol{\xi}_i, \boldsymbol{\xi}_{i'} \rangle}{\|\boldsymbol{\xi}_{i'}\|^2}}_{A_{10}}$$

$$+ \underbrace{\frac{1}{m} \sum_{j,r} j y_i \langle \mathbf{w}_{j,r}^k, \boldsymbol{\mu}_{y_i} \rangle \langle \boldsymbol{\mu}_{y_i}, \mathbf{w}_{j,r}^0 \rangle}_{A_{11}} + \underbrace{\frac{1}{m} \sum_{j,r} j y_i \langle \mathbf{w}_{j,r}^k, \boldsymbol{\xi}_i \rangle \langle \boldsymbol{\xi}_i, \mathbf{w}_{j,r}^0 \rangle}_{A_{12}}$$

where the second equality is by definition of $\mathbf{W}^*$. The third equality is by Lemma C.17 and Lemma C.15 on the sign invariance. We next bound based on the scale difference and Lemma B.2,

$$|A_{10}| = \widetilde{O}(nd^{-1/2}), \quad |A_{11}| = \widetilde{O}(\sigma_0 \|\boldsymbol{\mu}\|), \quad |A_{12}| \leq \widetilde{O}(\sigma_0 \sigma_\xi \sqrt{d})$$

where we use the global bound on the inner product by $\widetilde{O}(1)$. Next, by Theorem C.3 and Lemma C.17, we can show $\frac{1}{m}\sum_{r=1}^{m}|\langle \mathbf{w}_{y_i,r}^k, \boldsymbol{\mu}_{y_i}\rangle| \geq 2$ for all $i \in [n], k \geq T_1$ and we can bound

$$A_9 \geq 4\log(4/\epsilon)$$

Combining the bound for $A_9$, $A_{10}$, $A_{11}$, $A_{12}$, we have

$$y_i\langle \nabla f(\mathbf{W}^k, \mathbf{x}_i), \mathbf{W}^*\rangle \geq 2\log(4/\epsilon) \tag{20}$$

where we bound $|A_{10}| + |A_{11}| + |A_{12}| \leq 2\log(4/\epsilon)$ under Condition C.1.

Further, we derive

$$\begin{aligned}
&\|\mathbf{W}^k - \mathbf{W}^*\|^2 - \|\mathbf{W}^{k+1} - \mathbf{W}^*\|^2 \\
&= 2\eta\langle \nabla L_S(\mathbf{W}^k), \mathbf{W}^k - \mathbf{W}^*\rangle - \eta^2\|\nabla L_S(\mathbf{W}^k)\|^2 \\
&= \frac{2\eta}{n}\sum_{i=1}^{n}\ell_i'^k y_i\big(2f(\mathbf{W}^k, \mathbf{x}_i) - \langle \nabla f(\mathbf{W}^k, \mathbf{x}_i), \mathbf{W}^*\rangle\big) - \eta^2\|\nabla L_S(\mathbf{W}^k)\|^2 \\
&\geq \frac{2\eta}{n}\sum_{i=1}^{n}\ell_i'^k\big(2y_i f(\mathbf{W}^k, \mathbf{x}_i) - 2\log(2/\epsilon)\big) - \eta^2\|\nabla L_S(\mathbf{W}^k)\|^2 \\
&\geq \frac{4\eta}{n}\sum_{i=1}^{n}\big(\ell(y_i f(\mathbf{W}^k, \mathbf{x}_i)) - \epsilon/4\big) - \eta^2\|\nabla L_S(\mathbf{W}^k)\|^2 \\
&\geq 2\eta L_S(\mathbf{W}^k) - \eta\epsilon
\end{aligned}$$

where the first inequality is by (20) and the second inequality is by convexity of cross-entropy function and the last inequality is by Lemma C.8. $\qquad\square$

**Theorem C.4.** *Under the same settings as in Theorem C.3, let* $T = T_1 + \lfloor \frac{\|\mathbf{W}^{T_1} - \mathbf{W}^*\|^2}{\eta\epsilon} \rfloor = T_1 + O(\eta^{-1}\epsilon^{-1}mn\sigma_\xi^{-2}d^{-1})$. *Then we have*

- *there exists* $T_1 \leq k \leq T$ *such that* $L_S(\mathbf{W}^k) \leq 0.1$.

- $\max_{j,r,y}|\langle \mathbf{w}_{j,r}^k, \boldsymbol{\mu}_y\rangle| = o(1)$ *for all* $T_1 \leq k \leq T$.

- $\max_r|\langle \mathbf{w}_{y_i,r}^k, \boldsymbol{\xi}_i\rangle| \geq 1$ *for all* $i \in [n], T_1 \leq k \leq T$.

*Proof of Theorem C.4.* The proof is similar as in Theorem C.2. By Lemma C.19, for any $T_1 \leq k \leq T$, we have

$$\|\mathbf{W}^k - \mathbf{W}^*\|^2 - \|\mathbf{W}^{k+1} - \mathbf{W}^*\|^2 \geq 2\eta L_S(\mathbf{W}^k) - \eta\epsilon$$

for all $s \leq k$. Then summing over the inequality gives

$$\frac{1}{T - T_1 + 1}\sum_{k=T_1}^{T} L_S(\mathbf{W}^k) \leq \frac{\|\mathbf{W}^{T_1} - \mathbf{W}^*\|^2}{2\eta(T - T_1 + 1)} + \frac{\epsilon}{2} \leq \epsilon$$

where the last inequality is by the choice $T = T_1 + \lfloor \frac{\|\mathbf{W}^{T_1} - \mathbf{W}^*\|^2}{\eta\epsilon} \rfloor = T_1 + \Omega(\eta^{-1}\epsilon^{-1}nm^3\log(1/\epsilon)\sigma_\xi^{-2}d^{-1})$. Then we can claim that there exists a $k \in [T_1, T]$ such that $L_S(\mathbf{W}^k) \leq \epsilon$. Setting $\epsilon = 0.1$ shows the desired convergence.

Next, we show the upper bound on $\max_{j,y,r}|\langle \mathbf{w}_{j,r}^k, \boldsymbol{\mu}_y\rangle|$ for all $k \in [T_1, T]$. Notice that by Proposition C.1, we already have $\max_{j,r}|\langle \mathbf{w}_{-j,r}^k, \boldsymbol{\mu}_j\rangle| \leq \vartheta$, where we let

$$\vartheta := 3\max\{\max_{j,r}|\langle \mathbf{w}_{j,r}^{T_1}, \boldsymbol{\mu}_j\rangle|, \beta, 4\sqrt{\frac{\log(4n^2/\delta)}{d}}n\alpha\} = \widetilde{O}(\max\{n^{-1/2}, \sigma_0\sigma_\xi\sqrt{d}, \sigma_0\|\boldsymbol{\mu}\|, nd^{-1/2}\})$$

Subsequently, we use induction to prove $\max_{j,r}|\langle \mathbf{w}_{j,r}^k, \boldsymbol{\mu}_j\rangle| \leq 2\vartheta$. First we notice that

$$\sum_{k=T_1}^{T} L_S(\mathbf{W}^k) \leq \frac{\|\mathbf{W}^{T_1} - \mathbf{W}^*\|^2}{\eta} = O(\eta^{-1}nm\sigma_\xi^{-2}d^{-1}) \tag{21}$$

where the equality is by Lemma C.11 where we choose $\epsilon = 0.1$.

At $k = T_1$, we have $\max_{j,r} |\langle \mathbf{w}_{j,r}^{T_1}, \boldsymbol{\mu}_j \rangle| \leq \vartheta \leq 2\vartheta$. Suppose there $\widetilde{T} \in [T_1, T]$ such that $\max_{r,i} |\rho_{y_i,r,i}^{T_1}| \leq 2\vartheta$ for all $k \in [T_1, \widetilde{T} - 1]$. Now we let $\Psi^k := \max_{j,r} |\langle \mathbf{w}_{j,r}^k, \boldsymbol{\mu}_j \rangle|$ and thus by the update of inner product

$$
\begin{aligned}
\Psi^{k+1} &\leq \Psi^k + \frac{\eta}{nm} \sum_{i \in \mathcal{S}_j} |\ell_i'^k| \Psi^k \|\boldsymbol{\mu}\|^2 \\
&\leq \Psi^k + \frac{\eta}{nm} \sum_{i \in [n]} \ell_i^k \Psi^k \|\boldsymbol{\mu}\|^2 \\
&= \Psi^k + \frac{2\eta \|\boldsymbol{\mu}\|^2}{m} L_S(\mathbf{W}^k) \Psi^k.
\end{aligned}
$$

where we use $|\ell'| \leq \ell$ in the second inequality. Taking the summation from $T_1$ to $\widetilde{T}$ gives

$$
\begin{aligned}
\Psi^{\widetilde{T}} &\leq \Psi^{T_1} + \frac{2\eta \|\boldsymbol{\mu}\|^2}{m} \sum_{k=T_1}^{\widetilde{T}-1} L_S(\mathbf{W}^k) \cdot m^2 \vartheta \\
&\leq \Psi^{T_1} + O(n\text{SNR}^2) \cdot 2\vartheta \\
&\leq 2\vartheta
\end{aligned}
$$

where the second inequality is by (21) and the last inequality is by $n^{-1} \cdot \text{SNR}^{-2} \geq C'$ for sufficiently large constant $C' > 0$. The lower bound for noise inner product is directly from Lemma C.17. $\qquad\square$

## D  DIFFUSION MODEL

For the analysis of diffusion model, we restate Condition 3.1 specifically for the case of diffusion model.

**Condition D.1.** *Suppose $\delta > 0$ and the following conditions hold.*

1. *The dimension $d$ satisfies $d = \widetilde{\Omega}(\max\{n^4, n\sigma_\xi^{-1}\})$.*

2. *The training sample satisfies $n = \Omega(\log(m/\delta))$ and the network width satisfies $m = \Theta(1)$.*

3. *The initialization $\sigma_0$ satisfies $\widetilde{O}(n\sigma_\xi^{-1}d^{-5/4}) \leq \sigma_0 \leq \widetilde{O}(\min\{m^{-1/6}d^{-1/6}\sigma_\xi^{1/3}n^{-1/3},$ $m^{-1/6}d^{-7/12}\sigma_\xi^{-1/3}n^{1/3}, d^{-3/4}\sigma_\xi^{-1}n\})$*

4. *The signal strength satisfies $\|\boldsymbol{\mu}\| = \Theta(1)$.*

5. *$\mathrm{SNR}^{-1} = \widetilde{O}(d^{1/4})$.*

6. *The noise coefficients $\alpha_t, \beta_t$ satisfy $\alpha_t, \beta_t = \Theta(1)$.*

We make the following remarks on the conditions. Compared to the conditions required by classification, diffusion model requires $m = \Theta(1)$ for the analysis of stationary points. The lower bound on sample size $n$ is required for the concentration of $|\mathcal{S}_1|, |\mathcal{S}_{-1}|$. The lower bound on $\sigma_0$ is required to ensure the inner products of $\boldsymbol{\xi}_i$ across samples remain small relative to the initialization. The constant order of signal strength $\|\boldsymbol{\mu}\|$ and the bound for $n \cdot \mathrm{SNR}^2$ are utilized for simplifying the analysis. It is also worth mentioning that diffusion does not require a small learning rate for convergence.

### D.1  USEFUL LEMMAS

**Lemma D.1.** *Suppose $\delta > 0$. Then with probability at least $1 - \delta$, for any $t$,*

$$\sigma_0^2 d(1 - \widetilde{O}(d^{-1/2})) \leq \|\mathbf{w}_{r,t}^0\|^2 \leq \sigma_0^2 d(1 + \widetilde{O}(d^{-1/2}))$$
$$|\langle \mathbf{w}_{r,t}^0, \boldsymbol{\mu}_j \rangle| \leq \sqrt{2\log(16m/\delta)}\sigma_0\|\boldsymbol{\mu}_j\|,$$
$$|\langle \mathbf{w}_{r,t}^0, \boldsymbol{\xi}_i \rangle| \leq 2\sqrt{\log(16mn/\delta)}\sigma_0\sigma_\xi\sqrt{d}$$
$$|\langle \mathbf{w}_{r,t}^0, \mathbf{w}_{r',t}^0 \rangle| \leq 2\sqrt{\log(16m^2/\delta)}\sigma_0^2\sqrt{d}, \quad r \neq r'$$

*for all $r, r' \in [m]$ and $i \in [n]$, and $j = 1, 2$.*

*Proof of Lemma D.1.* The proof is the same as in (Kou et al., 2023) and we include here for completeness. Because at initialization $\mathbf{w}_{r,t}^0 \sim \mathcal{N}(0, \sigma_0^2\mathbf{I})$, by Bernstein's inequality, with probability at least $1 - \delta/(8m)$, we have

$$|\|\mathbf{w}_{r,t}^0\|_2^2 - \sigma_0^2 d| = O(\sigma_0^2\sqrt{d\log(16m/\delta)})$$

Then taking the union bound yields for all $r \in [m]$, we have with probability at least $1 - \delta/4$ that

$$\sigma_0^2 d(1 - \widetilde{O}(d^{-1/2})) \leq \|\mathbf{w}_{r,t}^0\|_2^2 \leq \sigma_0^2 d(1 + \widetilde{O}(d^{-1/2})).$$

Further, because $\langle \mathbf{w}_{r,t}^0, \boldsymbol{\mu}_j \rangle \sim \mathcal{N}(0, \sigma_0^2\|\boldsymbol{\mu}_j\|_2^2)$ for $j = 1, 2$, then by Gaussian tail bound and union bound, we have with probability at least $1 - \delta/4$, for all $j = 1, 2, r \in [m]$,

$$|\langle \mathbf{w}_{r,t}^0, \boldsymbol{\mu}_j \rangle| \leq \sqrt{2\log(16m/\delta)}\sigma_0\|\boldsymbol{\mu}\|_2$$

Finally, following similar argument and noticing that $\|\boldsymbol{\xi}_i\|_2^2 = \Theta(\sigma_\xi^2 d)$ and $\|\mathbf{w}_{r,t}^0\|_2^2 = \Theta(\sigma_0^2 d)$, we have with probability at least $1 - \delta/4$ that for all $i \in [n]$, $|\langle \mathbf{w}_{r,t}^0, \boldsymbol{\xi}_i \rangle| \leq 2\sqrt{\log(16mn/\delta)}\sigma_0\sigma_\xi\sqrt{d}$ and $|\langle \mathbf{w}_{r,t}^0, \mathbf{w}_{r',t}^0 \rangle| \leq 2\sqrt{\log(16m^2/\delta)}\sigma_0^2\sqrt{d}$. □

## D.2 DERIVATION OF LOSS FUNCTION AND GRADIENT

We first simplify the objective through taking the expectation over the added diffusion noise.

**Lemma D.2.** *The DDPM loss can be simplified under expectation as*

$$L(\mathbf{W}_t) = \frac{1}{2n} \sum_{i=1}^{n} \sum_{j \in [2]} \left( d + L_{1,i}^{(j)}(\mathbf{W}_t) + L_{2,i}^{(j)}(\mathbf{W}_t) \right),$$

*where*

$$L_{1,i}^{(j)}(\mathbf{W}_t) = \frac{1}{m} \sum_{r=1}^{m} \|\mathbf{w}_{r,t}\|^2 \left( \alpha_t^4 \langle \mathbf{w}_{r,t}, \mathbf{x}_{0,i}^{(j)} \rangle^4 + 6\alpha_t^2 \beta_t^2 \langle \mathbf{w}_{r,t}, \mathbf{x}_{0,i}^{(j)} \rangle^2 \|\mathbf{w}_{r,t}\|^2 + 3\beta_t^4 \|\mathbf{w}_{r,t}\|^4 \right.$$

$$\left. - 4\sqrt{m} \alpha_t \beta_t \langle \mathbf{w}_{r,t}, \mathbf{x}_{0,i}^{(j)} \rangle \right)$$

$$L_{2,i}^{(j)}(\mathbf{W}_t) = \frac{2}{m} \sum_{r=1}^{m} \sum_{r' \neq r} \langle \mathbf{w}_{r,t}, \mathbf{w}_{r',t} \rangle \left( \left( \alpha_t^2 \langle \mathbf{w}_{r,t}, \mathbf{x}_{0,i}^{(j)} \rangle^2 + \beta_t^2 \|\mathbf{w}_{r,t}\|^2 \right) \left( \alpha_t^2 \langle \mathbf{w}_{r',t}, \mathbf{x}_{0,i}^{(j)} \rangle^2 + \beta_t^2 \|\mathbf{w}_{r',t}\|^2 \right) \right.$$

$$\left. + 2\beta_t^4 \langle \mathbf{w}_{r,t}, \mathbf{w}_{r',t} \rangle^2 + 4\alpha_t^2 \beta_t^2 \langle \mathbf{w}_{r,t}, \mathbf{x}_{0,i} \rangle \langle \mathbf{w}_{r',t}, \mathbf{x}_{0,i} \rangle \langle \mathbf{w}_{r,t}, \mathbf{w}_{r',t} \rangle \right)$$

*corresponding to the learning of $r$-th neuron and alignment of $r$-th neuron with other neurons respectively.*

*Proof of Lemma D.2.* Without loss of generality, we consider for a single sample $\mathbf{x}_{t,i}$. We first write the objective as

$$\mathbb{E} \| \boldsymbol{f}_p(\mathbf{W}_t, \mathbf{x}_{t,i}^{(p)}) - \boldsymbol{\epsilon}_{t,i}^{(p)} \|^2$$

$$= \underbrace{\mathbb{E} \| \boldsymbol{\epsilon}_{t,i}^{(p)} \|^2}_{I_1} + \underbrace{\mathbb{E} \left\| \frac{1}{\sqrt{m}} \sum_{r=1}^{m} \sigma(\langle \mathbf{w}_{r,t}, \mathbf{x}_{t,i}^{(p)} \rangle) \mathbf{w}_{r,t} \right\|^2}_{I_2} - \underbrace{2\, \mathbb{E} \left[ \frac{1}{\sqrt{m}} \sum_{r=1}^{m} \sigma(\langle \mathbf{w}_{r,t}, \mathbf{x}_{t,i}^{(p)} \rangle) \langle \mathbf{w}_{r,t}, \boldsymbol{\epsilon}_{t,i} \rangle \right]}_{I_3}$$

where we omit the subscript for the expectation for clarity.

First, we can see $I_1 = d$. Then

$$I_3 = \frac{1}{\sqrt{m}} \sum_{r=1}^{m} \mathbb{E} \left[ (\langle \mathbf{w}_{r,t}, \mathbf{x}_{t,i}^{(p)} \rangle)^2 \langle \mathbf{w}_{r,t}, \boldsymbol{\epsilon}_{t,i} \rangle \right]$$

$$= \frac{1}{\sqrt{m}} \sum_{r=1}^{m} \sum_{i'=1}^{d} \mathbb{E} \left[ (\langle \mathbf{w}_{r,t}, \mathbf{x}_{t,i}^{(p)} \rangle)^2 \mathbf{w}_{r,t}[i'] \boldsymbol{\epsilon}_{t,i}[i'] \right]$$

$$= \frac{2\beta_t}{\sqrt{m}} \sum_{r=1}^{m} \sum_{i'=1}^{d} \mathbb{E} \left[ (\langle \mathbf{w}_{r,t}, \mathbf{x}_{t,i}^{(p)} \rangle) \mathbf{w}_{r,t}[i']^2 \right]$$

$$= \frac{2\beta_t}{\sqrt{m}} \sum_{r=1}^{m} \|\mathbf{w}_{r,t}\|^2 \mathbb{E} \left[ \langle \mathbf{w}_{r,t}, \mathbf{x}_{t,i}^{(p)} \rangle \right]$$

$$= \frac{2\alpha_t \beta_t}{\sqrt{m}} \sum_{r=1}^{m} \|\mathbf{w}_{r,t}\|^2 \langle \mathbf{w}_{r,t}, \mathbf{x}_{0,i}^{(p)} \rangle$$

where the third equality uses Stein's Lemma.

Next, we consider $I_2$ by writing

$$I_2 = \frac{1}{m} \sum_{r=1}^{m} \mathbb{E} \left[ (\langle \mathbf{w}_{r,t}, \mathbf{x}_{t,i}^{(p)} \rangle)^4 \right] \|\mathbf{w}_{r,t}\|^2 + \frac{2}{m} \sum_{r=1}^{m} \sum_{r' \neq r} \mathbb{E} \left[ (\langle \mathbf{w}_{r,t}, \mathbf{x}_{t,i}^{(p)} \rangle)^2 (\langle \mathbf{w}_{r',t}, \mathbf{x}_{t,i}^{(p)} \rangle)^2 \right] \langle \mathbf{w}_{r,t}, \mathbf{w}_{r',t} \rangle.$$

Next, we compute the two terms $\mathbb{E}\big[(\langle \mathbf{w}_{r,t}, \mathbf{x}_{t,i}^{(p)}\rangle)^4\big]$ and $\mathbb{E}\big[(\langle \mathbf{w}_{r,t}, \mathbf{x}_{t,i}^{(p)}\rangle)^2(\langle \mathbf{w}_{r',t}, \mathbf{x}_{t,i}^{(p)}\rangle)^2\big]$ respectively. For notation simplicity, we let $a_r := \alpha_t \langle \mathbf{w}_{r,t}, \mathbf{x}_{0,i}^{(p)}\rangle$, $b_r := \beta_t \|\mathbf{w}_{r,t}\|$ and $z_r := \beta_t \langle \mathbf{w}_{r,t}, \boldsymbol{\epsilon}_{t,i}\rangle$. We first compute $\mathbb{E}[z_r] = 0$ and $\mathbb{E}[z_r^2] = \beta_t^2 \|\mathbf{w}_{r,t}\|^2$, $\mathbb{E}[z_r^4] = 3\beta_t^4 \|\mathbf{w}_{r,t}\|^4$. For the first term,

$$
\begin{aligned}
\mathbb{E}\big[(\langle \mathbf{w}_{r,t}, \mathbf{x}_{t,i}^{(p)}\rangle)^4\big] &= \mathbb{E}\big[(a_r + z_r)^4\big] \\
&= \mathbb{E}[a_r^4 + 4a_r^3 z_r + 6a_r^2 z_r^2 + 4a_r z_r^3 + z_r^4] \\
&= a_r^4 + 6a_r^2 \mathbb{E}[z_r^2] + \mathbb{E}[z_r^4] \\
&= a_r^4 + 6a_r^2 b_r^2 + 3b_r^4 \\
&= \alpha_t^4 \langle \mathbf{w}_{r,t}, \mathbf{x}_{0,i}^{(p)}\rangle^4 + 6\alpha_t^2 \beta_t^2 \langle \mathbf{w}_{r,t}, \mathbf{x}_{0,i}^{(p)}\rangle^2 \|\mathbf{w}_{r,t}\|^2 + 3\beta_t^4 \|\mathbf{w}_{r,t}\|^4
\end{aligned}
$$

Next, for $\mathbb{E}_{\boldsymbol{\epsilon}_{t,i} \sim \mathcal{N}(0,\mathbf{I})}\big[\langle \mathbf{w}_{r,t}, \alpha_t \mathbf{x}_{0,i} + \beta_t \boldsymbol{\epsilon}_{t,i}\rangle^2 \langle \mathbf{w}_{r',t}, \alpha_t \mathbf{x}_{0,i} + \beta_t \boldsymbol{\epsilon}_{t,i}\rangle^2\big]$, we note that

$$
\begin{aligned}
\mathbb{E}[z_r z_{r'}] &= \beta_t^2 \mathbb{E}[\boldsymbol{\epsilon}_{t,i}^\top \mathbf{w}_{r,t} \mathbf{w}_{r',t}^\top \boldsymbol{\epsilon}_{t,i}] = \beta_t^2 \langle \mathbf{w}_{r,t}, \mathbf{w}_{r',t}\rangle, \\
\mathbb{E}[z_r z_{r'}^2] &= 0 \\
\mathbb{E}[z_r^2 z_{r'}^2] &= \mathbb{E}[z_r^2]\mathbb{E}[z_{r'}^2] + 2\mathbb{E}[z_r z_{r'}]^2 = \beta_t^4 \|\mathbf{w}_{r,t}\|^2 \|\mathbf{w}_{r',t}\|^2 + 2\beta_t^4 \langle \mathbf{w}_{r,t}, \mathbf{w}_{r',t}\rangle^2
\end{aligned}
$$

where the second and third results follow from Isserlis Theorem. Then we can simplify

$$
\begin{aligned}
&\mathbb{E}\big[\langle \mathbf{w}_{r,t}, \alpha_t \mathbf{x}_{0,i} + \beta_t \boldsymbol{\epsilon}_{t,i}\rangle^2 \langle \mathbf{w}_{r',t}, \alpha_t \mathbf{x}_{0,i} + \beta_t \boldsymbol{\epsilon}_{t,i}\rangle^2\big] \\
&= \mathbb{E}\big[(a_r + z_r)^2 (a_{r'} + z_{r'})^2\big] \\
&= a_r^2 a_{r'}^2 + a_r^2 \mathbb{E}[z_{r'}^2] + 4a_r a_{r'} \mathbb{E}[z_r z_{r'}] + a_{r'}^2 \mathbb{E}[z_r^2] + \mathbb{E}[z_r^2 z_{r'}^2] \\
&= \alpha_t^4 \langle \mathbf{w}_{r,t}, \mathbf{x}_{0,i}\rangle^2 \langle \mathbf{w}_{r',t}, \mathbf{x}_{0,i}\rangle^2 + \alpha_t^2 \beta_t^2 \langle \mathbf{w}_{r,t}, \mathbf{x}_{0,i}\rangle^2 \|\mathbf{w}_{r',t}\|^2 + 4\alpha_t^2 \beta_t^2 \langle \mathbf{w}_{r,t}, \mathbf{x}_{0,i}\rangle \langle \mathbf{w}_{r',t}, \mathbf{x}_{0,i}\rangle \langle \mathbf{w}_{r,t}, \mathbf{w}_{r',t}\rangle \\
&\quad + \alpha_t^2 \beta_t^2 \langle \mathbf{w}_{r',t}, \mathbf{x}_{0,i}\rangle^2 \|\mathbf{w}_{r,t}\|^2 + \beta_t^4 \|\mathbf{w}_{r,t}\|^2 \|\mathbf{w}_{r',t}\|^2 + 2\beta_t^4 \langle \mathbf{w}_{r,t}, \mathbf{w}_{r',t}\rangle^2
\end{aligned}
$$

Combining $I_1, I_2, I_3$ gives

$$
\begin{aligned}
&\mathbb{E}\|s_t(\mathbf{x}_{t,i}^{(p)}) - \boldsymbol{\epsilon}_{t,i}\|^2 \\
&= d + \underbrace{\frac{1}{m}\sum_{r=1}^m \|\mathbf{w}_{r,t}\|^2 \Big(\alpha_t^4 \langle \mathbf{w}_{r,t}, \mathbf{x}_{0,i}^{(p)}\rangle^4 + 6\alpha_t^2 \beta_t^2 \langle \mathbf{w}_{r,t}, \mathbf{x}_{0,i}^{(p)}\rangle^2 \|\mathbf{w}_{r,t}\|^2 + 3\beta_t^4 \|\mathbf{w}_{r,t}\|^4 - 4\sqrt{m}\alpha_t \beta_t \langle \mathbf{w}_{r,t}, \mathbf{x}_{0,i}^{(p)}\rangle\Big)}_{L_{1,i}^{(p)}(\mathbf{w}_{r,t})} \\
&\quad + \underbrace{\frac{2}{m}\sum_{r=1}^m \sum_{r' \neq r} \langle \mathbf{w}_{r,t}, \mathbf{w}_{r',t}\rangle \Big(\big(\alpha_t^2 \langle \mathbf{w}_{r,t}, \mathbf{x}_{0,i}^{(p)}\rangle^2 + \beta_t^2 \|\mathbf{w}_{r,t}\|^2\big)\big(\alpha_t^2 \langle \mathbf{w}_{r',t}, \mathbf{x}_{0,i}^{(p)}\rangle^2 + \beta_t^2 \|\mathbf{w}_{r',t}\|^2\big)}_{} \\
&\quad \underbrace{+ 2\beta_t^4 \langle \mathbf{w}_{r,t}, \mathbf{w}_{r',t}\rangle^2 + 4\alpha_t^2 \beta_t^2 \langle \mathbf{w}_{r,t}, \mathbf{x}_{0,i}\rangle \langle \mathbf{w}_{r',t}, \mathbf{x}_{0,i}\rangle \langle \mathbf{w}_{r,t}, \mathbf{w}_{r',t}\rangle\Big)}_{L_{2,i}^{(p)}(\mathbf{w}_{r,t})}
\end{aligned}
$$

where we respectively denote the two composing loss terms as $L_{1,i}^{(p)}$ (corresponding to the learning of $r$-th neuron) and $L_{2,i}^{(p)}$ (alignment with other neurons). $\qquad \square$

We next compute the gradient of the DDPM loss in expectation.

**Lemma D.3.** *The gradient of expected DDPM loss in Lemma D.2 can be computed as*

$$
\nabla L(\mathbf{W}_t) = \frac{1}{2n}\sum_{i=1}^n \sum_{p \in [2]} \big(\nabla L_{1,i}^{(p)}(\mathbf{W}_t) + \nabla L_{2,i}^{(p)}(\mathbf{W}_t)\big)
$$

*where*

$$
\begin{aligned}
&\nabla L_{1,i}^{(p)}(\mathbf{w}_{r,t}) \\
&= \frac{2}{m}\Big(\alpha_t^4 \langle \mathbf{w}_{r,t}, \mathbf{x}_{0,i}^{(p)}\rangle^4 + 12\alpha_t^2 \beta_t^2 \langle \mathbf{w}_{r,t}, \mathbf{x}_{0,i}^{(p)}\rangle^2 \|\mathbf{w}_{r,t}\|^2 + 9\beta_t^4 \|\mathbf{w}_{r,t}\|^4 - 4\sqrt{m}\alpha_t \beta_t \langle \mathbf{w}_{r,t}, \mathbf{x}_{0,i}^{(p)}\rangle\Big)\mathbf{w}_{r,t}
\end{aligned}
$$

$$+ \frac{2}{m}\Big(2\alpha_t^4\langle\mathbf{w}_{r,t}, \mathbf{x}_{0,i}^{(p)}\rangle^3\|\mathbf{w}_{r,t}\|^2 + 6\alpha_t^2\beta_t^2\|\mathbf{w}_{r,t}\|^4\langle\mathbf{w}_{r,t}, \mathbf{x}_{0,i}^{(p)}\rangle - 2\sqrt{m}\alpha_t\beta_t\|\mathbf{w}_{r,t}\|^2\Big)\mathbf{x}_{0,i}^{(p)}$$

$$\nabla L_{2,i}^{(p)}(\mathbf{w}_{r,t})$$
$$= \frac{2}{m}\sum_{r'\neq r}\Big((\alpha_t^2\langle\mathbf{w}_{r,t}, \mathbf{x}_{0,i}^{(p)}\rangle^2 + \beta_t^2\|\mathbf{w}_{r,t}\|^2)(\alpha_t^2\langle\mathbf{w}_{r',t}, \mathbf{x}_{0,i}^{(p)}\rangle^2 + \beta_t^2\|\mathbf{w}_{r',t}\|^2) + 2\beta_t^4\langle\mathbf{w}_{r,t}, \mathbf{w}_{r',t}\rangle^2$$

$$+ 4\alpha_t^2\beta_t^2\langle\mathbf{w}_{r,t}, \mathbf{x}_{0,i}\rangle\langle\mathbf{w}_{r',t}, \mathbf{x}_{0,i}\rangle\langle\mathbf{w}_{r,t}, \mathbf{w}_{r',t}\rangle\Big)\mathbf{w}_{r',t}$$

$$+ \frac{2}{m}\sum_{r'\neq r}(\alpha_t^2\langle\mathbf{w}_{r',t}, \mathbf{x}_{0,i}^{(p)}\rangle^2 + \beta_t^2\|\mathbf{w}_{r',t}\|^2)\langle\mathbf{w}_{r,t}, \mathbf{w}_{r',t}\rangle(2\alpha_t^2\langle\mathbf{w}_{r,t}, \mathbf{x}_{0,i}^{(p)}\rangle\mathbf{x}_{0,i}^{(p)} + 2\beta_t^2\mathbf{w}_{r,t})$$

$$+ \frac{2}{m}\sum_{r'\neq r}\langle\mathbf{w}_{r,t}, \mathbf{w}_{r',t}\rangle^2(4\beta_t^2\mathbf{w}_{r',t} + 8\alpha_t^2\beta_t^2\langle\mathbf{w}_{r,t}, \mathbf{x}_{0,i}\rangle\mathbf{x}_{0,i})$$

*Proof of Lemma D.3.* The proof is straightforward and thus omitted for clarity. $\qquad\square$

### D.3 FIRST STAGE

Before deriving the results for the first stage, we derive the following lemma that decomposes the weight norm given concentration of neurons.

**Lemma D.4.** *For any $k$ and $r \in [m]$, such that $\langle\mathbf{w}_{r,t}^k, \boldsymbol{\mu}_j\rangle = \Theta(\langle\mathbf{w}_{r,t}^k, \boldsymbol{\mu}_{j'}\rangle) \geq \widetilde{\Theta}(\sigma_0\|\boldsymbol{\mu}\|)$, $\langle\mathbf{w}_{r,t}^k, \boldsymbol{\xi}_i\rangle = \Theta(\langle\mathbf{w}_{r,t}^k, \boldsymbol{\xi}_{i'}\rangle) \geq \widetilde{\Theta}(\sigma_0\sigma_\xi\sqrt{d})$ and $\langle\mathbf{w}_{r,t}^k, \boldsymbol{\mu}_j\rangle, \langle\mathbf{w}_{r,t}^k, \boldsymbol{\xi}_i\rangle = \widetilde{O}(1)$, $\langle\mathbf{w}_{r,t}^k, \mathbf{w}_{r,t}^0\rangle = \Theta(\sigma_0^2 d)$ for any $j, j' = \pm 1, i, i' \in [n], r \in [m]$. Then we can show*
$$\|\mathbf{w}_{r,t}^k\|^2 = \Theta\big(\langle\mathbf{w}_{r,t}^k, \boldsymbol{\mu}_j\rangle^2\|\boldsymbol{\mu}\|^{-2} + n\cdot\mathrm{SNR}^2\langle\mathbf{w}_{r,t}^k, \boldsymbol{\xi}_i\rangle^2\|\boldsymbol{\mu}\|^{-2} + \|\mathbf{w}_{r,t}^0\|^2\big).$$
*and for $r \neq r'$, we have*
$$\langle\mathbf{w}_{r,t}^k, \mathbf{w}_{r',t}^k\rangle = \Theta\big(\langle\mathbf{w}_{r,t}^k, \boldsymbol{\mu}_j\rangle\langle\mathbf{w}_{r',t}^k, \boldsymbol{\mu}_j\rangle\|\boldsymbol{\mu}\|^{-2} + n\cdot\mathrm{SNR}^2\langle\mathbf{w}_{r,t}^k, \boldsymbol{\xi}_i\rangle\langle\mathbf{w}_{r',t}^k, \boldsymbol{\xi}_i\rangle\|\boldsymbol{\mu}\|^{-2} + \langle\mathbf{w}_{r,t}^0, \mathbf{w}_{r',t}^0\rangle\big)$$

*Proof of Lemma D.4.* We decompose the weight $\mathbf{w}_{r,t}^k$ as

$$\mathbf{w}_{r,t}^k = \phi_r^k\mathbf{w}_{r,t}^0 + \gamma_1^k\boldsymbol{\mu}_1\|\boldsymbol{\mu}_1\|^{-2} + \gamma_{-1}^k\boldsymbol{\mu}_{-1}\|\boldsymbol{\mu}_{-1}\|^{-2} + \sum_{i=1}^n\rho_{r,i}^k\boldsymbol{\xi}_i\|\boldsymbol{\xi}_i\|^{-2}, \tag{22}$$

based on the gradient descent updates of $\mathbf{w}_{r,t}^k$ starting from small initialization $\mathbf{w}_{r,t}^0$ and the direction of update only involves $\mathbf{w}_{r,t}^k$ and $\boldsymbol{\mu}_{\pm 1}, \boldsymbol{\xi}_i$, where $\gamma_1^0 = \gamma_{-1}^0 = \rho_{r,i}^0 = 0$ and $\phi_r^k = 1$.

Then given the assumption that $\langle\mathbf{w}_{r,t}^k, \mathbf{w}_{r,t}^0\rangle = \Theta(\sigma_0^2 d)$, we have $\phi_r^k = \Theta(1)$ because

$$\langle\mathbf{w}_{r,t}^k, \mathbf{w}_{r,t}^0\rangle = \phi_r^k\|\mathbf{w}_{r,t}^0\|^2 + \langle\gamma_1^k\boldsymbol{\mu}_1\|\boldsymbol{\mu}_1\|^{-2} + \gamma_{-1}^k\boldsymbol{\mu}_{-1}\|\boldsymbol{\mu}_{-1}\|^{-2} + \sum_{i=1}^n\rho_{r,i}^k\boldsymbol{\xi}_i\|\boldsymbol{\xi}_i\|^{-2}, \mathbf{w}_{r,t}^0\rangle$$
$$= \phi_r^k\Theta(\sigma_0^2 d),$$

where the second equality uses Lemma D.1. This suggests that $\phi_r^k = \Theta(1)$.

Then we can see
$$\langle\mathbf{w}_{r,t}^k, \boldsymbol{\mu}_j\rangle = \phi_r^k\langle\mathbf{w}_{r,t}^0, \boldsymbol{\mu}_j\rangle + \gamma_j^k$$
$$\langle\mathbf{w}_{r,t}^k, \boldsymbol{\xi}_i\rangle = \phi_r^k\langle\mathbf{w}_{r,t}^0, \boldsymbol{\xi}_i\rangle + \rho_{r,i}^k + \sum_{i'\neq i}\rho_{r,i'}^k\langle\boldsymbol{\xi}_i, \boldsymbol{\xi}_{i'}\rangle\|\boldsymbol{\xi}_i\|^{-2} = \phi_r^k\langle\mathbf{w}_{r,t}^0, \boldsymbol{\xi}_i\rangle + \rho_{r,i}^k + \widetilde{O}(nd^{-1/2}),$$

where the second equality for $\langle\mathbf{w}_{r,t}^k, \boldsymbol{\xi}_i\rangle$ is by Lemma B.2 and $\langle\mathbf{w}_{r,t}^k, \boldsymbol{\xi}_i\rangle = \widetilde{O}(1)$, thus $\rho_{r,i}^k = \widetilde{O}(1)$.

Then based on the assumptions that $|\langle\mathbf{w}_{r,t}^k, \boldsymbol{\mu}_j\rangle| \geq \widetilde{\Theta}(\sigma_0\|\boldsymbol{\mu}\|)$ and $|\langle\mathbf{w}_{r,t}^k, \boldsymbol{\xi}_i\rangle| \geq \widetilde{\Theta}(\sigma_0\sigma_\xi\sqrt{d})$ and $\phi_r^k = \Theta(1)$, we can simplify (22) as

$$\mathbf{w}_{r,t}^k = \Theta(\mathbf{w}_{r,t}^0) + \Theta(\langle\mathbf{w}_{r,t}^k, \boldsymbol{\mu}_j\rangle(\boldsymbol{\mu}_1 + \boldsymbol{\mu}_{-1})\|\boldsymbol{\mu}\|^{-2}) + \Theta(\langle\mathbf{w}_{r,t}^k, \boldsymbol{\xi}_i\rangle + \widetilde{O}(nd^{-1/2}))\sum_{i=1}^n\boldsymbol{\xi}_i\|\boldsymbol{\xi}_i\|^{-2}, \tag{23}$$

where we use $\langle \mathbf{w}_{r,t}^k, \boldsymbol{\mu}_j \rangle = \Theta(\langle \mathbf{w}_{r,t}^k, \boldsymbol{\mu}_{j'} \rangle) \geq \phi_r^k |\langle \mathbf{w}_{r,t}^0, \boldsymbol{\mu}_j \rangle|$ and $\langle \mathbf{w}_{r,t}^k, \boldsymbol{\xi}_i \rangle = \Theta(\langle \mathbf{w}_{r,t}^k, \boldsymbol{\xi}_{i'} \rangle) \geq \phi_r^k |\langle \mathbf{w}_{r,t}^0, \boldsymbol{\xi}_i \rangle|$ in the first equality. For the second equality, we use the assumption that $\langle \mathbf{w}_{r,t}^k, \boldsymbol{\xi}_i \rangle \geq \widetilde{\Theta}(\sigma_0 \sigma_\xi \sqrt{d})$.

Then we can show

$\|\mathbf{w}_{r,t}^k\|^2$

$$= \Theta(\|\mathbf{w}_{r,t}^0\|^2) + \Theta(\langle \mathbf{w}_{r,t}^k, \boldsymbol{\mu}_j \rangle^2)\|\boldsymbol{\mu}\|^{-2} + \Theta(\langle \mathbf{w}_{r,t}^k, \boldsymbol{\xi}_i \rangle^2 + \widetilde{O}(nd^{-1/2}))\|\sum_{i=1}^n \boldsymbol{\xi}_i \|\boldsymbol{\xi}_i\|^{-2}\|^2$$

$$+ \Theta(\langle \mathbf{w}_{r,t}^k, \boldsymbol{\mu}_j \rangle \langle \mathbf{w}_{r,t}^0, \boldsymbol{\mu}_j \rangle \|\boldsymbol{\mu}\|^{-2}) + \Theta((\langle \mathbf{w}_{r,t}^k, \boldsymbol{\xi}_i \rangle + \widetilde{O}(nd^{-1/2}))\langle \mathbf{w}_{r,t}^0, \boldsymbol{\xi}_i \rangle \sum_{i=1}^n \|\boldsymbol{\xi}_i\|^{-2})$$

$$= \Theta(\sigma_0^2 d) + \Theta(\langle \mathbf{w}_{r,t}^k, \boldsymbol{\mu}_j \rangle^2)\|\boldsymbol{\mu}\|^{-2} + \Theta(\langle \mathbf{w}_{r,t}^k, \boldsymbol{\xi}_i \rangle^2 + \widetilde{O}(nd^{-1/2}))(\Theta(n\sigma_\xi^{-2}d^{-1}) + \widetilde{O}(n^2\sigma_\xi^{-2}d^{-3/2}))$$

$$+ \Theta((\langle \mathbf{w}_{r,t}^k, \boldsymbol{\xi}_i \rangle + \widetilde{O}(nd^{-1/2}))\langle \mathbf{w}_{r,t}^0, \boldsymbol{\xi}_i \rangle n\sigma_\xi^{-2}d^{-1})$$

$$= \Theta(\sigma_0^2 d) + \Theta(\langle \mathbf{w}_{r,t}^k, \boldsymbol{\mu}_j \rangle^2)\|\boldsymbol{\mu}\|^{-2} + \Theta(n\sigma_\xi^{-2}d^{-1}\langle \mathbf{w}_{r,t}^k, \boldsymbol{\xi}_i \rangle^2) + \widetilde{O}(n^2\sigma_\xi^{-2}d^{-3/2}) + \widetilde{O}(n^2\sigma_0\sigma_\xi^{-1}d^{-1})$$

$$= \Theta(\langle \mathbf{w}_{r,t}^k, \boldsymbol{\mu}_j \rangle^2 \|\boldsymbol{\mu}\|^{-2} + n\mathrm{SNR}^2\langle \mathbf{w}_{r,t}^k, \boldsymbol{\xi}_i \rangle^2 \|\boldsymbol{\mu}\|^{-2} + \sigma_0^2 d),$$

where the second equality uses Lemma D.1, Lemma B.2 and $\langle \mathbf{w}_{r,t}^k, \boldsymbol{\mu}_j \rangle = \Theta(\langle \mathbf{w}_{r,t}^k, \boldsymbol{\mu}_{j'} \rangle) \geq \phi_r^k |\langle \mathbf{w}_{r,t}^0, \boldsymbol{\mu}_j \rangle|$. The third equality is by the condition on $d = \widetilde{\Omega}(n^2)$ and $\langle \mathbf{w}_{r,t}^k, \boldsymbol{\xi}_i \rangle = \Theta(\langle \mathbf{w}_{r,t}^k, \boldsymbol{\xi}_{i'} \rangle) \geq \phi_r^k |\langle \mathbf{w}_{r,t}^0, \boldsymbol{\xi}_i \rangle|$ and Lemma D.1. The last equality is by the condition $\sigma_0 \geq \widetilde{\Omega}(\max\{n\sigma_\xi^{-1}d^{-5/4}, n^2\sigma_\xi^{-1}d^{-2}\}) = \widetilde{\Omega}(n\sigma_\xi^{-1}d^{-5/4})$ given $d = \widetilde{\Omega}(n^2)$.

In addition, we can deduce from (23) that

$\langle \mathbf{w}_{r,t}^k, \mathbf{w}_{r',t}^k \rangle$

$$= \Theta(\langle \mathbf{w}_{r,t}^0, \mathbf{w}_{r',t}^0 \rangle) + \Theta(\langle \mathbf{w}_{r',t}^k, \boldsymbol{\mu}_j \rangle \langle \mathbf{w}_{r,t}^0, \boldsymbol{\mu}_j \rangle \|\boldsymbol{\mu}\|^{-2}) + \Theta(\langle \mathbf{w}_{r',t}^k, \boldsymbol{\xi}_i \rangle \sum_{i=1}^n \langle \mathbf{w}_{r,t}^0, \boldsymbol{\xi}_i \rangle \|\boldsymbol{\xi}_i\|^{-2})$$

$$+ \Theta(\langle \mathbf{w}_{r,t}^k, \boldsymbol{\mu}_j \rangle \langle \mathbf{w}_{r',t}^0, \boldsymbol{\mu}_j \rangle \|\boldsymbol{\mu}\|^{-2}) + \Theta(\langle \mathbf{w}_{r,t}^k, \boldsymbol{\xi}_i \rangle \sum_{i=1}^n \langle \mathbf{w}_{r',t}^0, \boldsymbol{\xi}_i \rangle \|\boldsymbol{\xi}_i\|^{-2})$$

$$+ \Theta(\langle \mathbf{w}_{r,t}^k, \boldsymbol{\mu}_j \rangle \langle \mathbf{w}_{r',t}^k, \boldsymbol{\mu}_j \rangle \|\boldsymbol{\mu}\|^{-2}) + \Theta(n\langle \mathbf{w}_{r,t}^k, \boldsymbol{\xi}_i \rangle \langle \mathbf{w}_{r',t}^k, \boldsymbol{\xi}_i \rangle \|\boldsymbol{\xi}_i\|^{-2})$$

$$= \Theta(\langle \mathbf{w}_{r,t}^0, \mathbf{w}_{r',t}^0 \rangle) + \Theta(\langle \mathbf{w}_{r,t}^k, \boldsymbol{\mu}_j \rangle \langle \mathbf{w}_{r',t}^k, \boldsymbol{\mu}_j \rangle \|\boldsymbol{\mu}\|^{-2}) + \Theta(n\langle \mathbf{w}_{r,t}^k, \boldsymbol{\xi}_i \rangle \langle \mathbf{w}_{r',t}^k, \boldsymbol{\xi}_i \rangle \|\boldsymbol{\xi}_i\|^{-2})$$

$$= \Theta(\langle \mathbf{w}_{r,t}^0, \mathbf{w}_{r',t}^0 \rangle) + \Theta(\langle \mathbf{w}_{r,t}^k, \boldsymbol{\mu}_j \rangle \langle \mathbf{w}_{r',t}^k, \boldsymbol{\mu}_j \rangle \|\boldsymbol{\mu}\|^{-2}) + \Theta(n\mathrm{SNR}^2\langle \mathbf{w}_{r,t}^k, \boldsymbol{\xi}_i \rangle \langle \mathbf{w}_{r',t}^k, \boldsymbol{\xi}_i \rangle \|\boldsymbol{\mu}\|^{-2})$$

where we use $\phi_r^k = \Theta(1)$ and the $\langle \mathbf{w}_{r,t}^k, \boldsymbol{\mu}_j \rangle = \Theta(\langle \mathbf{w}_{r,t}^k, \boldsymbol{\mu}_{j'} \rangle) \geq \phi_r^k |\langle \mathbf{w}_{r,t}^0, \boldsymbol{\mu}_j \rangle|$ and $\langle \mathbf{w}_{r,t}^k, \boldsymbol{\xi}_i \rangle = \Theta(\langle \mathbf{w}_{r,t}^k, \boldsymbol{\xi}_{i'} \rangle) \geq \phi_r^k |\langle \mathbf{w}_{r,t}^0, \boldsymbol{\xi}_i \rangle|$ for the equalities. $\qquad \square$

**Lemma D.5** (Restatement of Lemma 4.1). *Under Condition D.1, there exists an iteration $T_1 = \max\{T_\mu, T_\xi\}$, where $T_\mu = \widetilde{\Theta}(\sqrt{m}\sigma_0^{-1}d^{-1}\|\boldsymbol{\mu}\|^{-1}\eta^{-1})$ and $T_\xi = \widetilde{\Theta}(n\sqrt{m}\sigma_0^{-1}\sigma_\xi^{-1}d^{-3/2}\eta^{-1})$ such that for all $0 \leq k \leq T_1$, (1) $\|\mathbf{w}_{r,t}^k\|^2 = \Theta(\sigma_0^2 d)$ for all $r \in [m], j = \pm 1, i \in [n]$, (2) $\langle \mathbf{w}_{r,t}^k, \mathbf{w}_{r,t}^0 \rangle = \Theta(\sigma_0^2 d)$, and (3) the signal and noise learning dynamics can be simplified to*

$$\langle \mathbf{w}_{r,t}^{k+1}, \boldsymbol{\mu}_j \rangle = \langle \mathbf{w}_{r,t}^k, \boldsymbol{\mu}_j \rangle + \Theta\Big(\frac{\eta\alpha_t\beta_t|\mathcal{S}_j|}{n\sqrt{m}}\|\mathbf{w}_{r,t}^k\|^2\|\boldsymbol{\mu}_j\|^2\Big)$$

$$\langle \mathbf{w}_{r,t}^{k+1}, \boldsymbol{\xi}_i \rangle = \langle \mathbf{w}_{r,t}^k, \boldsymbol{\xi}_i \rangle + \Theta\Big(\frac{\eta\alpha_t\beta_t}{n\sqrt{m}}\|\mathbf{w}_{r,t}^k\|^2\|\boldsymbol{\xi}_i\|^2\Big)$$

*for all $j = \pm 1$, $r \in [m], i \in [n]$. Furthermore, we can show*

- $\langle \mathbf{w}_{r,t}^{T_1}, \boldsymbol{\mu}_j \rangle = \Theta(\langle \mathbf{w}_{r',t}^{T_1}, \boldsymbol{\mu}_{j'} \rangle) > 0$,

- $\langle \mathbf{w}_{r,t}^{T_1}, \boldsymbol{\xi}_i \rangle = \Theta(\langle \mathbf{w}_{r',t}^{T_1}, \boldsymbol{\xi}_{i'} \rangle) > 0$,

- $\|\mathbf{w}_{r,t}^{T_1}\|^2 = \Theta(\|\mathbf{w}_{r',t}^{T_1}\|^2)$,

- $\langle \mathbf{w}_{r,t}^{T_1}, \mathbf{w}_{r',t}^{T_1} \rangle = \Theta(\|\mathbf{w}_{r,t}^{T_1}\|^2), \textit{ for } r \neq r',$

- $\langle \nabla_{\mathbf{w}_{r,t}} L(\mathbf{W}_t^{T_1}), \mathbf{w}_{r,t}^0 \rangle = -\frac{1}{\sqrt{m}}\Theta\Big((\langle \mathbf{w}_{r,t}^{T_1}, \boldsymbol{\mu}_j + \overline{\boldsymbol{\xi}}\rangle - \sqrt{m}\|\mathbf{w}_{r,t}^{T_1}\|^4)\langle \mathbf{w}_{r,t}^{T_1}, \mathbf{w}_{r,t}^0 \rangle + \|\mathbf{w}_{r,t}^{T_1}\|^2\langle \mathbf{w}_{r,t}^0, \boldsymbol{\mu}_j + \overline{\boldsymbol{\xi}}\rangle\Big), \textit{ where we denote } \overline{\boldsymbol{\xi}} = \frac{1}{n}\sum_{i=1}^n \boldsymbol{\xi}_i.$

- $\langle \mathbf{w}_{r,t}^{T_1}, \boldsymbol{\mu}_j \rangle / \langle \mathbf{w}_{r',t}^{T_1}, \boldsymbol{\xi}_i \rangle = \Theta(n \cdot \mathrm{SNR}^2),$

*for all $j, j' = \pm 1, r, r' \in [m], i, i' \in [n]$.*

*Proof of Lemma D.5.* We prove the results by induction. To this end, we first compute the scale of the gradients projected to the space of $\boldsymbol{\mu}_1, \boldsymbol{\mu}_{-1}$ and $\boldsymbol{\xi}_i$, for $i \in [n]$ under the initialization scale. For notation clarity, we omit the index $k$.

As long as $\|\mathbf{w}_{r,t}\|^2 = \Theta(\sigma_0^2 d)$ and suppose $\langle \mathbf{w}_{r',t}, \boldsymbol{\mu}_j \rangle = O(\langle \mathbf{w}_{r,t}, \boldsymbol{\mu}_j \rangle)$, $\langle \mathbf{w}_{r',t}, \boldsymbol{\xi}_i \rangle = O(\langle \mathbf{w}_{r,t}, \boldsymbol{\xi}_i \rangle)$, we can identify the dominant terms as follows.

**Signal.** First for $\boldsymbol{\mu}_j$, and for any $i \in [n]$, we compute

$$\frac{1}{2n}\sum_{i=1}^n \langle \nabla L_{1,i}^{(1)}(\mathbf{w}_{r,t}), \boldsymbol{\mu}_j \rangle$$

$$= \frac{1}{m}\Theta(\langle \mathbf{w}_{r,t}, \boldsymbol{\mu}_j \rangle^5 + \langle \mathbf{w}_{r,t}, \boldsymbol{\mu}_j \rangle^3 \sigma_0^2 d + \sigma_0^4 d^2 \langle \mathbf{w}_{r,t}, \boldsymbol{\mu}_j \rangle) - \frac{1}{\sqrt{m}}\Theta(\langle \mathbf{w}_{r,t}, \boldsymbol{\mu}_j \rangle^2)$$

$$+ \frac{1}{m}\Theta(\sigma_0^2 d \langle \mathbf{w}_{r,t}, \boldsymbol{\mu}_j \rangle^3 + \sigma_0^4 d^2 \langle \mathbf{w}_{r,t}, \boldsymbol{\mu}_j \rangle) - \frac{1}{\sqrt{m}}\Theta(\sigma_0^2 d)$$

$$= \frac{1}{m}O(\sigma_0^4 d^2 \langle \mathbf{w}_{r,t}, \boldsymbol{\mu}_j \rangle) - \frac{1}{\sqrt{m}}\Theta(\sigma_0^2 d)$$

where the second equality is by $\langle \mathbf{w}_{r,t}, \boldsymbol{\mu}_j \rangle^2 \leq \|\mathbf{w}_{r,t}\|^2 \|\boldsymbol{\mu}\|^2 = \Theta(\sigma_0^2 d)$. It is clear the dominant term is $-\frac{1}{\sqrt{m}}4\alpha_t\beta_t\|\mathbf{w}_{r,t}\|^2\|\boldsymbol{\mu}\|^2$. The second dominant term comes from $\Theta(\frac{1}{m}\|\mathbf{w}_{r,t}\|^4 \langle \mathbf{w}_{r,t}, \boldsymbol{\mu}_j \rangle)$.

Further, we have due to the orthogonality between signal and noise vectors,

$$\frac{1}{2n}\sum_{i=1}^n \langle \nabla L_{1,i}^{(2)}(\mathbf{w}_{r,t}), \boldsymbol{\mu}_j \rangle = \frac{1}{m}O\big(\langle \mathbf{w}_{r,t}, \boldsymbol{\xi}_i \rangle^4 \langle \mathbf{w}_{r,t}, \boldsymbol{\mu}_j \rangle + \langle \mathbf{w}_{r,t}, \boldsymbol{\xi}_i \rangle^2 \sigma_0^2 d \langle \mathbf{w}_{r,t}, \boldsymbol{\mu}_j \rangle + \sigma_0^4 d^2 \langle \mathbf{w}_{r,t}, \boldsymbol{\mu}_j \rangle$$

$$- \sqrt{m}\langle \mathbf{w}_{r,t}, \boldsymbol{\xi}_i \rangle \langle \mathbf{w}_{r,t}, \boldsymbol{\mu}_j \rangle\big)$$

In addition, we have

$$\frac{1}{2n}\sum_{i=1}^n \langle \nabla L_{2,i}^{(1)}(\mathbf{w}_{r,t}), \boldsymbol{\mu}_j \rangle = \frac{m-1}{m}O\big(\langle \mathbf{w}_{r,t}, \boldsymbol{\mu}_j \rangle^5 + \langle \mathbf{w}_{r,t}, \boldsymbol{\mu}_j \rangle^3 \sigma_0^2 d + \sigma_0^4 d^2 \langle \mathbf{w}_{r,t}, \boldsymbol{\mu}_j \rangle\big)$$

$$= \frac{m-1}{m}O\big(\sigma_0^4 d^2 \langle \mathbf{w}_{r,t}, \boldsymbol{\mu}_j \rangle\big)$$

Further,

$$\frac{1}{2n}\sum_{i=1}^n \langle \nabla L_{2,i}^{(2)}(\mathbf{w}_{r,t}), \boldsymbol{\mu}_j \rangle = \frac{m-1}{m}O(\langle \mathbf{w}_{r,t}, \boldsymbol{\xi}_i \rangle^4 \langle \mathbf{w}_{r,t}, \boldsymbol{\mu}_j \rangle + \langle \mathbf{w}_{r,t}, \boldsymbol{\xi}_i \rangle^2 \sigma_0^2 d \langle \mathbf{w}_{r,t}, \boldsymbol{\mu}_j \rangle + \sigma_0^4 d^2 \langle \mathbf{w}_{r,t}, \boldsymbol{\mu}_j \rangle)$$

Then according to the definition of $|\mathcal{S}_{\pm 1}|$ and $\boldsymbol{\mu}_{\pm 1}$, we can simplify the gradient into the dominant terms as

$$\langle \nabla L(\mathbf{W}_t), \boldsymbol{\mu}_j \rangle = -\frac{1}{\sqrt{m}}\Theta(\|\mathbf{w}_{r,t}\|^2 + \langle \mathbf{w}_{r,t}, \boldsymbol{\xi}_i \rangle \langle \mathbf{w}_{r,t}, \boldsymbol{\mu}_j \rangle)$$

$$+ O\big(\sigma_0^4 d^2 \langle \mathbf{w}_{r,t}, \boldsymbol{\mu}_j \rangle + \langle \mathbf{w}_{r,t}, \boldsymbol{\xi}_i \rangle^4 \langle \mathbf{w}_{r,t}, \boldsymbol{\mu}_j \rangle + \langle \mathbf{w}_{r,t}, \boldsymbol{\xi}_i \rangle^2 \sigma_0^2 d \langle \mathbf{w}_{r,t}, \boldsymbol{\mu}_j \rangle\big)$$

**Noise.** Similarly, we can also show for the noise learning

$$\frac{1}{2n}\sum_{i'=1}^{n}\langle\nabla L_{1,i'}^{(1)}(\mathbf{w}_{r,t}),\boldsymbol{\xi}_i\rangle = \frac{1}{m}O\big(\langle\mathbf{w}_{r,t},\boldsymbol{\mu}_j\rangle^4\langle\mathbf{w}_{r,t},\boldsymbol{\xi}_i\rangle + \langle\mathbf{w}_{r,t},\boldsymbol{\mu}_j\rangle^2\sigma_0^2 d\langle\mathbf{w}_{r,t},\boldsymbol{\xi}_i\rangle + \sigma_0^4 d^2\langle\mathbf{w}_{r,t},\boldsymbol{\xi}_i\rangle\big)$$

$$-\frac{1}{\sqrt{m}}\Theta\big(\langle\mathbf{w}_{r,t},\boldsymbol{\xi}_i\rangle\langle\mathbf{w}_{r,t},\boldsymbol{\mu}_j\rangle\big)$$

$$=\frac{1}{m}O(\sigma_0^4 d^2\langle\mathbf{w}_{r,t},\boldsymbol{\xi}_i\rangle) - \frac{1}{\sqrt{m}}\Theta(\langle\mathbf{w}_{r,t},\boldsymbol{\xi}_i\rangle\langle\mathbf{w}_{r,t},\boldsymbol{\mu}_j\rangle)$$

where the dominating term is $-4\sqrt{m}\alpha_t\beta_t\langle\mathbf{w}_{r,t},\boldsymbol{\mu}_j\rangle\langle\mathbf{w}_{r,t},\boldsymbol{\xi}_i\rangle$.

In addition,

$$\frac{1}{2n}\sum_{i'=1}^{n}\langle\nabla L_{1,i'}^{(2)}(\mathbf{w}_{r,t}),\boldsymbol{\xi}_i\rangle$$

$$=\frac{1}{m}O\big(\langle\mathbf{w}_{r,t},\boldsymbol{\xi}_i\rangle^5 + \langle\mathbf{w}_{r,t},\boldsymbol{\xi}_i\rangle^3\sigma_0^2 d + \sigma_0^4 d^2\langle\mathbf{w}_{r,t},\boldsymbol{\xi}_i\rangle\big) - \frac{1}{\sqrt{m}}O(\langle\mathbf{w}_{r,t},\boldsymbol{\xi}_i\rangle^2)$$

$$+\frac{1}{m}\big(O(\langle\mathbf{w}_{r,t},\boldsymbol{\xi}_i\rangle^3\sigma_0^2 d + \sigma_0^4 d^2\langle\mathbf{w}_{r,t},\boldsymbol{\xi}_i\rangle) + \Theta(\sigma_0^2 d)\big)\big(\Theta(\sigma_\xi^2 dn^{-1}) - \sqrt{m}\widetilde{O}(\sigma_\xi^2\sqrt{d})\big)$$

$$=\frac{1}{m}O\big(\langle\mathbf{w}_{r,t},\boldsymbol{\xi}_i\rangle^5 + \langle\mathbf{w}_{r,t},\boldsymbol{\xi}_i\rangle^3\sigma_0^2 d + \sigma_0^4 d^2\langle\mathbf{w}_{r,t},\boldsymbol{\xi}_i\rangle\big) - \frac{1}{\sqrt{m}}O(\langle\mathbf{w}_{r,t},\boldsymbol{\xi}_i\rangle^2)$$

$$+\frac{1}{m}O(\langle\mathbf{w}_{r,t},\boldsymbol{\xi}_i\rangle^3\sigma_0^2\sigma_\xi^2 d^2 n^{-1} + \langle\mathbf{w}_{r,t},\boldsymbol{\xi}_i\rangle\sigma_0^4\sigma_\xi^2 d^3 n^{-1}) - \frac{1}{\sqrt{m}}\Theta(\sigma_0^2\sigma_\xi^2 d^2 n^{-1})$$

where we use Lemma B.2 in the first equality and the second equality is by $d = \widetilde{\Omega}(n^2)$.

Further we can show

$$\frac{1}{2n}\sum_{i'=1}^{n}\langle\nabla L_{2,i}^{(1)}(\mathbf{w}_{r,t}),\boldsymbol{\xi}_i\rangle$$

$$=\frac{m-1}{m}O\big(\langle\mathbf{w}_{r,t},\boldsymbol{\mu}_j\rangle^4\langle\mathbf{w}_{r,t},\boldsymbol{\xi}_i\rangle + \langle\mathbf{w}_{r,t},\boldsymbol{\mu}_j\rangle^2\sigma_0^2 d\langle\mathbf{w}_{r,t},\boldsymbol{\xi}_i\rangle + \sigma_0^4 d\langle\mathbf{w}_{r,t},\boldsymbol{\xi}_i\rangle\big)$$

$$=\frac{m-1}{m}O\big(\sigma_0^4 d\langle\mathbf{w}_{r,t},\boldsymbol{\xi}_i\rangle\big)$$

Lastly,

$$\frac{1}{2n}\sum_{i'=1}^{n}\langle\nabla L_{2,i}^{(2)}(\mathbf{w}_{r,t}),\boldsymbol{\xi}_i\rangle = \frac{m-1}{m}O\big(\langle\mathbf{w}_{r,t},\boldsymbol{\xi}_i\rangle^5 + \langle\mathbf{w}_{r,t},\boldsymbol{\xi}_i\rangle^3\sigma_0^2 d + \sigma_0^4 d^2\langle\mathbf{w}_{r,t},\boldsymbol{\xi}_i\rangle\big)$$

$$+\frac{m-1}{m}O(\langle\mathbf{w}_{r,t},\boldsymbol{\xi}_i\rangle^3\sigma_0^2\sigma_\xi^2 d^2 n^{-1} + \langle\mathbf{w}_{r,t},\boldsymbol{\xi}_i\rangle\sigma_0^4\sigma_\xi^2 d^3 n^{-1})$$

This suggests we can simplify the gradient along noise direction as

$$\langle\nabla L(\mathbf{W}_t),\boldsymbol{\xi}_i\rangle = -\frac{1}{\sqrt{m}}\Theta(\sigma_0^2\sigma_\xi^2 d^2 n^{-1} + \langle\mathbf{w}_{r,t},\boldsymbol{\xi}_i\rangle^2 + \langle\mathbf{w}_{r,t},\boldsymbol{\xi}_i\rangle\langle\mathbf{w}_{r,t},\boldsymbol{\mu}_j\rangle)$$

$$+O\big(\sigma_0^4 d^2\langle\mathbf{w}_{r,t},\boldsymbol{\xi}_i\rangle + \langle\mathbf{w}_{r,t},\boldsymbol{\xi}_i\rangle^5 + \langle\mathbf{w}_{r,t},\boldsymbol{\xi}_i\rangle^3\sigma_0^2 d\big)$$

$$+O\big(\langle\mathbf{w}_{r,t},\boldsymbol{\xi}_i\rangle^3\sigma_0^2\sigma_\xi^2 d^2 n^{-1} + \langle\mathbf{w}_{r,t},\boldsymbol{\xi}_i\rangle\sigma_0^4\sigma_\xi^2 d^3 n^{-1}\big)$$

In summary, as long as $\|\mathbf{w}_{r,t}\|^2 = \Theta(\sigma_0^2 d)$ and suppose $\langle\mathbf{w}_{r',t},\boldsymbol{\mu}_j\rangle = O(\langle\mathbf{w}_{r,t},\boldsymbol{\mu}_j\rangle)$, $\langle\mathbf{w}_{r',t},\boldsymbol{\xi}_i\rangle = O(\langle\mathbf{w}_{r,t},\boldsymbol{\xi}_i\rangle)$, we can simplify the gradient as

$$\langle\nabla_{\mathbf{w}_{r,t}}L(\mathbf{W}_t),\boldsymbol{\mu}_j\rangle = -\frac{1}{\sqrt{m}}\Theta(\sigma_0^2 d) - \frac{1}{\sqrt{m}}O(\langle\mathbf{w}_{r,t},\boldsymbol{\xi}_i\rangle\langle\mathbf{w}_{r,t},\boldsymbol{\mu}_j\rangle)$$

$$+O\big(\sigma_0^4 d^2\langle\mathbf{w}_{r,t},\boldsymbol{\mu}_j\rangle + \langle\mathbf{w}_{r,t},\boldsymbol{\xi}_i\rangle^4\langle\mathbf{w}_{r,t},\boldsymbol{\mu}_j\rangle + \langle\mathbf{w}_{r,t},\boldsymbol{\xi}_i\rangle^2\sigma_0^2 d\langle\mathbf{w}_{r,t},\boldsymbol{\mu}_j\rangle\big)$$

$$(24)$$

$$\langle \nabla_{\mathbf{w}_{r,t}} L(\mathbf{W}_t), \boldsymbol{\xi}_i \rangle = -\frac{1}{\sqrt{m}} \Theta(\sigma_0^2 \sigma_\xi^2 d^2 n^{-1}) - \frac{1}{\sqrt{m}} O(\langle \mathbf{w}_{r,t}, \boldsymbol{\xi}_i \rangle^2 + \langle \mathbf{w}_{r,t}, \boldsymbol{\xi}_i \rangle \langle \mathbf{w}_{r,t}, \boldsymbol{\mu}_j \rangle)$$
$$+ O(\sigma_0^4 d^2 \langle \mathbf{w}_{r,t}, \boldsymbol{\xi}_i \rangle + \langle \mathbf{w}_{r,t}, \boldsymbol{\xi}_i \rangle^5 + \langle \mathbf{w}_{r,t}, \boldsymbol{\xi}_i \rangle^3 \sigma_0^2 d)$$
$$+ O(\langle \mathbf{w}_{r,t}, \boldsymbol{\xi}_i \rangle^3 \sigma_0^2 \sigma_\xi^2 d^2 n^{-1} + \langle \mathbf{w}_{r,t}, \boldsymbol{\xi}_i \rangle \sigma_0^4 \sigma_\xi^2 d^3 n^{-1}) \tag{25}$$

In the initial phase where $\|\mathbf{w}_{r,t}^k\|^2 = \Theta(\sigma_0^2 d), |\langle \mathbf{w}_{r,t}^k, \boldsymbol{\mu}_j \rangle| = \widetilde{O}(\sigma_0 \|\boldsymbol{\mu}\|)$ and $|\langle \mathbf{w}_{r,t}^k, \boldsymbol{\xi}_i \rangle| = \widetilde{O}(\sigma_0 \sigma_\xi \sqrt{d})$ (by Lemma D.1), we can show (24) reduces to

$$\langle \nabla L(\mathbf{W}_t), \boldsymbol{\mu}_j \rangle$$
$$= -\frac{1}{\sqrt{m}} \Theta(\sigma_0^2 d) + O(\sigma_0^4 d^2 \langle \mathbf{w}_{r,t}, \boldsymbol{\mu}_j \rangle + \langle \mathbf{w}_{r,t}, \boldsymbol{\xi}_i \rangle^4 \langle \mathbf{w}_{r,t}, \boldsymbol{\mu}_j \rangle + \langle \mathbf{w}_{r,t}, \boldsymbol{\xi}_i \rangle^2 \sigma_0^2 d \langle \mathbf{w}_{r,t}, \boldsymbol{\mu}_j \rangle)$$
$$= -\frac{1}{\sqrt{m}} \Theta(\sigma_0^2 d)$$

where we use the condition that $\text{SNR}^{-1} = \widetilde{O}(d^{1/4})$, i.e., $\sigma_\xi = \widetilde{O}(d^{-1/4}) = o(1)$ in the first equality. The second equality is by condition $\sigma_0 \leq \widetilde{O}(m^{-1/6} d^{-1/3})$ and $\sigma_\xi^{-1} = \widetilde{\Omega}(d^{1/4})$. Further, we can show (25) reduces to

$$\langle \nabla L(\mathbf{W}_t), \boldsymbol{\xi}_i \rangle = -\frac{1}{\sqrt{m}} \Theta(\sigma_0^2 \sigma_\xi^2 d^2 n^{-1}) + O(\sigma_0^4 d^2 \langle \mathbf{w}_{r,t}, \boldsymbol{\xi}_i \rangle + \langle \mathbf{w}_{r,t}, \boldsymbol{\xi}_i \rangle^5 + \langle \mathbf{w}_{r,t}, \boldsymbol{\xi}_i \rangle^3 \sigma_0^2 d)$$
$$+ O(\langle \mathbf{w}_{r,t}, \boldsymbol{\xi}_i \rangle^3 \sigma_0^2 \sigma_\xi^2 d^2 n^{-1} + \langle \mathbf{w}_{r,t}, \boldsymbol{\xi}_i \rangle \sigma_0^4 \sigma_\xi^2 d^3 n^{-1})$$
$$= -\frac{1}{\sqrt{m}} \Theta(\sigma_0^2 \sigma_\xi^2 d^2 n^{-1}) + O(\langle \mathbf{w}_{r,t}, \boldsymbol{\xi}_i \rangle^3 \sigma_0^2 \sigma_\xi^2 d^2 n^{-1} + \langle \mathbf{w}_{r,t}, \boldsymbol{\xi}_i \rangle \sigma_0^4 \sigma_\xi^2 d^3 n^{-1})$$
$$= -\frac{1}{\sqrt{m}} \Theta(\sigma_0^2 \sigma_\xi^2 d^2 n^{-1})$$

where the first equality is by $d = \widetilde{\Omega}(n)$ and $d \geq \widetilde{\Omega}(n^{2/3} \sigma_\xi^{-2/3})$. The second equality is by $\sigma_0^3 \leq \widetilde{O}(m^{-1/2} d^{-1/2} \sigma_\xi n^{-1})$. The third equality is by $\sigma_0^3 \leq \widetilde{O}(m^{-1/2} d^{-3/2} \sigma_\xi^{-1})$.

In summary, we can show as long as $\|\mathbf{w}_{r,t}^k\|^2 = \Theta(\sigma_0^2 d), |\langle \mathbf{w}_{r,t}^k, \boldsymbol{\mu}_j \rangle| = \widetilde{O}(\sigma_0 \|\boldsymbol{\mu}\|)$ and $|\langle \mathbf{w}_{r,t}^k, \boldsymbol{\xi}_i \rangle| = \widetilde{O}(\sigma_0 \sigma_\xi \sqrt{d})$,

$$\langle \mathbf{w}_{r,t}^{k+1}, \boldsymbol{\mu}_j \rangle = \langle \mathbf{w}_{r,t}^k, \boldsymbol{\mu}_j \rangle + \frac{\eta \alpha_t \beta_t}{\sqrt{m}} \Theta(\sigma_0^2 d) \tag{26}$$

$$\langle \mathbf{w}_{r,t}^{k+1}, \boldsymbol{\xi}_i \rangle = \langle \mathbf{w}_{r,t}^k, \boldsymbol{\xi}_i \rangle + \frac{\eta \alpha_t \beta_t}{n \sqrt{m}} \Theta(\sigma_0^2 d) \|\boldsymbol{\xi}_i\|^2 \tag{27}$$

In addition, we similarly show that as long as $\|\mathbf{w}_{r,t}^k\|^2 = \Theta(\sigma_0^2 d), |\langle \mathbf{w}_{r,t}^k, \boldsymbol{\mu}_j \rangle|, |\langle \mathbf{w}_{r,t}^k, \boldsymbol{\xi}_i \rangle| = o(1)$,

$$\langle \mathbf{w}_{r,t}^{k+1}, \mathbf{w}_{r,t}^0 \rangle$$
$$= \langle \mathbf{w}_{r,t}^k, \mathbf{w}_{r,t}^0 \rangle + \eta O\left( (\sigma_0^4 d^2 + \langle \mathbf{w}_{r,t}^k, \boldsymbol{\mu}_j + \boldsymbol{\xi}_i \rangle) \langle \mathbf{w}_{r,t}^k, \mathbf{w}_{r,t}^0 \rangle + \sigma_0^2 d \langle \mathbf{w}_{r,t}^0, \boldsymbol{\mu}_j + \boldsymbol{\xi}_i \rangle \right), \tag{28}$$

Next, let $T_\mu = \Theta(\frac{\sqrt{m \log(16m/\delta)}}{\sigma_0 d \|\boldsymbol{\mu}\| \eta \alpha_t \beta_t})$ and $T_\xi = \Theta(\frac{n \sqrt{m \log(16mn/\delta)}}{\sigma_0 \sigma_\xi d^{3/2} \eta \alpha_t \beta_t})$ and $T_1 = \max\{T_\mu, T_\xi\}$. We prove the results hold for all $0 \leq k \leq T_1$ via induction. We partition the proof into two stages, namely when $0 \leq k \leq \min\{T_\mu, T_\xi\}$ and when $\min\{T_\mu, T_\xi\} \leq k \leq T_1$.

(1) We first show for all $0 \leq k \leq \min\{T_\mu, T_\xi\}$ that $\|\mathbf{w}_{r,t}^k\|^2 = \Theta(\sigma_0^2 d), \langle \mathbf{w}_{r,t}^k, \mathbf{w}_{r,t}^0 \rangle = \Theta(\sigma_0^2 d)$, $|\langle \mathbf{w}_{r,t}^k, \boldsymbol{\mu}_j \rangle| = \widetilde{O}(\sigma_0 \|\boldsymbol{\mu}\|)$ and $|\langle \mathbf{w}_{r,t}^k, \boldsymbol{\xi}_i \rangle| = \widetilde{O}(\sigma_0 \sigma_\xi \sqrt{d})$ hold and thus (26), (27), (28) are directly satisfied. We prove the claims by induction as follows.

It is clear that at $k = 0$, we have from Lemma D.1 that $\|\mathbf{w}_{r,t}^0\|^2 = \Theta(\sigma_0^2 d), \langle \mathbf{w}_{r,t}^k, \mathbf{w}_{r,t}^0 \rangle = \|\mathbf{w}_{r,t}^0\|^2 = \Theta(\sigma_0^2 d)$ and

$$|\langle \mathbf{w}_{r,t}^0, \boldsymbol{\mu}_j \rangle| \leq \sqrt{2 \log(16m/\delta)} \sigma_0 \|\boldsymbol{\mu}\| = \widetilde{O}(\sigma_0 \|\boldsymbol{\mu}\|)$$

$$|\langle \mathbf{w}_{r,t}^0, \boldsymbol{\xi}_i \rangle| \leq 2\sqrt{\log(16mn/\delta)}\sigma_0\sigma_\xi\sqrt{d} = \widetilde{O}(\sigma_0\sigma_\xi\sqrt{d})$$

Suppose there exists an iteration $\widetilde{T} \leq \min\{T_\mu, T_\xi\}$ such that $\|\mathbf{w}_{r,t}^k\|^2 = \Theta(\sigma_0^2 d)$, $\langle \mathbf{w}_{r,t}^k, \mathbf{w}_{r,t}^0 \rangle = \Theta(\sigma_0^2 d)$, $|\langle \mathbf{w}_{r,t}^0, \boldsymbol{\mu}_j \rangle| = \widetilde{O}(\sigma_0\|\boldsymbol{\mu}\|)$ and $|\langle \mathbf{w}_{r,t}^k, \boldsymbol{\xi}_i \rangle| = \widetilde{O}(\sigma_0\sigma_\xi\sqrt{d})$ for all $0 \leq k \leq \widetilde{T} - 1$. Then we have from (26) that

$$\langle \mathbf{w}_{r,t}^{\widetilde{T}}, \boldsymbol{\mu}_j \rangle = \langle \mathbf{w}_{r,t}^{\widetilde{T}-1}, \boldsymbol{\mu}_j \rangle + \frac{\eta\alpha_t\beta_t}{\sqrt{m}}\Theta(\sigma_0^2 d)\|\boldsymbol{\mu}\|^2 = \langle \mathbf{w}_{r,t}^0, \boldsymbol{\mu}_j \rangle + \frac{\eta\alpha_t\beta_t}{\sqrt{m}}\Theta(\sigma_0^2 d)\|\boldsymbol{\mu}\|^2\widetilde{T}$$

$$\leq \langle \mathbf{w}_{r,t}^0, \boldsymbol{\mu}_j \rangle + \frac{\eta\alpha_t\beta_t}{\sqrt{m}}\Theta(\sigma_0^2 d)\|\boldsymbol{\mu}\|^2 T_\mu$$

$$= \langle \mathbf{w}_{r,t}^0, \boldsymbol{\mu}_j \rangle + \widetilde{O}(\sigma_0\|\boldsymbol{\mu}\|)$$

$$= \widetilde{O}(\sigma_0\|\boldsymbol{\mu}\|) \tag{29}$$

where we use the Lemma B.1 that $|\mathcal{S}_j| = \Theta(n)$. In addition, we have from (27) that

$$\langle \mathbf{w}_{r,t}^{\widetilde{T}}, \boldsymbol{\xi}_i \rangle = \langle \mathbf{w}_{r,t}^{\widetilde{T}-1}, \boldsymbol{\xi}_i \rangle + \frac{\eta\alpha_t\beta_t}{n\sqrt{m}}\Theta(\sigma_0^2\sigma_\xi^2 d^2) \leq \langle \mathbf{w}_{r,t}^0, \boldsymbol{\xi}_i \rangle + \frac{\eta\alpha_t\beta_t}{n\sqrt{m}}\Theta(\sigma_0^2\sigma_\xi^2 d^2)T_\xi$$

$$= \widetilde{O}(\sigma_0\sigma_\xi\sqrt{d}) \tag{30}$$

where we use Lemma D.1 that $\|\boldsymbol{\xi}_i\|^2 = \Theta(\sigma_\xi^2 d)$ for all $i \in [n]$.

Finally, we deduce from (28) that

$$\langle \mathbf{w}_{r,t}^{\widetilde{T}}, \mathbf{w}_{r,t}^0 \rangle = \langle \mathbf{w}_{r,t}^{\widetilde{T}-1}, \mathbf{w}_{r,t}^0 \rangle + \eta O\Big( \big(\sigma_0^4 d^2 + \langle \mathbf{w}_{r,t}^k, \boldsymbol{\mu}_j + \boldsymbol{\xi}_i \rangle\big)\langle \mathbf{w}_{r,t}^k, \mathbf{w}_{r,t}^0 \rangle + \sigma_0^2 d\langle \mathbf{w}_{r,t}^0, \boldsymbol{\mu}_j + \boldsymbol{\xi}_i \rangle\Big)$$

$$= \langle \mathbf{w}_{r,t}^{\widetilde{T}-1}, \mathbf{w}_{r,t}^0 \rangle + \eta O\Big( \sigma_0^3 d^{3/2}(\|\boldsymbol{\mu}\| + \sigma_\xi\sqrt{d}) + \sigma_0^6 d^3\Big)$$

$$= \Theta(\sigma_0^2 d) + \eta O\Big( \sigma_0^3 d^{3/2}(\|\boldsymbol{\mu}\| + \sigma_\xi\sqrt{d}) + \sigma_0^6 d^3\Big)\widetilde{T} \tag{31}$$

where we use Cauchy-Schwarz inequality in the first inequality. When $\widetilde{T} = T_\mu$,

$$\eta O\Big( \sigma_0^3 d^{3/2}(\|\boldsymbol{\mu}\| + \sigma_\xi\sqrt{d}) + \sigma_0^6 d^3\Big)\widetilde{T} = \widetilde{O}\Big( (\|\boldsymbol{\mu}\| + \sigma_\xi\sqrt{d})\sigma_0^2\sqrt{d} + \sigma_0^5 d^2\Big) \leq \Theta(\sigma_0^2 d) \tag{32}$$

where the last inequality is by the condition on $\sigma_\xi^{-1} = \widetilde{\Omega}(d^{1/4}) \gg 1$ and $\sigma_0 \leq \widetilde{O}(d^{-1/3})$. When $\widetilde{T} = T_\xi$,

$$\eta O\Big( \sigma_0^3 d^{3/2}(\|\boldsymbol{\mu}\| + \sigma_\xi\sqrt{d}) + \sigma_0^6 d^3\Big)\widetilde{T} = \widetilde{O}\Big( n\sigma_\xi^{-1}\sigma_0^2(\|\boldsymbol{\mu}\| + \sigma_\xi\sqrt{d}) + n\sigma_0^5\sigma_\xi^{-1}d^{3/2}\Big) \leq \Theta(\sigma_0^2 d) \tag{33}$$

where the last inequality is by the condition on $d$ that $d = \widetilde{\Omega}(n\sigma_\xi^{-1}\|\boldsymbol{\mu}\|)$ and $d = \widetilde{\Omega}(n^2)$ and $\sigma_0 \leq \widetilde{O}(\sigma_\xi^{1/3}n^{-1/3}d^{-1/6})$. Hence we have proved the induction on $\langle \mathbf{w}_{r,t}^k, \mathbf{w}_{r,t}^0 \rangle$ and in fact proved a stronger result that $\langle \mathbf{w}_{r,t}^k, \mathbf{w}_{r,t}^0 \rangle = \Theta(\sigma_0^2 d)$ for all $k \leq \max\{T_\mu, T_\xi\}$ as long as $\|\mathbf{w}_{r,t}^k\|^2 = \Theta(\sigma_0^2 d)$, $|\langle \mathbf{w}_{r,t}^k, \boldsymbol{\mu}_j \rangle|, |\langle \mathbf{w}_{r,t}^k, \boldsymbol{\xi}_i \rangle| = o(1)$.

Next, we let $\mathbf{P}_{\boldsymbol{\xi}} = \frac{\boldsymbol{\xi}\boldsymbol{\xi}^\top}{\|\boldsymbol{\xi}\|^2}$ be the projection matrix onto the direction of $\boldsymbol{\xi}$ and we express $\mathbf{w}_{r,t}^{\widetilde{T}} = \mathbf{P}_{\boldsymbol{\mu}_1}\mathbf{w}_{r,t}^{\widetilde{T}} + \mathbf{P}_{\boldsymbol{\mu}_{-1}}\mathbf{w}_{r,t}^{\widetilde{T}} + \sum_{i=1}^n \mathbf{P}_{\boldsymbol{\xi}_i}\mathbf{w}_{r,t}^{\widetilde{T}} + \big(\mathbf{I} - \mathbf{P}_{\boldsymbol{\mu}_1} - \mathbf{P}_{\boldsymbol{\mu}_{-1}} - \sum_{i=1}^n \mathbf{P}_{\boldsymbol{\xi}_i}\big)\mathbf{w}_{r,t}^{\widetilde{T}}$ and due to the orthogonality of the decomposition, we have

$$\|\mathbf{w}_{r,t}^{\widetilde{T}}\|^2 = \frac{\langle \mathbf{w}_{r,t}^{\widetilde{T}}, \boldsymbol{\mu}_1 \rangle^2}{\|\boldsymbol{\mu}\|^2} + \frac{\langle \mathbf{w}_{r,t}^{\widetilde{T}}, \boldsymbol{\mu}_{-1} \rangle^2}{\|\boldsymbol{\mu}\|^2} + \left\|\sum_{i=1}^n \frac{\langle \mathbf{w}_{r,t}^{\widetilde{T}}, \boldsymbol{\xi}_i \rangle}{\|\boldsymbol{\xi}\|^2}\right\|^2 + \left\|\big(\mathbf{I} - \mathbf{P}_{\boldsymbol{\mu}_1} - \mathbf{P}_{\boldsymbol{\mu}_{-1}} - \sum_{i=1}^n \mathbf{P}_{\boldsymbol{\xi}_i}\big)\mathbf{w}_{r,t}^{\widetilde{T}}\right\|^2$$

$$= \widetilde{O}(\sigma_0^2) + \widetilde{O}(n\sigma_0^2) + \left\|\frac{\langle \mathbf{w}_{r,t}^{\widetilde{T}}, \mathbf{w}_{r,t}^0 \rangle}{\|\mathbf{w}_{r,t}^0\|^2}\mathbf{w}_{r,t}^0\right\|^2$$

$$= \Theta(\sigma_0^2 d)$$

where we use the induction results that $|\langle \mathbf{w}_{r,t}^{\widetilde{T}}, \boldsymbol{\mu}_j \rangle| = \widetilde{O}(\sigma_0 \|\boldsymbol{\mu}\|)$ and $|\langle \mathbf{w}_{r,t}^{\widetilde{T}}, \boldsymbol{\xi}_i \rangle| = \widetilde{O}(\sigma_0 \sigma_\xi \sqrt{d})$, and the $\left\| \left( \mathbf{I} - \mathbf{P}_{\boldsymbol{\mu}_1} - \mathbf{P}_{\boldsymbol{\mu}_{-1}} - \sum_{i=1}^n \mathbf{P}_{\boldsymbol{\xi}_i} \right) \mathbf{w}_{r,t}^{\widetilde{T}} \right\|^2$ is dominated by its projection to $\mathbf{w}_{r,t}^0$.

This completes the induction that for all $k \leq \min\{T_\mu, T_\xi\}$, we have $\|\mathbf{w}_{r,t}^k\|^2 = \Theta(\sigma_0^2 d)$, $\langle \mathbf{w}_{r,t}^k, \mathbf{w}_{r,t}^0 \rangle = \Theta(\sigma_0^2 d)$, $|\langle \mathbf{w}_{r,t}^k, \boldsymbol{\mu}_j \rangle| = \widetilde{O}(\sigma_0 \|\boldsymbol{\mu}\|)$ and $|\langle \mathbf{w}_{r,t}^k, \boldsymbol{\xi}_i \rangle| = \widetilde{O}(\sigma_0 \sigma_\xi \sqrt{d})$.

(2) Next, we examine the iteration $\min\{T_\mu, T_\xi\} \leq k \leq \max\{T_\mu, T_\xi\} = T_1$. The magnitude comparison between $T_\mu$ and $T_\xi$ depends on the condition on $n \cdot \mathrm{SNR}^2$. In particular, we can verify that $T_\mu/T_\xi = \widetilde{\Theta}(n^{-1/2}\sqrt{n^{-1}\mathrm{SNR}^{-2}}) = \widetilde{\Theta}(n^{-1}\mathrm{SNR}^{-1})$.

- When $T_\mu \leq T_\xi$, i.e., $n \cdot \mathrm{SNR}^2 = \widetilde{\Omega}(1)$, we use induction to show for all $\min\{T_\mu, T_\xi\} \leq k \leq T_1$, $\|\mathbf{w}_{r,t}^k\|^2 = \Theta(\sigma_0^2 d)$, $\langle \mathbf{w}_{r,t}^k, \mathbf{w}_{r,t}^0 \rangle = \Theta(\sigma_0^2 d)$, $|\langle \mathbf{w}_{r,t}^k, \boldsymbol{\mu}_j \rangle| = \widetilde{O}(\sigma_0 \|\boldsymbol{\mu}\| n \mathrm{SNR})$, $|\langle \mathbf{w}_{r,t}^k, \boldsymbol{\xi}_i \rangle| = \widetilde{O}(\sigma_0 \sigma_\xi \sqrt{d})$. It can be shown that under the condition $\sigma_0 \leq \widetilde{O}(n^{-1}\sigma_\xi d^{1/2})$, we have $|\langle \mathbf{w}_{r,t}^k, \boldsymbol{\xi}_i \rangle| = o(1)$, which suggests $\langle \mathbf{w}_{r,t}^k, \mathbf{w}_{r,t}^0 \rangle = \Theta(\sigma_0^2 d)$. Suppose there exists an iteration $T_\mu < \widetilde{T}_\xi \leq T_\xi$ such that the results hold for all $T_\mu \leq k \leq \widetilde{T}_\xi - 1$. Then we can derive the dominant terms in (24)

$$
\begin{aligned}
\langle \nabla L(\mathbf{W}_t), \boldsymbol{\mu}_j \rangle &= -\frac{1}{\sqrt{m}}\Theta(\sigma_0^2 d) \\
&\quad + O\big(\sigma_0^4 d^2 \langle \mathbf{w}_{r,t}, \boldsymbol{\mu}_j \rangle + \langle \mathbf{w}_{r,t}, \boldsymbol{\xi}_i \rangle^4 \langle \mathbf{w}_{r,t}, \boldsymbol{\mu}_j \rangle + \langle \mathbf{w}_{r,t}, \boldsymbol{\xi}_i \rangle^2 \sigma_0^2 d \langle \mathbf{w}_{r,t}, \boldsymbol{\mu}_j \rangle \big) \\
&= -\frac{1}{\sqrt{m}}\Theta(\sigma_0^2 d)
\end{aligned}
$$

where the first equality is by $d = \widetilde{\Omega}(n)$ and the second equality is by $\sigma_0^3 \leq \widetilde{O}(m^{-1/2}n^{-1}d^{-1/2}\sigma_\xi)$. This suggests that we can still leverage (26) to bound

$$
\begin{aligned}
\langle \mathbf{w}_{r,t}^{\widetilde{T}_\xi}, \boldsymbol{\mu}_j \rangle = \langle \mathbf{w}_{r,t}^{\widetilde{T}_\xi - 1}, \boldsymbol{\mu}_j \rangle + \frac{\eta \alpha_t \beta_t}{\sqrt{m}}\Theta(\sigma_0^2 d)\|\boldsymbol{\mu}\|^2 &\leq \langle \mathbf{w}_{r,t}^0, \boldsymbol{\mu}_j \rangle + \frac{\eta \alpha_t \beta_t}{\sqrt{m}}\Theta(\sigma_0^2 d)\|\boldsymbol{\mu}\|^2 T_\xi \\
&= \langle \mathbf{w}_{r,t}^0, \boldsymbol{\mu}_j \rangle + \widetilde{O}(\sigma_0 \|\boldsymbol{\mu}\| n \cdot \mathrm{SNR}) \\
&= \widetilde{O}(\sigma_0 \|\boldsymbol{\mu}\| n \cdot \mathrm{SNR})
\end{aligned}
$$

The bound on $|\langle \mathbf{w}_{r,t}^{\widetilde{T}_\xi}, \boldsymbol{\xi}_i \rangle|$ is the same as (30). Then by the same arguments in (31), and (32), (33), we can show $\langle \mathbf{w}_{r,t}^{\widetilde{T}_\xi}, \mathbf{w}_{r,t}^0 \rangle = \Theta(\sigma_0^2 d)$. Thus, we can compute

$$
\|\mathbf{w}_{r,t}^{\widetilde{T}_\xi}\|^2 = \widetilde{O}(\sigma_0^2 n^2 \cdot \mathrm{SNR}^2) + \widetilde{O}(n\sigma_0^2) + \Theta(\sigma_0^2 d) = \Theta(\sigma_0^2 d)
$$

where the last equality is by the condition on $d$ that $d = \widetilde{\Omega}(n\|\boldsymbol{\mu}\|\sigma_\xi^{-1})$. This verifies the induction on $\|\mathbf{w}_{r,t}^k\|^2 = \Theta(\sigma_0^2 d)$.

- When $T_\xi < T_\mu$, i.e., $n^{-1} \cdot \mathrm{SNR}^{-2} = \widetilde{\Omega}(1)$, we use induction to show for all $\min\{T_\mu, T_\xi\} \leq k \leq T_1$, $\|\mathbf{w}_{r,t}^k\|^2 = \Theta(\sigma_0^2 d)$, $|\langle \mathbf{w}_{r,t}^k, \boldsymbol{\mu}_j \rangle| = \widetilde{O}(\sigma_0 \|\boldsymbol{\mu}\|)$, $|\langle \mathbf{w}_{r,t}^k, \boldsymbol{\xi}_i \rangle| = \widetilde{O}(\sigma_0 \sigma_\xi \sqrt{d} n^{-1} \mathrm{SNR}^{-1})$. Under the condition that $\sigma_0 \leq \widetilde{O}(\sigma_\xi^{-1} d^{-3/4} n) \leq \widetilde{O}(\sigma_\xi^{-2} d^{-1} n)$, we have $|\langle \mathbf{w}_{r,t}^k, \boldsymbol{\xi}_i \rangle| = o(1)$, which suggests $\langle \mathbf{w}_{r,t}^k, \mathbf{w}_{r,t}^0 \rangle = \Theta(\sigma_0^2 d)$. Suppose there exists an iteration $T_\xi < \widetilde{T}_\mu \leq T_\mu$ such that the results hold for all $T_\xi \leq k \leq \widetilde{T}_\mu - 1$. Thus we can derive the dominant terms in (25) as

$$
\begin{aligned}
\langle \nabla L(\mathbf{W}_t), \boldsymbol{\xi}_i \rangle &= -\frac{1}{\sqrt{m}}\Theta(\sigma_0^2 \sigma_\xi^2 d^2 n^{-1}) + O\big(\sigma_0^4 d^2 \langle \mathbf{w}_{r,t}, \boldsymbol{\xi}_i \rangle + \langle \mathbf{w}_{r,t}, \boldsymbol{\xi}_i \rangle^5 + \langle \mathbf{w}_{r,t}, \boldsymbol{\xi}_i \rangle^3 \sigma_0^2 d \big) \\
&\quad + O\big(\langle \mathbf{w}_{r,t}, \boldsymbol{\xi}_i \rangle^3 \sigma_0^2 \sigma_\xi^2 d^2 n^{-1} + \langle \mathbf{w}_{r,t}, \boldsymbol{\xi}_i \rangle \sigma_0^4 \sigma_\xi^2 d^3 n^{-1}\big) \\
&= -\frac{1}{\sqrt{m}}\Theta(\sigma_0^2 \sigma_\xi^2 d^2 n^{-1}) + O\big(\langle \mathbf{w}_{r,t}, \boldsymbol{\xi}_i \rangle^3 \sigma_0^2 \sigma_\xi^2 d^2 n^{-1} + \langle \mathbf{w}_{r,t}, \boldsymbol{\xi}_i \rangle \sigma_0^4 \sigma_\xi^2 d^3 n^{-1}\big) \\
&= -\frac{1}{\sqrt{m}}\Theta(\sigma_0^2 \sigma_\xi^2 d^2 n^{-1})
\end{aligned}
$$

where the first equality is due to $\mathrm{SNR}^{-1} = \widetilde{O}(d^{1/4})$ and $d \geq \widetilde{\Omega}(n^{2/3}\sigma_\xi^{-2/3})$. The second equality is by $\sigma_0^3 \leq \widetilde{O}(d^{-1}m^{-1/2})$. The third equality is by $\sigma_0^3 \leq \widetilde{O}(\sigma_\xi^{-1}d^{-7/4}nm^{-1/2})$.

This suggests that we can still leverage (27) to bound

$$\langle \mathbf{w}_{r,t}^{\widetilde{T}_\mu}, \boldsymbol{\xi}_i \rangle = \langle \mathbf{w}_{r,t}^{\widetilde{T}_\mu - 1}, \boldsymbol{\xi}_i \rangle + \frac{\eta\alpha_t\beta_t}{n\sqrt{m}}\Theta(\sigma_0^2\sigma_\xi^2 d^2) \leq \langle \mathbf{w}_{r,t}^0, \boldsymbol{\xi}_i \rangle + \frac{\eta\alpha_t\beta_t}{n\sqrt{m}}\Theta(\sigma_0^2\sigma_\xi^2 d^2)T_\mu$$
$$= \widetilde{O}(\sigma_0\sigma_\xi\sqrt{d}n^{-1}\mathrm{SNR}^{-1})$$

The bound on $\langle \mathbf{w}_{r,t}^{\widetilde{T}_\xi}, \boldsymbol{\mu}_j \rangle$ is the same as (29). Then following the same argument, we can decompose

$$\|\mathbf{w}_{r,t}^{\widetilde{T}_\xi}\|^2 = \widetilde{O}(\sigma_0^2) + \widetilde{O}(\sigma_0^2 n^{-1}\mathrm{SNR}^{-2}) + \Theta(\sigma_0^2 d) = \Theta(\sigma_0^2 d)$$

where the last equality is by the condition that $\mathrm{SNR}^{-1} = \widetilde{O}(d^{1/4})$. This verifies the induction on $\|\mathbf{w}_{r,t}^k\|^2 = \Theta(\sigma_0^2 d)$.

Furthermore, at $k = T_1$, we have for all $r \in [m]$, $j = \pm 1$ and $i \in [n]$, the growth term dominates the initialization term and thus

$$\langle \mathbf{w}_{r,t}^{T_1}, \boldsymbol{\mu}_j \rangle = \Theta(\eta\alpha_t\beta_t m^{-1/2}\sigma_0^2 d\|\boldsymbol{\mu}\|^2 T_1) \geq \widetilde{\Theta}(\sigma_0\|\boldsymbol{\mu}\|) \geq \Theta(|\langle \mathbf{w}_{r,t}^0, \boldsymbol{\mu}_j \rangle|)$$
$$\langle \mathbf{w}_{r,t}^{T_1}, \boldsymbol{\xi}_i \rangle = \Theta(\eta\alpha_t\beta_t n^{-1}m^{-1/2}\sigma_0^2 d\sigma_\xi^2 dT_1) \geq \widetilde{\Theta}(\sigma_0\sigma_\xi\sqrt{d}) \geq \Theta(|\langle \mathbf{w}_{r,t}^0, \boldsymbol{\xi}_i \rangle|)$$

where the inequality is by the definition of $T_1$. Thus, we verify the concentration of inner products, i.e., $\langle \mathbf{w}_{r,t}^{T_1}, \boldsymbol{\mu}_j \rangle = \Theta(\langle \mathbf{w}_{r',t}^{T_1}, \boldsymbol{\mu}_{j'} \rangle)$ and $\langle \mathbf{w}_{r,t}^{T_1}, \boldsymbol{\xi}_i \rangle = \Theta(\langle \mathbf{w}_{r',t}^{T_1}, \boldsymbol{\xi}_{i'} \rangle)$, at the end of first stage as well as the ratio $\langle \mathbf{w}_{r,t}^{T_1}, \boldsymbol{\mu}_j \rangle / \langle \mathbf{w}_{r',t}^{T_1}, \boldsymbol{\xi}_i \rangle = \Theta(n \cdot \mathrm{SNR}^2)$ for any $r, r' \in [m]$. Then, we can see directly $\|\mathbf{w}_{r,t}^{T_1}\|^2 = \Theta(\|\mathbf{w}_{r',t}^{T_1}\|^2) = \Theta(\sigma_0^2 d)$ for all $r, r' \in [m]$.

Next, we verify at $T_1$, we have $\langle \mathbf{w}_{r,t}^{T_1}, \mathbf{w}_{r',t}^{T_1} \rangle = \Theta(\|\mathbf{w}_{r,t}^{T_1}\|^2)$ for all $r, r' \in [m]$ such that $r \neq r'$. To this end, we first notice that the conditions required by Lemma D.4 are readily satisfied at $k = T_1$ and thus applying Lemma D.4 yields

$$\|\mathbf{w}_{r,t}^{T_1}\|^2 = \Theta\big(\langle \mathbf{w}_{r,t}^{T_1}, \boldsymbol{\mu}_j \rangle^2 \|\boldsymbol{\mu}\|^{-2} + n \cdot \mathrm{SNR}^2 \langle \mathbf{w}_{r,t}^{T_1}, \boldsymbol{\xi}_i \rangle^2 \|\boldsymbol{\mu}\|^{-2} + \|\mathbf{w}_{r,t}^0\|^2\big)$$
$$\langle \mathbf{w}_{r,t}^{T_1}, \mathbf{w}_{r',t}^{T_1} \rangle = \Theta\big(\langle \mathbf{w}_{r,t}^{T_1}, \boldsymbol{\mu}_j \rangle \langle \mathbf{w}_{r',t}^{T_1}, \boldsymbol{\mu}_j \rangle \|\boldsymbol{\mu}\|^{-2} + n \cdot \mathrm{SNR}^2 \langle \mathbf{w}_{r,t}^{T_1}, \boldsymbol{\xi}_i \rangle \langle \mathbf{w}_{r',t}^{T_1}, \boldsymbol{\xi}_i \rangle \|\boldsymbol{\mu}\|^{-2} + \langle \mathbf{w}_{r,t}^0, \mathbf{w}_{r',t}^0 \rangle\big)$$
$$= \Theta\big(\langle \mathbf{w}_{r,t}^{T_1}, \boldsymbol{\mu}_j \rangle^2 \|\boldsymbol{\mu}\|^{-2} + n \cdot \mathrm{SNR}^2 \langle \mathbf{w}_{r,t}^{T_1}, \boldsymbol{\xi}_i \rangle^2 \|\boldsymbol{\mu}\|^{-2} + \langle \mathbf{w}_{r,t}^0, \mathbf{w}_{r',t}^0 \rangle\big)$$
$$= \Theta\big(\|\mathbf{w}_{r,t}^{T_1}\|^2 - \|\mathbf{w}_{r,t}^0\|^2 + \langle \mathbf{w}_{r,t}^0, \mathbf{w}_{r',t}^0 \rangle\big)$$
$$= \Theta\big(\|\mathbf{w}_{r,t}^{T_1}\|^2 - \sigma_0^2 d\big) + \widetilde{O}(\sigma_0^2\sqrt{d})$$
$$= \Theta(\|\mathbf{w}_{r,t}^{T_1}\|^2)$$

where the second equality for $\langle \mathbf{w}_{r,t}^{T_1}, \mathbf{w}_{r',t}^{T_1} \rangle$ is due to $\langle \mathbf{w}_{r,t}^{T_1}, \boldsymbol{\mu}_j \rangle = \Theta(\langle \mathbf{w}_{r',t}^{T_1}, \boldsymbol{\mu}_{j'} \rangle)$ and $\langle \mathbf{w}_{r,t}^{T_1}, \boldsymbol{\xi}_i \rangle = \Theta(\langle \mathbf{w}_{r',t}^{T_1}, \boldsymbol{\xi}_{i'} \rangle)$ and the second last equality is by Lemma D.1.

Finally we verify that at $T_1$,

$$\langle \nabla_{\mathbf{w}_{r,t}} L(\mathbf{W}_t^{T_1}), \mathbf{w}_{r,t}^0 \rangle$$
$$= -\frac{1}{\sqrt{m}}\Theta(\langle \mathbf{w}_{r,t}^{T_1}, \boldsymbol{\mu}_j + \overline{\boldsymbol{\xi}} \rangle \langle \mathbf{w}_{r,t}^{T_1}, \mathbf{w}_{r,t}^0 \rangle + \|\mathbf{w}_{r,t}^{T_1}\|^2 \langle \mathbf{w}_{r,t}^0, \boldsymbol{\mu}_j + \overline{\boldsymbol{\xi}} \rangle)$$
$$+ O\Big(\big((\langle \mathbf{w}_{r,t}^{T_1}, \boldsymbol{\mu}_j \rangle^4 + \langle \mathbf{w}_{r,t}^{T_1}, \boldsymbol{\xi}_i \rangle^4 + (\langle \mathbf{w}_{r,t}^{T_1}, \boldsymbol{\mu}_j \rangle^2 + \langle \mathbf{w}_{r,t}^{T_1}, \boldsymbol{\xi}_i \rangle^2)\|\mathbf{w}_{r,t}^{T_1}\|^2 + \|\mathbf{w}_{r,t}^{T_1}\|^4)\langle \mathbf{w}_{r,t}^{T_1}, \mathbf{w}_{r,t}^0 \rangle\Big)$$
$$+ O\Big(\langle \mathbf{w}_{r,t}^{T_1}, \boldsymbol{\mu}_j \rangle^3 \|\mathbf{w}_{r,t}^{T_1}\|^2 \langle \mathbf{w}_{r,t}^0, \boldsymbol{\mu}_j \rangle + \langle \mathbf{w}_{r,t}^{T_1}, \boldsymbol{\xi}_i \rangle^3 \|\mathbf{w}_{r,t}^{T_1}\|^2 \langle \mathbf{w}_{r,t}^0, \boldsymbol{\xi}_i \rangle\Big)$$
$$+ O\Big(\|\mathbf{w}_{r,t}^{T_1}\|^4 \langle \mathbf{w}_{r,t}^{T_1}, \boldsymbol{\mu}_j \rangle \langle \mathbf{w}_{r,t}^0, \boldsymbol{\mu}_j \rangle + \|\mathbf{w}_{r,t}^{T_1}\|^4 \langle \mathbf{w}_{r,t}^{T_1}, \boldsymbol{\xi}_i \rangle \langle \mathbf{w}_{r,t}^0, \boldsymbol{\xi}_i \rangle\Big)$$
$$= -\frac{1}{\sqrt{m}}\Theta((\langle \mathbf{w}_{r,t}^{T_1}, \boldsymbol{\mu}_j + \overline{\boldsymbol{\xi}} \rangle - \sqrt{m}\|\mathbf{w}_{r,t}^{T_1}\|^4)\langle \mathbf{w}_{r,t}^{T_1}, \mathbf{w}_{r,t}^0 \rangle + \|\mathbf{w}_{r,t}^{T_1}\|^2 \langle \mathbf{w}_{r,t}^0, \boldsymbol{\mu}_j + \overline{\boldsymbol{\xi}} \rangle)$$

where we use the concentration of neurons along directions $\boldsymbol{\mu}_j, \boldsymbol{\xi}_i$ at $T_1$ and the scale of $\langle \mathbf{w}_{r,t}^{T_1}, \boldsymbol{\mu}_j \rangle, \langle \mathbf{w}_{r,t}^{T_1}, \boldsymbol{\xi}_i \rangle$. $\qquad\square$

### D.4 SECOND STAGE

For the second stage, we derive an extension of Lemma D.4 given the scale of $\langle \mathbf{w}_{r,t}^k, \mathbf{w}_{r,t}^0 \rangle$ can escape initialization. We highlight that unlike $\langle \mathbf{w}_{r,t}^k, \boldsymbol{\mu}_j \rangle$ and $\langle \mathbf{w}_{r,t}^k, \boldsymbol{\xi}_i \rangle$ that increase monotonically, the dominant term of $\langle \nabla_{\mathbf{w}_{r,t}} L(\mathbf{W}_t^{T_1}), \mathbf{w}_{r,t}^0 \rangle$ suggests that $\langle \mathbf{w}_{r,t}^k, \mathbf{w}_{r,t}^0 \rangle$ can also decrease.

**Lemma D.6.** *For any $k$ and $r \in [m]$, such that $\langle \mathbf{w}_{r,t}^k, \boldsymbol{\mu}_j \rangle = \Theta(\langle \mathbf{w}_{r,t}^k, \boldsymbol{\mu}_{j'} \rangle) \geq \Theta(\langle \mathbf{w}_{r,t}^k, \mathbf{w}_{r,t}^0 \rangle \|\mathbf{w}_{r,t}^0\|^{-2} |\langle \mathbf{w}_{r,t}^0, \boldsymbol{\mu}_j \rangle|)$, $\langle \mathbf{w}_{r,t}^k, \boldsymbol{\xi}_i \rangle = \Theta(\langle \mathbf{w}_{r,t}^k, \boldsymbol{\xi}_{i'} \rangle) \geq \Theta(\langle \mathbf{w}_{r,t}^k, \mathbf{w}_{r,t}^0 \rangle \|\mathbf{w}_{r,t}^0\|^{-2} |\langle \mathbf{w}_{r,t}^0, \boldsymbol{\xi}_i \rangle|)$ and $\langle \mathbf{w}_{r,t}^k, \boldsymbol{\mu}_j \rangle, \langle \mathbf{w}_{r,t}^k, \boldsymbol{\xi}_i \rangle = \widetilde{O}(1)$, $\langle \mathbf{w}_{r,t}^k, \mathbf{w}_{r,t}^0 \rangle = \Theta(\langle \mathbf{w}_{r',t}^k, \mathbf{w}_{r',t}^0 \rangle) = \Omega(\min\{\sigma_0 \sigma_\xi^{-1} n^{1/2} m^{-1/6}, \sigma_0 \sqrt{d} m^{-1/6}\})$ for any $j, j' = \pm 1, i, i' \in [n], r, r' \in [m]$. Then we can show*

$$\|\mathbf{w}_{r,t}^k\|^2 = \Theta\big(\langle \mathbf{w}_{r,t}^k, \boldsymbol{\mu}_j \rangle^2 \|\boldsymbol{\mu}\|^{-2} + n \cdot \mathrm{SNR}^2 \langle \mathbf{w}_{r,t}^k, \boldsymbol{\xi}_i \rangle^2 \|\boldsymbol{\mu}\|^{-2} + \langle \mathbf{w}_{r,t}^k, \mathbf{w}_{r,t}^0 \rangle^2 \|\mathbf{w}_{r,t}^0\|^{-2}\big).$$

*And for $r \neq r'$, we have*

$$\langle \mathbf{w}_{r,t}^k, \mathbf{w}_{r',t}^k \rangle = \Theta\Big(\langle \mathbf{w}_{r,t}^k, \boldsymbol{\mu}_j \rangle \langle \mathbf{w}_{r',t}^k, \boldsymbol{\mu}_j \rangle \|\boldsymbol{\mu}\|^{-2} + n \cdot \mathrm{SNR}^2 \langle \mathbf{w}_{r,t}^k, \boldsymbol{\xi}_i \rangle \langle \mathbf{w}_{r',t}^k, \boldsymbol{\xi}_i \rangle \|\boldsymbol{\mu}\|^{-2}$$
$$+ \langle \mathbf{w}_{r,t}^k, \mathbf{w}_{r,t}^0 \rangle^2 \frac{\langle \mathbf{w}_{r,t}^0, \mathbf{w}_{r',t}^0 \rangle}{\|\mathbf{w}_{r,t}^0\|^4}\Big)$$

*Proof of Lemma D.6.* Similar to the proof of Lemma D.4, we can decompose the weight $\mathbf{w}_{r,t}^k$ as

$$\mathbf{w}_{r,t}^k = \phi_r^k \mathbf{w}_{r,t}^0 + \gamma_1^k \boldsymbol{\mu}_1 \|\boldsymbol{\mu}_1\|^{-2} + \gamma_{-1}^k \boldsymbol{\mu}_{-1} \|\boldsymbol{\mu}_{-1}\|^{-2} + \sum_{i=1}^n \rho_{r,i}^k \boldsymbol{\xi}_i \|\boldsymbol{\xi}_i\|^{-2}.$$

First, we show that $\phi_r^k = \Theta(\langle \mathbf{w}_{r,t}^k, \mathbf{w}_{r,t}^0 \rangle \|\mathbf{w}_{r,t}^0\|^{-2})$ as follows. We compute

$$\langle \mathbf{w}_{r,t}^k, \mathbf{w}_{r,t}^0 \rangle = \phi_r^k \|\mathbf{w}_{r,t}^0\|^2 + \Theta(\langle \mathbf{w}_{r,t}^k, \boldsymbol{\mu}_j \rangle \langle \mathbf{w}_{r,t}^0, \boldsymbol{\mu}_j \rangle \|\boldsymbol{\mu}\|^{-2} + \langle \mathbf{w}_{r,t}^k, \boldsymbol{\xi}_i \rangle \sum_{i=1}^n \langle \mathbf{w}_{r,t}^0, \boldsymbol{\xi}_i \rangle \|\boldsymbol{\xi}_i\|^{-2})$$

$$= \phi_r^k \|\mathbf{w}_{r,t}^0\|^2 + \widetilde{O}(\sigma_0 + n\sigma_0 \sigma_\xi^{-1} d^{-1/2})$$

$$= \Theta(\phi_r^k \|\mathbf{w}_{r,t}^0\|^2)$$

where the second equality is by the assumption that $\langle \mathbf{w}_{r,t}^k, \boldsymbol{\mu}_j \rangle, \langle \mathbf{w}_{r,t}^k, \boldsymbol{\xi}_i \rangle = \widetilde{O}(1)$ and the last equality is by the assumption $\langle \mathbf{w}_{r,t}^k, \mathbf{w}_{r,t}^0 \rangle = \Omega(\min\{\sigma_0 \sigma_\xi^{-1} n^{1/2} m^{-1/6}, \sigma_0 \sqrt{d} m^{-1/6}\})$ and the condition that $\sigma_\xi^{-1} = \Omega(d^{1/4})$, $d \geq \widetilde{O}(nm^{1/3})$ and $d \geq \widetilde{O}(nm^{1/6}\sigma_\xi^{-1})$. Then based on the assumption, we can still bound $\langle \mathbf{w}_{r,t}^k, \boldsymbol{\mu}_j \rangle \geq \phi_r^k |\langle \mathbf{w}_{r,t}^0, \boldsymbol{\mu}_j \rangle| = \Theta(\langle \mathbf{w}_{r,t}^k, \mathbf{w}_{r,t}^0 \rangle \|\mathbf{w}_{r,t}^0\|^{-2} |\langle \mathbf{w}_{r,t}^0, \boldsymbol{\mu}_j \rangle|)$, and similarly we can bound $\langle \mathbf{w}_{r,t}^k, \boldsymbol{\xi}_i \rangle \geq \phi_r^k |\langle \mathbf{w}_{r,t}^0, \boldsymbol{\xi}_i \rangle| = \Theta(\langle \mathbf{w}_{r,t}^k, \mathbf{w}_{r,t}^0 \rangle \|\mathbf{w}_{r,t}^0\|^{-2} |\langle \mathbf{w}_{r,t}^0, \boldsymbol{\xi}_i \rangle|)$. This allows to simplify

$$\mathbf{w}_{r,t}^k = \Theta(\langle \mathbf{w}_{r,t}^k, \mathbf{w}_{r,t}^0 \rangle \|\mathbf{w}_{r,t}^0\|^{-2}) \mathbf{w}_{r,t}^0 + \Theta(\langle \mathbf{w}_{r,t}^k, \boldsymbol{\mu}_j \rangle (\boldsymbol{\mu}_1 + \boldsymbol{\mu}_{-1}) \|\boldsymbol{\mu}\|^{-2})$$
$$+ \Theta\big(\langle \mathbf{w}_{r,t}^k, \boldsymbol{\xi}_i \rangle\big) \sum_{i=1}^n \boldsymbol{\xi}_i \|\boldsymbol{\xi}_i\|^{-2} \tag{34}$$

Consequently, the assumption that $\langle \mathbf{w}_{r,t}^k, \mathbf{w}_{r,t}^0 \rangle = \Theta(\langle \mathbf{w}_{r',t}^k, \mathbf{w}_{r',t}^0 \rangle)$, combined with (34), we can derive that $\phi_r^k = \Theta(\phi_{r'}^k)$ given $\langle \mathbf{w}_{r,t}^k, \boldsymbol{\mu}_j \rangle = \Theta(\langle \mathbf{w}_{r',t}^k, \boldsymbol{\mu}_j \rangle)$ and $\langle \mathbf{w}_{r,t}^k, \boldsymbol{\xi}_i \rangle = \Theta(\langle \mathbf{w}_{r',t}^k, \boldsymbol{\xi}_i \rangle)$. Thus, we can compute

$$\|\mathbf{w}_{r,t}^k\|^2 = \Theta\Big((\phi_r^k)^2 \|\mathbf{w}_{r,t}^0\|^2 + \langle \mathbf{w}_{r,t}^k, \boldsymbol{\mu}_j \rangle^2 \|\boldsymbol{\mu}\|^{-2} + n\mathrm{SNR}^2 \langle \mathbf{w}_{r,t}^k, \boldsymbol{\xi}_i \rangle^2 \|\boldsymbol{\mu}\|^{-2}\Big) = \Theta(\|\mathbf{w}_{r',t}^k\|^2)$$

In addition, we can derive for $r \neq r'$

$$\langle \mathbf{w}_{r,t}^k, \mathbf{w}_{r',t}^k \rangle$$
$$= \Theta\big((\phi_r^k)^2 \langle \mathbf{w}_{r,t}^0, \mathbf{w}_{r',t}^0 \rangle + \langle \mathbf{w}_{r,t}^k, \boldsymbol{\mu}_j \rangle \langle \mathbf{w}_{r',t}^k, \boldsymbol{\mu}_j \rangle \|\boldsymbol{\mu}\|^{-2} + n\mathrm{SNR}^2 \langle \mathbf{w}_{r,t}^k, \boldsymbol{\xi}_i \rangle \langle \mathbf{w}_{r',t}^k, \boldsymbol{\xi}_i \rangle \|\boldsymbol{\mu}\|^{-2}\big)$$

which completes the proof. $\qquad\square$

**Lemma D.7.** *Let $T_1^+ \geq T_1$ and suppose for all $T_1 \leq k < T_1^+$, it satisfies that for all $j = \pm 1, i \in [n], r \in [m]$, $\langle \mathbf{w}_{r,t}^{k+1}, \boldsymbol{\mu}_j \rangle, \langle \mathbf{w}_{r,t}^{k+1}, \boldsymbol{\xi}_i \rangle = \widetilde{O}(1)$, $\langle \mathbf{w}_{r,t}^k, \mathbf{w}_{r,t}^0 \rangle = \Omega(\min\{\sigma_0 \sigma_\xi^{-1} n^{1/2} m^{-1/6}, \sigma_0 \sqrt{d} m^{-1/6}\})$ and*

$$\langle \mathbf{w}_{r,t}^{k+1}, \boldsymbol{\mu}_j \rangle = \langle \mathbf{w}_{r,t}^k, \boldsymbol{\mu}_j \rangle + \Theta\left(\frac{\eta}{\sqrt{m}} \|\mathbf{w}_{r,t}^k\|^2 \|\boldsymbol{\mu}\|^2\right) \tag{35}$$

$$\langle \mathbf{w}_{r,t}^{k+1}, \boldsymbol{\xi}_i \rangle = \langle \mathbf{w}_{r,t}^k, \boldsymbol{\xi}_i \rangle + \Theta\left(\frac{\eta}{n\sqrt{m}} \|\mathbf{w}_{r,t}^k\|^2 \|\boldsymbol{\xi}_i\|^2\right). \tag{36}$$

$$\langle \mathbf{w}_{r,t}^{k+1}, \mathbf{w}_{r,t}^0 \rangle = \langle \mathbf{w}_{r,t}^k, \mathbf{w}_{r,t}^0 \rangle$$
$$+ \frac{\eta}{\sqrt{m}} \Theta\left( \left(\langle \mathbf{w}_{r,t}^k, \boldsymbol{\mu}_j + \overline{\boldsymbol{\xi}} \rangle - \sqrt{m}\|\mathbf{w}_{r,t}^k\|^4\right) \langle \mathbf{w}_{r,t}^k, \mathbf{w}_{r,t}^0 \rangle + \|\mathbf{w}_{r,t}^k\|^2 \langle \mathbf{w}_{r,t}^0, \boldsymbol{\mu}_j + \overline{\boldsymbol{\xi}} \rangle \right) \tag{37}$$

*Then we have for all $T_1 \leq k \leq T_1^+$,*

(1) $\langle \mathbf{w}_{r,t}^k, \boldsymbol{\mu}_j \rangle = \Theta(\langle \mathbf{w}_{r',t}^k, \boldsymbol{\mu}_{j'} \rangle)$

(2) $\langle \mathbf{w}_{r,t}^k, \boldsymbol{\xi}_i \rangle = \Theta(\langle \mathbf{w}_{r',t}^k, \boldsymbol{\xi}_{i'} \rangle)$

(3) $\langle \mathbf{w}_{r,t}^k, \mathbf{w}_{r,t}^0 \rangle = \Theta(\langle \mathbf{w}_{r',t}^k, \mathbf{w}_{r',t}^0 \rangle)$

(4) $\langle \mathbf{w}_{r,t}^k, \boldsymbol{\mu}_j \rangle \geq \Theta(\langle \mathbf{w}_{r,t}^k, \mathbf{w}_{r,t}^0 \rangle \|\mathbf{w}_{r,t}^0\|^{-2} |\langle \mathbf{w}_{r,t}^0, \boldsymbol{\mu}_j \rangle|)$,
$\langle \mathbf{w}_{r,t}^k, \boldsymbol{\xi}_i \rangle \geq \Theta(\langle \mathbf{w}_{r,t}^k, \mathbf{w}_{r,t}^0 \rangle \|\mathbf{w}_{r,t}^0\|^{-2} |\langle \mathbf{w}_{r,t}^0, \boldsymbol{\xi}_i \rangle|)$,

(5) $\|\mathbf{w}_{r,t}^k\|^2 = \Theta(\|\mathbf{w}_{r',t}^k\|^2)$

(6) $\langle \mathbf{w}_{r,t}^k, \mathbf{w}_{r',t}^k \rangle = \Theta(\|\mathbf{w}_{r,t}^k\|^2)$ *for $r' \neq r$*

(7) $|\langle \mathbf{w}_{r,t}^k, \boldsymbol{\mu}_j \rangle| / |\langle \mathbf{w}_{r',t}^k, \boldsymbol{\xi}_i \rangle| = \Theta(n \cdot \mathrm{SNR}^2)$

*for all $j = \pm 1, r, r' \in [m], i \in [n]$.*

*Proof of Lemma D.7.* The proof is by induction. First, when $k = T_1$, claims (1-8) are satisfied by Lemma D.5 with $\langle \mathbf{w}_{r,t}^{T_1}, \mathbf{w}_{r,t}^0 \rangle = \Theta(\sigma_0^2 d)$, $\langle \mathbf{w}_{r,t}^{T_1}, \mathbf{w}_{r,t}^0 \rangle \|\mathbf{w}_{r,t}^0\|^{-2} = \Theta(1)$. Now suppose there exists $\widetilde{T}_1^+ < T_1^+$ such that for all $T_1 \leq k \leq \widetilde{T}_1^+$, (1-6) are satisfied. We aim to show for it is also satisfied for $k + 1$. By the assumption that for any $r \in [m]$

$$\langle \mathbf{w}_{r,t}^{k+1}, \boldsymbol{\mu}_j \rangle = \langle \mathbf{w}_{r,t}^k, \boldsymbol{\mu}_j \rangle + \Theta\left(\frac{\eta}{\sqrt{m}} \|\mathbf{w}_{r,t}^k\|^2 \|\boldsymbol{\mu}\|^2\right)$$

$$\langle \mathbf{w}_{r,t}^{k+1}, \boldsymbol{\xi}_i \rangle = \langle \mathbf{w}_{r,t}^k, \boldsymbol{\xi}_i \rangle + \Theta\left(\frac{\eta}{n\sqrt{m}} \|\mathbf{w}_{r,t}^k\|^2 \|\boldsymbol{\xi}_i\|^2\right),$$

$$\langle \mathbf{w}_{r,t}^{k+1}, \mathbf{w}_{r,t}^0 \rangle = \langle \mathbf{w}_{r,t}^k, \mathbf{w}_{r,t}^0 \rangle$$
$$+ \frac{\eta}{\sqrt{m}} \Theta\left( \left(\langle \mathbf{w}_{r,t}^k, \boldsymbol{\mu}_j + \boldsymbol{\xi}_i \rangle - \sqrt{m}\|\mathbf{w}_{r,t}^k\|^4\right) \langle \mathbf{w}_{r,t}^k, \mathbf{w}_{r,t}^0 \rangle + \|\mathbf{w}_{r,t}^k\|^2 \langle \mathbf{w}_{r,t}^0, \boldsymbol{\mu}_j + \boldsymbol{\xi}_i \rangle \right)$$

we can show

$$\langle \mathbf{w}_{r,t}^{k+1}, \boldsymbol{\mu}_j \rangle = \langle \mathbf{w}_{r,t}^k, \boldsymbol{\mu}_j \rangle + \Theta\left(\frac{\eta}{\sqrt{m}} \|\mathbf{w}_{r,t}^k\|^2 \|\boldsymbol{\mu}\|^2\right) = \Theta\left( \langle \mathbf{w}_{r',t}^k, \boldsymbol{\mu}_{j'} \rangle + \frac{\eta}{\sqrt{m}} \|\mathbf{w}_{r',t}^k\|^2 \|\boldsymbol{\mu}\|^2 \right)$$
$$= \Theta(\langle \mathbf{w}_{r',t}^{k+1}, \boldsymbol{\mu}_{j'} \rangle)$$

where the second equality is by induction condition, thus verifying the induction for claim (1). Similarly, we can use the same argument for verifying claim (2). For the claim (3)

$$\langle \mathbf{w}_{r,t}^{k+1}, \mathbf{w}_{r,t}^0 \rangle$$
$$= \langle \mathbf{w}_{r,t}^k, \mathbf{w}_{r,t}^0 \rangle + \frac{\eta}{\sqrt{m}} \Theta\left( \left(\langle \mathbf{w}_{r,t}^k, \boldsymbol{\mu}_j + \overline{\boldsymbol{\xi}} \rangle - \sqrt{m}\|\mathbf{w}_{r,t}^k\|^2\right) \langle \mathbf{w}_{r,t}^k, \mathbf{w}_{r,t}^0 \rangle + \|\mathbf{w}_{r,t}^k\|^2 \langle \mathbf{w}_{r,t}^0, \boldsymbol{\mu}_j + \overline{\boldsymbol{\xi}} \rangle \right)$$
$$= \Theta(\langle \mathbf{w}_{r',t}^k, \mathbf{w}_{r',t}^0 \rangle)$$

$$+ \frac{\eta}{\sqrt{m}} \Theta\Big( \big(\langle \mathbf{w}^k_{r',t}, \boldsymbol{\mu}_j + \overline{\boldsymbol{\xi}}\rangle - \sqrt{m}\|\mathbf{w}^k_{r',t}\|^4\big)\langle \mathbf{w}^k_{r',t}, \mathbf{w}^0_{r',t}\rangle + \|\mathbf{w}^k_{r',t}\|^2\langle \mathbf{w}^0_{r',t}, \boldsymbol{\mu}_j + \overline{\boldsymbol{\xi}}\rangle \Big)$$

$$= \Theta(\langle \mathbf{w}^{k+1}_{r',t}, \mathbf{w}^0_{r',t}\rangle)$$

where the second equality is due to induction claim that $\langle \mathbf{w}^k_{r,t}, \mathbf{w}^0_{r,t}\rangle = \Theta(\langle \mathbf{w}^k_{r',t}, \mathbf{w}^0_{r',t}\rangle)$, $\langle \mathbf{w}^k_{r,t}, \boldsymbol{\mu}_j + \overline{\boldsymbol{\xi}}\rangle = \Theta(\langle \mathbf{w}^k_{r',t}, \boldsymbol{\mu}_j + \overline{\boldsymbol{\xi}}\rangle)$, and we can show that $\Theta(\langle \mathbf{w}^0_{r,t}, \mathbf{v}\rangle) = \langle \mathbf{w}^0_{r',t}, \mathbf{v}\rangle$ holds for any $\mathbf{v}$, and any $r, r' \in [m]$ with constant probability due to $m = \Theta(1)$. Next, we verify claim (7)

$$\frac{\langle \mathbf{w}^{k+1}_{r,t}, \boldsymbol{\mu}_j\rangle}{\langle \mathbf{w}^{k+1}_{r',t}, \boldsymbol{\xi}_i\rangle} = \frac{\langle \mathbf{w}^k_{r,t}, \boldsymbol{\mu}_j\rangle + \Theta\big(\frac{\eta}{\sqrt{m}}\|\mathbf{w}^k_{r,t}\|^2\|\boldsymbol{\mu}\|^2\big)}{\langle \mathbf{w}^k_{r',t}, \boldsymbol{\xi}_i\rangle + \Theta\big(\frac{\eta}{n\sqrt{m}}\|\mathbf{w}^k_{r',t}\|^2\|\boldsymbol{\xi}_i\|^2\big)} = \Theta(n \cdot \mathrm{SNR}^2)$$

where the last equality follows from the induction condition and $\|\boldsymbol{\mu}\|^2/\|\boldsymbol{\xi}_i\|^2 = \Theta(\mathrm{SNR}^2)$ by Lemma B.2 and $\|\mathbf{w}^k_{r,t}\|^2 = \Theta(\|\mathbf{w}^k_{r',t}\|^2)$ by induction condition. Thus the induction for (7) is verified.

Next in order to verify (4), we only need to show the growth of $\langle \mathbf{w}^k_{r,t}, \boldsymbol{\mu}_j\rangle$, $\langle \mathbf{w}^k_{r,t}, \boldsymbol{\xi}_i\rangle$ is larger than the growth of $\langle \mathbf{w}^k_{r,t}, \mathbf{w}^0_{r,t}\rangle \|\mathbf{w}^k_{r,t}\|^{-2}|\langle \mathbf{w}^0_{r,t}, \boldsymbol{\mu}_j\rangle|$ and $\langle \mathbf{w}^k_{r,t}, \mathbf{w}^0_{r,t}\rangle\|\mathbf{w}^k_{r,t}\|^{-2}|\langle \mathbf{w}^0_{r,t}, \boldsymbol{\xi}_i\rangle|$ respectively. To this end, we consider upper bounding the update of $\langle \mathbf{w}^k_{r,t}, \mathbf{w}^0_{r,t}\rangle$ as

$$|\langle \mathbf{w}^k_{r,t}, \boldsymbol{\mu}_j + \overline{\boldsymbol{\xi}}\rangle\langle \mathbf{w}^k_{r,t}, \mathbf{w}^0_{r,t}\rangle| \le \Theta\big(\|\mathbf{w}^k_{r,t}\|^2(\|\boldsymbol{\mu}\| + \sigma_\xi\sqrt{d})\|\mathbf{w}^0_{r,t}\|\big)$$

$$|\|\mathbf{w}^k_{r,t}\|^2\langle \mathbf{w}^0_{r,t}, \boldsymbol{\mu}_j + \overline{\boldsymbol{\xi}}\rangle| \le \Theta\big(\|\mathbf{w}^k_{r,t}\|^2(\|\boldsymbol{\mu}\| + \sigma_\xi\sqrt{d})\|\mathbf{w}^0_{r,t}\|\big).$$

Then we consider two cases depending on the magnitude of $\|\boldsymbol{\mu}\|$ and $\sigma_\xi\sqrt{d}$:

- When $\|\boldsymbol{\mu}\| \ge \sigma_\xi\sqrt{d}$, i.e., $\sigma_\xi\sqrt{d} = O(1)$. Then

$$\|\mathbf{w}^k_{r,t}\|^2(\|\boldsymbol{\mu}\| + \sigma_\xi\sqrt{d})\|\mathbf{w}^0_{r,t}\|^{-1}|\langle \mathbf{w}^0_{r,t}, \boldsymbol{\mu}_j\rangle| = \widetilde{O}(\|\mathbf{w}^k_{r,t}\|^2 d^{-1/2}) \le \Theta(\|\mathbf{w}^k_{r,t}\|^2\|\boldsymbol{\mu}\|^2) \quad (38)$$

$$\|\mathbf{w}^k_{r,t}\|^2(\|\boldsymbol{\mu}\| + \sigma_\xi\sqrt{d})\|\mathbf{w}^0_{r,t}\|^{-1}|\langle \mathbf{w}^0_{r,t}, \boldsymbol{\xi}_i\rangle| = \widetilde{O}(\|\mathbf{w}^k_{r,t}\|^2\sigma_\xi) \le \Theta(\frac{1}{n}\|\mathbf{w}^k_{r,t}\|^2\|\boldsymbol{\xi}_i\|^2) \quad (39)$$

  where we use the condition on $d = \widetilde{\Omega}(n\sigma_\xi^{-1})$ and $\|\boldsymbol{\xi}_i\|^2 = \Theta(\sigma_\xi^2 d)$ for (39).

- When $\|\boldsymbol{\mu}\| \le \sigma_\xi\sqrt{d}$, i.e., we have $\sigma_\xi\sqrt{d} = \Omega(1)$. Then

$$\|\mathbf{w}^k_{r,t}\|^2(\|\boldsymbol{\mu}\| + \sigma_\xi\sqrt{d})\|\mathbf{w}^0_{r,t}\|^{-1}|\langle \mathbf{w}^0_{r,t}, \boldsymbol{\mu}_j\rangle|$$
$$= \Theta(\|\mathbf{w}^k_{r,t}\|^2\sigma_0^{-1}\sigma_\xi\langle \mathbf{w}^0_{r,t}, \boldsymbol{\mu}_j\rangle) = \widetilde{O}(\|\mathbf{w}^k_{r,t}\|^2\sigma_\xi) = \widetilde{O}(\|\mathbf{w}^k_{r,t}\|^2 nd^{-1/2}) \le \Theta(\|\mathbf{w}^k_{r,t}\|^2\|\boldsymbol{\mu}\|^2) \quad (40)$$

$$\|\mathbf{w}^k_{r,t}\|^2(\|\boldsymbol{\mu}\| + \sigma_\xi\sqrt{d})\|\mathbf{w}^0_{r,t}\|^{-1}|\langle \mathbf{w}^0_{r,t}, \boldsymbol{\xi}_i\rangle|$$
$$= \Theta(\|\mathbf{w}^k_{r,t}\|^2\sigma_0^{-1}\sigma_\xi\langle \mathbf{w}^0_{r,t}, \boldsymbol{\xi}_i\rangle) = \widetilde{O}(\|\mathbf{w}^k_{r,t}\|^2\sigma_\xi^2\sqrt{d}) \le \Theta(\frac{1}{n}\|\mathbf{w}^k_{r,t}\|^2\sigma_\xi^2 d) = \Theta(\frac{1}{n}\|\mathbf{w}^k_{r,t}\|^2\|\boldsymbol{\xi}_i\|^2) \quad (41)$$

  where the second last equality of (40) is by the condition that $\mathrm{SNR}^{-1} = \widetilde{O}(n)$ which implies that $\sigma_\xi = \widetilde{O}(nd^{-1/2})$. The second last inequality of (41) is by $d = \widetilde{\Omega}(n^2)$.

This suggests that

$$|\langle \mathbf{w}^k_{r,t}, \boldsymbol{\mu}_j + \overline{\boldsymbol{\xi}}\rangle\langle \mathbf{w}^k_{r,t}, \mathbf{w}^0_{r,t}\rangle| \le \Theta\big(\|\mathbf{w}^k_{r,t}\|^2\|\boldsymbol{\mu}\|^2\big)$$

$$|\|\mathbf{w}^k_{r,t}\|^2\langle \mathbf{w}^0_{r,t}, \boldsymbol{\mu}_j + \overline{\boldsymbol{\xi}}\rangle| \le \Theta(\frac{1}{n}\|\mathbf{w}^k_{r,t}\|^2\|\boldsymbol{\xi}_i\|^2).$$

which verifies the claim (4) by combining with the update (35), (36), (37).

Next, in order to verify (5,6), we leverage Lemma D.6. First, it is easy to verify that at $k + 1$, the conditions for Lemma D.6 are satisfied by the induction claims (1-4) at $k + 1$. Then we have

$$\|\mathbf{w}^{k+1}_{r,t}\|^2 = \Theta\big(\langle \mathbf{w}^{k+1}_{r,t}, \boldsymbol{\mu}_j\rangle^2\|\boldsymbol{\mu}\|^{-2} + n \cdot \mathrm{SNR}^2\langle \mathbf{w}^{k+1}_{r,t}, \boldsymbol{\xi}_i\rangle^2\|\boldsymbol{\mu}\|^{-2} + \langle \mathbf{w}^k_{r,t}, \mathbf{w}^0_{r,t}\rangle^2\|\mathbf{w}^0_{r,t}\|^{-2}\big)$$
$$= \Theta(\langle \mathbf{w}^{k+1}_{r',t}, \boldsymbol{\mu}_j\rangle^2\|\boldsymbol{\mu}\|^{-2} + n \cdot \mathrm{SNR}^2\langle \mathbf{w}^{k+1}_{r',t}, \boldsymbol{\xi}_i\rangle^2\|\boldsymbol{\mu}\|^{-2} + \langle \mathbf{w}^k_{r',t}, \mathbf{w}^0_{r',t}\rangle^2\|\mathbf{w}^0_{r',t}\|^{-2})$$

$$= \Theta(\|\mathbf{w}_{r',t}^{k+1}\|^2).$$

Finally, to verify (4) for $k+1$, we have from Lemma D.6 that

$$\langle \mathbf{w}_{r,t}^{k+1}, \mathbf{w}_{r',t}^{k+1} \rangle$$

$$= \Theta\big( \langle \mathbf{w}_{r,t}^{k+1}, \boldsymbol{\mu}_j \rangle \langle \mathbf{w}_{r',t}^{k+1}, \boldsymbol{\mu}_j \rangle \|\boldsymbol{\mu}\|^{-2} + n \cdot \mathrm{SNR}^2 \langle \mathbf{w}_{r,t}^{k+1}, \boldsymbol{\xi}_i \rangle \langle \mathbf{w}_{r',t}^{k+1}, \boldsymbol{\xi}_i \rangle \|\boldsymbol{\mu}\|^{-2} + \langle \mathbf{w}_{r,t}^{k+1}, \mathbf{w}_{r,t}^0 \rangle^2 \frac{\langle \mathbf{w}_{r,t}^0, \mathbf{w}_{r',t}^0 \rangle}{\|\mathbf{w}_{r,t}^0\|^4} \big)$$

$$= \Theta\big( \langle \mathbf{w}_{r,t}^{k+1}, \boldsymbol{\mu}_j \rangle^2 \|\boldsymbol{\mu}\|^{-2} + n \cdot \mathrm{SNR}^2 \langle \mathbf{w}_{r,t}^{k+1}, \boldsymbol{\xi}_i \rangle^2 \|\boldsymbol{\mu}\|^{-2} + \langle \mathbf{w}_{r,t}^{k+1}, \mathbf{w}_{r,t}^0 \rangle^2 \frac{\langle \mathbf{w}_{r,t}^0, \mathbf{w}_{r',t}^0 \rangle}{\|\mathbf{w}_{r,t}^0\|^4} \big)$$

$$= \Theta\big( \|\mathbf{w}_{r,t}^{k+1}\|^2 - \langle \mathbf{w}_{r,t}^{k+1}, \mathbf{w}_{r,t}^0 \rangle^2 \|\mathbf{w}_{r,t}^0\|^{-2} + \langle \mathbf{w}_{r,t}^{k+1}, \mathbf{w}_{r,t}^0 \rangle^2 \frac{\langle \mathbf{w}_{r,t}^0, \mathbf{w}_{r',t}^0 \rangle}{\|\mathbf{w}_{r,t}^0\|^4} \big)$$

$$= \Theta(\|\mathbf{w}_{r,t}^{k+1}\|^2)$$

where we use the induction claims (1-2) for $k+1$ and Lemma D.1. The last equality is by $\langle \mathbf{w}_{r,t}^{k+1}, \mathbf{w}_{r,t}^0 \rangle^2 \|\mathbf{w}_{r,t}^0\|^{-2} \le \Theta(\|\mathbf{w}_{r,t}^{k+1}\|^2)$. which completes all the induction. □

From Lemma D.5 and Lemma D.7, we know that for $T_1 \le k \le T_1^+$ we can decompose the gradient into two parts, the dominant term and the residual term:

$$\langle \nabla_{\mathbf{w}_{r,t}} L(\mathbf{W}_t^k), \boldsymbol{\mu}_j \rangle = -\frac{1}{\sqrt{m}} \Theta\Big( \|\mathbf{w}_{r,t}^k\|^2 \|\boldsymbol{\mu}\|^2 \Big) + E_{r,t,\mu_j}^k \tag{42}$$

$$\langle \nabla_{\mathbf{w}_{r,t}} L(\mathbf{W}_t^k), \boldsymbol{\xi}_i \rangle = -\frac{1}{n\sqrt{m}} \Theta\Big( \|\mathbf{w}_{r,t}^k\|^2 \|\boldsymbol{\xi}_i\|^2 \Big) + E_{r,t,\xi_i}^k \tag{43}$$

$$\langle \nabla_{\mathbf{w}_{r,t}} L(\mathbf{W}_t^k), \mathbf{w}_{r,t}^0 \rangle$$
$$= -\frac{1}{\sqrt{m}} \Theta\Big( \big( \langle \mathbf{w}_{r,t}^k, \boldsymbol{\mu}_j + \overline{\boldsymbol{\xi}} \rangle - \sqrt{m} \|\mathbf{w}_{r,t}^k\|^4 \big) \langle \mathbf{w}_{r,t}^k, \mathbf{w}_{r,t}^0 \rangle + \|\mathbf{w}_{r,t}^k\|^2 \langle \mathbf{w}_{r,t}^0, \boldsymbol{\mu}_j + \overline{\boldsymbol{\xi}} \rangle \Big) + E_{r,t,w^0}^k \tag{44}$$

where we let $E_{r,t,\mu_j}^k$, $E_{r,t,\xi_i}^k$, $E_{r,t,w^0}^k$ denote the residual terms. Therefore, before $E_{r,t,\mu_j}^k$, $E_{r,t,\xi_i}^k$ grow to reach $E_{r,t,\mu_j}^k = \Theta(\frac{1}{\sqrt{m}} \|\mathbf{w}_{r,t}^k\|^2 \|\boldsymbol{\mu}\|^2)$, $E_{r,t,\xi_i}^k = \Theta(\frac{1}{n\sqrt{m}} \|\mathbf{w}_{r,t}^k\|^2 \|\boldsymbol{\xi}_i\|^2)$, $E_{r,t,w^0}^k = \frac{1}{\sqrt{m}} \Theta((\langle \mathbf{w}_{r,t}^k, \boldsymbol{\mu}_j + \overline{\boldsymbol{\xi}} \rangle - \sqrt{m} \|\mathbf{w}_{r,t}^k\|^4) \langle \mathbf{w}_{r,t}^k, \mathbf{w}_{r,t}^0 \rangle + \|\mathbf{w}_{r,t}^k\|^2 \langle \mathbf{w}_{r,t}^0, \boldsymbol{\mu}_j + \overline{\boldsymbol{\xi}} \rangle)$, it can be verified that (35), (36), (37) are satisfied respectively. If further, $\langle \mathbf{w}_{r,t}^k, \boldsymbol{\mu}_j \rangle, \langle \mathbf{w}_{r,t}^k, \boldsymbol{\xi}_i \rangle = \widetilde{O}(1)$, $\langle \mathbf{w}_{r,t}^k, \mathbf{w}_{r,t}^0 \rangle = \Omega(\min\{\sigma_0 \sigma_\xi^{-1} n^{1/2} m^{-1/6}, \sigma_0 \sqrt{d} m^{-1/6}\})$ are satisfied, then we readily have $|\langle \mathbf{w}_{r,t}^k, \boldsymbol{\mu}_j \rangle| / |\langle \mathbf{w}_{r',t}^k, \boldsymbol{\xi}_i \rangle| = \Theta(n \cdot \mathrm{SNR}^2)$ by Lemma D.7.

The next lemma characterizes the end of second stage where the residual term reaches the same order as the dominant term.

**Lemma D.8** (Restatement of Lemma 4.2). *Consider the gradient decomposition defined in* (42), (43) *and* (44). *There exists $T_2 > T_1$ with $T_2 = \Theta(\max\{\eta^{-1} m^{1/3} \sigma_0^{-2} d^{-1}, \eta^{-1} m^{1/3} n \sigma_0^{-2} \sigma_\xi^2\})$ such that for all $j = \pm 1, r \in [m], i \in [n]$,*

*(1) If $n \cdot \mathrm{SNR}^2 = \Omega(1)$,*

$$\langle \mathbf{w}_{r,t}^{T_2}, \boldsymbol{\mu}_j \rangle = \Theta(m^{-1/6}), \quad \langle \mathbf{w}_{r,t}^{T_2}, \boldsymbol{\xi}_i \rangle = \Theta(\langle \mathbf{w}_{r,t}^{T_2}, \boldsymbol{\mu}_j \rangle),$$
$$\langle \mathbf{w}_{r,t}^{T_2}, \mathbf{w}_{r,t}^0 \rangle \|\mathbf{w}_{r,t}^0\|^{-2} \le \Theta(n \cdot \mathrm{SNR}^2 \langle \mathbf{w}_{r,t}^{T_2}, \boldsymbol{\xi}_i \rangle)$$

*If $n^{-1} \cdot \mathrm{SNR}^{-2} = \Omega(1)$,*

$$\langle \mathbf{w}_{r,t}^{T_2}, \boldsymbol{\mu}_j \rangle = \Theta(n \cdot \mathrm{SNR}^2 \cdot m^{-1/6}), \quad \langle \mathbf{w}_{r,t}^{T_2}, \boldsymbol{\xi}_i \rangle = \Theta(m^{-1/6})$$
$$\langle \mathbf{w}_{r,t}^{T_2}, \mathbf{w}_{r,t}^0 \rangle \|\mathbf{w}_{r,t}^0\|^{-2} \le \Theta(\sqrt{n \cdot \mathrm{SNR}^2} \langle \mathbf{w}_{r,t}^{T_2}, \boldsymbol{\xi}_i \rangle)$$

*(2) $E_{r,t,\mu_j}^{T_2} = \Theta(\frac{1}{\sqrt{m}} \|\mathbf{w}_{r,t}^{T_2}\|^2 \|\boldsymbol{\mu}\|^2)$, $E_{r,t,\xi_i}^{T_2} = \Theta(\frac{1}{n\sqrt{m}} \|\mathbf{w}_{r,t}^{T_2}\|^2 \|\boldsymbol{\xi}_i\|^2)$, $E_{r,t,w^0}^{T_2} = \frac{1}{\sqrt{m}} \Theta(\langle \mathbf{w}_{r,t}^k, \boldsymbol{\mu}_j + \boldsymbol{\xi}_i \rangle \langle \mathbf{w}_{r,t}^k, \mathbf{w}_{r,t}^0 \rangle + \|\mathbf{w}_{r,t}^k\|^2 \langle \mathbf{w}_{r,t}^0, \boldsymbol{\mu}_j + \boldsymbol{\xi}_i \rangle)$.*

*In addition, for any $T_1 \leq k \leq T_2$ and for all $j = \pm 1, r \in [m], i \in [n]$,*

(3) $\langle \mathbf{w}_{r,t}^k, \boldsymbol{\mu}_j \rangle = \Theta(\langle \mathbf{w}_{r',t}^k, \boldsymbol{\mu}_{j'} \rangle)$ *and* $\langle \mathbf{w}_{r,t}^k, \boldsymbol{\xi}_i \rangle = \Theta(\langle \mathbf{w}_{r',t}^k, \boldsymbol{\xi}_{i'} \rangle)$,

(4) $\|\mathbf{w}_{r,t}^k\|^2 = \Theta(\|\mathbf{w}_{r',t}^k\|^2)$ *and* $\langle \mathbf{w}_{r,t}^k, \mathbf{w}_{r',t}^k \rangle = \Theta(\|\mathbf{w}_{r,t}^k\|^2)$,

(5) $\langle \mathbf{w}_{r,t}^k, \boldsymbol{\mu}_j \rangle / \langle \mathbf{w}_{r,t}^k, \boldsymbol{\xi}_i \rangle = \Theta(n \cdot \mathrm{SNR}^2)$.

*Proof of Lemma D.8.* Here we let $T_2$ be the *first* time such that $E_{r,t,\mu_j}^{T_2} = \Theta(\frac{1}{\sqrt{m}}\|\mathbf{w}_{r,t}^{T_2}\|^2\|\boldsymbol{\mu}\|^2)$ or $E_{r,t,\xi_i}^{T_2} = \Theta(\frac{1}{n\sqrt{m}}\|\mathbf{w}_{r,t}^{T_2}\|^2\|\boldsymbol{\xi}_i\|^2)$ or $E_{r,t,w^0}^{T_2} = \frac{1}{\sqrt{m}}\Theta((\langle \mathbf{w}_{r,t}^k, \boldsymbol{\mu}_j + \overline{\boldsymbol{\xi}} \rangle - \sqrt{m}\|\mathbf{w}_{r,t}^k\|^4)\langle \mathbf{w}_{r,t}^k, \mathbf{w}_{r,t}^0 \rangle + \|\mathbf{w}_{r,t}^k\|^2\langle \mathbf{w}_{r,t}^0, \boldsymbol{\mu}_j + \overline{\boldsymbol{\xi}} \rangle)$. In order to prove the results, we use induction $k$ to simultaneously prove the following conditions $\mathscr{A}(k), \mathscr{B}(k), \mathscr{C}(k), \mathscr{D}(T_2), \mathscr{E}(T_2)$, for $T_1 \leq k \leq T_2$:

- $\mathscr{A}(k)$: $\langle \mathbf{w}_{r,t}^k, \boldsymbol{\mu}_j \rangle, \langle \mathbf{w}_{r,t}^k, \boldsymbol{\xi}_i \rangle = \widetilde{O}(1)$, $\langle \mathbf{w}_{r,t}^k, \mathbf{w}_{r,t}^0 \rangle = \Omega(\min\{\sigma_0 \sigma_\xi^{-1} n^{1/2} m^{-1/6}, \sigma_0 \sqrt{d} m^{-1/6}\})$ for all $j = \pm 1, r \in [m], i \in [n]$.

- $\mathscr{B}(k)$: $\langle \nabla_{\mathbf{w}_{r,t}} L(\mathbf{W}_t^k), \boldsymbol{\mu}_j \rangle = \Theta\left(-\frac{1}{\sqrt{m}}\|\mathbf{w}_{r,t}^k\|^2\|\boldsymbol{\mu}\|^2\right)$, $\langle \nabla_{\mathbf{w}_{r,t}} L(\mathbf{W}_t^k), \boldsymbol{\xi}_i \rangle = \Theta\left(-\frac{1}{n\sqrt{m}}\|\mathbf{w}_{r,t}^k\|^2\|\boldsymbol{\xi}_i\|^2\right)$ and $\langle \nabla_{\mathbf{w}_{r,t}} L(\mathbf{W}_t^k), \mathbf{w}_{r,t}^0 \rangle = -\frac{1}{\sqrt{m}}\Theta\left(\langle \mathbf{w}_{r,t}^k, \boldsymbol{\mu}_j + \overline{\boldsymbol{\xi}} \rangle \langle \mathbf{w}_{r,t}^k, \mathbf{w}_{r,t}^0 \rangle + \|\mathbf{w}_{r,t}^k\|^2\langle \mathbf{w}_{r,t}^0, \boldsymbol{\mu}_j + \overline{\boldsymbol{\xi}} \rangle\right)$ for all $j = \pm 1, r \in [m], i \in [n]$.

- $\mathscr{C}(k)$: Claims (3-5), i.e., $\langle \mathbf{w}_{r,t}^k, \boldsymbol{\mu}_j \rangle = \Theta(\langle \mathbf{w}_{r',t}^k, \boldsymbol{\mu}_{j'} \rangle)$, $\langle \mathbf{w}_{r,t}^k, \boldsymbol{\xi}_i \rangle = \Theta(\langle \mathbf{w}_{r',t}^k, \boldsymbol{\xi}_{i'} \rangle)$, $\|\mathbf{w}_{r,t}^k\|^2 = \Theta(\|\mathbf{w}_{r',t}^k\|^2)$, $\langle \mathbf{w}_{r,t}^k, \mathbf{w}_{r',t}^k \rangle = \Theta(\|\mathbf{w}_{r,t}^k\|^2)$, and $\langle \mathbf{w}_{r,t}^k, \boldsymbol{\mu}_j \rangle / \langle \mathbf{w}_{r,t}^k, \boldsymbol{\xi}_i \rangle = \Theta(n \cdot \mathrm{SNR}^2)$.

- $\mathscr{D}(T_2)$: Claim (1), i.e., If $n \cdot \mathrm{SNR}^2 = \Omega(1)$, $\langle \mathbf{w}_{r,t}^{T_2}, \boldsymbol{\mu}_j \rangle = \Theta(m^{-1/6})$, $\langle \mathbf{w}_{r,t}^{T_2}, \boldsymbol{\xi}_i \rangle = \Theta(n^{-1} \cdot \mathrm{SNR}^{-2} \cdot m^{-1/6})$, and if $n^{-1} \cdot \mathrm{SNR}^{-2} = \Omega(1)$, $\langle \mathbf{w}_{r,t}^{T_2}, \boldsymbol{\mu}_j \rangle = \Theta(n \cdot \mathrm{SNR}^2 \cdot m^{-1/6})$, $\langle \mathbf{w}_{r,t}^{T_2}, \boldsymbol{\xi}_i \rangle = \Theta(m^{-1/6})$.

- $\mathscr{E}(T_2)$: Claim (2), i.e., $E_{r,t,\mu_j}^{T_2} = \Theta(\frac{1}{\sqrt{m}}\|\mathbf{w}_{r,t}^{T_2}\|^2\|\boldsymbol{\mu}\|^2)$, $E_{r,t,\xi_i}^{T_2} = \Theta(\frac{1}{n\sqrt{m}}\|\mathbf{w}_{r,t}^{T_2}\|^2\|\boldsymbol{\xi}_i\|^2)$, and $E_{r,t,w^0}^{T_2} = \frac{1}{\sqrt{m}}\Theta\left((\langle \mathbf{w}_{r,t}^{T_2}, \boldsymbol{\mu}_j + \overline{\boldsymbol{\xi}} \rangle - \sqrt{m}\|\mathbf{w}_{r,t}^{T_2}\|^4)\langle \mathbf{w}_{r,t}^{T_2}, \mathbf{w}_{r,t}^0 \rangle + \|\mathbf{w}_{r,t}^{T_2}\|^2\langle \mathbf{w}_{r,t}^0, \boldsymbol{\mu}_j + \overline{\boldsymbol{\xi}} \rangle\right)$.

The initial conditions $\mathscr{A}(T_1), \mathscr{B}(T_1), \mathscr{C}(T_1)$ are satisfied by Lemma D.5 at the end of the first stage. In order to show $\mathscr{C}(k), \mathscr{D}(T_2), \mathscr{E}(T_2)$, we show the following claims respectively.

**Claim D.1.** $\mathscr{A}(k), \mathscr{B}(k) \Rightarrow \mathscr{C}(k)$, *for any* $T_1 \leq k \leq T_2$.

**Claim D.2.** $\mathscr{C}(T_1), ..., \mathscr{C}(T_2) \Rightarrow \mathscr{D}(T_2), \mathscr{E}(T_2)$.

**Claim D.3.** $\mathscr{D}(T_2), \mathscr{E}(T_2) \Rightarrow \mathscr{A}(T_1), ..., \mathscr{A}(T_2), \mathscr{B}(T_1), ..., \mathscr{B}(T_2)$.

**Proof of Claim D.1.** Claim D.1 directly follows from Lemma D.7.

**Proof of Claim D.2.** First, when $\mathscr{C}(k)$ is satisfied, we can simplify $\|\mathbf{w}_{r,t}^k\|^2$ from Lemma D.6

$$\|\mathbf{w}_{r,t}^k\|^2 = \Theta\left(\langle \mathbf{w}_{r,t}^k, \boldsymbol{\mu}_j \rangle^2\|\boldsymbol{\mu}\|^{-2} + n \cdot \mathrm{SNR}^2\langle \mathbf{w}_{r,t}^k, \boldsymbol{\xi}_i \rangle^2\|\boldsymbol{\mu}\|^{-2} + \langle \mathbf{w}_{r,t}^k, \mathbf{w}_{r,t}^0 \rangle^2\|\mathbf{w}_{r,t}^0\|^{-2}\right).$$
$$= \Theta\left((n^2\mathrm{SNR}^4 + n\mathrm{SNR}^2)\|\boldsymbol{\mu}\|^{-2}\langle \mathbf{w}_{r,t}^k, \boldsymbol{\xi}_i \rangle^2 + \langle \mathbf{w}_{r,t}^k, \mathbf{w}_{r,t}^0 \rangle^2\|\mathbf{w}_{r,t}^0\|^{-2}\right)$$
$$= \Theta\left((\chi^2 + \chi)\|\boldsymbol{\mu}\|^{-2}\langle \mathbf{w}_{r,t}^k, \boldsymbol{\xi}_i \rangle^2 + \psi_{r,t}^k\right)$$

where we temporarily denote $\chi := n \cdot \mathrm{SNR}^2$ and $\psi_{r,t}^k := \langle \mathbf{w}_{r,t}^k, \mathbf{w}_{r,t}^0 \rangle^2\|\mathbf{w}_{r,t}^0\|^{-2}$ for notation clarity.

Then, for the update of $\langle \mathbf{w}_{r,t}^k, \boldsymbol{\mu}_j \rangle$, we can compute

$$\frac{1}{2n}\sum_{i=1}^n \langle \nabla L_{1,i}^{(1)}(\mathbf{w}_{r,t}^k), \boldsymbol{\mu}_j \rangle$$
$$= \frac{1}{m}\Theta\left(\langle \mathbf{w}_{r,t}^k, \boldsymbol{\mu}_j \rangle^5 + \langle \mathbf{w}_{r,t}^k, \boldsymbol{\mu}_j \rangle^3\|\mathbf{w}_{r,t}^k\|^2 + \langle \mathbf{w}_{r,t}^k, \boldsymbol{\mu}_j \rangle\|\mathbf{w}_{r,t}^k\|^4 - \sqrt{m}\langle \mathbf{w}_{r,t}^k, \boldsymbol{\mu}_j \rangle^2\right)$$

$$
+ \frac{1}{m}\Theta\big(\langle \mathbf{w}_{r,t}^k, \boldsymbol{\mu}_j\rangle^3 \|\mathbf{w}_{r,t}^k\|^2 \|\boldsymbol{\mu}\|^2 + \langle \mathbf{w}_{r,t}^k, \boldsymbol{\mu}_j\rangle \|\mathbf{w}_{r,t}^k\|^4 \|\boldsymbol{\mu}\|^2 - \sqrt{m}\|\mathbf{w}_{r,t}^k\|^2 \|\boldsymbol{\mu}\|^2\big)
$$

$$
= \frac{1}{m}\Theta\big(\chi^5\langle \mathbf{w}_{r,t}^k, \boldsymbol{\xi}_i\rangle^5 + (\chi^5+\chi^4)\langle \mathbf{w}_{r,t}^k, \boldsymbol{\xi}_i\rangle^5 \|\boldsymbol{\mu}\|^{-2} + (\chi^5+\chi^3)\langle \mathbf{w}_{r,t}^k, \boldsymbol{\xi}_i\rangle^5 \|\boldsymbol{\mu}\|^{-4} - \sqrt{m}\chi^2\langle \mathbf{w}_{r,t}^k, \boldsymbol{\xi}_i\rangle^2\big)
$$

$$
+ \frac{1}{m}\Theta\big((\chi^5+\chi^4)\langle \mathbf{w}_{r,t}^k, \boldsymbol{\xi}_i\rangle^5 + (\chi^5+\chi^3)\langle \mathbf{w}_{r,t}^k, \boldsymbol{\xi}_i\rangle^5 \|\boldsymbol{\mu}\|^{-2} - \sqrt{m}(\chi^2+\chi)\langle \mathbf{w}_{r,t}^k, \boldsymbol{\xi}_i\rangle^2\big)
$$

$$
+ \frac{1}{m}\Theta\big(\psi_{r,t}^k\chi^3\langle \mathbf{w}_{r,t}^k, \boldsymbol{\xi}_i\rangle^3 \|\boldsymbol{\mu}\|^2 + (\psi_{r,t}^k)^2\chi\langle \mathbf{w}_{r,t}^k, \boldsymbol{\xi}_i\rangle\|\boldsymbol{\mu}\|^2 - \sqrt{m}\psi_{r,t}^k\|\boldsymbol{\mu}\|^2\big)
$$

$$
= \frac{1}{m}\Theta\big(-\sqrt{m}(\chi^2+\chi)\langle \mathbf{w}_{r,t}^k, \boldsymbol{\xi}_i\rangle^2 + (\chi^5+\chi^4)\langle \mathbf{w}_{r,t}^k, \boldsymbol{\xi}_i\rangle^5 + (\chi^5+\chi^3)\langle \mathbf{w}_{r,t}^k, \boldsymbol{\xi}_i\rangle^5 \|\boldsymbol{\mu}\|^{-2}\big)
$$

$$
+ \frac{1}{m}\Theta\big(\psi_{r,t}^k\chi^3\langle \mathbf{w}_{r,t}^k, \boldsymbol{\xi}_i\rangle^3 \|\boldsymbol{\mu}\|^2 + (\psi_{r,t}^k)^2\chi\langle \mathbf{w}_{r,t}^k, \boldsymbol{\xi}_i\rangle\|\boldsymbol{\mu}\|^2 - \sqrt{m}\psi_{r,t}^k\|\boldsymbol{\mu}\|^2\big)
$$

Similarly, we obtain

$$
\frac{1}{2n}\sum_{i=1}^n \langle \nabla L_{1,i}^{(2)}(\mathbf{w}_{r,t}), \boldsymbol{\mu}_j\rangle
$$

$$
= \frac{1}{m}\Theta\big(\langle \mathbf{w}_{r,t}^k, \boldsymbol{\xi}_i\rangle^4\langle \mathbf{w}_{r,t}^k, \boldsymbol{\mu}_j\rangle + \langle \mathbf{w}_{r,t}^k, \boldsymbol{\xi}_i\rangle^2\langle \mathbf{w}_{r,t}^k, \boldsymbol{\mu}_j\rangle\|\mathbf{w}_{r,t}^k\|^2 + \langle \mathbf{w}_{r,t}^k, \boldsymbol{\mu}_j\rangle\|\mathbf{w}_{r,t}^k\|^4
$$

$$
- \sqrt{m}\langle \mathbf{w}_{r,t}^k, \boldsymbol{\xi}_i\rangle\langle \mathbf{w}_{r,t}^k, \boldsymbol{\mu}_j\rangle\big)
$$

$$
= \frac{1}{m}\Theta\big(\chi\langle \mathbf{w}_{r,t}^k, \boldsymbol{\xi}_i\rangle^5 + (\chi^3+\chi^2)\langle \mathbf{w}_{r,t}^k, \boldsymbol{\xi}_i\rangle^5 \|\boldsymbol{\mu}\|^{-2} + (\chi^5+\chi^3)\langle \mathbf{w}_{r,t}^k, \boldsymbol{\xi}_i\rangle^5 \|\boldsymbol{\mu}\|^{-4}
$$

$$
- \sqrt{m}\chi\langle \mathbf{w}_{r,t}^k, \boldsymbol{\xi}_i\rangle^2\big) + \frac{1}{m}\Theta\big(\chi\psi_{r,t}^k\langle \mathbf{w}_{r,t}^k, \boldsymbol{\xi}_i\rangle^3 + \chi(\psi_{r,t}^k)^2\langle \mathbf{w}_{r,t}^k, \boldsymbol{\xi}_i\rangle\big)
$$

$$
= \frac{1}{m}\Theta\big(-\sqrt{m}\chi\langle \mathbf{w}_{r,t}^k, \boldsymbol{\xi}_i\rangle^2 + \chi\langle \mathbf{w}_{r,t}^k, \boldsymbol{\xi}_i\rangle^5 + (\chi^3+\chi^2)\langle \mathbf{w}_{r,t}^k, \boldsymbol{\xi}_i\rangle^5 \|\boldsymbol{\mu}\|^{-2} + (\chi^5+\chi^3)\langle \mathbf{w}_{r,t}^k, \boldsymbol{\xi}_i\rangle^5 \|\boldsymbol{\mu}\|^{-4}\big)
$$

$$
+ \frac{1}{m}\Theta\big(\chi\psi_{r,t}^k\langle \mathbf{w}_{r,t}^k, \boldsymbol{\xi}_i\rangle^3 + \chi(\psi_{r,t}^k)^2\langle \mathbf{w}_{r,t}^k, \boldsymbol{\xi}_i\rangle\big)
$$

$$
\frac{1}{2n}\sum_{i=1}^n \langle \nabla L_{2,i}^{(1)}(\mathbf{w}_{r,t}), \boldsymbol{\mu}_j\rangle
$$

$$
= \frac{m-1}{m}\Theta\big(\langle \mathbf{w}_{r,t}^k, \boldsymbol{\mu}_j\rangle^5 + \langle \mathbf{w}_{r,t}^k, \boldsymbol{\mu}_j\rangle^3\|\mathbf{w}_{r,t}^k\|^2 + \langle \mathbf{w}_{r,t}^k, \boldsymbol{\mu}_j\rangle\|\mathbf{w}_{r,t}^k\|^4 + \langle \mathbf{w}_{r,t}^k, \boldsymbol{\mu}_j\rangle\|\mathbf{w}_{r,t}^k\|^4\|\boldsymbol{\mu}\|^2
$$

$$
+ \langle \mathbf{w}_{r,t}^k, \boldsymbol{\mu}_j\rangle^3\|\mathbf{w}_{r,t}^k\|^2\|\boldsymbol{\mu}\|^2\big)
$$

$$
= \frac{m-1}{m}\Theta\big((\chi^5+\chi^4)\langle \mathbf{w}_{r,t}^k, \boldsymbol{\xi}_i\rangle^5 + (\chi^5+\chi^3)\langle \mathbf{w}_{r,t}^k, \boldsymbol{\xi}_i\rangle^5 \|\boldsymbol{\mu}\|^{-2}\big)
$$

$$
+ \frac{m-1}{m}\Theta\big(\psi_{r,t}^k\chi^3\langle \mathbf{w}_{r,t}^k, \boldsymbol{\xi}_i\rangle^3 \|\boldsymbol{\mu}\|^2 + (\psi_{r,t}^k)^2\chi\langle \mathbf{w}_{r,t}^k, \boldsymbol{\xi}_i\rangle\|\boldsymbol{\mu}\|^2 - \sqrt{m}\psi_{r,t}^k\|\boldsymbol{\mu}\|^2\big)
$$

$$
\frac{1}{2n}\sum_{i=1}^n \langle \nabla L_{2,i}^{(2)}(\mathbf{w}_{r,t}), \boldsymbol{\mu}_j\rangle
$$

$$
= \frac{m-1}{m}\Theta\big(\langle \mathbf{w}_{r,t}^k, \boldsymbol{\mu}_j\rangle\langle \mathbf{w}_{r,t}^k, \boldsymbol{\xi}_i\rangle^4 + \langle \mathbf{w}_{r,t}^k, \boldsymbol{\mu}_j\rangle\|\mathbf{w}_{r,t}^k\|^4 + \langle \mathbf{w}_{r,t}^k, \boldsymbol{\mu}_j\rangle\langle \mathbf{w}_{r,t}^k, \boldsymbol{\xi}_i\rangle^2\|\mathbf{w}_{r,t}^k\|^2\big)
$$

$$
= \frac{m-1}{m}\Theta\big(\chi\langle \mathbf{w}_{r,t}^k, \boldsymbol{\xi}_i\rangle^5 + (\chi^3+\chi^2)\langle \mathbf{w}_{r,t}^k, \boldsymbol{\xi}_i\rangle^5 \|\boldsymbol{\mu}\|^{-2} + (\chi^5+\chi^3)\langle \mathbf{w}_{r,t}^k, \boldsymbol{\xi}_i\rangle^5 \|\boldsymbol{\mu}\|^{-4}\big)
$$

$$
+ \frac{m-1}{m}\Theta\big(\chi\psi_{r,t}^k\langle \mathbf{w}_{r,t}^k, \boldsymbol{\xi}_i\rangle^3 + \chi(\psi_{r,t}^k)^2\langle \mathbf{w}_{r,t}^k, \boldsymbol{\xi}_i\rangle\big)
$$

Combining the above results, we have

$$
\langle \nabla_{\mathbf{w}_{r,t}} L(\mathbf{W}_t^k), \boldsymbol{\mu}_j\rangle = -\frac{1}{\sqrt{m}}\Theta\Big(\|\mathbf{w}_{r,t}^k\|^2\|\boldsymbol{\mu}\|^2\Big) + E_{r,t,\mu_j}^k
$$

$$
= -\frac{1}{\sqrt{m}}\Theta\big((\chi^2+\chi)\langle \mathbf{w}_{r,t}^k, \boldsymbol{\xi}_i\rangle^2 + \psi_{r,t}^k\|\boldsymbol{\mu}\|^2\big)
$$

$$+ \Theta\Big((\chi^5 + \chi^4)\langle \mathbf{w}_{r,t}^k, \boldsymbol{\xi}_i \rangle^5 + (\chi^5 + \chi^3)\langle \mathbf{w}_{r,t}^k, \boldsymbol{\xi}_i \rangle^5 \|\boldsymbol{\mu}\|^{-2}\Big)$$

$$+ \Theta\Big(\chi\langle \mathbf{w}_{r,t}^k, \boldsymbol{\xi}_i \rangle^5 + (\chi^3 + \chi^2)\langle \mathbf{w}_{r,t}^k, \boldsymbol{\xi}_i \rangle^5 \|\boldsymbol{\mu}\|^{-2} + (\chi^5 + \chi^3)\langle \mathbf{w}_{r,t}^k, \boldsymbol{\xi}_i \rangle^5 \|\boldsymbol{\mu}\|^{-4}\Big)$$

$$+ \Theta(\psi_{r,t}^k \chi^3 \langle \mathbf{w}_{r,t}^k, \boldsymbol{\xi}_i \rangle^3 \|\boldsymbol{\mu}\|^2 + (\psi_{r,t}^k)^2 \chi \langle \mathbf{w}_{r,t}^k, \boldsymbol{\xi}_i \rangle \|\boldsymbol{\mu}\|^2 + \chi \psi_{r,t}^k \langle \mathbf{w}_{r,t}^k, \boldsymbol{\xi}_i \rangle^3)$$

Similarly, we can derive for the update of $\langle \mathbf{w}_{r,t}^k, \boldsymbol{\xi}_i \rangle$ as follows:

$$\frac{1}{2n} \sum_{i=1}^n \langle \nabla L_{1,i}^{(1)}(\mathbf{w}_{r,t}^k), \boldsymbol{\xi}_i \rangle$$

$$= \frac{1}{m} \Theta\Big( \langle \mathbf{w}_{r,t}^k, \boldsymbol{\mu}_j \rangle^4 \langle \mathbf{w}_{r,t}^k, \boldsymbol{\xi}_i \rangle + \langle \mathbf{w}_{r,t}^k, \boldsymbol{\mu}_j \rangle^2 \|\mathbf{w}_{r,t}^k\|^2 \langle \mathbf{w}_{r,t}^k, \boldsymbol{\xi}_i \rangle + \|\mathbf{w}_{r,t}^k\|^4 \langle \mathbf{w}_{r,t}^k, \boldsymbol{\xi}_i \rangle$$

$$- \sqrt{m}\langle \mathbf{w}_{r,t}^k, \boldsymbol{\mu}_j \rangle \langle \mathbf{w}_{r,t}^k, \boldsymbol{\xi}_i \rangle \Big)$$

$$= \frac{1}{m} \Theta\big(\chi^4 \langle \mathbf{w}_{r,t}^k, \boldsymbol{\xi}_i \rangle^5 + (\chi^4 + \chi^3)\langle \mathbf{w}_{r,t}^k, \boldsymbol{\xi}_i \rangle^5 \|\boldsymbol{\mu}\|^{-2} + (\chi^4 + \chi^2)\langle \mathbf{w}_{r,t}^k, \boldsymbol{\xi}_i \rangle^5 \|\boldsymbol{\mu}\|^{-4} - \sqrt{m}\chi \langle \mathbf{w}_{r,t}^k, \boldsymbol{\xi}_i \rangle^2\big)$$

$$+ \frac{1}{m} \Theta(\psi_{r,t}^k \chi^2 \langle \mathbf{w}_{r,t}^k, \boldsymbol{\xi}_i \rangle^3 + (\psi_{r,t}^k)^2 \langle \mathbf{w}_{r,t}^k, \boldsymbol{\xi}_i \rangle)$$

$$= \frac{1}{m} \Theta\big( - \sqrt{m}\chi \langle \mathbf{w}_{r,t}^k, \boldsymbol{\xi}_i \rangle^2 + \chi^4 \langle \mathbf{w}_{r,t}^k, \boldsymbol{\xi}_i \rangle^5 + (\chi^4 + \chi^3)\langle \mathbf{w}_{r,t}^k, \boldsymbol{\xi}_i \rangle^5 \|\boldsymbol{\mu}\|^{-2} + (\chi^4 + \chi^2)\langle \mathbf{w}_{r,t}^k, \boldsymbol{\xi}_i \rangle^5 \|\boldsymbol{\mu}\|^{-4}\big)$$

$$+ \frac{1}{m} \Theta(\psi_{r,t}^k \chi^2 \langle \mathbf{w}_{r,t}^k, \boldsymbol{\xi}_i \rangle^3 + (\psi_{r,t}^k)^2 \langle \mathbf{w}_{r,t}^k, \boldsymbol{\xi}_i \rangle)$$

$$\frac{1}{2n} \sum_{i=1}^n \langle \nabla L_{1,i}^{(2)}(\mathbf{w}_{r,t}^k), \boldsymbol{\xi}_i \rangle$$

$$= \frac{1}{m} \Theta\Big( \langle \mathbf{w}_{r,t}^k, \boldsymbol{\xi}_i \rangle^5 + \langle \mathbf{w}_{r,t}^k, \boldsymbol{\xi}_i \rangle^3 \|\mathbf{w}_{r,t}^k\|^2 + \|\mathbf{w}_{r,t}^k\|^4 \langle \mathbf{w}_{r,t}^k, \boldsymbol{\xi}_i \rangle - \sqrt{m}\langle \mathbf{w}_{r,t}^k, \boldsymbol{\xi}_i \rangle^2 \Big)$$

$$+ \frac{1}{nm} \Theta\Big( \langle \mathbf{w}_{r,t}^k, \boldsymbol{\xi}_i \rangle^3 \|\mathbf{w}_{r,t}^k\|^2 \|\boldsymbol{\xi}_i\|^2 + \|\mathbf{w}_{r,t}^k\|^4 \langle \mathbf{w}_{r,t}^k, \boldsymbol{\xi}_i \rangle \|\boldsymbol{\xi}_i\|^2 - \sqrt{m}\|\mathbf{w}_{r,t}^k\|^2 \|\boldsymbol{\xi}_i\|^2 \Big)$$

$$= \frac{1}{m} \Theta\Big( \langle \mathbf{w}_{r,t}^k, \boldsymbol{\xi}_i \rangle^5 + (\chi^2 + \chi)\langle \mathbf{w}_{r,t}^k, \boldsymbol{\xi}_i \rangle^5 \|\boldsymbol{\mu}\|^{-2} + (\chi^4 + \chi^2)\langle \mathbf{w}_{r,t}^k, \boldsymbol{\xi}_i \rangle^5 \|\boldsymbol{\mu}\|^{-4} - \sqrt{m}\langle \mathbf{w}_{r,t}^k, \boldsymbol{\xi}_i \rangle^2 \Big)$$

$$+ \frac{1}{\chi m} \Theta\Big( (\chi^2 + \chi)\langle \mathbf{w}_{r,t}^k, \boldsymbol{\xi}_i \rangle^5 + (\chi^4 + \chi^2)\langle \mathbf{w}_{r,t}^k, \boldsymbol{\xi}_i \rangle^5 \|\boldsymbol{\mu}\|^{-2} - \sqrt{m}(\chi^2 + \chi)\langle \mathbf{w}_{r,t}^k, \boldsymbol{\xi}_i \rangle^2 \Big)$$

$$+ \frac{1}{m} \Theta\big(\psi_{r,t}^k \langle \mathbf{w}_{r,t}^k, \boldsymbol{\xi}_i \rangle^3 + (\psi_{r,t}^k)^2 \langle \mathbf{w}_{r,t}^k, \boldsymbol{\xi}_i \rangle + \chi^{-1}\psi_{r,t}^k \langle \mathbf{w}_{r,t}^k, \boldsymbol{\xi}_i \rangle^3 \|\boldsymbol{\mu}\|^2 + \chi^{-1}(\psi_{r,t}^k)^2 \langle \mathbf{w}_{r,t}^k, \boldsymbol{\xi}_i \rangle \|\boldsymbol{\mu}\|^2$$

$$- \chi^{-1}\sqrt{m}\psi_{r,t}^k \|\boldsymbol{\mu}\|^2\big)$$

$$= \frac{1}{m} \Theta\big( - \sqrt{m}(\chi + 1)\langle \mathbf{w}_{r,t}^k, \boldsymbol{\xi}_i \rangle^2 + (\chi^2 + \chi)\langle \mathbf{w}_{r,t}^k, \boldsymbol{\xi}_i \rangle^5 \|\boldsymbol{\mu}\|^{-2} + (\chi^4 + \chi^2)\langle \mathbf{w}_{r,t}^k, \boldsymbol{\xi}_i \rangle^5 \|\boldsymbol{\mu}\|^{-4}$$

$$+ (\chi + 1)\langle \mathbf{w}_{r,t}^k, \boldsymbol{\xi}_i \rangle^5 + (\chi^3 + \chi)\langle \mathbf{w}_{r,t}^k, \boldsymbol{\xi}_i \rangle^5 \|\boldsymbol{\mu}\|^{-2}\big)$$

$$+ \frac{1}{m} \Theta\big(\psi_{r,t}^k \langle \mathbf{w}_{r,t}^k, \boldsymbol{\xi}_i \rangle^3 + (\psi_{r,t}^k)^2 \langle \mathbf{w}_{r,t}^k, \boldsymbol{\xi}_i \rangle + \chi^{-1}\psi_{r,t}^k \langle \mathbf{w}_{r,t}^k, \boldsymbol{\xi}_i \rangle^3 \|\boldsymbol{\mu}\|^2 + \chi^{-1}(\psi_{r,t}^k)^2 \langle \mathbf{w}_{r,t}^k, \boldsymbol{\xi}_i \rangle \|\boldsymbol{\mu}\|^2$$

$$- \chi^{-1}\sqrt{m}\psi_{r,t}^k \|\boldsymbol{\mu}\|^2\big)$$

where the second equality follows from $\sum_{i'=1}^n \langle \boldsymbol{\xi}_{i'}, \boldsymbol{\xi}_i \rangle = (1 + \widetilde{O}(nd^{-1/2}))\|\boldsymbol{\xi}_i\|^2 = \Theta(\|\boldsymbol{\xi}_i\|^2)$ by Lemma B.2 and condition on $d$. Further,

$$\frac{1}{2n} \sum_{i'=1}^n \langle \nabla L_{2,i'}^{(1)}(\mathbf{w}_{r,t}^k), \boldsymbol{\xi}_i \rangle$$

$$= \frac{m-1}{m} \Theta\Big( \langle \mathbf{w}_{r,t}^k, \boldsymbol{\mu}_j \rangle^4 \langle \mathbf{w}_{r,t}^k, \boldsymbol{\xi}_i \rangle + \|\mathbf{w}_{r,t}^k\|^4 \langle \mathbf{w}_{r,t}^k, \boldsymbol{\xi}_i \rangle + \langle \mathbf{w}_{r,t}^k, \boldsymbol{\mu}_j \rangle^2 \|\mathbf{w}_{r,t}^k\|^2 \langle \mathbf{w}_{r,t}^k, \boldsymbol{\xi}_i \rangle \Big)$$

$$= \frac{m-1}{m} \Theta\big(\chi^4 \langle \mathbf{w}_{r,t}^k, \boldsymbol{\xi}_i \rangle^5 + (\chi^4 + \chi^2)\langle \mathbf{w}_{r,t}^k, \boldsymbol{\xi}_i \rangle^5 \|\boldsymbol{\mu}\|^{-4} + (\chi^4 + \chi^3)\langle \mathbf{w}_{r,t}^k, \boldsymbol{\xi}_i \rangle^5 \|\boldsymbol{\mu}\|^{-2}\big)$$

$$+ \frac{m-1}{m} \Theta\big(\psi_{r,t}^k \chi^2 \langle \mathbf{w}_{r,t}^k, \boldsymbol{\xi}_i \rangle^3 + (\psi_{r,t}^k)^2 \langle \mathbf{w}_{r,t}^k, \boldsymbol{\xi}_i \rangle\big)$$

$$\frac{1}{2n}\sum_{i'=1}^{n}\langle\nabla L_{2,i'}^{(2)}(\mathbf{w}_{r,t}^k),\boldsymbol{\xi}_i\rangle$$

$$=\frac{m-1}{m}\Theta\Big(\langle\mathbf{w}_{r,t}^k,\boldsymbol{\xi}_i\rangle^5+\|\mathbf{w}_{r,t}^k\|^4\langle\mathbf{w}_{r,t}^k,\boldsymbol{\xi}_i\rangle+\langle\mathbf{w}_{r,t}^k,\boldsymbol{\xi}_i\rangle^3\|\mathbf{w}_{r,t}^k\|^2\Big)$$

$$+\frac{m-1}{nm}\Theta\Big(\langle\mathbf{w}_{r,t}^k,\boldsymbol{\xi}_i\rangle^3\|\mathbf{w}_{r,t}^k\|^2\|\boldsymbol{\xi}_i\|^2+\|\mathbf{w}_{r,t}^k\|^4\langle\mathbf{w}_{r,t}^k,\boldsymbol{\xi}_i\rangle\|\boldsymbol{\xi}_i\|^2\Big)$$

$$=\frac{m-1}{m}\Theta\big(\langle\mathbf{w}_{r,t}^k,\boldsymbol{\xi}_i\rangle^5+(\chi^4+\chi^2)\langle\mathbf{w}_{r,t}^k,\boldsymbol{\xi}_i\rangle^5\|\boldsymbol{\mu}\|^{-4}+(\chi^2+\chi)\langle\mathbf{w}_{r,t}^k,\boldsymbol{\xi}_i\rangle^5\|\boldsymbol{\mu}\|^{-2}\big)$$

$$+\frac{m-1}{\chi m}\Theta\big((\chi^2+\chi)\langle\mathbf{w}_{r,t}^k,\boldsymbol{\xi}_i\rangle^5+(\chi^4+\chi^2)\langle\mathbf{w}_{r,t}^k,\boldsymbol{\xi}_i\rangle^5\|\boldsymbol{\mu}\|^{-2}\big)$$

$$+\frac{m-1}{m}\Theta\big(\psi_{r,t}^k\langle\mathbf{w}_{r,t}^k,\boldsymbol{\xi}_i\rangle^3+(\psi_{r,t}^k)^2\langle\mathbf{w}_{r,t}^k,\boldsymbol{\xi}_i\rangle+\chi^{-1}\psi_{r,t}^k\langle\mathbf{w}_{r,t}^k,\boldsymbol{\xi}_i\rangle^3\|\boldsymbol{\mu}\|^2+\chi^{-1}(\psi_{r,t}^k)^2\langle\mathbf{w}_{r,t}^k,\boldsymbol{\xi}_i\rangle\|\boldsymbol{\mu}\|^2$$

$$-\chi^{-1}\sqrt{m}\psi_{r,t}^k\|\boldsymbol{\mu}\|^2\big)$$

$$=\frac{m-1}{m}\Theta\big((\chi^2+\chi)\langle\mathbf{w}_{r,t}^k,\boldsymbol{\xi}_i\rangle^5\|\boldsymbol{\mu}\|^{-2}+(\chi^4+\chi^2)\langle\mathbf{w}_{r,t}^k,\boldsymbol{\xi}_i\rangle^5\|\boldsymbol{\mu}\|^{-4}+(\chi+1)\langle\mathbf{w}_{r,t}^k,\boldsymbol{\xi}_i\rangle^5$$

$$+(\chi^3+\chi)\langle\mathbf{w}_{r,t}^k,\boldsymbol{\xi}_i\rangle^5\|\boldsymbol{\mu}\|^{-2}\big)$$

$$+\frac{m-1}{m}\Theta\big(\psi_{r,t}^k\langle\mathbf{w}_{r,t}^k,\boldsymbol{\xi}_i\rangle^3+(\psi_{r,t}^k)^2\langle\mathbf{w}_{r,t}^k,\boldsymbol{\xi}_i\rangle+\chi^{-1}\psi_{r,t}^k\langle\mathbf{w}_{r,t}^k,\boldsymbol{\xi}_i\rangle^3\|\boldsymbol{\mu}\|^2+\chi^{-1}(\psi_{r,t}^k)^2\langle\mathbf{w}_{r,t}^k,\boldsymbol{\xi}_i\rangle\|\boldsymbol{\mu}\|^2$$

$$-\chi^{-1}\sqrt{m}\psi_{r,t}^k\|\boldsymbol{\mu}\|^2\big)$$

Combining the above results, we have

$$\langle\nabla_{\mathbf{w}_{r,t}}L(\mathbf{W}_t^k),\boldsymbol{\xi}_i\rangle=-\frac{1}{n\sqrt{m}}\Theta\Big(\|\mathbf{w}_{r,t}^k\|^2\|\boldsymbol{\xi}_i\|^2\Big)+E_{r,t,\xi_i}^k$$

$$=-\frac{1}{\sqrt{m}}\Theta\Big((\chi+1)\langle\mathbf{w}_{r,t}^k,\boldsymbol{\xi}_i\rangle^2+\chi^{-1}\psi_{r,t}^k\|\boldsymbol{\mu}\|^2\Big)$$

$$+\Theta\Big(\chi^4\langle\mathbf{w}_{r,t}^k,\boldsymbol{\xi}_i\rangle^5+(\chi^4+\chi^3)\langle\mathbf{w}_{r,t}^k,\boldsymbol{\xi}_i\rangle^5\|\boldsymbol{\mu}\|^{-2}+(\chi^4+\chi^2)\langle\mathbf{w}_{r,t}^k,\boldsymbol{\xi}_i\rangle^5\|\boldsymbol{\mu}\|^{-4}\Big)$$

$$+\Theta\Big((\chi^2+\chi)\langle\mathbf{w}_{r,t}^k,\boldsymbol{\xi}_i\rangle^5\|\boldsymbol{\mu}\|^{-2}+(\chi^4+\chi^2)\langle\mathbf{w}_{r,t}^k,\boldsymbol{\xi}_i\rangle^5\|\boldsymbol{\mu}\|^{-4}$$

$$+\Theta\Big((\chi+1)\langle\mathbf{w}_{r,t}^k,\boldsymbol{\xi}_i\rangle^5+(\chi^3+\chi)\langle\mathbf{w}_{r,t}^k,\boldsymbol{\xi}_i\rangle^5\|\boldsymbol{\mu}\|^{-2}\Big)$$

$$+\Theta\Big(\psi_{r,t}^k\chi^2\langle\mathbf{w}_{r,t}^k,\boldsymbol{\xi}_i\rangle^3+(\psi_{r,t}^k)^2\langle\mathbf{w}_{r,t}^k,\boldsymbol{\xi}_i\rangle+\psi_{r,t}^k\langle\mathbf{w}_{r,t}^k,\boldsymbol{\xi}_i\rangle^3+\chi^{-1}\psi_{r,t}^k\langle\mathbf{w}_{r,t}^k,\boldsymbol{\xi}_i\rangle^3\|\boldsymbol{\mu}\|^2$$

$$+\chi^{-1}(\psi_{r,t}^k)^2\langle\mathbf{w}_{r,t}^k,\boldsymbol{\xi}_i\rangle\|\boldsymbol{\mu}\|^2\Big)$$

In summary, we finally arrive at

$$\langle\nabla_{\mathbf{w}_{r,t}}L(\mathbf{W}_t^k),\boldsymbol{\mu}_j\rangle=-\frac{1}{\sqrt{m}}\Theta\Big(\|\mathbf{w}_{r,t}^k\|^2\|\boldsymbol{\mu}\|^2\Big)+E_{r,t,\mu_j}^k$$

$$=-\frac{1}{\sqrt{m}}\Theta\big((\chi^2+\chi)\langle\mathbf{w}_{r,t}^k,\boldsymbol{\xi}_i\rangle^2+\psi_{r,t}^k\|\boldsymbol{\mu}\|^2\big)$$

$$+\Theta\Big((\chi^5+\chi^4)\langle\mathbf{w}_{r,t}^k,\boldsymbol{\xi}_i\rangle^5+(\chi^5+\chi^3)\langle\mathbf{w}_{r,t}^k,\boldsymbol{\xi}_i\rangle^5\|\boldsymbol{\mu}\|^{-2}\Big)$$

$$+\Theta\Big(\chi\langle\mathbf{w}_{r,t}^k,\boldsymbol{\xi}_i\rangle^5+(\chi^3+\chi^2)\langle\mathbf{w}_{r,t}^k,\boldsymbol{\xi}_i\rangle^5\|\boldsymbol{\mu}\|^{-2}+(\chi^5+\chi^3)\langle\mathbf{w}_{r,t}^k,\boldsymbol{\xi}_i\rangle^5\|\boldsymbol{\mu}\|^{-4}\Big)$$

$$+\Theta(\psi_{r,t}^k\chi^3\langle\mathbf{w}_{r,t}^k,\boldsymbol{\xi}_i\rangle^3\|\boldsymbol{\mu}\|^2+(\psi_{r,t}^k)^2\chi\langle\mathbf{w}_{r,t}^k,\boldsymbol{\xi}_i\rangle\|\boldsymbol{\mu}\|^2+\chi\psi_{r,t}^k\langle\mathbf{w}_{r,t}^k,\boldsymbol{\xi}_i\rangle^3)\qquad(45)$$

$$\langle\nabla_{\mathbf{w}_{r,t}}L(\mathbf{W}_t^k),\boldsymbol{\xi}_i\rangle=-\frac{1}{n\sqrt{m}}\Theta\Big(\|\mathbf{w}_{r,t}^k\|^2\|\boldsymbol{\xi}_i\|^2\Big)+E_{r,t,\xi_i}^k$$

$$=-\frac{1}{\sqrt{m}}\Theta\Big((\chi+1)\langle\mathbf{w}_{r,t}^k,\boldsymbol{\xi}_i\rangle^2+\chi^{-1}\psi_{r,t}^k\|\boldsymbol{\mu}\|^2\Big)$$

$$+\Theta\Big(\chi^4\langle\mathbf{w}_{r,t}^k,\boldsymbol{\xi}_i\rangle^5+(\chi^4+\chi^3)\langle\mathbf{w}_{r,t}^k,\boldsymbol{\xi}_i\rangle^5\|\boldsymbol{\mu}\|^{-2}+(\chi^4+\chi^2)\langle\mathbf{w}_{r,t}^k,\boldsymbol{\xi}_i\rangle^5\|\boldsymbol{\mu}\|^{-4}\Big)$$

$$+\Theta\Big((\chi^2+\chi)\langle\mathbf{w}_{r,t}^k,\boldsymbol{\xi}_i\rangle^5\|\boldsymbol{\mu}\|^{-2}+(\chi^4+\chi^2)\langle\mathbf{w}_{r,t}^k,\boldsymbol{\xi}_i\rangle^5\|\boldsymbol{\mu}\|^{-4}$$

$$+ \Theta\Big((\chi+1)\langle \mathbf{w}_{r,t}^k, \boldsymbol{\xi}_i\rangle^5 + (\chi^3+\chi)\langle \mathbf{w}_{r,t}^k, \boldsymbol{\xi}_i\rangle^5 \|\boldsymbol{\mu}\|^{-2}\Big)$$

$$+ \Theta\Big(\psi_{r,t}^k \chi^2 \langle \mathbf{w}_{r,t}^k, \boldsymbol{\xi}_i\rangle^3 + (\psi_{r,t}^k)^2 \langle \mathbf{w}_{r,t}^k, \boldsymbol{\xi}_i\rangle + \psi_{r,t}^k \langle \mathbf{w}_{r,t}^k, \boldsymbol{\xi}_i\rangle^3 + \chi^{-1}\psi_{r,t}^k\langle \mathbf{w}_{r,t}^k, \boldsymbol{\xi}_i\rangle^3 \|\boldsymbol{\mu}\|^2$$

$$+ \chi^{-1}(\psi_{r,t}^k)^2 \langle \mathbf{w}_{r,t}^k, \boldsymbol{\xi}_i\rangle \|\boldsymbol{\mu}\|^2\Big) \tag{46}$$

We also examine $\langle \nabla_{\mathbf{w}_{r,t}} L(\mathbf{W}_t^k), \mathbf{w}_{r,t}^0\rangle$ as follows. We first upper bound for $r' \neq r$

$$\langle \mathbf{w}_{r,t}^k, \mathbf{w}_{r',t}^0\rangle = \Theta(\langle \mathbf{w}_{r,t}^k, \mathbf{w}_{r,t}^0\rangle \|\mathbf{w}_{r,t}^0\|^{-2})\langle \mathbf{w}_{r,t}^0, \mathbf{w}_{r',t}^0\rangle + \Theta(\langle \mathbf{w}_{r,t}^k, \boldsymbol{\mu}_j\rangle\langle \mathbf{w}_{r',t}^0, \boldsymbol{\mu}_j\rangle\|\boldsymbol{\mu}\|^{-2})$$

$$+ \Theta(\langle \mathbf{w}_{r,t}^k, \boldsymbol{\xi}_i\rangle)\sum_{i=1}^n \langle \mathbf{w}_{r',t}^0, \boldsymbol{\xi}_i\rangle\|\boldsymbol{\xi}_i\|^{-2}$$

$$= \widetilde{O}(\sigma_0) + \widetilde{O}(\sigma_0) + \widetilde{O}(n\sigma_0\sigma_\xi^{-1}d^{-1/2})$$

$$= \widetilde{O}(\sigma_0) + \widetilde{O}(n\sigma_0\sigma_\xi^{-1}d^{-1/2})$$

where we use (34) in the first equality and Lemma D.1, Lemma D.7 in the second equality.

Next we simplify the gradient as

$$\langle \nabla_{\mathbf{w}_{r,t}} L(\mathbf{W}_t^k), \mathbf{w}_{r,t}^0\rangle$$

$$= -\frac{1}{\sqrt{m}}\Theta\Big(\langle \mathbf{w}_{r,t}^k, \boldsymbol{\mu}_j + \boldsymbol{\xi}_i\rangle\langle \mathbf{w}_{r,t}^k, \mathbf{w}_{r,t}^0\rangle + \|\mathbf{w}_{r,t}^k\|^2\langle \mathbf{w}_{r,t}^0, \boldsymbol{\mu}_j + \boldsymbol{\xi}_i\rangle\Big)$$

$$+ \frac{1}{m}\Theta\Big(\big((\langle \mathbf{w}_{r,t}^k, \boldsymbol{\mu}_j\rangle^4 + \langle \mathbf{w}_{r,t}^k, \boldsymbol{\xi}_i\rangle^4\big)\langle \mathbf{w}_{r,t}^k, \mathbf{w}_{r,t}^0\rangle + \big(\langle \mathbf{w}_{r,t}^k, \boldsymbol{\mu}_j\rangle^2 + \langle \mathbf{w}_{r,t}^k, \boldsymbol{\xi}_i\rangle^2\big)\|\mathbf{w}_{r,t}^k\|^2\langle \mathbf{w}_{r,t}^k, \mathbf{w}_{r,t}^0\rangle\Big)$$

$$+ \frac{1}{m}\Theta\Big(\|\mathbf{w}_{r,t}^k\|^4\langle \mathbf{w}_{r,t}^k, \mathbf{w}_{r,t}^0\rangle + \big(\langle \mathbf{w}_{r,t}^k, \boldsymbol{\mu}_j\rangle^3\langle \mathbf{w}_{r,t}^0, \boldsymbol{\mu}_j\rangle + \langle \mathbf{w}_{r,t}^k, \boldsymbol{\xi}_i\rangle^3\langle \mathbf{w}_{r,t}^0, \boldsymbol{\xi}_i\rangle\big)\|\mathbf{w}_{r,t}^k\|^2\Big)$$

$$+ \frac{1}{m}\Theta\Big(\big(\langle \mathbf{w}_{r,t}^k, \boldsymbol{\mu}_j\rangle\langle \mathbf{w}_{r,t}^0, \boldsymbol{\mu}_j\rangle + \langle \mathbf{w}_{r,t}^k, \boldsymbol{\xi}_i\rangle\langle \mathbf{w}_{r,t}^0, \boldsymbol{\xi}_i\rangle\big)\|\mathbf{w}_{r,t}^k\|^4\Big)$$

$$+ \frac{m-1}{m}\Theta\Big(\langle \mathbf{w}_{r,t}^k, \boldsymbol{\mu}_j\rangle^4 + \langle \mathbf{w}_{r,t}^k, \boldsymbol{\xi}_i\rangle^4 + \|\mathbf{w}_{r,t}^k\|^2\big(\langle \mathbf{w}_{r,t}^k, \boldsymbol{\mu}_j\rangle^2 + \langle \mathbf{w}_{r,t}^k, \boldsymbol{\xi}_i\rangle^2\big)$$

$$+ \|\mathbf{w}_{r,t}^k\|^4\Big)\widetilde{O}(\sigma_0 + n\sigma_0\sigma_\xi^{-1}d^{-1/2})$$

$$+ \frac{m-1}{m}\Theta\Big(\big(\langle \mathbf{w}_{r,t}^k, \boldsymbol{\mu}_j\rangle^3\langle \mathbf{w}_{r,t}^0, \boldsymbol{\mu}_j\rangle + \langle \mathbf{w}_{r,t}^k, \boldsymbol{\xi}_i\rangle^3\langle \mathbf{w}_{r,t}^0, \boldsymbol{\xi}_i\rangle\big)\|\mathbf{w}_{r,t}^k\|^2\Big)$$

$$+ \frac{m-1}{m}\Theta\Big(\big(\langle \mathbf{w}_{r,t}^k, \boldsymbol{\mu}_j\rangle\langle \mathbf{w}_{r,t}^0, \boldsymbol{\mu}_j\rangle + \langle \mathbf{w}_{r,t}^k, \boldsymbol{\xi}_i\rangle\langle \mathbf{w}_{r,t}^0, \boldsymbol{\xi}_i\rangle\big)\|\mathbf{w}_{r,t}^k\|^4\Big)$$

$$+ \frac{m-1}{m}\Theta\Big(\big(\langle \mathbf{w}_{r,t}^k, \boldsymbol{\mu}_j\rangle^2 + \langle \mathbf{w}_{r,t}^k, \boldsymbol{\xi}_i\rangle^2\big)\|\mathbf{w}_{r,t}^k\|^2\langle \mathbf{w}_{r,t}^k, \mathbf{w}_{r,t}^0\rangle\Big)$$

$$+ \frac{m-1}{m}\Theta\Big(\|\mathbf{w}_{r,t}^k\|^4\langle \mathbf{w}_{r,t}^k, \mathbf{w}_{r,t}^0\rangle\Big)$$

$$= -\frac{1}{\sqrt{m}}\Theta\Big(\langle \mathbf{w}_{r,t}^k, \boldsymbol{\mu}_j + \boldsymbol{\xi}_i\rangle\langle \mathbf{w}_{r,t}^k, \mathbf{w}_{r,t}^0\rangle + \|\mathbf{w}_{r,t}^k\|^2\langle \mathbf{w}_{r,t}^0, \boldsymbol{\mu}_j + \boldsymbol{\xi}_i\rangle\Big)$$

$$+ \frac{1}{m}\Theta\Big(\langle \mathbf{w}_{r,t}^k, \boldsymbol{\mu}_j\rangle^4 + \langle \mathbf{w}_{r,t}^k, \boldsymbol{\xi}_i\rangle^4\Big)\langle \mathbf{w}_{r,t}^k, \mathbf{w}_{r,t}^0\rangle$$

$$+ \Theta\Big(\big(\langle \mathbf{w}_{r,t}^k, \boldsymbol{\mu}_j\rangle^2 + \langle \mathbf{w}_{r,t}^k, \boldsymbol{\xi}_i\rangle^2\big)\|\mathbf{w}_{r,t}^k\|^2 + \|\mathbf{w}_{r,t}^k\|^4\Big)\langle \mathbf{w}_{r,t}^k, \mathbf{w}_{r,t}^0\rangle$$

$$+ \Theta\Big(\big(\langle \mathbf{w}_{r,t}^k, \boldsymbol{\mu}_j\rangle^3\langle \mathbf{w}_{r,t}^0, \boldsymbol{\mu}_j\rangle + \langle \mathbf{w}_{r,t}^k, \boldsymbol{\xi}_i\rangle^3\langle \mathbf{w}_{r,t}^0, \boldsymbol{\xi}_i\rangle\big)\|\mathbf{w}_{r,t}^k\|^2\Big)$$

$$+ \Theta\Big(\big(\langle \mathbf{w}_{r,t}^k, \boldsymbol{\mu}_j\rangle\langle \mathbf{w}_{r,t}^0, \boldsymbol{\mu}_j\rangle + \langle \mathbf{w}_{r,t}^k, \boldsymbol{\xi}_i\rangle\langle \mathbf{w}_{r,t}^0, \boldsymbol{\xi}_i\rangle\big)\|\mathbf{w}_{r,t}^k\|^4\Big) \tag{47}$$

where we use that $\langle \mathbf{w}_{r,t}^k, \mathbf{w}_{r,t}^0\rangle \geq \Theta(\sigma_0^2 d) \geq \widetilde{O}(\sigma_0 + n\sigma_0\sigma_\xi^{-1}d^{-1/2})$ due to the condition that $\sigma_0 \geq \widetilde{O}(\max\{n\sigma_\xi^{-1}d^{-3/2}, d^{-1}\})$.

In order to identify the dominant terms of (45) and (46), we separate the analysis for three cases depending on the scale of $n \cdot \mathrm{SNR}^2$. For each case, we also consider two sub-cases depending on the scale of $\langle \mathbf{w}_{r,t}^k, \boldsymbol{\xi}_i\rangle$. Recall that we define $\psi_{r,t}^k = \langle \mathbf{w}_{r,t}^k, \mathbf{w}_{r,t}^0\rangle^2 \|\mathbf{w}_{r,t}^0\|^{-2}$

- When $\chi = n \cdot \mathrm{SNR}^2 = \Theta(1)$,

  - If $\langle \mathbf{w}_{r,t}^k, \boldsymbol{\xi}_i \rangle^2 \geq \Theta(\psi_{r,t}^k)$, we can identify the dominant terms as

$$\frac{1}{\sqrt{m}} \|\mathbf{w}_{r,t}^k\| \|\boldsymbol{\mu}\|^2 = -\frac{1}{\sqrt{m}} \Theta\big(\langle \mathbf{w}_{r,t}^k, \boldsymbol{\xi}_i \rangle^2\big), \quad E_{r,t,\mu_j}^k = \Theta(\langle \mathbf{w}_{r,t}^k, \boldsymbol{\xi}_i \rangle^5)$$

$$\frac{1}{n\sqrt{m}} \|\mathbf{w}_{r,t}^k\|^2 \|\boldsymbol{\xi}_i\|^2 = -\frac{1}{\sqrt{m}} \Theta\big(\langle \mathbf{w}_{r,t}^k, \boldsymbol{\xi}_i \rangle^2\big), \quad E_{r,t,\xi_i}^k = \Theta(\langle \mathbf{w}_{r,t}^k, \boldsymbol{\xi}_i \rangle^5)$$

Hence, we can see at $k = T_2$, when $\langle \mathbf{w}_{r,t}^{T_2}, \boldsymbol{\xi}_i \rangle = \Theta(m^{-1/6})$ and $\langle \mathbf{w}_{r,t}^{T_2}, \boldsymbol{\mu}_j \rangle = \Theta(\chi m^{-1/6}) = \Theta(m^{-1/6})$ (by the condition of $\chi$), we have $E_{r,t,\mu_j}^{T_2} = \Theta(\frac{1}{\sqrt{m}} \|\mathbf{w}_{r,t}^{T_2}\| \|\boldsymbol{\mu}\|^2)$ and $E_{r,t,\xi_i}^{T_2} = \Theta(\frac{1}{n\sqrt{m}} \|\mathbf{w}_{r,t}^{T_2}\|^2 \|\boldsymbol{\xi}_i\|^2)$.

It remains to show that $E_{r,t,w^0}^{T_2} = \frac{1}{\sqrt{m}} \Theta\big((\langle \mathbf{w}_{r,t}^{T_2}, \boldsymbol{\mu}_j + \overline{\boldsymbol{\xi}} \rangle - \sqrt{m} \|\mathbf{w}_{r,t}^{T_2}\|^4) \langle \mathbf{w}_{r,t}^{T_2}, \mathbf{w}_{r,t}^0 \rangle + \|\mathbf{w}_{r,t}^{T_2}\|^2 \langle \mathbf{w}_{r,t}^0, \boldsymbol{\mu}_j + \overline{\boldsymbol{\xi}} \rangle\big)$. We compute that $\|\mathbf{w}_{r,t}^{T_2}\|^2 = \Theta\big(\langle \mathbf{w}_{r,t}^{T_2}, \boldsymbol{\xi}_i \rangle^2 + \psi_{r,t}^{T_2}\big) = \Theta(m^{-1/3})$, which implies

$$\frac{1}{\sqrt{m}} \Big( (\langle \mathbf{w}_{r,t}^{T_2}, \boldsymbol{\mu}_j + \overline{\boldsymbol{\xi}} \rangle - \sqrt{m} \|\mathbf{w}_{r,t}^{T_2}\|^2) \langle \mathbf{w}_{r,t}^{T_2}, \mathbf{w}_{r,t}^0 \rangle + \|\mathbf{w}_{r,t}^{T_2}\|^2 \langle \mathbf{w}_{r,t}^0, \boldsymbol{\mu}_j + \overline{\boldsymbol{\xi}} \rangle \Big)$$

$$= \frac{1}{\sqrt{m}} \Theta\big(m^{-1/6} \langle \mathbf{w}_{r,t}^{T_2}, \mathbf{w}_{r,t}^0 \rangle + m^{-1/3} \langle \mathbf{w}_{r,t}^0, \boldsymbol{\mu}_j + \overline{\boldsymbol{\xi}} \rangle\big)$$

$$= \Theta\big(m^{-2/3} \langle \mathbf{w}_{r,t}^{T_2}, \mathbf{w}_{r,t}^0 \rangle + m^{-5/6} \langle \mathbf{w}_{r,t}^0, \boldsymbol{\mu}_j + \overline{\boldsymbol{\xi}} \rangle\big)$$

$$= E_{r,t,w^0}^{T_2}$$

based on the derivation in (47). In this case, we can show $\psi_{r,t}^k \leq \langle \mathbf{w}_{r,t}^k, \boldsymbol{\xi}_i \rangle^2 = \Theta(m^{-1/3})$ is feasible given the lower bound on $\langle \mathbf{w}_{r,t}^k, \mathbf{w}_{r,t}^0 \rangle = \Omega(\sigma_0 \sqrt{d} m^{-1/6})$ in $\mathscr{A}(k)$. This verifies $\mathscr{D}(T_2), \mathscr{E}(T_2)$ in the case where $\langle \mathbf{w}_{r,t}^k, \boldsymbol{\xi}_i \rangle^2 \geq \Theta(\psi_{r,t}^k)$.

  - If $\langle \mathbf{w}_{r,t}^k, \boldsymbol{\xi}_i \rangle^2 = o(\psi_{r,t}^k)$, we show $\langle \nabla_{\mathbf{w}_{r,t}} L(\mathbf{W}_t^k), \mathbf{w}_{r,t}^0 \rangle \neq 0$ and thus cannot converge. More specifically, we can identify the dominant terms as

$$\frac{1}{\sqrt{m}} \|\mathbf{w}_{r,t}^k\| \|\boldsymbol{\mu}\|^2 = -\frac{1}{\sqrt{m}} \Theta\big(\psi_{r,t}^k\big), \quad E_{r,t,\mu_j}^k = \Theta((\psi_{r,t}^k)^2 \langle \mathbf{w}_{r,t}^k, \boldsymbol{\xi}_i \rangle)$$

$$\frac{1}{n\sqrt{m}} \|\mathbf{w}_{r,t}^k\|^2 \|\boldsymbol{\xi}_i\|^2 = -\frac{1}{\sqrt{m}} \Theta\big(\psi_{r,t}^k\big), \quad E_{r,t,\xi_i}^k = \Theta((\psi_{r,t}^k)^2 \langle \mathbf{w}_{r,t}^k, \boldsymbol{\xi}_i \rangle).$$

Thus, we see $E_{r,t,\mu_j}^k = \Theta(\frac{1}{\sqrt{m}} \|\mathbf{w}_{r,t}^k\| \|\boldsymbol{\mu}\|^2)$ and $E_{r,t,\xi_i}^k = \Theta(\frac{1}{n\sqrt{m}} \|\mathbf{w}_{r,t}^k\|^2 \|\boldsymbol{\xi}_i\|^2)$ only when $\langle \mathbf{w}_{r,t}^k, \boldsymbol{\xi}_i \rangle = \Theta(m^{-1/2} (\psi_{r,t}^k)^{-1}) = \langle \mathbf{w}_{r,t}^k, \boldsymbol{\mu}_j \rangle$. This implies that $\psi_{r,t}^k \geq \Theta(m^{-1/3})$ and thus $\|\mathbf{w}_{r,t}^k\|^2 = \Theta(\psi_{r,t}^k) \geq \Theta(m^{-1/3})$. We show in this case, the gradient along direction $\mathbf{w}_{r,t}^0$ is negative. Examining $\langle \nabla_{\mathbf{w}_{r,t}} L(\mathbf{W}_t^k), \mathbf{w}_{r,t}^0 \rangle$, we see

$$\frac{1}{\sqrt{m}} \Big( (\langle \mathbf{w}_{r,t}^k, \boldsymbol{\mu}_j + \overline{\boldsymbol{\xi}} \rangle - \sqrt{m} \|\mathbf{w}_{r,t}^k\|^4) \langle \mathbf{w}_{r,t}^k, \mathbf{w}_{r,t}^0 \rangle + \|\mathbf{w}_{r,t}^k\|^2 \langle \mathbf{w}_{r,t}^0, \boldsymbol{\mu}_j + \overline{\boldsymbol{\xi}} \rangle \Big)$$

$$= \frac{1}{\sqrt{m}} \big( \Theta(m^{-1/2}(\psi_{r,t}^k)^{-1} - \sqrt{m}(\psi_{r,t}^k)^2) \langle \mathbf{w}_{r,t}^k, \mathbf{w}_{r,t}^0 \rangle + \Theta(\psi_{r,t}^k) \langle \mathbf{w}_{r,t}^0, \boldsymbol{\mu}_j + \overline{\boldsymbol{\xi}} \rangle \big)$$

$$= \Theta\big( m^{-1}(\psi_{r,t}^k)^{-1} \langle \mathbf{w}_{r,t}^k, \mathbf{w}_{r,t}^0 \rangle - (\psi_{r,t}^k)^2 \langle \mathbf{w}_{r,t}^k, \mathbf{w}_{r,t}^0 \rangle + m^{-1/2} \psi_{r,t}^k \langle \mathbf{w}_{r,t}^0, \boldsymbol{\mu}_j + \overline{\boldsymbol{\xi}} \rangle \big)$$

$$E_{r,t,w^0}^k = \Theta\big( (m^{-3}(\psi_{r,t}^k)^{-4} + m^{-1}(\psi_{r,t}^k)^{-1} + (\psi_{r,t}^k)^2) \langle \mathbf{w}_{r,t}^k, \mathbf{w}_{r,t}^0 \rangle \big)$$

$$\qquad + \Theta\big( (m^{-3/2}(\psi_{r,t}^k)^{-2} + m^{-1/2} \psi_{r,t}^k) \langle \mathbf{w}_{r,t}^0, \boldsymbol{\mu}_j + \overline{\boldsymbol{\xi}} \rangle \big)$$

$$= \Theta((\psi_{r,t}^k)^2 \langle \mathbf{w}_{r,t}^k, \mathbf{w}_{r,t}^0 \rangle + m^{-1/2} \psi_{r,t}^k \langle \mathbf{w}_{r,t}^0, \boldsymbol{\mu}_j + \overline{\boldsymbol{\xi}} \rangle)$$

where we use that $\psi_{r,t}^k \geq \Theta(m^{-1/3})$. It is clear that that due to the condition $\psi_{r,t}^k \geq \Theta(m^{-1/3})$, it satisfies $\langle \nabla_{\mathbf{w}_{r,t}} L(\mathbf{W}_t^k), \mathbf{w}_{r,t}^0 \rangle < 0$, which then concludes that $\psi_{r,t}^k$ would decrease and cannot reach stationary point.

- When $\chi = n \cdot \mathrm{SNR}^2 = \widetilde{\Omega}(1)$,

- If $\chi^2\langle\mathbf{w}_{r,t}^k,\boldsymbol{\xi}_i\rangle^2 \geq \Theta(\psi_{r,t}^k)$, we can simplify (45) and (46) to

$$\frac{1}{\sqrt{m}}\|\mathbf{w}_{r,t}^k\|\|\boldsymbol{\mu}\|^2 = -\frac{1}{\sqrt{m}}\Theta\big(\chi^2\langle\mathbf{w}_{r,t}^k,\boldsymbol{\xi}_i\rangle^2\big), \quad E_{r,t,\mu_j}^k = \Theta(\chi^5\langle\mathbf{w}_{r,t}^k,\boldsymbol{\xi}_i\rangle^5)$$

$$\frac{1}{n\sqrt{m}}\|\mathbf{w}_{r,t}^k\|^2\|\boldsymbol{\xi}_i\|^2 = -\frac{1}{\sqrt{m}}\Theta\big(\chi\langle\mathbf{w}_{r,t}^k,\boldsymbol{\xi}_i\rangle^2\big), \quad E_{r,t,\xi_i}^k = \Theta(\chi^4\langle\mathbf{w}_{r,t}^k,\boldsymbol{\xi}_i\rangle^5)$$

Hence, we can see at $k = T_2$, when $\langle\mathbf{w}_{r,t}^{T_2},\boldsymbol{\xi}_i\rangle = \Theta(\chi^{-1}m^{-1/6})$ and thus $\langle\mathbf{w}_{r,t}^{T_2},\boldsymbol{\mu}_j\rangle = \Theta(m^{-1/6})$, we have $E_{r,t,\mu_j}^{T_2} = \Theta(\frac{1}{\sqrt{m}}\|\mathbf{w}_{r,t}^{T_2}\|\|\boldsymbol{\mu}\|^2)$ and $E_{r,t,\xi_i}^{T_2} = \Theta(\frac{1}{n\sqrt{m}}\|\mathbf{w}_{r,t}^{T_2}\|^2\|\boldsymbol{\xi}_i\|^2)$. Next we show $E_{r,t,w^0}^{T_2} = \frac{1}{\sqrt{m}}\Theta\big((\langle\mathbf{w}_{r,t}^{T_2},\boldsymbol{\mu}_j + \overline{\boldsymbol{\xi}}\rangle - \sqrt{m}\|\mathbf{w}_{r,t}^{T_2}\|^4)\langle\mathbf{w}_{r,t}^{T_2},\mathbf{w}_{r,t}^0\rangle + \|\mathbf{w}_{r,t}^{T_2}\|^2\langle\mathbf{w}_{r,t}^0,\boldsymbol{\mu}_j + \overline{\boldsymbol{\xi}}\rangle\big)$. We first compute that $\|\mathbf{w}_{r,t}^{T_2}\|^2 = \Theta\big(\chi^2\langle\mathbf{w}_{r,t}^{T_2},\boldsymbol{\xi}_i\rangle^2 + \psi_{r,t}^{T_2}\big) = \Theta(m^{-1/3})$, which implies

$$\frac{1}{\sqrt{m}}\Big((\langle\mathbf{w}_{r,t}^{T_2},\boldsymbol{\mu}_j + \overline{\boldsymbol{\xi}}\rangle - \sqrt{m}\|\mathbf{w}_{r,t}^{T_2}\|^2)\langle\mathbf{w}_{r,t}^{T_2},\mathbf{w}_{r,t}^0\rangle + \|\mathbf{w}_{r,t}^{T_2}\|^2\langle\mathbf{w}_{r,t}^0,\boldsymbol{\mu}_j + \overline{\boldsymbol{\xi}}\rangle\Big)$$

$$= \frac{1}{\sqrt{m}}\Theta\big(m^{-1/6}\langle\mathbf{w}_{r,t}^{T_2},\mathbf{w}_{r,t}^0\rangle + m^{-1/3}\langle\mathbf{w}_{r,t}^0,\boldsymbol{\mu}_j + \overline{\boldsymbol{\xi}}\rangle\big)$$

$$= \Theta\big(m^{-2/3}\langle\mathbf{w}_{r,t}^{T_2},\mathbf{w}_{r,t}^0\rangle + m^{-5/6}\langle\mathbf{w}_{r,t}^0,\boldsymbol{\mu}_j + \overline{\boldsymbol{\xi}}\rangle\big)$$

$$= E_{r,t,w^0}^{T_2}$$

based on the derivation in (47). In this case, we can show $\psi_{r,t}^k \leq \chi^2\langle\mathbf{w}_{r,t}^k,\boldsymbol{\xi}_i\rangle^2 = \Theta(m^{-1/3})$ is feasible given the lower bound on $\langle\mathbf{w}_{r,t}^k,\mathbf{w}_{r,t}^0\rangle = \Omega(\sigma_0\sqrt{d}m^{-1/6})$ in $\mathscr{A}(k)$. This verifies $\mathscr{D}(T_2),\mathscr{E}(T_2)$ in the case where $\chi^2\langle\mathbf{w}_{r,t}^k,\boldsymbol{\xi}_i\rangle^2 \geq \Theta(\psi_{r,t}^k)$.

- If $\chi^2\langle\mathbf{w}_{r,t}^k,\boldsymbol{\xi}_i\rangle^2 = o(\psi_{r,t}^k)$, we can follow a similar argument to show that $\langle\nabla_{\mathbf{w}_{r,t}}L(\mathbf{W}_t^k),\mathbf{w}_{r,t}^0\rangle \neq 0$. Specifically, we can simplify (45) and (46) to

$$\frac{1}{\sqrt{m}}\|\mathbf{w}_{r,t}^k\|\|\boldsymbol{\mu}\|^2 = -\frac{1}{\sqrt{m}}\Theta\big(\psi_{r,t}^k\big), \quad E_{r,t,\mu_j}^k = \Theta((\psi_{r,t}^k)^2\chi\langle\mathbf{w}_{r,t}^k,\boldsymbol{\xi}_i\rangle)$$

$$\frac{1}{n\sqrt{m}}\|\mathbf{w}_{r,t}^k\|^2\|\boldsymbol{\xi}_i\|^2 = -\frac{\eta}{\sqrt{m}}\Theta\big(\chi^{-1}\psi_{r,t}^k\big), \quad E_{r,t,\xi_i}^k = \Theta((\psi_{r,t}^k)^2\langle\mathbf{w}_{r,t}^k,\boldsymbol{\xi}_i\rangle).$$

Thus, we see $E_{r,t,\mu_j}^k = \Theta(\frac{1}{\sqrt{m}}\|\mathbf{w}_{r,t}^k\|\|\boldsymbol{\mu}\|^2)$ and $E_{r,t,\xi_i}^k = \Theta(\frac{1}{n\sqrt{m}}\|\mathbf{w}_{r,t}^k\|^2\|\boldsymbol{\xi}_i\|^2)$ only when $\chi\langle\mathbf{w}_{r,t}^k,\boldsymbol{\xi}_i\rangle = \Theta((\psi_{r,t}^k)^{-1}m^{-1/2})$, which implies $\psi_{r,t}^k \geq \Theta(m^{-1/3})$. However, by a similar argument as the case $n \cdot \text{SNR}^2 = \Theta(1)$, we can show $\langle\nabla_{\mathbf{w}_{r,t}}L(\mathbf{W}_t^k),\mathbf{w}_{r,t}^0\rangle < 0$, which then concludes that $\psi_{r,t}^k$ would decrease and cannot reach stationary point.

- When $\chi^{-1} = n^{-1}\text{SNR}^{-2} = \widetilde{\Omega}(1)$,

  - If $\langle\mathbf{w}_{r,t}^k,\boldsymbol{\xi}_i\rangle^2 \geq \Theta(\chi^{-1}\psi_{r,t}^k)$, we can simplify (45) and (46) into

$$\frac{1}{\sqrt{m}}\|\mathbf{w}_{r,t}^k\|\|\boldsymbol{\mu}\|^2 = -\frac{1}{\sqrt{m}}\Theta\big(\chi\langle\mathbf{w}_{r,t}^k,\boldsymbol{\xi}_i\rangle^2\big), \quad E_{r,t,\mu_j}^k = \Theta(\chi\langle\mathbf{w}_{r,t}^k,\boldsymbol{\xi}_i\rangle^5)$$

$$\frac{1}{n\sqrt{m}}\|\mathbf{w}_{r,t}^k\|^2\|\boldsymbol{\xi}_i\|^2 = -\frac{1}{\sqrt{m}}\Theta\big(\langle\mathbf{w}_{r,t}^k,\boldsymbol{\xi}_i\rangle^2\big), \quad E_{r,t,\xi_i}^k = \Theta(\langle\mathbf{w}_{r,t}^k,\boldsymbol{\xi}_i\rangle^5)$$

Hence, we can see at $k = T_2$, when $\langle\mathbf{w}_{r,t}^{T_2},\boldsymbol{\xi}_i\rangle = \Theta(m^{-1/6})$ and $\langle\mathbf{w}_{r,t}^{T_2},\boldsymbol{\mu}_j\rangle = \Theta(\chi m^{-1/6})$, we have $E_{r,t,\mu_j}^{T_2} = \Theta(\frac{1}{\sqrt{m}}\|\mathbf{w}_{r,t}^{T_2}\|\|\boldsymbol{\mu}\|^2)$ and $E_{r,t,\xi_i}^{T_2} = \Theta(\frac{1}{n\sqrt{m}}\|\mathbf{w}_{r,t}^{T_2}\|^2\|\boldsymbol{\xi}_i\|^2)$. In this case, we can show $\psi_{r,t}^k \leq \chi\langle\mathbf{w}_{r,t}^k,\boldsymbol{\xi}_i\rangle^2$ is feasible given the lower bound on $\langle\mathbf{w}_{r,t}^k,\mathbf{w}_{r,t}^0\rangle = \Omega(\sigma_0\sigma_\xi^{-1}n^{1/2}m^{-1/6})$ in $\mathscr{A}(k)$ when $\chi^{-1} = \widetilde{\Omega}(1)$.

Next we can check that at $T_2$, $\max\{\langle\mathbf{w}_{r,t}^{T_2},\boldsymbol{\mu}_j\rangle,\langle\mathbf{w}_{r,t}^{T_1},\boldsymbol{\xi}_i\rangle\} = \langle\mathbf{w}_{r,t}^{T_2},\boldsymbol{\xi}_i\rangle = \Theta(m^{-1/6})$. Further,

$$\|\mathbf{w}_{r,t}^{T_2}\|^2 = \Theta\big(\langle\mathbf{w}_{r,t}^{T_2},\boldsymbol{\mu}_j\rangle^2 + \langle\mathbf{w}_{r,t}^{T_2},\boldsymbol{\mu}_j\rangle\langle\mathbf{w}_{r,t}^{T_2},\boldsymbol{\xi}_i\rangle + \langle\mathbf{w}_{r,t}^{T_2},\mathbf{w}_{r,t}^0\rangle^2\|\mathbf{w}_{r,t}^0\|^{-2}\big) = \Theta(\chi m^{-1/3})$$

where the second equality is by $\langle \mathbf{w}_{r,t}^{T_2}, \mathbf{w}_{r,t}^0 \rangle^2 \|\mathbf{w}_{r,t}^0\|^{-2} = \psi_{r,t}^{T_2} \le \Theta(\chi m^{-1/3})$. This leads to

$$\frac{1}{\sqrt{m}}\Big( (\langle \mathbf{w}_{r,t}^{T_2}, \boldsymbol{\mu}_j + \overline{\boldsymbol{\xi}} \rangle - \sqrt{m}\|\mathbf{w}_{r,t}^{T_2}\|^4) \langle \mathbf{w}_{r,t}^{T_2}, \mathbf{w}_{r,t}^0 \rangle + \|\mathbf{w}_{r,t}^{T_2}\|^2 \langle \mathbf{w}_{r,t}^0, \boldsymbol{\mu}_j + \overline{\boldsymbol{\xi}} \rangle \Big)$$

$$= \Theta\big( m^{-2/3} \langle \mathbf{w}_{r,t}^{T_2}, \mathbf{w}_{r,t}^0 \rangle + \chi m^{-5/6} \langle \mathbf{w}_{r,t}^0, \boldsymbol{\mu}_j + \overline{\boldsymbol{\xi}} \rangle \big)$$

$$E_{r,t,w^0}^{T_2} = \Theta\Big( (m^{-1} + \chi)m^{-2/3} \langle \mathbf{w}_{r,t}^{T_2}, \mathbf{w}_{r,t}^0 \rangle + \chi m^{-5/6} \langle \mathbf{w}_{r,t}^0, \boldsymbol{\mu}_j + \overline{\boldsymbol{\xi}} \rangle \Big)$$

where $E_{r,t,w^0}^{T_2} = \frac{1}{\sqrt{m}}\Big( (\langle \mathbf{w}_{r,t}^{T_2}, \boldsymbol{\mu}_j + \overline{\boldsymbol{\xi}} \rangle - \sqrt{m}\|\mathbf{w}_{r,t}^{T_2}\|^4) \langle \mathbf{w}_{r,t}^{T_2}, \mathbf{w}_{r,t}^0 \rangle + \|\mathbf{w}_{r,t}^{T_2}\|^2 \langle \mathbf{w}_{r,t}^0, \boldsymbol{\mu}_j + \overline{\boldsymbol{\xi}} \rangle \Big)$
holds due to $m = \Theta(1)$. This verifies $\mathscr{D}(T_2), \mathscr{E}(T_2)$.

– If $\langle \mathbf{w}_{r,t}^k, \boldsymbol{\xi}_i \rangle^2 < \Theta(\chi^{-1}\psi_{r,t}^k)$, we can simplify (45) and (46) into

$$\frac{1}{\sqrt{m}}\|\mathbf{w}_{r,t}^k\|\|\boldsymbol{\mu}\|^2 = -\frac{1}{\sqrt{m}}\Theta\big(\psi_{r,t}^k\big), \quad E_{r,t,\mu_j}^k = \Theta((\psi_{r,t}^k)^2 \langle \mathbf{w}_{r,t}^k, \boldsymbol{\xi}_i \rangle \chi + \psi_{r,t}^k \langle \mathbf{w}_{r,t}^k, \boldsymbol{\xi}_i \rangle^3 \chi)$$

$$\frac{1}{n\sqrt{m}}\|\mathbf{w}_{r,t}^k\|^2 \|\boldsymbol{\xi}_i\|^2 = -\frac{1}{\sqrt{m}}\Theta\big(\chi^{-1}\psi_{r,t}^k\big), \quad E_{r,t,\xi_i}^k = \Theta(\chi^{-1}\psi_{r,t}^k \langle \mathbf{w}_{r,t}^k, \boldsymbol{\xi}_i \rangle^3 + \chi^{-1}(\psi_{r,t}^k)^2 \langle \mathbf{w}_{r,t}^k, \boldsymbol{\xi}_i \rangle)$$

* If $\langle \mathbf{w}_{r,t}^k, \boldsymbol{\xi}_i \rangle^2 \ge \Theta(\psi_{r,t}^k)$, the equalities become

$$\frac{1}{\sqrt{m}}\|\mathbf{w}_{r,t}^k\|\|\boldsymbol{\mu}\|^2 = -\frac{1}{\sqrt{m}}\Theta\big(\psi_{r,t}^k\big), \quad E_{r,t,\mu_j}^k = \Theta(\psi_{r,t}^k \langle \mathbf{w}_{r,t}^k, \boldsymbol{\xi}_i \rangle^3 \chi)$$

$$\frac{1}{n\sqrt{m}}\|\mathbf{w}_{r,t}^k\|^2 \|\boldsymbol{\xi}_i\|^2 = -\frac{1}{\sqrt{m}}\Theta\big(\chi^{-1}\psi_{r,t}^k\big), \quad E_{r,t,\xi_i}^k = \Theta(\chi^{-1}\psi_{r,t}^k \langle \mathbf{w}_{r,t}^k, \boldsymbol{\xi}_i \rangle^3),$$

and $E_{r,t,\mu_j}^k = \Theta(\frac{1}{\sqrt{m}}\|\mathbf{w}_{r,t}^k\|\|\boldsymbol{\mu}\|^2)$ when $\langle \mathbf{w}_{r,t}^k, \boldsymbol{\xi}_i \rangle = \Theta(m^{-1/6}\chi^{-1/3})$ and $E_{r,t,\xi_i}^k = \Theta(\frac{1}{n\sqrt{m}}\|\mathbf{w}_{r,t}^k\|^2 \|\boldsymbol{\xi}_i\|^2)$ when $\langle \mathbf{w}_{r,t}^k, \boldsymbol{\xi}_i \rangle = \Theta(m^{-1/6})$. Due to the scale that $\chi^{-1} = \widetilde{\Omega}(1)$, two equalities cannot hold at the same time.

* If $\langle \mathbf{w}_{r,t}^k, \boldsymbol{\xi}_i \rangle^2 < \Theta(\psi_{r,t}^k)$, the equalities become

$$\frac{1}{\sqrt{m}}\|\mathbf{w}_{r,t}^k\|\|\boldsymbol{\mu}\|^2 = -\frac{1}{\sqrt{m}}\Theta\big(\psi_{r,t}^k\big), \quad E_{r,t,\mu_j}^k = \Theta((\psi_{r,t}^k)^2 \langle \mathbf{w}_{r,t}^k, \boldsymbol{\xi}_i \rangle \chi)$$

$$\frac{1}{n\sqrt{m}}\|\mathbf{w}_{r,t}^k\|^2 \|\boldsymbol{\xi}_i\|^2 = -\frac{1}{\sqrt{m}}\Theta\big(\chi^{-1}\psi_{r,t}^k\big), \quad E_{r,t,\xi_i}^k = \Theta(\chi^{-1}(\psi_{r,t}^k)^2 \langle \mathbf{w}_{r,t}^k, \boldsymbol{\xi}_i \rangle).$$

Thus $E_{r,t,\mu_j}^k = \Theta(\frac{1}{\sqrt{m}}\|\mathbf{w}_{r,t}^k\|\|\boldsymbol{\mu}\|^2)$ when $\langle \mathbf{w}_{r,t}^k, \boldsymbol{\xi}_i \rangle = \Theta(\chi^{-1}(\psi_{r,t}^k)^{-1})$ and $E_{r,t,\xi_i}^k = \Theta(\frac{1}{n\sqrt{m}}\|\mathbf{w}_{r,t}^k\|^2 \|\boldsymbol{\xi}_i\|^2)$ when $\langle \mathbf{w}_{r,t}^k, \boldsymbol{\xi}_i \rangle = \Theta((\psi_{r,t}^k)^{-1})$, which clearly cannot be satisfied at the same time given $\chi^{-1} = \widetilde{\Omega}(1)$.

To conclude, we verify that when $\langle \mathbf{w}_{r,t}^k, \boldsymbol{\xi}_i \rangle^2 \le \Theta(\chi^{-1}\psi_{r,t}^k)$, $E_{r,t,\mu_j}^k = \Theta(\frac{1}{\sqrt{m}}\|\mathbf{w}_{r,t}^k\|\|\boldsymbol{\mu}\|^2)$ or $E_{r,t,\xi_i}^k = \Theta(\frac{1}{n\sqrt{m}}\|\mathbf{w}_{r,t}^k\|^2 \|\boldsymbol{\xi}_i\|^2)$ cannot be satisfied.

In summary, we obtain the following results:

* When $n \cdot \mathrm{SNR}^2 = \Theta(1)$, we can show $E_{r,t,\mu_j}^k = \Theta(\frac{1}{\sqrt{m}}\|\mathbf{w}_{r,t}^k\|\|\boldsymbol{\mu}\|^2)$ and $E_{r,t,\xi_i}^k = \Theta(\frac{1}{n\sqrt{m}}\|\mathbf{w}_{r,t}^k\|^2\|\boldsymbol{\xi}_i\|^2)$ when $\langle \mathbf{w}_{r,t}^k, \boldsymbol{\xi}_i \rangle = \Theta(m^{-1/6})$.

* When $n \cdot \mathrm{SNR}^2 = \widetilde{\Omega}(1)$, we can show $E_{r,t,\mu_j}^k = \Theta(\frac{1}{\sqrt{m}}\|\mathbf{w}_{r,t}^k\|\|\boldsymbol{\mu}\|^2)$ and $E_{r,t,\xi_i}^k = \Theta(\frac{1}{n\sqrt{m}}\|\mathbf{w}_{r,t}^k\|^2\|\boldsymbol{\xi}_i\|^2)$ when $\langle \mathbf{w}_{r,t}^k, \boldsymbol{\xi}_i \rangle = \Theta(\chi^{-1}m^{-1/6})$.

* When $n^{-1} \cdot \mathrm{SNR}^{-2} = \widetilde{\Omega}(1)$, we can show $E_{r,t,\mu_j}^k = \Theta(\frac{1}{\sqrt{m}}\|\mathbf{w}_{r,t}^k\|\|\boldsymbol{\mu}\|^2)$ and $E_{r,t,\xi_i}^k = \Theta(\frac{1}{n\sqrt{m}}\|\mathbf{w}_{r,t}^k\|^2\|\boldsymbol{\xi}_i\|^2)$ when $\langle \mathbf{w}_{r,t}^k, \boldsymbol{\xi}_i \rangle = \Theta(m^{-1/6})$.

Combining the definition of $T_2$, we complete the proof for Claim D.2.

**Proof of Claim D.3.** By the definition of $T_2$ and Lemma D.5, we know that for all $T_1 \le k \le T_2$, the gradients can be written as

$$\langle \nabla_{\mathbf{w}_{r,t}} L(\mathbf{W}_t^k), \boldsymbol{\mu}_j \rangle = \Theta\left( -\frac{1}{\sqrt{m}} \|\mathbf{w}_{r,t}^k\|^2 \|\boldsymbol{\mu}\|^2 \right)$$

$$\langle \nabla_{\mathbf{w}_{r,t}} L(\mathbf{W}_t^k), \boldsymbol{\xi}_i \rangle = \Theta\left( -\frac{1}{n\sqrt{m}} \|\mathbf{w}_{r,t}^k\|^2 \|\boldsymbol{\xi}_i\|^2 \right)$$

and thus (35), (36) are satisfied, which verifies $\mathscr{B}(k)$. In addition, this suggests, for all $T_1 \le k \le T_2$, the increase in $\langle \mathbf{w}_{r,t}^k, \boldsymbol{\xi}_i \rangle, \langle \mathbf{w}_{r,t}^k, \boldsymbol{\mu}_j \rangle$ is monotonic. Combining with $\mathscr{D}(T_2)$, we have $\langle \mathbf{w}_{r,t}^k, \boldsymbol{\mu}_j \rangle, \langle \mathbf{w}_{r,t}^k, \boldsymbol{\xi}_i \rangle = O(m^{-1/6}) = \widetilde{O}(1)$ for all $T_1 \le k \le T_2$, thus verifying $\mathscr{A}(k)$.

Hence, the proof completes the induction on $k$ and verify the claims $\mathscr{A}(k), \mathscr{B}(k), \mathscr{C}(k), \mathscr{D}(T_2), \mathscr{E}(T_2), T_1 \le k \le T_2$.

Finally, we derive an upper bound on $T_2$. Because for all $T_1 \le k \le T_2$, we can decompose $\|\mathbf{w}_{r,t}^k\|^2$ from Lemma D.4 as

$$\|\mathbf{w}_{r,t}^k\|^2 = \Theta\left( \langle \mathbf{w}_{r,t}^k, \boldsymbol{\mu}_j \rangle^2 \|\boldsymbol{\mu}\|^{-2} + \chi \langle \mathbf{w}_{r,t}^k, \boldsymbol{\xi}_i \rangle^2 \|\boldsymbol{\mu}\|^{-2} + \|\mathbf{w}_{r,t}^0\|^2 \right) \ge \Theta(\sigma_0^2 d)$$

Therefore, we can upper bound the update in (35), (36) for $T_1 \le k \le T_2$ by a liner growth as

$$\langle \mathbf{w}_{r,t}^{k+1}, \boldsymbol{\mu}_j \rangle = \langle \mathbf{w}_{r,t}^k, \boldsymbol{\mu}_j \rangle + \Theta\left( \frac{\eta}{\sqrt{m}} \|\mathbf{w}_{r,t}^k\|^2 \|\boldsymbol{\mu}\|^2 \right) \ge \langle \mathbf{w}_{r,t}^k, \boldsymbol{\mu}_j \rangle + \Theta\left( \eta m^{-1/2} \sigma_0^2 d \right)$$

$$\langle \mathbf{w}_{r,t}^{k+1}, \boldsymbol{\xi}_i \rangle = \langle \mathbf{w}_{r,t}^k, \boldsymbol{\xi}_i \rangle + \Theta\left( \frac{\eta}{n\sqrt{m}} \|\mathbf{w}_{r,t}^k\|^2 \|\boldsymbol{\xi}_i\|^2 \right) \ge \langle \mathbf{w}_{r,t}^k, \boldsymbol{\xi}_i \rangle + \Theta\left( \eta \chi^{-1} m^{-1/2} \sigma_0^2 d \right).$$

Therefore, we can upper bound as $T_2 \le \Theta(\max\{\eta^{-1} m^{1/3} \sigma_0^{-2} d^{-1}, \eta^{-1} m^{1/3} n \sigma_0^{-2} \sigma_\xi^2\})$. $\qquad\square$

### D.5 STATIONARY POINT

This section analyzes the stationary point with the conditions at the end of the second stage.

**Theorem D.1.** *Under Condition D.1, suppose (1)* $\langle \mathbf{w}_{r,t}^*, \boldsymbol{\mu}_j \rangle = \Theta(\langle \mathbf{w}_{r',t}^*, \boldsymbol{\mu}_{j'} \rangle) = \widetilde{O}(1)$, *(2)* $\langle \mathbf{w}_{r,t}^*, \boldsymbol{\xi}_i \rangle = \Theta(\langle \mathbf{w}_{r',t}^*, \boldsymbol{\xi}_{i'} \rangle) = \widetilde{O}(1)$, *(3)* $\|\mathbf{w}_{r,t}^*\|^2 = \Theta(\|\mathbf{w}_{r',t}^*\|^2)$ *and (4)* $\langle \mathbf{w}_{r,t}^*, \mathbf{w}_{r',t}^* \rangle = \Theta(\|\mathbf{w}_{r,t}^*\|^2)$ *(5)* $\langle \mathbf{w}_{r,t}^*, \mathbf{w}_{r,t}^0 \rangle = \Theta(\langle \mathbf{w}_{r',t}^*, \mathbf{w}_{r',t}^0 \rangle) = \Omega(\min\{\sigma_0 \sigma_\xi^{-1} n^{1/2} m^{-1/6}, \sigma_0 \sqrt{d} m^{-1/6}\})$ *hold for all* $j = \pm 1, r \in [m], i \in [m]$. *Then there exists a stationary point* $\mathbf{W}_t^*$, *i.e.,* $\nabla_{\mathbf{w}_{r,t}} L(\mathbf{W}_t^*) = 0$ *that satisfies*

$$|\langle \mathbf{w}_{r,t}^*, \boldsymbol{\mu}_j \rangle| / |\langle \mathbf{w}_{r,t}^*, \boldsymbol{\xi}_i \rangle| = \Theta(n \cdot \mathrm{SNR}^2),$$

*with* $\langle \mathbf{w}_{r,t}^*, \boldsymbol{\xi}_i \rangle = \Theta(n^{-1} \cdot \mathrm{SNR}^{-2} \cdot m^{-1/6})$, $\langle \mathbf{w}_{r,t}^*, \mathbf{w}_{r,t}^0 \rangle \|\mathbf{w}_{r,t}^0\|^{-1} \le \Theta(\langle \mathbf{w}_{r,t}^*, \boldsymbol{\mu}_j \rangle)$ *if* $n \cdot \mathrm{SNR}^2 = \Omega(1)$, *and* $\langle \mathbf{w}_{r,t}^*, \boldsymbol{\xi}_i \rangle = \Theta(m^{-1/6})$, $\langle \mathbf{w}_{r,t}^*, \mathbf{w}_{r,t}^0 \rangle \|\mathbf{w}_{r,t}^0\|^{-1} \le \Theta(\sqrt{n} \cdot \mathrm{SNR}^2 \langle \mathbf{w}_{r,t}^*, \boldsymbol{\xi}_i \rangle)$ *if* $n^{-1} \cdot \mathrm{SNR}^{-2} = \Omega(1)$.

*Proof of Theorem D.1.* The analysis mostly follows from Lemma D.8. Due to the concentration of neurons, we can derive

$\langle \nabla_{\mathbf{w}_{r,t}} L(\mathbf{W}_t^*), \boldsymbol{\mu}_j \rangle$

$= -\frac{1}{\sqrt{m}} \Theta\left( \langle \mathbf{w}_{r,t}^*, \boldsymbol{\mu}_j \rangle^2 + \|\mathbf{w}_{r,t}^*\|^2 \|\boldsymbol{\mu}_j\|^2 + \langle \mathbf{w}_{r,t}^*, \boldsymbol{\xi}_i \rangle \langle \mathbf{w}_{r,t}^*, \boldsymbol{\mu}_j \rangle \right)$

$\quad + \Theta\left( \langle \mathbf{w}_{r,t}^*, \boldsymbol{\mu}_j \rangle^5 + \langle \mathbf{w}_{r,t}^*, \boldsymbol{\mu}_j \rangle^3 \|\mathbf{w}_{r,t}^*\|^2 + \|\mathbf{w}_{r,t}^*\|^4 \langle \mathbf{w}_{r,t}^*, \boldsymbol{\mu}_j \rangle + \langle \mathbf{w}_{r,t}^*, \boldsymbol{\mu}_j \rangle^3 \|\mathbf{w}_{r,t}^*\|^2 \|\boldsymbol{\mu}_j\|^2 \right.$

$\quad\quad \left. + \|\mathbf{w}_{r,t}^*\|^4 \langle \mathbf{w}_{r,t}^*, \boldsymbol{\mu}_j \rangle \|\boldsymbol{\mu}_j\|^2 + \langle \mathbf{w}_{r,t}^*, \boldsymbol{\xi}_i \rangle^4 \langle \mathbf{w}_{r,t}^*, \boldsymbol{\mu}_j \rangle + \langle \mathbf{w}_{r,t}^*, \boldsymbol{\xi}_i \rangle^2 \|\mathbf{w}_{r,t}^*\|^2 \langle \mathbf{w}_{r,t}^*, \boldsymbol{\mu}_j \rangle \right)$

$\langle \nabla_{\mathbf{w}_{r,t}} L(\mathbf{W}_t^*), \boldsymbol{\xi}_i \rangle$

$= -\frac{1}{\sqrt{m}} \Theta\left( \langle \mathbf{w}_{r,t}^*, \boldsymbol{\mu}_j \rangle \langle \mathbf{w}_{r,t}^*, \boldsymbol{\xi}_i \rangle + \langle \mathbf{w}_{r,t}^*, \boldsymbol{\xi}_i \rangle^2 + \frac{1}{n} \|\mathbf{w}_{r,t}^*\|^2 \|\boldsymbol{\xi}_i\|^2 \right)$

$\quad + \Theta\left( \langle \mathbf{w}_{r,t}^*, \boldsymbol{\xi}_i \rangle^5 + \langle \mathbf{w}_{r,t}^*, \boldsymbol{\xi}_i \rangle^3 \|\mathbf{w}_{r,t}^*\|^2 + \langle \mathbf{w}_{r,t}^*, \boldsymbol{\mu}_j \rangle^4 \langle \mathbf{w}_{r,t}^*, \boldsymbol{\xi}_i \rangle + \langle \mathbf{w}_{r,t}^*, \boldsymbol{\mu}_j \rangle^2 \|\mathbf{w}_{r,t}^*\|^2 \langle \mathbf{w}_{r,t}^*, \boldsymbol{\xi}_i \rangle \right.$

$$+ \|\mathbf{w}_{r,t}^*\|^4 \langle \mathbf{w}_{r,t}^*, \boldsymbol{\xi}_i \rangle + \frac{1}{n} \langle \mathbf{w}_{r,t}^*, \boldsymbol{\xi}_i \rangle^3 \|\mathbf{w}_{r,t}^*\|^2 \|\boldsymbol{\xi}_i\|^2 + \frac{1}{n} \|\mathbf{w}_{r,t}^*\|^4 \langle \mathbf{w}_{r,t}^*, \boldsymbol{\xi}_i \rangle \|\boldsymbol{\xi}_i\|^2 \Big)$$

$$\langle \nabla_{\mathbf{w}_{r,t}} L(\mathbf{W}_t^*), \mathbf{w}_{r,t}^0 \rangle$$

$$= -\frac{1}{\sqrt{m}} \Theta \Big( \langle \mathbf{w}_{r,t}^*, \boldsymbol{\mu}_j + \boldsymbol{\xi}_i \rangle \langle \mathbf{w}_{r,t}^*, \mathbf{w}_{r,t}^0 \rangle + \|\mathbf{w}_{r,t}^*\|^2 \langle \mathbf{w}_{r,t}^0, \boldsymbol{\mu}_j + \boldsymbol{\xi}_i \rangle \Big)$$

$$+ \frac{1}{m} \Theta \Big( \langle \mathbf{w}_{r,t}^*, \boldsymbol{\mu}_j \rangle^4 + \langle \mathbf{w}_{r,t}^*, \boldsymbol{\xi}_i \rangle^4 \Big) \langle \mathbf{w}_{r,t}^*, \mathbf{w}_{r,t}^0 \rangle + \Theta \Big( \big( \langle \mathbf{w}_{r,t}^*, \boldsymbol{\mu}_j \rangle^2 + \langle \mathbf{w}_{r,t}^*, \boldsymbol{\xi}_i \rangle^2 \big) \|\mathbf{w}_{r,t}^*\|^2 \Big) \langle \mathbf{w}_{r,t}^*, \mathbf{w}_{r,t}^0 \rangle$$

$$+ \Theta \Big( \|\mathbf{w}_{r,t}^*\|^4 \langle \mathbf{w}_{r,t}^*, \mathbf{w}_{r,t}^0 \rangle + \big( \langle \mathbf{w}_{r,t}^*, \boldsymbol{\mu}_j \rangle^3 \langle \mathbf{w}_{r,t}^0, \boldsymbol{\mu}_j \rangle + \langle \mathbf{w}_{r,t}^*, \boldsymbol{\xi}_i \rangle^3 \langle \mathbf{w}_{r,t}^0, \boldsymbol{\xi}_i \rangle \big) \|\mathbf{w}_{r,t}^*\|^2 \Big)$$

$$+ \Theta \Big( \big( \langle \mathbf{w}_{r,t}^*, \boldsymbol{\mu}_j \rangle \langle \mathbf{w}_{r,t}^0, \boldsymbol{\mu}_j \rangle + \langle \mathbf{w}_{r,t}^*, \boldsymbol{\xi}_i \rangle \langle \mathbf{w}_{r,t}^0, \boldsymbol{\xi}_i \rangle \big) \|\mathbf{w}_{r,t}^*\|^4 \Big)$$

And we can verify that $\mathbf{W}_t^*$ is a stationary point if and only if for all $j = \pm 1, r \in [m], i \in [n]$, $\langle \nabla_{\mathbf{w}_{r,t}} L(\mathbf{W}_t^*), \boldsymbol{\mu}_j \rangle = \langle \nabla_{\mathbf{w}_{r,t}} L(\mathbf{W}_t^*), \boldsymbol{\xi}_i \rangle = \langle \nabla_{\mathbf{w}_{r,t}} L(\mathbf{W}_t^*), \mathbf{w}_{r,t}^0 \rangle = 0$. This leads to the following equation system:

$$\sqrt{m} \Theta \Big( \langle \mathbf{w}_{r,t}^*, \boldsymbol{\mu}_j \rangle^5 + \langle \mathbf{w}_{r,t}^*, \boldsymbol{\mu}_j \rangle^3 \|\mathbf{w}_{r,t}^*\|^2 + \langle \mathbf{w}_{r,t}^*, \boldsymbol{\xi}_i \rangle^4 \langle \mathbf{w}_{r,t}^*, \boldsymbol{\mu}_j \rangle + \|\mathbf{w}_{r,t}^*\|^4 \langle \mathbf{w}_{r,t}^*, \boldsymbol{\mu}_j \rangle$$

$$+ \langle \mathbf{w}_{r,t}^*, \boldsymbol{\xi}_i \rangle^2 \|\mathbf{w}_{r,t}^*\|^2 \langle \mathbf{w}_{r,t}^*, \boldsymbol{\mu}_j \rangle + \langle \mathbf{w}_{r,t}^*, \boldsymbol{\mu}_j \rangle^3 \|\mathbf{w}_{r,t}^*\|^2 \|\boldsymbol{\mu}_j\|^2 + \|\mathbf{w}_{r,t}^*\|^4 \langle \mathbf{w}_{r,t}^*, \boldsymbol{\mu}_j \rangle \|\boldsymbol{\mu}_j\|^2 \Big)$$

$$= \Theta \Big( \langle \mathbf{w}_{r,t}^*, \boldsymbol{\xi}_i \rangle \langle \mathbf{w}_{r,t}^*, \boldsymbol{\mu}_j \rangle + \langle \mathbf{w}_{r,t}^*, \boldsymbol{\mu}_j \rangle^2 + \|\mathbf{w}_{r,t}^*\|^2 \|\boldsymbol{\mu}_j\|^2 \Big) \tag{48}$$

$$\sqrt{m} \Theta \Big( \langle \mathbf{w}_{r,t}^*, \boldsymbol{\xi}_i \rangle^5 + \langle \mathbf{w}_{r,t}^*, \boldsymbol{\xi}_i \rangle^3 \|\mathbf{w}_{r,t}^*\|^2 + \langle \mathbf{w}_{r,t}^*, \boldsymbol{\mu}_j \rangle^4 \langle \mathbf{w}_{r,t}^*, \boldsymbol{\xi}_i \rangle + \|\mathbf{w}_{r,t}^*\|^4 \langle \mathbf{w}_{r,t}^*, \boldsymbol{\xi}_i \rangle$$

$$+ \langle \mathbf{w}_{r,t}^*, \boldsymbol{\mu}_j \rangle^2 \|\mathbf{w}_{r,t}^*\|^2 \langle \mathbf{w}_{r,t}^*, \boldsymbol{\xi}_i \rangle + \frac{1}{n} \langle \mathbf{w}_{r,t}^*, \boldsymbol{\xi}_i \rangle^3 \|\mathbf{w}_{r,t}^*\|^2 \|\boldsymbol{\xi}_i\|^2 + \frac{1}{n} \|\mathbf{w}_{r,t}^*\|^4 \langle \mathbf{w}_{r,t}^*, \boldsymbol{\xi}_i \rangle \|\boldsymbol{\xi}_i\|^2 \Big)$$

$$= \Theta \Big( \langle \mathbf{w}_{r,t}^*, \boldsymbol{\mu}_j \rangle \langle \mathbf{w}_{r,t}^*, \boldsymbol{\xi}_i \rangle + \langle \mathbf{w}_{r,t}^*, \boldsymbol{\xi}_i \rangle^2 + \frac{1}{n} \|\mathbf{w}_{r,t}^*\|^2 \|\boldsymbol{\xi}_i\|^2 \Big) \tag{49}$$

$$\sqrt{m} \Theta \Big( \frac{1}{m} \big( \langle \mathbf{w}_{r,t}^*, \boldsymbol{\mu}_j \rangle^4 + \langle \mathbf{w}_{r,t}^*, \boldsymbol{\xi}_i \rangle^4 \big) \langle \mathbf{w}_{r,t}^*, \mathbf{w}_{r,t}^0 \rangle + \big( \langle \mathbf{w}_{r,t}^*, \boldsymbol{\mu}_j \rangle^2 + \langle \mathbf{w}_{r,t}^*, \boldsymbol{\xi}_i \rangle^2 \big) \|\mathbf{w}_{r,t}^*\|^2 \langle \mathbf{w}_{r,t}^*, \mathbf{w}_{r,t}^0 \rangle$$

$$+ \|\mathbf{w}_{r,t}^*\|^4 \langle \mathbf{w}_{r,t}^*, \mathbf{w}_{r,t}^0 \rangle + \big( \langle \mathbf{w}_{r,t}^*, \boldsymbol{\mu}_j \rangle^3 \langle \mathbf{w}_{r,t}^0, \boldsymbol{\mu}_j \rangle + \langle \mathbf{w}_{r,t}^*, \boldsymbol{\xi}_i \rangle^3 \langle \mathbf{w}_{r,t}^0, \boldsymbol{\xi}_i \rangle \big) \|\mathbf{w}_{r,t}^*\|^2$$

$$+ \big( \langle \mathbf{w}_{r,t}^*, \boldsymbol{\mu}_j \rangle \langle \mathbf{w}_{r,t}^0, \boldsymbol{\mu}_j \rangle + \langle \mathbf{w}_{r,t}^*, \boldsymbol{\xi}_i \rangle \langle \mathbf{w}_{r,t}^0, \boldsymbol{\xi}_i \rangle \big) \|\mathbf{w}_{r,t}^*\|^4 \Big)$$

$$= \Theta \Big( \langle \mathbf{w}_{r,t}^*, \boldsymbol{\mu}_j + \boldsymbol{\xi}_i \rangle \langle \mathbf{w}_{r,t}^*, \mathbf{w}_{r,t}^0 \rangle + \|\mathbf{w}_{r,t}^*\|^2 \langle \mathbf{w}_{r,t}^0, \boldsymbol{\mu}_j + \boldsymbol{\xi}_i \rangle \Big) \tag{50}$$

In order to solve the system, we let $\tau_{i,j} := \frac{\langle \mathbf{w}_{r,t}^*, \boldsymbol{\mu}_j \rangle}{\langle \mathbf{w}_{r,t}^*, \boldsymbol{\xi}_i \rangle}$ for any $i \in [n], j = \pm 1$. We let $\tau = \Theta(\tau_{i,j})$. We first consider solving (48) and (49) and then analyze (50).

Furthermore, because the claims (1-4) are assumed, we can leverage Lemma D.4 to decompose

$$\|\mathbf{w}_{r,t}^*\|^2 = \Theta \big( \langle \mathbf{w}_{r,t}^*, \boldsymbol{\mu}_j \rangle^2 \|\boldsymbol{\mu}\|^{-2} + n \cdot \mathrm{SNR}^2 \langle \mathbf{w}_{r,t}^*, \boldsymbol{\xi}_i \rangle^2 \|\boldsymbol{\mu}\|^{-2} + \langle \mathbf{w}_{r,t}^*, \mathbf{w}_{r,t}^0 \rangle^2 \|\mathbf{w}_{r,t}^0\|^{-2} \big)$$

$$= \Theta \big( (\tau^2 + n \cdot \mathrm{SNR}^2) \langle \mathbf{w}_{r,t}^*, \boldsymbol{\xi}_i \rangle^2 \|\boldsymbol{\mu}\|^{-2} + \langle \mathbf{w}_{r,t}^*, \mathbf{w}_{r,t}^0 \rangle^2 \|\mathbf{w}_{r,t}^0\|^{-2} \big)$$

where the third equality is by the scale of $\langle \mathbf{w}_{r,t}^*, \boldsymbol{\mu}_j \rangle$ and $\langle \mathbf{w}_{r,t}^*, \boldsymbol{\xi}_i \rangle$.

Next, we separately consider three SNR conditions, namely (1) $n \cdot \mathrm{SNR}^2 = \Theta(1)$; (2) $n \cdot \mathrm{SNR}^2 \geq \widetilde{\Omega}(1)$; and (3) $n^{-1} \cdot \mathrm{SNR}^{-2} \geq \widetilde{\Omega}(1)$.

1. When $n \cdot \mathrm{SNR}^2 = \Theta(1)$: we first can bound $\langle \mathbf{w}_{r,t}^*, \mathbf{w}_{r,t}^0 \rangle \|\mathbf{w}_{r,t}^0\|^{-1} \leq \Theta(\langle \mathbf{w}_{r,t}^*, \boldsymbol{\mu}_j \rangle)$ and derive

$$\|\mathbf{w}_{r,t}^*\|^2 = \max\{\Theta(\langle \mathbf{w}_{r,t}^*, \boldsymbol{\mu} \rangle^2), \Theta(\langle \mathbf{w}_{r,t}^*, \boldsymbol{\xi}_i \rangle^2)\} \|\boldsymbol{\mu}\|^{-2}$$

Next, we can simplify (48) and (49) depending on the scale of $\tau$.

- When $\tau = \widetilde{\Omega}(1)$, we have $\|\mathbf{w}_{r,t}^*\|^2 = \Theta(\langle \mathbf{w}_{r,t}^*, \boldsymbol{\mu}_j \rangle^2) \|\boldsymbol{\mu}_j\|^{-2}$ and the equations reduce to

$$\begin{cases} \Theta(\sqrt{m} \tau^5 \langle \mathbf{w}_{r,t}^*, \boldsymbol{\xi}_i \rangle^5) = \Theta(\tau^2 \langle \mathbf{w}_{r,t}^*, \boldsymbol{\xi}_i \rangle^2) \\ \Theta(\sqrt{m} \tau^4 \langle \mathbf{w}_{r,t}^*, \boldsymbol{\xi}_i \rangle^5) = \Theta(\tau^2 \langle \mathbf{w}_{r,t}^*, \boldsymbol{\xi}_i \rangle^2) \end{cases}$$

It is clear to see for $\tau = \widetilde{\Omega}(1)$, the equations cannot be jointly satisfied.

- When $\tau^{-1} = \widetilde{\Omega}(1)$, we have $\|\mathbf{w}_{r,t}^*\|^2 = \Theta(\langle \mathbf{w}_{r,t}^*, \boldsymbol{\xi}_i \rangle^2) \|\boldsymbol{\mu}_j\|^{-2}$ and the equations reduce to

$$\begin{cases} \Theta(\sqrt{m}\tau \langle \mathbf{w}_{r,t}^*, \boldsymbol{\xi}_i \rangle^5) = \Theta(\langle \mathbf{w}_{r,t}^*, \boldsymbol{\xi}_i \rangle^2) \\ \Theta(\sqrt{m} \langle \mathbf{w}_{r,t}^*, \boldsymbol{\xi}_i \rangle^5) = \Theta(\langle \mathbf{w}_{r,t}^*, \boldsymbol{\xi}_i \rangle^2) \end{cases}$$

which cannot be satisfied simultaneously for $\tau^{-1} = \widetilde{\Omega}(1)$.
- When $\tau = \Theta(1)$, $\|\mathbf{w}_{r,t}^*\|^2 = \Theta(\langle \mathbf{w}_{r,t}^*, \boldsymbol{\mu}_j \rangle^2) \|\boldsymbol{\mu}_j\|^{-2} = \Theta(\langle \mathbf{w}_{r,t}^*, \boldsymbol{\xi}_i \rangle^2) \|\boldsymbol{\mu}_j\|^{-2}$ and thus we can simplify the equations to

$$\begin{cases} \Theta(\sqrt{m} \langle \mathbf{w}_{r,t}^*, \boldsymbol{\xi}_i \rangle^5) = \Theta(\langle \mathbf{w}_{r,t}^*, \boldsymbol{\xi}_i \rangle^2) \\ \Theta(\sqrt{m} \langle \mathbf{w}_{r,t}^*, \boldsymbol{\xi}_i \rangle^5) = \Theta(\langle \mathbf{w}_{r,t}^*, \boldsymbol{\xi}_i \rangle^2) \end{cases}$$

which has a solution with $\langle \mathbf{w}_{r,t}^*, \boldsymbol{\xi}_i \rangle = \Theta(m^{-1/6}) = \langle \mathbf{w}_{r,t}^*, \boldsymbol{\mu}_j \rangle$, thus verifying the scale and $\tau = \Theta(n \cdot \text{SNR}^2)$. Then we can verify (50) holds under the scale of $\langle \mathbf{w}_{r,t}^*, \boldsymbol{\xi}_i \rangle, \langle \mathbf{w}_{r,t}^*, \boldsymbol{\mu}_j \rangle = \Theta(m^{-1/6})$ and $\|\mathbf{w}_{r,t}^*\|^2 = \Theta(m^{-1/3})$. With the same argument as in Lemma D.8, we can show $\langle \nabla_{\mathbf{w}_{r,t}} L(\mathbf{W}_t^*), \mathbf{w}_{r,t}^0 \rangle = 0$.

2. When $n \cdot \text{SNR}^2 = \widetilde{\Omega}(1)$: we first can bound $\langle \mathbf{w}_{r,t}^*, \mathbf{w}_{r,t}^0 \rangle \|\mathbf{w}_{r,t}^0\|^{-1} \le \Theta(\langle \mathbf{w}_{r,t}^*, \boldsymbol{\mu}_j \rangle)$ and derive

$$\|\mathbf{w}_{r,t}^*\|^2 = \max\{\Theta(\tau^2), \Theta(n\text{SNR}^2)\} \langle \mathbf{w}_{r,t}^*, \boldsymbol{\xi}_i \rangle^2 \|\boldsymbol{\mu}\|^{-2}.$$

We only consider the scale when $\tau = \Theta(n \cdot \text{SNR}^2)$, where we can simplify $\|\mathbf{w}_{r,t}^*\|^2 = \Theta(\tau^2) \langle \mathbf{w}_{r,t}^*, \boldsymbol{\xi}_i \rangle^2 \|\boldsymbol{\mu}_j\|^{-2}$ and thus the system becomes

$$\begin{cases} \Theta(\sqrt{m}\tau^5 \langle \mathbf{w}_{r,t}^*, \boldsymbol{\xi}_i \rangle^5) = \Theta(\tau^2 \langle \mathbf{w}_{r,t}^*, \boldsymbol{\xi}_i \rangle^2) \\ \Theta(\sqrt{m}\tau^4 \langle \mathbf{w}_{r,t}^*, \boldsymbol{\xi}_i \rangle^5) = \Theta(\tau \langle \mathbf{w}_{r,t}^*, \boldsymbol{\xi}_i \rangle^2) \end{cases}$$

In order to satisfy both equations, we require $\langle \mathbf{w}_{r,t}^*, \boldsymbol{\xi}_i \rangle = \Theta(\tau^{-1} m^{-1/6})$ and $\langle \mathbf{w}_{r,t}^*, \boldsymbol{\mu}_j \rangle = \Theta(m^{-1/6})$, which verifies the scale. We can then verify (50) holds under such condition. With the same argument as in Lemma D.8, we can show $\langle \nabla_{\mathbf{w}_{r,t}} L(\mathbf{W}_t^*), \mathbf{w}_{r,t}^0 \rangle = 0$.

3. When $n^{-1} \cdot \text{SNR}^{-2} = \widetilde{\Omega}(1)$: we first can bound $\langle \mathbf{w}_{r,t}^*, \mathbf{w}_{r,t}^0 \rangle \|\mathbf{w}_{r,t}^0\|^{-1} \le \Theta(\sqrt{n \cdot \text{SNR}^2} \langle \mathbf{w}_{r,t}^*, \boldsymbol{\xi}_i \rangle)$ and derive

$$\|\mathbf{w}_{r,t}^*\|^2 = \max\{\Theta(\tau^2), \Theta(n\text{SNR}^2)\} \langle \mathbf{w}_{r,t}^*, \boldsymbol{\xi}_i \rangle^2 \|\boldsymbol{\mu}_j\|^{-2}.$$

We only consider the scale when $\tau = \Theta(n \cdot \text{SNR}^2)$, where we can simplify $\|\mathbf{w}_{r,t}^*\|^2 = \Theta(n \cdot \text{SNR}^2) \langle \mathbf{w}_{r,t}^*, \boldsymbol{\xi}_i \rangle^2 \|\boldsymbol{\mu}_j\|^{-2}$ and thus the system becomes

$$\begin{cases} \Theta(\sqrt{m}\tau \langle \mathbf{w}_{r,t}^*, \boldsymbol{\xi}_i \rangle^5) = \Theta(n\text{SNR}^2 \langle \mathbf{w}_{r,t}^*, \boldsymbol{\xi}_i \rangle^2) \\ \Theta(\sqrt{m} \langle \mathbf{w}_{r,t}^*, \boldsymbol{\xi}_i \rangle^5) = \Theta(\langle \mathbf{w}_{r,t}^*, \boldsymbol{\xi}_i \rangle^2) \end{cases}$$

which can be satisfied when $\langle \mathbf{w}_{r,t}^*, \boldsymbol{\xi}_i \rangle = \Theta(m^{-1/6})$ and $\langle \mathbf{w}_{r,t}^*, \boldsymbol{\mu}_j \rangle = \Theta(\tau m^{-1/6})$ and thus verify the scale. In this case, we can also verify that (50) holds under the condition that $m = \Theta(1)$. With the same argument as in Lemma D.8, we can show $\langle \nabla_{\mathbf{w}_{r,t}} L(\mathbf{W}_t^*), \mathbf{w}_{r,t}^0 \rangle = 0$.

This concludes the proof that suppose the scales and concentration are the same as the end of second stage, then there exists a stationary point where $\langle \mathbf{w}_{r,t}^*, \boldsymbol{\mu}_j \rangle / \langle \mathbf{w}_{r,t}^*, \boldsymbol{\xi}_i \rangle = \Theta(n \cdot \text{SNR}^2)$. $\qquad \square$

