# OpenReview forum: "On the Feature Learning in Diffusion Models"
_ICLR.cc/2025/Conference — ICLR 2025 Poster_

### Official Review · Reviewer_Cb9H · 2024-11-01

**Soundness:** 4
**Presentation:** 4
**Contribution:** 3
**Rating:** 6
**Confidence:** 3

**Summary:**

This paper investigates feature learning for diffusion models through a theoritical approach. Under assumptions of two-class label setup and using certrain two-layer neural networks, the authors reveal an interesting difference between diffusion models and classification models.  Diffusion models learn both signal and noise in a "balanced way", where the "learning" grows proportionally to $\text{sample size} \times SNR^2$. In contrast, classification models tend to focus on certain features and exhist a "jump" learning behavior when $\text{sample size} \times SNR^2$ exceeds some threshold.

This paper provides theoritical supports for their findings as well as experiments on synthetical data and MNIST data.

**Strengths:**

1. The paper provides a noval perspective to understand the behavior of diffusion model: the feature learning. Defusion model learns feature over noise in a balanced manner. This addresses a gap in existing theoritical study that focuses mainly on distribution learning and convergance. This provides insights on why diffusion model could learn weak features.

2. The paper provides a rigourous theoritical framework and validate its results with numerical experiments. Experiments results clearly illustrate the feature learning patterns.

3. The theoretical results are well-organized and explained.

**Weaknesses:**

1. Using SNR to demonstrate the feature learning pattern may oversimplify other factors. Real-world data can be heterogeneous, where signal structure and noise can vary significantly, making it unlikely to have a conistent SNR.

2. The assumptions of orthogonal signals and a single feature seem quite strong. Is there any justification provided for these choices?

3. The synthetic data experiment settings are somewhat limited. Including more SNR settings could provide a better illustration of how feature learning behavior changes gradually with different SNR values.

**Questions:**

1. In Figure 6 of Appendix A.3, the seeting is described as high SNR, but the caption states "In this low-SNR case, we see classification tends to predominately learn noise while diffusion learns both signal and noise." Seems the caption is not consistent with the setting in the figure. Is my understanding correct?

2. Comparing the plot of difuusion model in left and right panel of figue 2, the $n \, SNR^2$ change significantly from 0.75 to 6.75. However, the ratio between  $max(w_r, \mu)$ and $max(wr, \xi)$ does not appear to change as dramatically much. Could you explain how this aligns with your theory results?  What is the motivation behind selecting 0.75 and 6.75 as low and high SNR settings? I am a bit curious if you have experimental results for other SNR values?

**Details Of Ethics Concerns:**

No concern.

---

> ### Author Response · Authors · 2024-11-21
> **Responses to Reviewer Cb9H**
>
> We are grateful for the reviewer's positive comments on our work. We also appreciate the reviewer for raising constructive comments and questions. Below are our responses to each comment and question.
>
> ----
> **1. (W1) On the heterogeneous data structure where signal structure and noise can vary significantly.**
>
> We believe our analysis can be extended to multiple features by following a similar proof strategy. For instance, consider three feature patches $\mu_1, \mu_2, ..., \mu_k$ along with a noise patch where the SNRs are different (i.e., $||\mu_1|| \neq ||\mu_2|| \dots\neq ||\mu_k||$), we can still use the same analysis to write the gradient descent updates (in early stages) for diffusion models as follows:
> $$\langle w_{r,t}^{k+1} , \mu_j  \rangle = \langle w_{r,t}^k , \mu_j\rangle + \Theta(  \frac{\eta}{\sqrt{m}}||w_{r,t} ||^2 ||\mu_j||^2 ), j = 1,2,\dots,k$$
> $$\langle w_{r,t}^{k+1} , \xi_i  \rangle = \langle w_{r,t}^k , \xi_i\rangle + \Theta( \frac{\eta}{n \sqrt{m}} ||w_{r,t} ||^2 ||\xi_i||^2 )$$
> Then following a similar analysis, we can show even with varying SNRs, the feature learning outcomes of diffusion models would still be determined by their relative SNRs. Thus diffusion models still learn all the features. In contrast, existing studies [5,6] found that under the presence of multiple features with varying magnitude, classification models only lean partial features that are more significant.
>
> Therefore, we believe the conclusions drawn in this paper that diffusion models learn more balanced features than classification models would not change with more diverse features. We have conducted additional experiments on 10-class Noisy-MNIST dataset (in **Appendix I of the revised manuscript**) to numerically confirm this claim. See more details in Reply 2 below.
>
> ----
> **2. (W2) On the assumption of orthogonal signals and a single feature.**
>
> We have followed prior analysis on classification models [1,2,3,4] to introduce the orthogonality assumption, in order to facilitate theoretical analysis. The orthogonality assumption is used to simplify the analysis and enables clear insights into the training dynamics of diffusion models and classification.
>
> Furthermore, our analysis can be extended to address multiple features, as discussed in Reply 1 above.
>
> Numerically, in **Appendix I of the revised manuscript**, we conducted additional experiments on Noisy-MNIST dataset with *10 classes* (compared to 2 classes in the main paper). In this case, the features are more diverse and are *not* orthogonal. From the results, we see despite with more features, diffusion models learn all the features to relatively the same level whereas classification models predominantly learn subset of features. This verifies the applicability of the theoretical results in this paper to more realistic settings.
>
> ----
> **3. (W3) Experiments on different SNR values.**
>
> We have now conducted additional experiments over a broader range of SNR values, as detailed in **Appendix G of the revised manuscript**. These include $n SNR^2 = 1.92$, $n SNR^2 = 3$, and $n SNR^2 = 4.32$, complementing the main experiments on $n SNR^2 = 0.75$ and $n SNR^2 = 6.75$.
>
> From the results, we observe that classification models are more sensitive to changes in $n SNR^2$, often overfitting to one feature while ignoring others. In contrast, diffusion models consistently learn more balanced features across the range of SNR values. These findings are consistent with our theoretical analysis and further support our conclusions.
>
> ----
> **4. (Q1) Caption not consistent with setting in Figure 6.**
>
> Thank you for pointing this out. We have now corrected the caption for Figure 6 in the revised version.
>
> ----
> **5. (Q2) On the ratio of signal and noise learning in relation to $nSNR^2$.**
>
> This is consistent with our theory because we only claim  the ratio of signal to noise learning is on the same *order* of $n SNR^2$ (rather than exactly matching it). The absolute magnitude of changes in signal and noise inner products of diffusion models when switching from $nSNR^2 = 0.75$ to $6.75$ is not meaningful without benchmarking against the changes for classification models. In comparison, we see diffusion models are more aligned with the order of $n SNR^2$ compared to classification models.
>
> -----
> **Reference**
>
> [1] Cao et al. Benign overfitting in two-layer convolutional neural networks. In *NeurIPS 2022.*
>
> [2] Chen et al. Why does sharpness-aware minimization generalize better than SGD?. In *NeurIPS 2023*.
>
> [3] Chen et al. Understanding and improving feature learning for out-of-distribution generalization. In *NeurIPS 2023*.
>
> [4] Lu et al. Benign Oscillation of Stochastic Gradient Descent with Large Learning Rate. In *ICLR 2024*.
>
> [5] Zou et al. The benefits of mixup for feature learning. *ICML 2023*.
>
> [6] Shen et al. Data augmentation as feature manipulation. *ICML 2022*.

---

> > ### Comment · Reviewer_Cb9H · 2024-11-25
> >
> > Thank you to the authors for their response. It has addressed my concerns, and I continue to hold a positive overall view of this work. I have updated my ratings of soundness and presentation of this paper to reflect their improvement.

---

### Official Review · Reviewer_2j1t · 2024-11-03

**Soundness:** 3
**Presentation:** 3
**Contribution:** 2
**Rating:** 6
**Confidence:** 2

**Summary:**

This paper focuses on the theoretical analysis of the feature learning framework of diffusion models. While previous studies have primarily analyzed the generative capabilities of diffusion models, this work examines how diffusion models differ from traditional supervised learning in terms of feature learning. Specifically, they show that, unlike classification models that tend to focus dominantly on either signal or noise, diffusion models learn features that are balanced with respect to signal and noise. They experimentally validate their theoretical claims using synthetic datasets and the MNIST dataset.

**Strengths:**

* They provide a theoretical analysis of the features learned by diffusion models, a relatively unexplored area. While diffusion models are often expected to have good representational capabilities due to their powerful generative capabilities, they provide a preliminary theoretical grounding for this expectation.

* The comparison between diffusion models and classification models, showing how they differ in feature learning, is interesting and meaningful. This work could contribute to the long-standing research on the discriminative vs. generative perspective in machine learning.

**Weaknesses:**

* Comparison with other generative models, such as VAE, could be more informative. Diffusion models have unique features, such as the diffusion timesteps, that distinguish them from other generative models; an analysis of how these differences affect feature learning would add significant value.
* It would be beneficial to discuss how the theoretical results can be applied to real-world applications, even if only briefly, to provide practical relevance.
* The meaning of formulations could be further explained. For example, in Definition 2.1, it would be helpful to explain the rationale or significance of setting the variance of $\xi$ as defined.  Additionally, in Condition 3.1 (5), it would be useful to clarify whether the assumption of constant-order $\alpha_t$ and $\beta_t$ in diffusion models differs from conventional diffusion model formulations.
* A detailed explanation of the theoretical differences between diffusion and classification models would be valuable. For example, diffusion models tie two layers, while classification models fix the second layer to ±1; it needs to be explained whether this difference affects the theoretical results.
* It would be helpful to know whether the theoretical analysis could also be applied to the score matching loss [1] and the reconstruction loss [2] in diffusion models.

[1] Song et al., Score-Based Generative Modeling through Stochastic Differential Equations, ICLR 2021
[2] Karras et al., Elucidating the Design Space of Diffusion-Based Generative Models, NeurIPS 2022

**Questions:**

* Could the authors provide their thoughts on the issues raised in Weaknesses?

---

> ### Author Response · Authors · 2024-11-21
> **Responses to Reviewer 2j1t (Part 1)**
>
> We sincerely appreciate the reviewer for acknowledging our contributions and below we provide detailed responses to all the comments and questions.
>
> -----
> **1. (W1) Comparison to other generative models.**
>
> We would like to emphasize that the goal of this work is to understand the training dynamics of diffusion models, rather than all generative models. While we agree that comparing diffusion models to other generative models is interesting, it is beyond the scope of this work.
>
> Nevertheless, we believe the theoretical frameworks developed in this work can be extended to other generative models, such as VAEs. We have added **Appendix F in the revised version** discussing the feature learning of VAEs, both theoretically and numerically.
>
> We find that under the current setup, the objective of VAE  simplified to include a reconstruction term and some regularization terms. The reconstruction compels the model to learn both the signals and noise present in the data, thus promoting a balanced feature learning. To test this claim, we conducted an experiment, examining the feature learning dynamics of VAEs. We notice that similar to diffusion model, VAEs learn more balanced features compared to classification. However, a thorough comparison between diffusion models and VAEs would require dedicated efforts and additional analysis, which are beyond the scope of this paper.
>
> ----
> **2. (W1) On the effect of diffusion timesteps on feature learning.**
>
> In this work, we consider a fixed range of timesteps rather than all timesteps in order to focusing on the training dynamics of diffusion model at specific timesteps and their effects on feature learning. Nevertheless, extending the analysis to cover broader scales of timesteps is indeed our next step. This extension would allow us to further investigate how features are reconstructed throughout the entire reverse diffusion process.
>
> -----
> **3. (W2) Discussion on how the theoretical results apply to real-world applications.**
>
> We believe the theoretical results in this paper have real-world substance. For instance, the more balanced feature learning of diffusion models may be able to explain the inherent adversarial robustness of diffusion models used for downstream classification (as shown in [3,4]). Because the learning outcomes of diffusion models are less sensitive to changes in SNR compared to models trained for classification, the feature learning of diffusion models tend to be more resistant to adversarial perturbation.
>
> We have now added such discussions in the conclusion section (**Section 6 of the revised version**).
>
> ----
> **4. (W3) Explanation on the meaning of formulations.**
>
> (1) The noise $\xi$ is generated from the Gaussian distribution such that it is orthogonal to the signal vectors $\mu_1, \mu_{-1}$ for simplicity of analysis. Such an orthogonal signal-noise setup is common in existing works, such as [5,6].
>
> (2) We believe that the constant order of $\alpha_t$ and $\beta_t$ reflects the standard practice in diffusion models. For instance, in DDPM [7], the noise schedules are pre-defined, and the number of timesteps is typically fixed. This ensures that $\alpha_t$ and $\beta_t$ remain of constant order, independent of the data or model size. Nevertheless, we agree that extending the analysis to cover broader range of $t$ is an important future direction as we have discussed in Reply 2.
>
> We have added the explanations in the revised version accordingly.
>
> -----
> **5. (W4) On the effect of different networks for classification and diffusion models on feature learning.**
>
> Due to the differences in the objectives and output sizes between diffusion models and classification, it is difficult to employ exactly the same model setup. The current model setup, i.e., weight-tying for diffusion model and weight-fixing for classification model ensures the first-layer and the trainable parameters are consistent across two models.
>
>
> We believe such a difference in networks *does not affect the theoretical results*. To validate such a claim, we have conducted an additional experiment in **Appendix H of the revised manuscript**, comparing feature learning dynamics between diffusion model and classification on a three-layer networks. The results indicate that while adding more layers, the similar learning patterns still maintain, i.e., diffusion model learns more balanced features while classification focuses on learning partial features.

---

> > ### Author Response · Authors · 2024-11-21
> > **Responses to Reviewer 2j1t (Part 2)**
> >
> > **6. (W5) Applicability of theoretical analysis to score matching loss [1] and reconstruction loss [2].**
> >
> > We believe our analysis can be adapted to other objectives of diffusion models. In fact, under the current setup, the DDPM loss employed in this paper is equivalent to the (denoising) score matching loss [1] up to some re-scaling that depends on the diffusion coefficients (see e.g., [8]). This is because the conditional score is equal to the re-scaled Gaussian noise added, i.e., $\nabla \log_{x}p(x|x_0) = - \epsilon_t/\beta_t$, which suggests the (denoising) score matching loss is equivalent to $E||  f(x_t) + \epsilon_t/\beta_t||^2$ . Thus, within the current framework where $\alpha_t$ and $\beta_t$ are in constant order, our analysis is directly applicable to the score matching loss.
> >
> > In addition, we can apply a similar analysis (developed in this work) for the reconstruction loss [2]. Specifically, if we replace the added noise  by input in the objective, we can still see the gradient descent dynamics follows in the direction of signal and noise. Below we show the dominate terms in the gradient updates in the early-stages:
> > $$\langle w_{r}^{k+1} , \mu_j  \rangle = \langle w_r^k , \mu_j\rangle + \Theta( \eta \langle w_r^k, \mu_j \rangle^3 )$$
> > $$\langle w_{r}^{k+1} , \xi_i  \rangle = \langle w_r^k , \xi_i\rangle + \Theta( \frac{\eta}{n} \langle w_r^k, \xi_i \rangle^3 )$$
> > Then following a similar analysis, we can analyze the feature learning dynamics. The reconstruction loss forces the model to learn all features and we believe the key finding holds: diffusion models still learn balanced features when trained with the reconstruction loss.
> >
> > -----
> > **Reference**
> >
> > [1] Song et al., Score-Based Generative Modeling through Stochastic Differential Equations, *ICLR 2021*.
> >
> > [2] Karras et al., Elucidating the Design Space of Diffusion-Based Generative Models, *NeurIPS 2022*.
> >
> > [3] Li et al. Your diffusion model is secretly a zero-shot classifier. *ICCV 2023*.
> >
> > [4] Chen et al. Your diffusion model is secretly a certifiably robust classifier. *NeurIPS 2024.*
> >
> > [5] Cao et al. Benign overfitting in two-layer convolutional neural networks. NeurIPS 2022.
> >
> > [6] Chen et al. Understanding and improving feature learning for out-of-distribution generalization. In NeurIPS 2023.
> >
> > [7] Ho et al. Denoising diffusion probabilistic models. *NeurIPS 2020.*
> >
> > [8] Chen et al. Sampling is as easy as learning the score: theory for diffusion models with minimal data assumptions. *ICLR 2023.*

---

> > > ### Comment · Reviewer_2j1t · 2024-11-26
> > >
> > > Thank you for the authors' response. The authors have adequately addressed most of my concerns, and accordingly I am raising my rating from 5 to 6, leaning toward acceptance.

---

### Official Review · Reviewer_T68k · 2024-11-03

**Soundness:** 4
**Presentation:** 4
**Contribution:** 3
**Rating:** 6
**Confidence:** 2

**Summary:**

This paper provides a theoretical analysis of feature learning in diffusion models and contrasts that with feature learning in classification models. This work finds that diffusion models tend to prioritize learn balanced representations of the data whereas classification tends to prioritize learning easy to learn features of the data.

As an applied researcher with no experience in the theory behind feature learning or theoretical analysis of neural networks, I honestly can’t explain how I was assigned this paper. As such it is difficult for me to judge the novelty, impact, and soundness of this work.

**Strengths:**

* The motivation and essential results were clear even to a non-expert in this subfield
* To the best of my knowledge no one has characterized the differences in feature learning in these settings.

**Weaknesses:**

* I don’t find it particularly surprising that when training on Gaussian noised samples that the features learned would be broader than those trained on clean samples where small details that will be destroyed by Gaussian noise will still appear.

**Questions:**

* Would this analysis apply to the training of a classifier on the same levels of noise as a diffusion model? I.e. my question is whether these results apply specifically to diffusion models, or if they also apply to any model trained on all levels of (Gaussian?) noise.
* Does this analysis also apply to flow-based models used in flow-matching? I would guess that this is likely and it may increase the impact of this work to study more general frameworks.

---

> ### Author Response · Authors · 2024-11-21
> **Responses to Reviewer T68k**
>
> We are grateful for the reviewer’s acknowledgment of our work’s contributions. We also appreciate all the comments and questions raised by the reviewer. Below are our detailed responses to each comment.
>
> ----
> **1. (W1 + Q1) On the feature learning when training on noised inputs.**
>
> We would like to clarify that the difference between diffusion model and classification models in feature learning lies in their different training objectives, rather than the additive Gaussian noise to the data. In fact, when performing the classification on the Gaussian noisy data, the feature learning results will not be changed. In particular, this can be justified both from both *theory* and *experiments*:
>
> * **Theoretically**, in **Appendix J of the revised manuscript**, we show that if we replace $\mu$ and $\xi$ with $\mu + \epsilon$ and $\xi + \epsilon$, and take expectation of the gradient with respect to the noise added, we find that
> $$E_{\epsilon_{i} } [\nabla_{w_{j,r}} \tilde L_S(W^k)] \approx \nabla_{w_{j,r}} L_S(W^k) + \frac{1}{nm} (\sum_{i=1}^n \ell_i'^k) w_{j,r}^k$$
> where $\tilde L_S(W) = \frac{1}{n}\sum_{i=1}^n \ell(y_i f(W, x_i + \epsilon_i))$ and $L_S(W) = \frac{1}{n}\sum_{i=1}^n \ell(y_i f(W, x_i))$ denote the noise-added objective and noise-free objective respectively. The above expression of gradient implies that the Gaussian noise perturbation is equivalent to (in expectation) adding an $L_2$-type regularization. This regularization will perform the same penalization for both learning signals and noises, thus will not alter our current results for classification models.
>
> * **Numerically**, we have conducted an **additional experiment** in **Appendix J of the revised manuscript**, examining the feature learning dynamics of classification models with the same Gaussian noise injected (as for diffusion model). The results suggest that even with the presence of Gaussian noise in the inputs, classification still overly rely on partial features, rather than the balanced feature learning performed by diffusion model training. This validates our claim that training classifiers on Gaussian-noised inputs *does not* promote the learning of more balanced features.
>
> ----
> **2. (Q2) Applicability of analysis to flow-based models.**
>
> Thank you for your question. We agree that extending our analysis to flow-based models is an interesting direction for future work, particularly given the importance of comparing diffusion models with flow-based models. The key differences between these approaches lie in (1) diffusion paths and (2) their prediction targets, which consequently result in different training objectives. This naturally aligns with our next step, which aims to investigate the role of training objectives in feature learning dynamics of diffusion models.

---

> > ### Comment · Reviewer_T68k · 2024-11-22
> >
> > I thank the authors for their response. The response has addressed my concerns, and I maintain my overall positive view of this work.

---

### Official Review · Reviewer_cG9c · 2024-11-04

**Soundness:** 3
**Presentation:** 3
**Contribution:** 3
**Rating:** 6
**Confidence:** 3

**Summary:**

The paper introduces a new theoretical framework specifically for analyzing the feature learning dynamics in diffusion models compared to classification models. This is a significant contribution to the existing body of literature, which has primarily focused on the generative aspects of diffusion models.

**Strengths:**

- The authors not only provide a theoretical analysis of the feature learning process in diffusion models but also validate their findings through experiments. This strengthens the credibility and practical relevance of the proposed framework.
- A key contribution of the work is the comparison between diffusion models and classification models in terms of feature learning dynamics. The results show that diffusion models tend to learn more balanced representations of both signal and noise, which contrasts with the classification models' tendency to focus on one or the other, depending on the signal-to-noise ratio (SNR).
- The paper provides clear mathematical results (e.g., Theorem 1.1 and Theorem 3.1) that demonstrate key differences between diffusion and classification models, particularly in terms of feature learning speed and final outcomes. These results are well-supported by rigorous proofs and visualization (e.g., Figure 1).

**Weaknesses:**

- While the theoretical insights are valuable, the paper does not sufficiently discuss how these findings might impact practical applications of diffusion models. The focus is more on theoretical analysis, and less on how these insights could be used to improve real-world tasks like image generation, classification, or other downstream applications.
- The theoretical results depend on several strict assumptions (e.g., Condition 3.1), particularly regarding the dimensionality and network initialization. In real-world scenarios, these assumptions may not hold, limiting the practical applicability of the theoretical findings.

**Questions:**

- The theoretical analysis relies on specific assumptions about high-dimensional spaces and signal-to-noise ratios. How would these theoretical insights hold up when applied to real-world datasets like ImageNet or COCO? Is there any plan to test these results on more practical benchmarks?
- The analysis is based on a simple two-layer convolutional network. If the architecture changes—say, using deeper networks or different types like ResNets or Transformers—would the feature learning dynamics of diffusion models change? Are there any experimental or theoretical results to explore this?
- The paper suggests that classification models tend to overfit noise when the SNR is low. However, in certain tasks, could learning noise or background features also be beneficial (e.g., in robustness or generalization)? Is there any discussion or analysis of cases where noise learning might be advantageous?
- Diffusion models are typically slower to train due to their iterative denoising process. The paper highlights the balanced feature learning in diffusion models, but are there any methods to accelerate training while maintaining this advantage? Would techniques like accelerated sampling or reduced time steps affect the feature learning dynamics?
- Besides classification, how do diffusion models fare in other tasks like object detection, semantic segmentation, or even reinforcement learning? Are there any theoretical or empirical results that support the use of diffusion models in these more complex tasks?

**Details Of Ethics Concerns:**

None.

---

> ### Author Response · Authors · 2024-11-21
> **Responses to Reviewer cG9c (Part 1)**
>
> We sincerely appreciate the reviewer’s positive feedback on our work, as well as the comments and questions. We would like to take this opportunity to further clarify the concerns raised and highlight our contributions. Below are our detailed responses to all the comments.
>
> ----
> **1. (W1 + Q1) How these findings impact practical applications of diffusion models. How the insights hold up when applied to real-world datasets.**
>
> We believe our findings provide valuable insights into the properties of diffusion models. For instance, the results in the paper potentially offer an explanation for the inherent adversarial robustness of (pre-trained) diffusion model when used as classifiers (as in [1,2]). Since the learning outcomes of diffusion models are less sensitive to changes in SNRs compared to classification models, diffusion models may be more resistant to adversarial perturbation.  We have now added such discussions in the conclusion section (**Section 6 of the revised manuscript**).
>
>
> While experimenting on large-scale datasets, such as ImageNet, is difficult within the limited rebuttal period, we conducted an **additional experiment** with 10-class Noisy-MNIST dataset, which contains more diverse features (than the 2-class Noisy-MNIST used in the main paper). We have included the experiment settings and results in **Appendix I of the revised manuscript**. The results indicate similar learning dynamics to those observed in the 2-class setup: diffusion models learn more balanced features, whereas classification models predominantly rely on a subset of features. This suggests the insights in the paper hold for more real-world scenarios.
>
> In theory, such a more complicated dataset can be modelled as the data distribution with multiple signals and we plan to extend our theoretical analysis to this more challenging setting. Intuitively, our main technical analysis can still be adapted. For instance, let $\mu_1,\mu_2,\dots,\mu_k$ be the signals that have different strengths, we may still obtain their early-phase learning dynamics as follows:
> $$\langle w_{r,t}^{k+1} , \mu_j  \rangle = \langle w_{r,t}^k , \mu_j\rangle + \Theta(  \frac{\eta}{\sqrt{m}}||w_{r,t} ||^2 ||\mu_j||^2 ), j = 1,2,\dots,k$$
> $$\langle w_{r,t}^{k+1} , \xi_i  \rangle = \langle w_{r,t}^k , \xi_i\rangle + \Theta( \frac{\eta}{n \sqrt{m}} ||w_{r,t} ||^2 ||\xi_i||^2 )$$
> When the first-order terms dominate the learning dynamics, we may still prove balanced feature learning in this setting. Such a more general result can further trigger a number of follow-up works on understanding more detailed training designs of diffusion models.
>
> Finally, we believe that this *first* attempt at analyzing the training dynamics of diffusion models establishes a theoretical foundation for advancing our understanding of diffusion model training and paves the way for exploring more complex problem setups. For instance, we expect the theoretical framework can inspire research on the roles of the loss function, model architecture, and optimizers in diffusion models. This can improve our understanding of the strengths and weaknesses of current practices, such as the Adam optimizer, U-Net architecture, and flow-matching versus score-matching losses.
>
> ----
> **2. (W2) On the assumptions being strict, particularly the dimensionality and initialization.**
>
> Our theoretical setups follow a series of prior theoretical works on classification models [3,5,6,9,10], where similar assumptions were widely adopted to simplify the analysis and highlight key phenomena.
>
> Regarding dimensionality and initialization, our work seeks to provide a theoretical comparison of the feature learning dynamics of diffusion models and classification in the over-parameterization regime. The high-dimensional setting we adopted ensures sufficient over-parameterization, as the number of trainable parameters is $d \times m$. Additionally, we require small Gaussian initialization to enable successful feature learning and convergence analysis, as demonstrated in [1]. Using a large initialization would place the model in the lazy/NTK regime [4], which does not allow for meaningful comparisons of feature learning dynamics.

---

> > ### Author Response · Authors · 2024-11-21
> > **Responses to Reviewer cG9c (Part 2)**
> >
> > **3. (Q2) On the feature learning dynamics of diffusion models with deeper networks.**
> >
> > Thank you for the question. We believe the key insights into the feature learning dynamics presented in this paper *would not change*  with deeper networks. To support this claim, in **Appendix H of the revised manuscript**, we have conducted an **additional experiment** examining the feature learning dynamics of three-layer diffusion and classification models. The results show similar patterns to those observed in the two-layer setup: diffusion models tend to learn all features, whereas classification models predominantly rely on learning partial features. These findings suggest that the insights derived in this paper are likely to hold for deeper models and potentially for more advanced architectures.
> >
> > However, a rigorous theoretical analysis of such more complex setups would require dedicated efforts, which is beyond the scope of this paper. Without our preliminary studies on shallow networks, it would be even infeasible to analyze deeper networks or other more advanced architectures, such as Transformers. We believe this work lays the foundation for future analysis of deeper models and advanced architectures.
> >
> > -----
> > **4. (Q3) On the potential benefits of learning noise or background features.**
> >
> > Thank you for the thoughtful question. In general, learning noise will *not* bring advantage in performing better classification as it does not include useful information. It is true that in some tasks, learning noise or background features may improve generalization and robustness. Nevertheless, for such cases, noise or background features may need to carry *task-relevant* information, such as the labels in classification.
> >
> > However, in our setting, the noise patch is purely Gaussian and is independent of data label, making it *irrelevant* to the classification task. Consequently, overfitting to such noise would be *harmful* for generalization rather than beneficial.
> >
> > We agree that extending the analysis to include label-relevant noise would be an interesting direction for future research.
> >
> > -----
> > **5. (Q4) On the effect of accelerated training/sampling on feature learning.**
> >
> > We believe the balanced feature learning in diffusion model arises from the training with denoising objective. This suggests the use of accelerated sampling *would not affect* the key finding: diffusion models learn more balanced features. Similarly, as demonstrated in Theorem 3.1 that a class of stationary points satisfies the balanced feature learning property, accelerated training also *would not affect* feature learning outcome, although the speed of convergence to these points could be different.
> >
> > ----
> > **6. (Q5) On the use of diffusion models for other tasks, such as object detection.**
> >
> > Diffusion models can indeed be leveraged for downstream tasks beyond classification, including object detection [7], semantic segmentation [8]. As demonstrated in this work, diffusion models learn diverse features from the input images and thus naturally capture the important object-level structure and semantic-level information.
> >
> > Based on these findings, our next step is to explore the advantages of diffusion models with more diverse and complex feature distributions, and to establish feature learning guarantees. Specifically, we will consider data with multiple patches and features. These feature learning results can then be leveraged to discover the capabilities of diffusion models in these more challenging downstream tasks. For instance, the object detection and semantic segmentation can be formulated as identifying the location and separations of features at different patches.
> >
> > ----
> > **Reference**
> >
> > [1] Li et al. Your diffusion model is secretly a zero-shot classifier. *ICCV 2023*.
> >
> > [2] Chen et al. Your diffusion model is secretly a certifiably robust classifier. *NeurIPS 2024.*
> >
> > [3] Cao et al. Benign overfitting in two-layer convolutional neural networks. *NeurIPS 2022.*
> >
> > [4] Jacot et al.   Neural tangent kernel: Convergence and generalization in neural networks. *NeurIPS 2018*.
> >
> > [5] Kou et al. Benign overfitting in two-layer ReLU convolutional neural networks. *ICML 2023.*
> >
> > [6] Chen et al.  Why does sharpness-aware minimization generalize better than SGD?. *NeurIPS 2023.*
> >
> > [7] Zhang et al. Diffusionengine: Diffusion model is scalable data engine for object detection. *Preprint 2023.*
> >
> > [8] Baranchuk et al. Label-efficient semantic segmentation with diffusion models. *ICLR 2022.*
> >
> > [9] Meng et al. Benign overfitting in two-layer relu convolutional neural networks for xor data. *ICML 2024*.
> >
> > [10] Huang et al. Understanding convergence and generalization in federated learning through feature learning theory. *ICLR 2024*.

---

> > > ### Author Response · Authors · 2024-11-30
> > > **Looking forward to your reply**
> > >
> > > Dear Reviewer cG9c
> > >
> > > We have not heard back from you since our last response. We are keen to know whether our responses have addressed your concerns. Given the deadline of discussion phase is approaching, we would be greatly appreciated if you could take a look at our responses and let us know whether there are further questions.
> > >
> > > Best
> > >
> > > Authors

---

> > > > ### Comment · Reviewer_cG9c · 2024-11-30
> > > >
> > > > Thank you for the authors' detailed response. The authors have satisfactorily addressed the majority of my concerns. Consequently, I am inclined to raise my evaluation from 5 to 6, with a favorable inclination toward acceptance

---

### Official Review · Reviewer_VpM3 · 2024-11-04

**Soundness:** 3
**Presentation:** 4
**Contribution:** 3
**Rating:** 6
**Confidence:** 3

**Summary:**

This work study the feature learning mechanism behind the performance of diffusion models in fitting complex distributions and their out-of-distribution transferability. The authors attribute these capabilities to the denoising objective, which results in balanced training and comprehensive representations, in contrast to a classification objective that focuses on specific patterns. A theoretical and empirical analysis are conducted which show that diffusion models (DMs) tend to learn both the signal and noise in a balanced manner, in contrast to supervised classification, which focuses either on signal or noise. These claims are supported using a synthetic dataset and an edited MINTS dataset.

**Strengths:**

This work addresses a highly interesting problem: elucidating the underlying mechanisms in diffusion models that lead to their performance and transferability.
The authors have done a good job in building a framework where the mechanism and learning dynamics can be analyzed both empirically and theoretically.
The problem is well-motivated, and the paper is generally pleasant to read.

**Weaknesses:**

1) The paper’s objective is to analyze feature learning in diffusion models, which could explain their success. A detailed comparison with supervised classification is provided, however a comparison to other generative approaches such as GANs or VAEs is missing. Therefore, it is unclear if the balanced learning regime is due to DMs' specific denoising based learning, or due to a more abstract or fundamental difference between generative and discriminative approaches.

2) The paper could benefit of more exhaustive experimental setting such a case with larger feature space rather than one label.

3) The organization and the notation of the paper could be improved.

**Questions:**

1) It is interesting to analyze the learning regimes of diffusion models in comparison to classification objectives. However, it is somehow expected that a generative learning objective (as in diffusion models) results in more balanced learning, unlike the classification objective, which is designed to focus on learning features related to specific labels. A natural question that arises is how DMs behave in comparison to other generative approaches, such as GANs or VAEs. Could the authors discuss this point or provide experimental results?

2) In the diffusion case: Since the neural network input is a noisy version of the data sample, the SNR should be altered as additional noise is introduced. Consequently, the SNR of the neural network input differs between the diffusion model and the classification case. Could you please elaborate on this point?

3) DMs can also be viewed as maximizing the Evidence Lower Bound (ELBO) with a form of data augmentation [1]. Since the neural network input is a noisy version of the data sample, unlike in the classification case, it may be unclear which factors are really influential. Could you elaborate on this point? If data augmentation were applied in the classification case, what behavior would be expected?

4) Redundant content between Sections 1.1 and 2 should be revised.

5) The notation is sometimes difficult to follow in several parts of the paper and needs to be revised:

   * In 1.1, some elements are used without being introduced, such as \(d\). I assume \(d\) and \(n\) refer to vector dimensions, but it would be better not to rely on guesswork. Other variables like \(r\), \(\gamma\), and \(k\) are first used without any introduction.

   * In Section 2, \(W\) and \(x\) sometimes have one or two subscripts, which is confusing.

   * Figures 2 and 4 should be improved (Y Axis labels missing ).

6) In Section 5.1, Figure A.1 should be moved to the main text for better cohesion with the section.

[1] Kingma, D., & Gao, R. (2024). Understanding diffusion objectives as the elbo with simple data augmentation. Advances in Neural Information Processing Systems, 36.

---

> ### Author Response · Authors · 2024-11-21
> **Responses to Reviewer VpM3 (Part 1)**
>
> We are grateful for the positive feedback and recognition of the strengths of our work. We also thank the reviewer for providing feedback and comments. Below, we provide detailed responses to all the comments and questions.
>
> ----
> **1. (W1 + Q1) Comparison to other generative models.**
>
> We would like to emphasize that the focus of this paper is to understand the training and feature learning dynamics of *diffusion models*, rather than all generative models. While characterizing the feature learning capabilities of other generative models is an interesting direction, it is beyond the scope of this paper.
>
> Although the focus is on diffusion models, we believe our analysis in this paper can be generalized to other generative models, such as VAEs.  We have now included a discussion in **Appendix F of the revised manuscript**, where we explore the feature learning dynamics of VAEs. Under the current model setup (where we take expectation over the latent sample), we show the objective of VAE includes a reconstruction term  with a specific regularization:
> $$L = \frac{1}{n} \sum_{i=1}^n \sum_{p=1}^2   || x_i^{(p)} - W^\top (W x_i^{(p)})^2 ||^2 + L_{\rm reg}.$$
> The reconstruction loss (the first term of the above equation) forces the network to learn both the signals and noise. Specifically, if we compute the gradient of the reconstruction loss, we can see in the early stage, the updates can be written as
> $$\langle w_{r}^{k+1} , \mu_j  \rangle = \langle w_r^k , \mu_j\rangle + \Theta( \eta \langle w_r^k, \mu_j \rangle^3 )$$
> $$\langle w_{r}^{k+1} , \xi_i  \rangle = \langle w_r^k , \xi_i\rangle + \Theta( \frac{\eta}{n} \langle w_r^k, \xi_i \rangle^3 )$$
> Then we can follow a similar analysis as for diffusion models to analyze the feature learning dynamics of VAEs and show that VAEs also learn balanced features.
>
> To  validate the claim numerically, we conducted **additional experiments training a VAE** on the same synthetic dataset used in the main paper. The results, presented in **Appendix F**, demonstrate that under both low and high SNR settings, the VAE behaves similarly to diffusion models, learning more balanced features compared to classification models.
>
> -----
> **2. (W2) More exhaustive experimental setting, such as with larger feature space rather than one label.**
>
> We have now included an **additional experiment** on 10-class MNIST dataset in **Appendix I of the revised version**. This dataset contains more diverse features compared to the 2-class MNIST dataset (in the main paper). Even with a larger feature space and more labels, we observe similar learning patterns as for the two-class setup, i.e., classification models tend to learn some partial features over the others, while diffusion models learn all the features.
>
> ----
> **3. (W3 + Q4 + Q5 + Q6) "Organization and notation of the paper could be improved."**
>
> Thank you for the suggestions on improving the clarity of the paper. We have incorporated your suggestions in the revised version.
>
> (**Q4**): We have now removed some redundancy in Section 1.1 and Section 2. Nevertheless, we believe  Definition 2.1 in Section 2 is necessary to formalize the data distribution, as briefly introduced in Section 1.1.
>
> (**Q5**): (a) In Section 1.1, the notation $d$ is already introduced in Line 69, which corresponds to the feature dimension and the notation $n$ is introduced in Line 85, which refers to sample size. Variables $r$ are now introduced in Line 190.
> (b) We have now removed the subscript $t$ when introducing neural network functions for diffusion model for clarity purpose. (c ) We have now added y-axis label to all figures thanks to your suggestion.
>
> (**Q6**): Thank you for the suggestion, we have now moved the Figure in Appendix A.1 to Section 5.1.
>
> We hope we have addressed you concerns regarding organization and notation.
>
> ----
> **4. (Q2) SNR of input differs between the diffusion model and classification case.**
>
> In our work, SNR is defined as the ratio between the signal and noise in the *clean data*, which is used as a criterion to quantify the difficulty of a learning problem. It is *not correct* to claim that SNRs of inputs differ between diffusion and classification models. This is because the noise addition process is part of the diffusion model, known as the forward diffusion process. As a result, the input to diffusion models is the clean data, which is consistent with the classification models. Thus, we ensure the SNR of inputs is consistent across both settings.

---

> > ### Author Response · Authors · 2024-11-21
> > **Responses to Reviewer VpM3 (Part 2)**
> >
> > **5. (Q3) On the effect of data augmentation.**
> >
> >
> > We would like to clarify that the balanced feature learning in diffusion model is due to the training via denoising loss rather than the Gaussian noise data augmentation (as shown in reference [1]). While some non-trivial data-specific augmentation strategies, such as Mixup [2] or feature permutation [3] are able to improve feature learning of some less significant features, simple Gaussian noise perturbation (as in diffusion model) cannot achieve the same effect. This is because Gaussian noise perturbation effectively introduces a regularization term that drives weights towards zero. Hence, we believe using noised inputs for classification *would not* enable the model to learn more balanced features. We verify this claim both *theoretically* and *numerically*.
> >
> > * **Theory**: We derived the gradient of a noise added logistic loss in **Appendix J of the revised manuscript**. Taking expectation of the gradient with respect to the noise added, we find that
> > $$E_{\epsilon_{i} } [\nabla_{w_{j,r}} \tilde L_S(W^k)] \approx \nabla_{w_{j,r}} L_S(W^k) + \frac{1}{nm} (\sum_{i=1}^n \ell_i'^k) w_{j,r}^k$$
> > where $\tilde L_S(W) = \frac{1}{n}\sum_{i=1}^n \ell(y_i f(W, x_i + \epsilon_i))$ and $L_S(W) = \frac{1}{n}\sum_{i=1}^n \ell(y_i f(W, x_i))$ denote the noise-added objective and noise-free objective respectively. The above expression of gradient implies that the Gaussian noise perturbation is equivalent to (in expectation) adding an $L_2$-type regularization. This regularization will perform the same penalization for both learning signals and noises, thus will not alter our current results for classification models.
> >
> > * **Experiments**: We also conducted an **additional experiment** in **Appendix J** , examining the feature learning dynamics of classification models with the random Gaussian noise perturbation. The results demonstrate that even with Gaussian noise present, *classification models predominantly learn partial features*. This validates the claim that simple data augmentation, such as Gaussian perturbation, does not encourage classification to learn more balanced features.
> >
> > ----
> > **Reference**
> >
> > [1] Kingma, D., & Gao, R. Understanding diffusion objectives as the elbo with simple data augmentation. *NeurIPS 2024*.
> >
> > [2] Zou et al. The benefits of mixup for feature learning. *ICML 2023.*
> >
> > [3] Shen, et al.   Data augmentation as feature manipulation. *ICML 2022.*

---

> > > ### Comment · Reviewer_VpM3 · 2024-11-25
> > >
> > > I thank the authors for their response. The presentation of the paper has improved, and the additional answers and content addressed my concerns. Therefore, I have raised the score and updated the ratings.

---

### Official Review · Reviewer_nXhk · 2024-11-07

**Soundness:** 3
**Presentation:** 2
**Contribution:** 3
**Rating:** 6
**Confidence:** 3

**Summary:**

This work presents a novel theoretical framework for analyzing the feature learning dynamics in diffusion models, marking the first such contribution to the existing literature. Through rigorous analysis, the authors demonstrate that diffusion models inherently promote the learning of more balanced features, in contrast to traditional classification methods, which tend to prioritize certain features over others. Additionally, the framework can be adapted to accommodate more complex data settings. This theoretical insight has significant implications for downstream tasks, such as out-of-distribution classification, where only these rare or weak features may be present.

**Strengths:**

1. The theoretical findings are solid and  innovative.
2. The contribution appears valuable for comprehending the internal mechanisms of diffusion models and neural networks.

**Weaknesses:**

1. The presentation requires enhancement. Some notations are introduced without prior definition upon their first occurrence, e.g., what does $m$ refer to in line 188?
2. The classifier considered in this manuscript lacks sufficient generality.

**Questions:**

1. My main concern is how applicable the theoretical findings are to a broader range of classifiers and diffusion models. To be honest, the shallow neural network function defined in this manuscript appears unsuitable for real-world data.
2. Is the requirement for orthogonality in data setup common in practice?

---

> ### Author Response · Authors · 2024-11-21
> **Responses to Reviewer nXhk**
>
> We sincerely thank the reviewer for all the positive comments and general appreciation of our work. Below, we address the questions and concerns raised in detail.
>
> ----
> **1. (W1) Presentation requires enhancement. Meaning of $m$ in Line 188.**
>
> We have proofread the paper and have added introduction to the notations, including $m$ in Line 188, which refers to the network width, $k,r,t$ in Line 220, $\omega(\cdot)$ in Line 60.
>
> -----
> **2. (W2 + Q1) Classifier considered lacks sufficient generality. Applicability of theoretical findings to broader range of models.**
>
> Given there is no prior work analyzing the feature learning dynamics of diffusion models, we follow existing theoretical works on classification models [1,2,3,4,5] to establish the model setups. Although we focus on simplified models, they successfully capture the primary factors necessary to understand and compare the training dynamics of both classification and diffusion models. Furthermore, we believe the model setup is suitable for theoretical studies, given the *non-linear, two layer structure* and the *non-convex* nature of neural network optimization.
>
> To further support the applicability of our theoretical findings to broader model classes, we have conducted **additional experiments** on *three-layer classifier and diffusion model with ReLU activation*. Details of the setup and the results are included in **Appendix H of the revised version**. Notably we observe a similar pattern as for the two-layer network setup. That is, classification tends to bias learning towards one of the features while diffusion model learns more balanced features. These findings verify the generality of our findings.
>
> As the first study on diffusion model training dynamics, we believe our work lays a foundational framework that can inspire future research into more general problem setups and more complex model architectures.
>
> ----
>
> **3. (Q2) "Requirement on orthogonality in data setup common in practice?"**
>
> The orthogonality assumption is introduced to simplify the analysis, enabling clear insights into the training dynamics of diffusion models and classification. While perfect orthogonality is often not common in practice, this assumption has been commonly adopted in prior theoretical works [1,3,4] to facilitate the study of complex dynamics in high-dimensional spaces.
>
> We believe extending the analysis to account for non-orthogonal features could follow a similar approach (as done in this paper), like the previous work [6] that studies the non-orthogonal XOR data models for classification problems. However, such an extension would require a more rigorous and detailed analysis, which we leave for future work.
>
> -----
> **References**
>
>
> [1] Cao et al. Benign overfitting in two-layer convolutional neural networks. In *NeurIPS 2022.*
>
> [2] Chen et al. Why does sharpness-aware minimization generalize better than SGD?. In *NeurIPS 2023*.
>
> [3] Chen et al. Understanding and improving feature learning for out-of-distribution generalization. In *NeurIPS 2023*.
>
> [4] Lu et al. Benign Oscillation of Stochastic Gradient Descent with Large Learning Rate. In *ICLR 2024*.
>
> [5] Kou et al. Benign overfitting in two-layer ReLU convolutional neural networks. In *ICML 2023*.
>
> [6] Meng et al. Benign Overfitting in Two-Layer ReLU Convolutional Neural Networks for XOR Data. In *ICML 2024*.

---

> > ### Comment · Reviewer_nXhk · 2024-11-26
> > **Re: Responses to Reviewer nXhk**
> >
> > I would like to express my gratitude to the authors for their detailed response. It would be helpful if they could discuss the potential technical challenges involved in establishing theoretical results for more complex neural networks. As the first study on diffusion model training dynamics, I believe this work is a good advancement. I remain positive about this work.

---

### Author Response · Authors · 2024-12-03
**Thank you for your efforts**

Dear ACs and Reviewers

We sincerely thank the ACs for managing our submission and the reviewers for their thoughtful feedback and appreciation of our work. We are glad that our responses have resolved the concerns.

To *Reviewer nXhk*: we comment that extending to complex networks indeed presents significant challenges due to the intricate interplay of multiple weights, which cannot be fully captured by dynamics based on fixed patterns. We will include a more detailed discussion on this aspect in the next version of the manuscript.

Best

Authors

---

### Meta-Review · Area_Chair_zJbi · 2024-12-20

**Metareview:**

During the review phase the reviewers raised several concerns about the paper including the assumptions being made, the exposition, and the generality of applications of the proposed work. After the rebuttal discussions the reviewers are largely in agreement that the paper makes significant contritbutions in fowarding the theoretical basis and understanding of diffusion models.

**Additional Comments On Reviewer Discussion:**

No separate reviewer discussions were done, or needed since the reviewers are largely in agreement.

---

### Decision · Program_Chairs · 2025-01-22

Accept (Poster)